# Learning to Execute Graph Algorithms Exactly with Graph Neural Networks

**Muhammad Fetrat Qharabagh** [1]   **Artur Back de Luca** [1]   **George Giapitzakis** [1]   **Kimon Fountoulakis** [1]

## Abstract

Understanding what graph neural networks can learn, especially their ability to learn to execute algorithms, remains a central theoretical challenge. In this work, we prove exact learnability results for graph algorithms under bounded-degree and finite-precision constraints. Our approach follows a two-step process. First, we train an ensemble of multi-layer perceptrons (MLPs) to execute the local instructions of a single node. Second, during inference, we use the trained MLP ensemble as the update function within a graph neural network (GNN). Leveraging Neural Tangent Kernel (NTK) theory, we show that local instructions can be learned from a small training set, enabling the complete graph algorithm to be executed during inference without error and with high probability. To illustrate the learning power of our setting, we establish a rigorous learnability result for the LOCAL model of distributed computation. We further demonstrate positive learnability results for widely studied algorithms such as message flooding, breadth-first and depth-first search, and Bellman-Ford.

## 1. Introduction

Algorithmic tasks are a critical frontier for testing the performance of neural networks. As models become more capable, algorithmic computation benchmarks have been increasingly used to evaluate neural networks on progressively more challenging tasks (Saxton et al., 2019; Dua et al., 2019; Hendrycks et al., 2021; Cobbe et al., 2021; Anil et al., 2022; Veličković et al., 2022; Markeeva et al., 2024; Estermann et al., 2024; Gao et al., 2025). This has contributed to a renewed interest in understanding how neural networks can learn and reliably compute algorithms.

While the computational capabilities of neural networks have been theoretically established (Siegelmann & Sontag, 1992; Pérez et al., 2021; Wei et al., 2022), less is known about when training can reliably recover desired representations. Many results provide probabilistic learning guarantees (Wei et al., 2022; Malach, 2024), but these are typically approximate. This is inadequate for computational tasks requiring iterative execution, where approximation errors can compound and invalidate computations. This concern is echoed by György et al. (2025), who argue that reliable deductive reasoning in learning-based systems requires moving beyond high performance on sampled reasoning tasks and toward exact learning, where learned rules are expected to apply correctly to all well-formed inputs.

In more recent work, Back de Luca et al. (2025a) moves toward non-approximate learning guarantees for binary algorithms. Rather than training on standard end-to-end input–output examples, they encode algorithmic instructions in the training set. However, their results apply to feedforward networks, which are poorly suited to variable-size inputs, limiting the scope of the guarantees, as discussed in Section 6.

In this work, we extend these results to graph neural networks (GNNs). Using Neural Tangent Kernel (NTK) theory in the infinite-width regime, we derive exact learnability results for graph algorithms. We show that, under a particular encoding scheme, a GNN can implement the local update rules of distributed graph algorithms by training the node-level MLPs on an efficient set of binary instructions. Unlike the feedforward results of Back de Luca et al. (2025a), our architecture is better suited to graphs, as every node shares the same local model. Consequently, the number of instructions remains constant or grows only logarithmically with the maximum graph size imposed by finite-memory constraints. In contrast, executing the same algorithms with a feedforward model would require encoding the entire graph in the input vector, leading to feature dimensions and instruction counts that scale linearly or even quadratically with the maximum node count.

Our approach applies to graphs of arbitrary size, but any fixed feature dimension limits the largest graph that can be processed correctly. In general, our implementations impose a bound on the maximum node degree, and some algorithms

---

[1]David R. Cheriton School of Computer Science, University of Waterloo, Waterloo, Ontario, Canada. Correspondence to: Muhammad Fetrat Qharabagh <m2fetrat@uwaterloo.ca>.

*Proceedings of the $43^{rd}$ International Conference on Machine Learning*, Seoul, South Korea. PMLR 306, 2026. Copyright 2026 by the author(s).

additionally require a bound on the node count. Under these conditions, we establish exact learnability for Message Flooding, Breadth-First Search, Depth-First Search, Bellman-Ford, and more generally, any algorithm within the LOCAL model of distributed computation (Angluin, 1980; Linial, 1992; Naor & Stockmeyer, 1993).

## 2. Related Work

The closest work to ours is Back de Luca et al. (2025a), which establishes exact learning guarantees via Neural Tangent Kernel (NTK) theory. Both our work and Back de Luca et al. (2025a) build on the NTK framework introduced in Jacot et al. (2018); Lee et al. (2019). We adopt a similar proof strategy, but extend the analysis to graph neural networks, whose architecture is more naturally aligned with graph algorithms than the feedforward networks in Back de Luca et al. (2025a). This alignment yields a more efficient training dataset and embedding dimension, as extensively discussed in Section 6.

As illustrated in Figure 1, our framework is initialized by training a set of shared local MLPs on the same data to exactly execute local instructions for every node. Note that all MLPs are trained on the same data. We then use these trained MLPs within a message passing architecture, which enables processing entire graphs using the same local rules. This local training and global inference perspective is conceptually related to classical label propagation and its connections to graph neural networks (Wang & Leskovec, 2020; Jia & Benson, 2020; Huang et al., 2021), which are more commonly studied in the context of node classification.

Beyond exact learnability, Wei et al. (2022); Malach (2024) provide probabilistic guarantees for approximating Turing-computable functions. However, these guarantees are non-exact and rely on distributional assumptions over training and test data. Such assumptions are natural when learning from input–output samples, but because we train directly on instructions, we avoid input distributions altogether. Furthermore, for graph algorithms, Nerem et al. (2025); Garnier et al. (2025) study specific GNN variants trained on hand-crafted datasets. They show that achieving sufficiently low error on a sparsity-regularized objective forces the learned GNN to implement the target algorithm and to extrapolate to arbitrarily large graphs under structural constraints (Bellman-Ford in Nerem et al. (2025) and a finite-volume operator on PDE meshes in Garnier et al. (2025)). However, these results do not guarantee that standard gradient-based training reaches this near-optimal regime.

More broadly, other theoretical work studies neural-network computation by relating architectural expressive power to formal models of computation (Siegelmann & Sontag, 1992; Loukas, 2020; Pérez et al., 2021; Wei et al., 2022; Sanford

et al., 2024; Back de Luca et al., 2025b). Closest to ours is Loukas (2020), which establishes an expressive-power equivalence between GNNs and the LOCAL model of distributed computing (Angluin, 1980; Linial, 1992; Naor & Stockmeyer, 1993). We discuss this connection in Section 5.

Along this line of work, other papers establish expressive power by demonstrating simulation of specific algorithms (Giannou et al., 2023; Cho et al., 2025; Yang et al., 2024). In the graph setting, such simulation has been shown for graph algorithms including DFS/BFS and shortest paths (Back de Luca & Fountoulakis, 2024), for maximum flow (Hertrich & Sering, 2025), and for heuristics on harder problems such as graph isomorphism (Xu et al., 2019; Müller & Morris, 2024), knapsack (Hertrich & Skutella, 2023) and other combinatorial optimization problems (Hashemi et al., 2025). However, these expressive-power results typically prove existence of suitable parameters without guaranteeing they can be found efficiently by standard training.

## 3. Background and Notation

We begin by establishing the following conventions: the symbols $\mathbf{1}$ and $\mathbf{0}$ denote, respectively, the all-ones and all-zeros vectors of appropriate dimension, while $[n]$ denotes the set $\{1, 2, \ldots, n\}$. For two vectors $\boldsymbol{u}_1 \in \mathbb{R}^{n_1}$ and $\boldsymbol{u}_2 \in \mathbb{R}^{n_2}$ we denote their concatenation by $\boldsymbol{u}_1 \oplus \boldsymbol{u}_2 \in \mathbb{R}^{n_1+n_2}$. We let $I_n$ denote the $n \times n$ identity matrix. While the statements of our theorems in the main text do not rely on NTK results, their proofs do. Accordingly, a brief review of the necessary NTK theory is given in Appendix B. Next, we establish notation for computation over attributed graphs, which constitutes the focus of this work. Let $G = (V, A, X)$ denote an attributed graph, where $V = \{1, \ldots, N\}$ is the set of nodes with arbitrarily chosen ordering, $A \in \{0, 1\}^{n \times n}$ is the binary adjacency matrix with self-loops, and $X = \{\boldsymbol{x}_u : u \in V\}$ is the collection of node attributes. Each node $u \in V$ has a binary attributes vector $\boldsymbol{x}_u \in \{0, 1\}^k$, which may encode node-level information (e.g., local IDs) as well as edge attributes. For a node $u$, we define its open neighborhood as $\mathcal{N}_u = \{v \neq u : A_{u,v} = 1\}$ and its closed neighborhood as $\mathcal{N}_u^+ = \mathcal{N}_u \cup \{u\}$.

**Distributed computation:** In the context of computation over graphs, we also recall the LOCAL distributed computational model (Linial, 1992). In this model, nodes act as processors, and edges establish communication links between them. Computation unfolds in discrete rounds: in each round, every node may send and receive arbitrarily large messages to and from its neighbors (the message received by node $u$ from neighbor $v \in \mathcal{N}_u^+$ is denoted by $\boldsymbol{h}_{u \leftarrow v}$ and the message sent by $u$ to its neighbors is denoted by $\boldsymbol{h}_{u \rightarrow}$ ), and perform local computations over its current state $\boldsymbol{h}_u$ together with the received messages $\boldsymbol{h}_{u \leftarrow v}$. The only constraints are the graph topology, the maximum

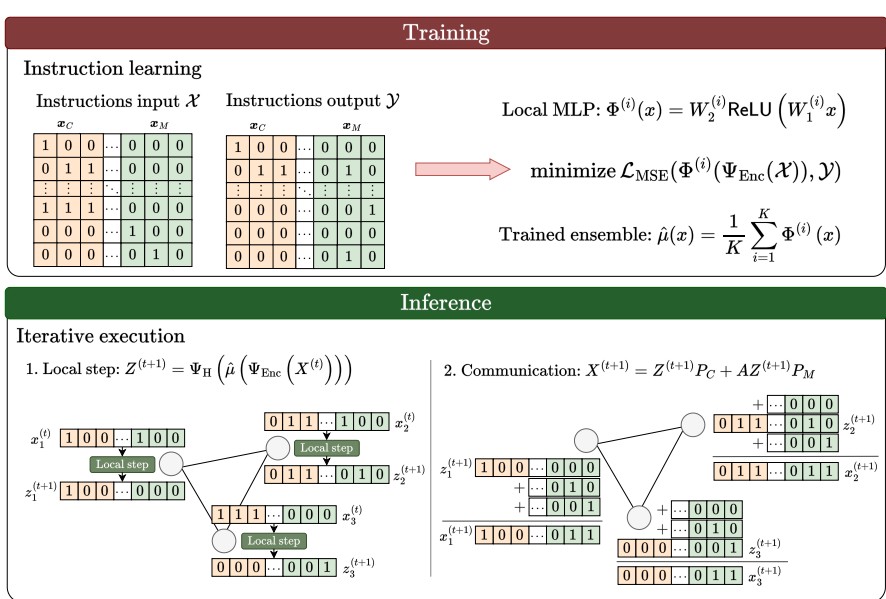

*Figure 1.* Outline of our approach. During training (top), we train $K$ MLP instances $\Phi$ on binary, block-structured instructions that teach local operations. The bits are split into a computation section (yellow) and a message/communication section (green). Inputs are encoded by a fixed $\Psi_{\text{Enc}}$, processed by $\Phi$, and trained by minimizing MSE to ground-truth instruction outputs. We then form an ensemble $\hat{\mu}$ by averaging MLP predictions on each data point. At inference time (bottom), we apply $\hat{\mu}$ to an attributed graph. In Step 1, we compute local node outputs from features using $\Psi_{\text{Enc}}$ and $\hat{\mu}$, then binarize with the step function $\Psi_H$. In Step 2, communication is carried out by message passing using masking matrices $P_C$ and $P_M$, which keep computation blocks local and transmit only message blocks. This procedure yields a GNN that can learn and execute multiple graph algorithms, including Message Flooding, Breadth-First Search, Depth-First Search, Bellman–Ford, and more generally, algorithms representable by the LOCAL model of distributed computation.

number of rounds $L$, and the initial local information available at each node. A pseudocode[1] is provided in Model 1.

A connection between the expressive power of LOCAL

---

**Model 1** LOCAL Distributed Computational Model

---

1: **Initialization:**
2: Set $\boldsymbol{h}_u^{(0)}$ (initial local state) and $\boldsymbol{h}_{u \leftarrow v}^{(0)}$ (initial neighbor messages) for all $u \in V, v \in \mathcal{N}_u^+$.
3: **for** round $\ell \in [L]$ **do**
4:    **Computation Step:**
5:    Compute new state and messages in one operation:

$$\left(\boldsymbol{h}_u^{(\ell)}, \boldsymbol{h}_{u\rightarrow}^{(\ell)}\right) = \text{ALG}\left(\boldsymbol{h}_u^{(\ell-1)}, \{\boldsymbol{h}_{u\leftarrow v}^{(\ell-1)}\}_{v \in \mathcal{N}_u^+}\right).$$

6:    **Message Passing Step:**
7:    Send $\boldsymbol{h}_{u\rightarrow}^{(\ell)}$ to each neighbor $v \in \mathcal{N}_u^+$.
8: **end for**
9: **Return** $\boldsymbol{h}_u^{(L)}, \forall u \in V$

---

and GNNs has already been established in Loukas (2020). However, that work only focuses on comparing expressive

[1]This matches the LOCAL model in (Linial, 1992) up to two conventions: (i) we use a single update that computes both the new state and the outgoing message, and (ii) each node broadcasts one message rather than sending per-neighbor messages. These are equivalent since per-neighbor messages can be encoded in a broadcast message and multiple updates can be composed.

capabilities. In contrast, we provide exact learnability results by showing that our proposed GNN architecture can learn any algorithmic execution in LOCAL, with explicit upper bounds on both message size and local memory.

## 4. The Neural Network Architecture

We now describe the neural network architecture used in this work. Our approach constructs a GNN from an ensemble of MLPs. Each MLP is trained on the same non-graph binary data to perform local operations and differs only by its random initialization. The central idea is to use the ensemble's mean local prediction within a GNN structure to execute steps of graph algorithms exactly. Formally, define

$$\hat{\mu}_K(X) = \frac{1}{K}\sum_{i=1}^{K} \Phi^{(i)}(X), \tag{1}$$

where $\Phi^{(i)} \forall i \in [K]$ are distinct trained MLPs and $K$ is the ensemble size (we omit $K$ when not needed for clarity). Using this ensemble, we define our GNN $F_{\text{GNN}} : \mathbb{R}^{n \times k} \rightarrow \mathbb{R}^{n \times k}$ as:

$$F_{\text{GNN}}(X) = F_{\text{node}}(X)P_C + AF_{\text{node}}(X)P_M,$$
$$\text{where } F_{\text{node}}(X) = \Psi_H(\hat{\mu}(\Psi_{\text{Enc}}(X))) \tag{2}$$

Here $F_{\text{node}}$ represents node-local behavior. The function

$\Psi_{\text{Enc}}$ applies an encoding on the node features by orthogonalizing subsets of input coordinates as in Back de Luca et al. (2025a) and as explained in detail in Appendix D.4.1. The function $\Psi_H : \mathbb{R}^{n \times k} \to \{0,1\}^{n \times k}$ is the entry-wise Heaviside step function, $\Psi_H(X)_{i,j} = 1$ if $X_{i,j} \geq 0$ and $0$ otherwise.

The local predictions are then partially aggregated via the GNN. The matrices $P_C, P_M \in \{0,1\}^{k \times k}$ are column-preserving projection operators masking complementary coordinate subsets, such that $\hat{\mu}(X)_C + \hat{\mu}(X)_M = \hat{\mu}(X)$. These are formally defined as:

$$P_C = \begin{pmatrix} I_{k-k_M} & \mathbf{0} \\ \mathbf{0} & \mathbf{0} \end{pmatrix}, \qquad P_M = \begin{pmatrix} \mathbf{0} & \mathbf{0} \\ \mathbf{0} & I_{k_M} \end{pmatrix}, \quad (3)$$

where $k_M$ denotes the portion of the local predictions (in terms of dimension) that is aggregated across neighboring nodes. This workflow is illustrated in Figure 1, where "Instruction learning" refers to learning local operations via MLPs. The figure shows how the components of our GNN interact during training and inference. Each component in (2) plays a distinct role in ensuring the model executes the intended algorithmic steps:

**Encoding $\Psi_{\text{Enc}}$:** Inputs are structured into functional blocks, each encoding part of the target algorithm. Before being processed by the local models, we apply $\Psi_{\text{Enc}}$ to orthogonalize values within each block. This removes unwanted correlations and is crucial for correct algorithmic execution (Back de Luca et al., 2025a). The procedure remains efficient because it is applied blockwise rather than across the entire input. Details are provided in Appendix D.4.1.

**Heaviside $\Psi_H$:** Before message passing, we apply the step function $\Psi_H$ to the ensemble outputs to binarize them. This prevents conflicting signs from being mixed during aggregation that could otherwise produce incorrect outputs.

**Ensembling:** Instead of relying on a single trained local model, we average across multiple independently trained MLPs. From the perspective of NTK theory, this ensemble better approximates the NTK predictor that captures the desired algorithmic behavior.

**Masking:** The complementary masks in Equation (2) separate which parts of the representation are propagated versus kept local. This reduces unwanted correlations during message passing, preserving locality while still allowing neighborhoods to exchange information. The masks provide a structural bias, while the learned algorithmic instructions determine how this separation is utilized. The parameter $k_M$ is application-dependent. This is a useful feature as the amount of information that needs to be shared is not necessarily the same for all graph algorithms.

**Local training / Global inference:** We train MLPs on non-graph data for local operations, then integrate them into a GNN architecture. This separation between local training and global inference via aggregation ensures that the target graph-level algorithm can be realized by learning only simple local computations. This is conceptually related to label propagation methods for node classification (Wang & Leskovec, 2020; Jia & Benson, 2020; Huang et al., 2021).

**Static training / Iterative inference:** Rather than training on an algorithm's initial input and final output, we train at the level of individual instructions. In classical stepwise training, each sample pairs an input with its one-step output. In our method, the instruction applied in a step is embedded in the sample itself, enabling exact learning of a single step. During inference, we apply this learned step iteratively over multiple rounds to execute the full algorithm. This static training and iterative execution pattern is common in algorithmic learning (Yan et al., 2020), where repeated updates yield the final result.

# 5. Learnability of LOCAL Model

In this section, we present our main learnability result, showing that the GNN architecture of Equation (2) can learn to execute exactly any algorithm expressible in the LOCAL model up to an arbitrarily chosen memory, on graphs with maximum degree $D$. We begin by presenting an informal statement to clarify the implications. The precise statement, along with the full proof, is given in Theorem D.7.

**Theorem 5.1** (Learnability and execution of any LOCAL model using the GNN in Equation (2) (Informal)). *Let $G$ be an input graph. Consider a GNN as in Equation (2) with an ensemble of infinite-width MLPs and any LOCAL-model algorithm $\mathcal{A}$, both operating on graph $G$ with maximum degree $D$. Assume that $\mathcal{A}$ runs for $L$ rounds with finite bounds on the local state size $\left| h_u^{(\ell)} \right|$ and message size $\left| h_{u \to}^{(\ell)} \right|$ as well as the memory size used for the computation, for every round $\ell \in [L]$ and every node $u \in V(G)$. There exists a training dataset whose size scales linearly with the local state size and message size, and quadratically with the graph's maximum degree, such that the GNN can learn to perfectly execute $\mathcal{A}$ in $\mathcal{O}(L)$ iterations. This exact execution is guaranteed with arbitrarily high probability assuming the ensemble size $K$ is polynomial to $D$ and logarithmic to $L$ and the number of vertices $|V|$.*

*Proof outline.* To begin proving this theorem, we utilize a proxy computational model provided in Model 2, the graph template matching framework. This model executes algorithms on graph-structured data using a local function that operates on binary vectors, together with an aggregation mechanism that enables message passing. We prove the theorem in two steps: (i) we show that any algorithm expressed in the LOCAL model with an arbitrarily bounded memory requirement, on a graph of maximum degree $D$, can also

be expressed in the graph template matching framework; and (ii) we show that the GNN in Equation (2) can learn to execute the graph template matching framework. In the following, we present a proof outline of each step.

**Simulation of LOCAL by the graph template matching framework**

*The framework:* In the graph template matching framework of Model 2, each node $u$ has a binary vector $\boldsymbol{x}_u$ with two sections: a *computation section* $\boldsymbol{x}_{u_C}$ and a *message section* $\boldsymbol{x}_{u_M}$. The computation section stores local state and variables that track the algorithm, and the message section is used solely for communication. A local function $f$ operates on $\boldsymbol{x}_u$, and is defined by a set of *templates* $\mathcal{T} = \{(\boldsymbol{z}, \boldsymbol{y})_i\}_{i=1}^p$, each corresponding to a distinct, non-overlapping subset of input bits called a block. Each block may admit certain bit configurations, each associated with an output label. Here, $\boldsymbol{z}$ denotes a specific configuration of a block in the input vector, and $\boldsymbol{y}$ denotes the corresponding output label, which is a vector with the same dimension and structure as $\boldsymbol{x}_u$. The configuration of each template either matches the current state of its block or does not. The output of $f$ is the bitwise OR of the labels of all matching templates.

---

**Model 2** Graph template matching framework

1: **Initialization:**
2: Set $\boldsymbol{x}_{u_C}^{(0)}$ (initial local state) and $\boldsymbol{x}_{u_M}^{(0)}$ (initial neighbor messages) for all $u \in V$.
3: **for** round $\ell' \in [L']$ **do**
4:     compute: $(\boldsymbol{x}_{u_C}^{(\ell')}, \boldsymbol{x}_{u_M}^{(\ell')}) = f\big( \underbrace{\boldsymbol{x}_{u_C}^{(\ell'-1)} \oplus \boldsymbol{x}_{u_M}^{(\ell'-1)}}_{\boldsymbol{x}_u^{(\ell'-1)}} \big)$
5:     aggregate: $\boldsymbol{x}_{u_M}^{(\ell')} = \sum_{v \in \mathcal{N}_u^+} \boldsymbol{x}_{v_M}^{(\ell')}$
6: **end for**
7: **Return** $\boldsymbol{x}_u^{(L')}, \forall u \in V$

---

At each round, after applying $f$, the $\boldsymbol{x}_{u_M}$ sections of the neighboring nodes is aggregated for message passing. Each node is assigned a local ID $u_{\mathcal{L}} \in [D^2 + 1]$, unique within its 2-hop neighborhood. Accordingly, $\boldsymbol{x}_{u_M}$ is divided into $D^2 + 1$ subsections, referred to as communication slots. The templates in $f$ ensure that a node writes its outgoing message only to the slot corresponding to its local ID, preventing distortion during binary aggregation. After aggregation, the updated $\boldsymbol{x}_{u_M}$, containing all received messages, is merged with $\boldsymbol{x}_{u_C}$ to form $\boldsymbol{x}_u$ for the next round. An example of execution is illustrated in Figure 2. We refer the reader to Appendix D.1 for details of the graph template matching framework.

Similar to the message passing phase of the Message Passing Neural Networks framework of Gilmer et al. (2017),

our framework uses an aggregation mechanism that sums features from the neighborhood and an update function that updates the local features based on a combination of the aggregated features as the message and the local features from the previous round.

*Simulation of LOCAL:* Although the workflows of Model 1 and Model 2 appear similar, there is a key distinction: the computation step in Model 1 is assumed to be able to compute any Turing-computable function in a single iteration, whereas a single application of $f$ cannot. Nevertheless, iterative applications of $f$ can compute any Turing-computable function (Lemma D.1).

To show that Model 2 can simulate any LOCAL model, we first characterize a Turing machine that, given a node's previous state and received messages in the LOCAL model, computes the next state and the outgoing message in finitely many steps (Appendix D.3.1). We call this a LOCAL Turing machine (LTM). We then construct a set of templates for $f$ in Model 2. Their number scales linearly with the sizes of the LOCAL state and messages, and quadratically with $D$ (Lemma D.3). In a fixed number of iterations, $f$ proceeds as follows. First, it copies received messages from $\boldsymbol{x}_{u_M}$ into $\boldsymbol{x}_{u_C}$ and runs a binary LTM in $\boldsymbol{x}_{u_C}$ to compute the next state and outgoing messages. During this computation, $\boldsymbol{x}_{u_M} = 0$ (no message passing). Second, it copies the computed messages into the designated slot of $\boldsymbol{x}_{u_M}$. The subsequent aggregation step implements the message passing at the end of a LOCAL round. By initializing $\boldsymbol{x}_{u_C}$ and $\boldsymbol{x}_{u_M}$ identically to the LOCAL model and applying induction, we conclude that Model 2 simulates the LOCAL model with a number of iterations proportional to the number of LOCAL rounds.

**GNN learnability of the graph template matching**

We show that one iteration of the GNN in Equation (2) is equivalent to one iteration of Model 2. The proof proceeds in three steps. First, using Theorem D.5, we show that a training dataset containing each template-label pair of $f$ as a sample, with inputs encoded by $\Psi_{\text{Enc}}$ in Equation (2), suffices to train an NTK predictor that, when followed by a Heaviside function, exactly recovers the ground-truth for any input at inference. Second, assuming that the ensemble of MLPs in Equation (2) is trained on the same dataset, we use NTK theory and a concentration bound for Gaussian random variables to show that the ensemble mean, followed by Heaviside, matches the NTK predictor output (also followed by Heaviside) with arbitrarily high probability, determined by the ensemble size. Finally, note that $P_M$ in Equation (2) preserves the $\boldsymbol{x}_{u_M}$ section of each node and zeros out the rest. Hence, $AF_{\text{node}}(X)P_M$ implements the aggregation step in Model 2. Meanwhile, $P_C$ preserves the $\boldsymbol{x}_{u_C}$ component, so $F_{\text{node}}(X)P_C + AF_{\text{node}}(X)P_M$ yields the same node-wise output as Model 2 after one iteration. $\qquad\square$

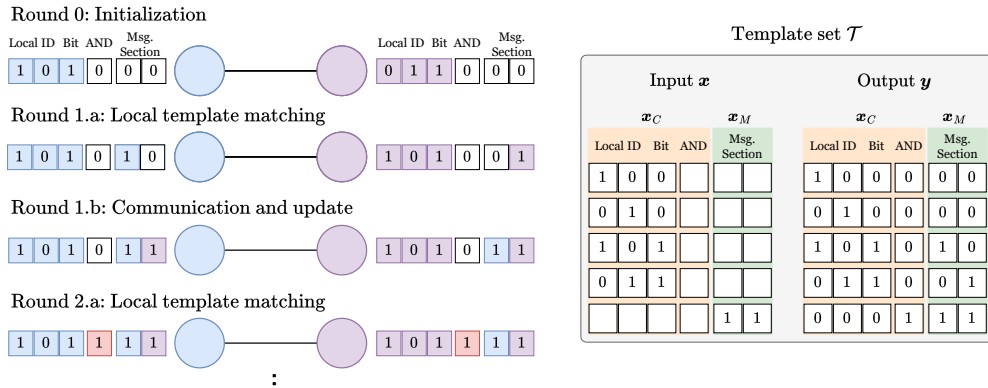

*Figure 2.* Illustration of the template-matching framework on a two-node line graph: the left (blue) node and the right (purple) node. The task computes the logical AND of the bits stored at both nodes in the Bit coordinate. Top-left shows the initial node states, followed by alternating steps of local template matching and communication. Each node begins with a local ID encoding its neighborhood position, followed by its bit value. In Round 1.a, local template matching uses the template set (right), which specifies each block's input–output relation. The computation ($\boldsymbol{x}_C$) and message ($\boldsymbol{x}_M$) components (orange and green) indicate which entries are retained locally and which are shared. Each node sends its Bit value through a designated slot in the Msg. component, determined by its local ID. Here each slot is one bit, and the first Msg. coordinate corresponds to the blue node and the second to the purple node. In Round 1.b, one communication round fills these slots so each node receives its neighbor's bit. In the next matching step (Round 2.a), the AND is applied, and the system converges to a steady state with the result stored in the AND coordinate.

## 6. Results

The conclusion of Theorem 5.1 shows that our architecture can learn to execute any LOCAL-model graph algorithm on graphs with bounded maximum degree and Turing-computable local steps. However, it does not give a direct way to construct training data for a given algorithm. To address this, we derive concrete learnability results for several algorithms bypassing the need for their LOCAL model implementations. The implementations follow the workflow in Model 2. Also, instead of a Turing machine executing the local operations, they are encoded directly in our template-matching framework (outlined in Section 5 and detailed in Appendix D.1). We consider *Message Flooding*, *Breadth First Search* (BFS), *Depth First Search* (DFS), and *Bellman-Ford*. These are canonical primitives spanning traversal (BFS/DFS), local message propagation (Message Flooding), and shortest paths (Bellman-Ford). In what follows, $l$ denotes the bit precision for storing algorithm variables and $d_G$ denotes the diameter of a graph $G$ when all edges are assumed to have unit length. The proofs of Theorems 6.1-6.4 appear in Appendix E.

**Theorem 6.1** (Message Flooding). *The GNN given in Equation* (2) *in the infinite-width regime can learn to exactly simulate the Message Flooding algorithm with arbitrarily high probability on graphs with maximum degree at most $D$, after $\mathcal{O}(D^2 \cdot d_G)$ iterations, using a training dataset of size $\mathcal{O}(l \cdot D^2)$, an embedding dimension $k' = \mathcal{O}(l \cdot D^2)$, and an ensemble size $K = \mathcal{O}\left(l^2 D^4 + lD^2 \log\left(n D^2 d_G\right)\right)$.*

**Theorem 6.2** (BFS). *The GNN given in Equation* (2) *in the infinite-width regime can learn to exactly simu-*

*late BFS with arbitrarily high probability on connected graphs with maximum degree at most $D$ and node count at most $2^l - 1$, after $\mathcal{O}\left(n(l \cdot D + l \cdot d_G + D^2 \cdot d_G)\right)$ iterations, using a training dataset of size $\mathcal{O}(l \cdot D^3)$, an embedding dimension $k' = \mathcal{O}(l \cdot D^3)$, and an ensemble size $K = \mathcal{O}\left(l^2 D^6 + lD^3 \log\left(n^2(lD + ld_G + D^2 d_G)\right)\right)$.*

**Theorem 6.3** (DFS). *The GNN given in Equation* (2) *in the infinite-width regime can learn to exactly simulate DFS with arbitrarily high probability on connected graphs with maximum degree at most $D$ and node count at most $2^l - 1$, after $\mathcal{O}\left(n \cdot D(l + D^2)\right)$ iterations, using a training dataset of size $\mathcal{O}(l \cdot D^3)$, an embedding dimension $k' = \mathcal{O}(l \cdot D^3)$, and an ensemble size $K = \mathcal{O}\left(l^2 D^6 + lD^3 \log\left(n^2 D(l + D^2)\right)\right)$.*

**Theorem 6.4** (Bellman-Ford). *The GNN given in Equation* (2) *in the infinite-width regime can learn to exactly simulate the Bellman-Ford algorithm with arbitrarily high probability on connected weighted graphs with no negative edge weights, longest simple path length $2^l - 1$, maximum degree at most $D$, and node count at most $2^l - 1$, after $\mathcal{O}\left(n^2(l \cdot D + l \cdot d_G + D^2 \cdot d_G)\right)$ iterations, using a training dataset of size $\mathcal{O}(l \cdot D^2)$, an embedding dimension $k' = \mathcal{O}(l \cdot D^2)$, and an ensemble size $K = \mathcal{O}\left(l^2 D^4 + lD^2 \log\left(n^3(lD + ld_G + D^2 d_G)\right)\right)$.*

Note that, assuming the maximum degree is bounded by $D$, test-graph size is unrestricted unless the algorithm necessitates unique global IDs. For example, Message Flooding does not use global IDs, so it applies to arbitrarily large graphs. By contrast, BFS, DFS, and Bellman-Ford use global IDs, so the test-graph size is limited by the bit preci-

sion $l$ used to store the binary variables. With $l$-bit precision, at most $\mathcal{O}(2^l)$ distinct global identifiers can be represented, hence any algorithm that requires unique identifiers is confined to graphs with at most $\mathcal{O}(2^l)$ nodes.

It is also worth noting that the number of GNN iterations needed to simulate these algorithms can exceed the step complexity of their classical centralized counterparts. This gap arises from two factors: (i) *Distributed simulation*: information propagates locally rather than via global access, so operations that are instantaneous with centralized memory may require time proportional to the graph diameter; and (ii) *Execution granularity*: the GNN executes bitwise micro-instructions rather than high-level primitives, so one algorithmic step may require multiple GNN iterations for exact binary execution. The latter point also makes it inappropriate to directly compare the number of GNN iterations with the number of rounds in the LOCAL distributed computing. In the LOCAL model, round complexity abstracts away all local computation: each communication step counts as one round, regardless of the amount of local processing performed between communication steps.

**Comparison with (Back de Luca et al., 2025a).** The main difference lies in the architectures. The architecture in Back de Luca et al. (2025a) is a feedforward network that operates on a single input vector, whereas our architecture is a graph neural network (GNN) with message-passing capabilities. This distinction has important implications for the graph size, instruction count, and ensemble complexity. We discuss each aspect below and show the advantages of our approach.

*(a) Graph size.* In Back de Luca et al. (2025a) the entire graph must be encoded in the input vector: if that vector has $B$ bits then the graph size is upper-bounded by $B$. For test graphs with up to $N$ nodes, encoding the node set requires $\Omega(N)$ bits, hence $B = O(N)$. By contrast, our model does not need to explicitly encode the graph in node feature vectors because the structure is captured via message passing. Our design has two parameters, $D$ (maximum degree) and $l$ (bit precision); the node-feature length $B$ scales linearly with $l$. If an algorithm needs global node IDs represented with $l$ bits and the test graphs have up to $N$ nodes, then $B = O(\log N)$. If global IDs are unnecessary, $B$ does not grow with $N$; for fixed $B$ we can handle arbitrarily many nodes provided the maximum degree is $\leq D$.

*(b) Instruction count.* In Back de Luca et al. (2025a), graph instances may require node-specific instruction blocks that operate on different portions of the global state, so the instruction count scales at least linearly with the maximum node count. In our setting, all nodes share the same local processor, so the instruction count grows polynomially with the maximum degree and only logarithmically with the

maximum node count when IDs are required.

*(c) Ensemble size.* In both frameworks, the ensemble complexity scales quadratically with the embedding dimension, so for a fixed graph size it suffices to compare the required input dimensions. Theorems 6.1-6.4 show that our embedding dimension grows polynomially in the maximum degree and only logarithmically in the maximum node count when IDs are required. In contrast, the encodings for graph algorithms in Back de Luca et al. (2025a) grow at least linearly in the maximum node count. As a result, our framework yields asymptotically smaller ensembles in the large-graph regime, particularly for bounded-degree graphs.

## 7. Experiments

We empirically validate the results of Section 6 and include an ablation on the architecture of Equation (2). Since our theory concerns infinite-width MLPs and may require ensemble sizes that can be prohibitively large, all experiments use Message Flooding, the only algorithm feasible with our computational resources. Nevertheless, we empirically verify the correctness of the instructions used in Theorems 6.1–6.4, as discussed in Appendix C.1. The code for replicating these experiments is available in our GitHub repository[2].

### 7.1. Evaluating Ensemble Complexity

We compute numerical and theoretical lower bounds on the ensemble size in Equation (2) required to learn Message Flooding. The goal is to illustrate the ensemble-complexity scaling outlined in the theorems in Section 6. For the numerical bound, we compute an NTK predictor using the dataset constructed from the instructions used in the proof of Theorem 6.1 and provided in Appendix E. From the NTK output, we extract $\mu$ and $\Sigma$ (defined in equations 6-7) for the output viewed as a Gaussian random variable. We then apply a Gaussian concentration bound to the output, together with a union bound over all iterations, to obtain a sufficient ensemble size for correct execution.

The theoretical bound is obtained by analyzing the output of the NTK mathematically when the input dimension scales as $\mathcal{O}(l \cdot D^2)$, where $l$ is the bit precision and $D$ is the maximum degree. To this end, we leverage the theoretical NTK analysis of Back de Luca et al. (2025a) (Lemma 6.1), which, when applied to our setting, implies that maintaining the high-probability guarantee for exact learning requires the ensemble size to scale as $\mathcal{O}\left(k'^2 + k' \log(nL)\right)$, where $k'$ is the input dimension, $n$ is the number of vertices, and $L$ is the number of algorithm iterations. The same approach can be used to derive ensemble-complexity bounds for other

---

[2]Our GitHub repository is available at `https://github.com/watcl-lab/exact_gnn`

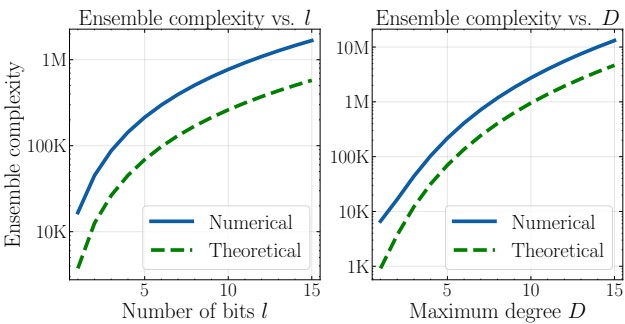

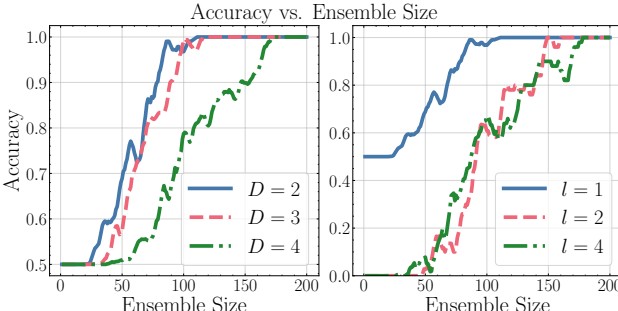

*Figure 3.* Numerical and theoretical lower bounds on the ensemble size required for Message Flooding. Left: ensemble size vs. message bits $l$ with $D = 2$. Right: ensemble size vs. $D$ with $l = 1$. When all other parameters are fixed, the bound grows as $\mathcal{O}(l^2)$ in $l$ and as $\mathcal{O}(D^4)$ in $D$, respectively.

*Figure 4.* Impact of ensemble size (number of trained models) on Message Flooding accuracy as message size $l$ (left) and maximum node degree $D$ (right) increase. In the left plot, we fix $D = 2$ and in the right, $l = 1$. Accuracy is averaged over all trees with $n = 7$ nodes. Accuracy improves with ensemble size, and larger $l$ or $D$ typically requires larger ensembles to match simpler settings.

applications. However, for BFS, DFS, and Bellman-Ford, computing the NTK for large $l$ and $D$ becomes computationally intensive: the input dimension is substantially larger and grows quickly, making NTK training particularly challenging for us.

As shown in Figure 3, when all other parameters are fixed, both the theoretical and numerical bounds grow as $\mathcal{O}(l^2)$ in $l$ (left subplot) and as $\mathcal{O}(D^4)$ in $D$ (right subplot), respectively. They also indicate that correct execution can require large ensembles, already in the thousands, even when either $l$ or $D$ is below 10.

### 7.2. Empirical Evaluation of the Learning Theorem

We empirically validate the NTK predictor approximation for algorithmic execution by training the MLP ensemble in Equation (2), showing that accuracy rises with ensemble size until full accuracy is reached. Because of its simplicity, we use Message Flooding as a suitable testbed for this validation.

We train over increasing bit precision $l$ and maximum degree $D$, sweeping all graphs with such parameters and a fixed node count. To keep evaluation tractable at larger values, we perform Message Flooding on trees, which typically require smaller ensembles and are therefore feasible for larger $l$ and $D$. We also provide a broader evaluation in Appendix C.2 on a randomly selected subset of general graphs with a random number of nodes. When graphs are restricted to trees with maximum degree at most $D$, the number of iterations reduces to $\mathcal{O}(D \cdot d_G)$, the training set size and input dimension can be taken as $\mathcal{O}(l \cdot D)$, and the required ensemble size becomes $K = \mathcal{O}\left((lD)^2 + (lD)\log(nDd_G)\right)$. Thus, the training set size, input dimension, and required ensemble size are all smaller for trees than for general graphs.

We take the set of all trees with $n = 7$ nodes for testing and train the local MLPs on a dataset of binary instructions

from the proof of Theorem 6.1 (see Appendix E.2). We then form a local ensemble as in Equation (1) and apply the GNN layer in Equation (2), executed iteratively. Figure 4 reports accuracy versus ensemble size for increasing message bits and maximum degree, varying each factor independently. Accuracy is computed over all valid messages and trees. We observe perfect accuracy convergence with ensemble growth, and slower convergence for larger $l$ or $D$. We observe the expected increase in required ensemble size with $l$ and $D$, consistent with Section 6. However, the empirical results in Figure 4 are substantially more optimistic than the bounds in Figure 3. We attribute this mainly to the fact that the union bound is conservative, since it sums per-iteration failure probabilities and ignores dependencies across iterations.

### 7.3. Input Encoding and Architecture Design Choices

We investigate whether our architectural choices are necessary for the results of our theoretical framework to hold, using an ablation study on Message Flooding. We ablate the binary encoding $\Psi_{\text{Enc}}$ and the Heaviside $\Psi_H$, also comparing the application of $\Psi_H$ before vs. after aggregation. We omit any ablations on the masking scheme in Equation (2) as it breaks the separation between local computation and communication, directly invalidating any execution. To eliminate approximation noise, we replace the ensemble in Equation (2) with the NTK predictor.

We use complete graphs where all but one node start with a three-bit message. The goal is for the empty node to receive that message. We test graphs with 3-15 nodes over all $2^l$ messages for $l = 3$, excluding the all-zero message. Figure 5 reports accuracy for each variant. Accuracy decreases with node count for all variants except the main model. Because Message Flooding training samples are independent of tree size (see Section 6), the baseline GNN generalizes

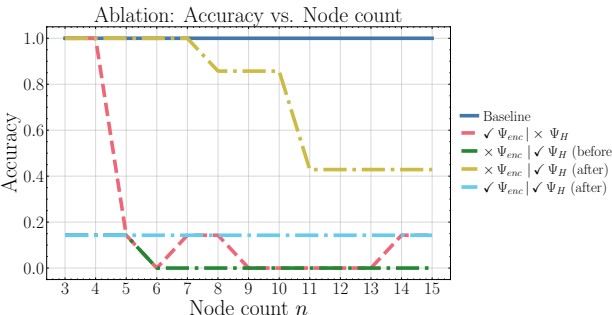

*Figure 5.* Accuracy on the ablation task for variants of Equation (2). We run Message Flooding with $l = 3$-bit messages, and measure accuracy over all non-empty messages. Variants enable or disable $\Psi_{\mathrm{enc}}$, disable $\Psi_H$, or apply $\Psi_H$ before (baseline) or after aggregation. Omitted curves indicate zero accuracy throughout.

to arbitrary node counts, as shown in Figure 5.

We also briefly explain the failure modes of the other variants. Without $\Psi_{\mathrm{enc}}$, correlations with non-matching training samples can shift predictions away from the ground truth. The NTK predictor produces real-valued outputs, with non-positive values mapped to 0 and positive values to 1. If $\Psi_H$ is applied after aggregation, negative neighbor contributions can flip positive bits in a communication slot and corrupt the message. Without $\Psi_H$, these distortions accumulate across iterations and degrade performance further.

## 8. Limitations and Future Work

Our results rely on assumptions, in particular, bounded degree graphs, finite-precision identifiers, and infinite-width NTK-based analyses, which naturally lead to large ensemble requirements and restrict the class of graphs and algorithms that can be handled efficiently. A promising direction for future work is to relax these assumptions by analyzing finite-width training dynamics, improving scalability, and extending our framework and learnability guarantees to other architectures, such as transformers and attention-based graph models.

## Impact Statement

This work provides theoretical results on when graph neural networks can execute algorithms, proving exact learnability under practical constraints. Although the analysis is theoretical and uses synthetic data, it lays a foundation that may later inform neural network design for algorithmic reasoning in routing, network analysis, and large-scale graph processing. While we identify no immediate societal impacts, this work may support longer-term progress toward reliable, interpretable neural systems for graph data.

## Acknowledgments

K. Fountoulakis would like to acknowledge the support of the Natural Sciences and Engineering Research Council of Canada (NSERC). Cette recherche a été financée par le Conseil de recherches en sciences naturelles et en génie du Canada (CRSNG), [RGPIN-2019-04067, DGECR-2019-00147].

G. Giapitzakis would like to acknowledge the support of the Onassis Foundation - Scholarship ID: F ZU 020-1/2024-2025.

A. Back de Luca would like to acknowledge the support of the Natural Sciences and Engineering Research Council of Canada (NSERC).

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

# Appendix

This appendix contains additional related work on empirical efforts in algorithmic execution using neural networks, along with an overview of Neural Tangent Kernel (NTK) theory that is important for the proofs in the main paper. It also includes empirical validations of the correctness of the binary instructions in Section 6, as well as additional results on approximating the NTK predictor as the ensemble size increases. We further discuss and demonstrate the effect of instruction collisions and the resulting performance degradation as a function of the collision rate.

On the theoretical side, we present a proof of learnability for general computations under the LOCAL model of distributed computation. The proof involves multiple intermediate steps, which we explain progressively.

Finally, we provide proofs of the theorems underlying the applications presented in the main paper. We begin by describing the overall proof structure, which is common across applications, and then cover the instructions for each application in detail. Although these instruction sets are extensive, readers do not need to track the low-level logical details to understand the proofs. Only the implications of the instruction sets, such as (i) the resulting input dimension, (ii) the size of the constructed training dataset, and (iii) the number of iterations required to execute the algorithm, are used in deriving the theorems. Accordingly, after presenting the instructions, we include dedicated subsections that analyze the instruction sets and explicitly state the resulting parameter values.

Because the instructions rely on fine-grained binary-level operations, verifying their correctness can be time-consuming. To ensure the binary logic is correct for each application, we implement the corresponding NTK predictor and confirm that our GNN executes the algorithms exactly. We discuss these verification experiments in Appendix C.1 and provide the supporting code in our GitHub repository.

**Table of contents**

# A. Additional related work

Beyond the theoretical works mentioned in Section 2, empirical studies have explored the intersection of neural networks and algorithms. One line of work uses this intersection to motivate architectural or training modifications that incorporate useful computational priors (Graves et al., 2014; Joulin & Mikolov, 2015; Reed & De Freitas, 2016; Kaiser & Sutskever, 2016; Zaremba & Sutskever, 2014; Wang et al., 2017; Trask et al., 2018), and more recently, modifications that directly target the Transformer architecture (Nogueira et al., 2021; Ontañón et al., 2022; Jelassi et al., 2023; Dziri et al., 2023; Liu

et al., 2023; Lee et al., 2024; Cho et al., 2025; McLeish et al., 2024; Zhou et al., 2024). A related line of work trains neural networks, especially message-passing architectures, to execute specific algorithms, often utilizing intermediate supervision. This work spans a range of applications, including search, dynamic programming, and graph problems Tang et al. (2020); Yan et al. (2020); Veličković et al. (2020); Veličković et al. (2022); Ibarz et al. (2022); Bevilacqua et al. (2023); Engelmayer et al. (2023); Rodionov & Prokhorenkova (2025). This paradigm is also leveraged for combinatorial optimization, aiming for more efficient solutions (Cappart et al., 2023; Požgaj et al., 2025). Across these studies, success is typically measured by generalization beyond the training distribution, often evaluated through increased input size (i.e., length generalization).

## B. Neural Tangent Kernel Theory Overview

In this section, we present a brief overview of the Neural Tangent Kernel (NTK) theory results used in this work. For a broader survey of the NTK literature, we refer the reader to Golikov et al. (2022).

For a vector $\boldsymbol{x} \in \mathbb{R}^n$, we write $\|\boldsymbol{x}\| = \sqrt{\sum_{i=1}^n x_i^2}$ for its Euclidean norm. Let $\mathcal{D} \subseteq \mathbb{R}^{k'} \times \mathbb{R}^k$ denote the training set with inputs $\mathcal{X} = \{\boldsymbol{x} : (\boldsymbol{x}, \boldsymbol{y}) \in \mathcal{D}\}$ and labels $\mathcal{Y} = \{\boldsymbol{y} : (\boldsymbol{x}, \boldsymbol{y}) \in \mathcal{D}\}$. We consider a two-layer MLP $\Phi : \mathbb{R}^{k'} \to \mathbb{R}^k$ with no bias terms and hidden width $n_h$ given by $\Phi(\boldsymbol{x}) = W_2 \operatorname{ReLU}(W_1 \boldsymbol{x})$. The parameters are initialized according to the NTK initialization (Lee et al., 2019), namely $W_1^{i,j} = \frac{\sigma_\omega}{\sqrt{k'}} \omega_1^{i,j}$ and $W_2^{i,j} = \frac{\sigma_\omega}{\sqrt{n_h}} \omega_2^{i,j}$ with $\omega_{i,j}^{\{1,2\}} \sim \mathcal{N}(0,1)$. We denote by $\mathbf{W} = \operatorname{vec}(W_1, W_2)$ the parameter vector. At training time $t$, the empirical tangent kernel is $\hat{\Theta}_t(\mathcal{X}, \mathcal{X}) = \nabla_{\mathbf{W}} \Phi_t(\mathcal{X}) \nabla_{\mathbf{W}} \Phi_t(\mathcal{X})^\top \in \mathbb{R}^{k|\mathcal{D}| \times k|\mathcal{D}|}$, where $\Phi_t(\mathcal{X}) = \operatorname{vec}([\Phi_t(\boldsymbol{x})]_{\boldsymbol{x} \in \mathcal{X}}) \in \mathbb{R}^{k|D|}$ denotes the vector resulting by stacking the outputs $\Phi_t(\boldsymbol{x})$ for all $\boldsymbol{x} \in \mathcal{X}$. In the infinite-width limit $n_h \to \infty$, this kernel converges to the deterministic NTK:

$$\Theta(\boldsymbol{x}, \boldsymbol{x}') = \left[\frac{\boldsymbol{x}^\top \boldsymbol{x}'}{2\pi k'}(\pi - \theta) + \frac{\|\boldsymbol{x}\| \cdot \|\boldsymbol{x}'\|}{2\pi k'} \times \big((\pi - \theta)\cos\theta + \sin\theta\big)\right] I_k \in \mathbb{R}^{k \times k}, \tag{4}$$

where $\theta = \arccos\left(x^\top x'/\|x\| \cdot \|x'\|\right)$. At initialization, the outputs converge in distribution to a Gaussian process, i.e. $\Phi_0(\mathcal{X}) \sim \mathcal{N}(0, \mathcal{K}(\mathcal{X}, \mathcal{X}))$, where the NNGP kernel is

$$\mathcal{K}(\boldsymbol{x}, \boldsymbol{x}') = \left[\frac{\|\boldsymbol{x}\| \cdot \|\boldsymbol{x}'\|}{2\pi k'} \times \big((\pi - \theta)\cos\theta + \sin\theta\big)\right] I_k \in \mathbb{R}^{k \times k}. \tag{5}$$

The following result, adapted from Theorem 2.2 in (Lee et al., 2019), states that if $\Theta(\mathcal{X}, \mathcal{X})$ is positive definite and the network is trained with gradient descent (with sufficiently small step size) or gradient flow on the empirical MSE loss $\mathcal{L}(\mathcal{D}) = (2|\mathcal{D}|)^{-1} \sum_{(\boldsymbol{x}, \boldsymbol{y}) \in \mathcal{D}} \|f_t(\boldsymbol{x}) - \boldsymbol{y}\|^2$, then for any test input $\hat{\boldsymbol{x}} \in \mathbb{R}^{k'}$ with $\|\hat{\boldsymbol{x}}\| \leq 1$, as $n_h \to \infty$ the output at training time $t$, $\Phi_t(\hat{\boldsymbol{x}})$, converges in distribution to a Gaussian with mean and variance

$$\mu(\hat{\boldsymbol{x}}) = \Theta(\hat{\boldsymbol{x}}, \mathcal{X})\Theta^{-1}\mathcal{Y} \tag{6}$$

$$\Sigma(\hat{\boldsymbol{x}}) = \mathcal{K}(\hat{\boldsymbol{x}}, \hat{\boldsymbol{x}}) + \Theta(\hat{\boldsymbol{x}}, \mathcal{X})\Theta^{-1}\mathcal{K}(\mathcal{X}, \mathcal{X})\Theta^{-1}\Theta(\mathcal{X}, \hat{\boldsymbol{x}}) - \big(\Theta(\hat{\boldsymbol{x}}, \mathcal{X})\Theta^{-1}\mathcal{K}(\mathcal{X}, \hat{\boldsymbol{x}}) + \text{h.c.}\big) \tag{7}$$

where $\mu$ is referred to as the NTK predictor, $\mathcal{Y}$ is the vectorized collection of labels and "$h.c.$" denotes the Hermitian conjugate.

## C. Experiments

### C.1. NTK Evaluation of Results for Flooding, BFS, DFS, and Bellman-Ford

We verify the correctness of the instructions used in the applications in Theorems 6.1-6.4. This verification also checks the stated dataset sizes and iteration counts. To do so, we use the architecture in Equation (2), replacing the MLP ensemble with an NTK predictor.

We construct the training dataset from the template instructions in Appendix E and use it to build the NTK predictor. We then run the resulting GNN for the number of iterations specified in Section 6. Correctness is validated by comparing the final outputs to ground truth and accepting only exact matches.

For each application, we test the instructions on 100 random graphs. The number of nodes, maximum degree, and (for Bellman-Ford) edge weights are sampled uniformly from $[2, 7]$, $[1, 3]$, and $[0, 3]$, respectively. Conditioned on the number of nodes, each potential edge is included independently with probability 0.5, subject to the degree constraints. Across all test sets, the GNN achieves perfect accuracy on all cases. The verification code is provided in the supplementary material.

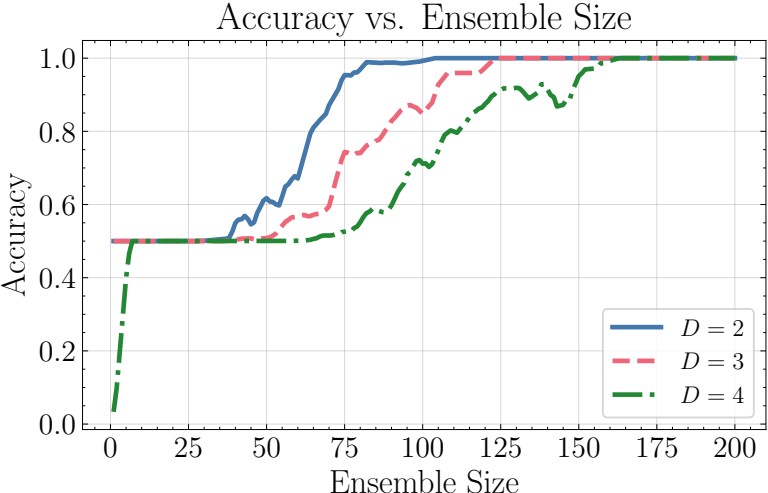

*Figure 6.* The figure shows the number of MLPs that need to be trained as the ensemble in the GNN architecture of Equation (2) to achieve a certain accuracy for executing the Message Flooding algorithm with a 1 bit message on random graphs with maximum degrees $D = 1$, 2, or 3. As the diagrams show, the number of the required ensembles increase as the maximum degree of the graphs grows.

### C.2. Empirical Evaluation of Learning Message Flooding Task for Random General Graphs

Complementing the results of Section 7.2, we empirically verify the learnability of Equation (2) on the Message Flooding algorithm for general graphs. We create three test sets, each with a different maximum degree, $D \in \{1, 2, 3\}$. In all cases, the message passed consists of a single bit. Graphs are generated by first selecting the number of nodes uniformly at random between 5 and 20. Edges are then added randomly, subject to the specified maximum-degree constraint. Each test set contains 2000 generated graphs. Figure 6 reports the number of ensembles required to achieve various accuracy targets for each value of $D$. As expected, the number of ensembles needed to reach a given accuracy increases with $D$.

### C.3. Effect of the Collisions in Templates

In our framework, a *collision* is the maximum number of templates that output 1 at the same coordinate. Collisions can affect execution correctness. We empirically demonstrate this by showing how increasing collisions degrades the accuracy of the NTK predictor. To study this degradation, we consider a simple variant of the Message Flooding algorithm on star graphs. One leaf node holds a non-zero 1-bit message that must be delivered to the center. The center's local register should be set to 1 once it receives the message from any neighbor.

We compare two template strategies. The first (baseline) writes directly to the target bit. Specifically, suppose templates $T_i$ for $i \in [N]$ all write to coordinate $a$ (the target local register), setting $a \leftarrow 1$. Then the conflict at $a$ is $N$. Even when only one leaf contains a non-zero message, this induces collisions on the order of the number of incoming slots, which for a star graph scales with the number of nodes.

We also consider an alternative that mitigates conflicts by trading parallel writes for a sequential cascade. We introduce auxiliary bits, one per template: let $b_i$ denote the bit associated with $T_i$. We modify each $T_i$ to set $b_i \leftarrow 1$ instead of writing to $a$. We then add templates $T'_i$ for $i \in [N-1]$ that propagate activation along the chain: if $b_i = 1$, set $b_{i+1} \leftarrow 1$. Finally, $T'_N$ sets the target bit: if $b_N = 1$, set $a \leftarrow 1$. This replaces the original parallel write with a cascade chain, ensuring that if any template matches, $a$ is eventually set to 1 after a bounded number of iterations. Under this strategy, collisions remain constant, at the cost of additional sequential steps. We use this trick in our applications to control growing conflicts.

In this experiment, we use the architecture in Equation (2), replacing the MLP ensemble with the NTK predictor. We run the model for a large number of iterations to ensure that information propagates to every node in the graph. We then report the average of the center node's local-register over the final three iterations, before applying the Heaviside function.In this setting, a non-positive value corresponds to a zero-valued bit and indicates that execution has failed, whereas a positive value corresponds to a bit equal to 1, indicating that the message is correctly stored in the local register. We repeat this experiment on star graphs with $n = 3$ to 20 nodes.

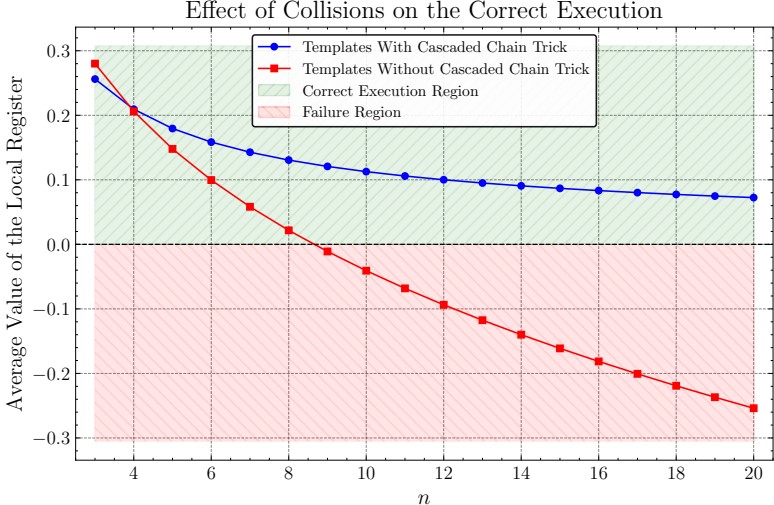

*Figure 7.* Effect of collisions on the execution of Message Flooding on a star graph. The $y$-axis shows the center node's average local-register value. Values above 0 indicate correct execution, while values below or equal to 0 indicate failure. When templates write directly from communication slots to the local register (red), the register value quickly decreases and execution fails for $n \geq 9$, since the number of collisions grows with $n$. In contrast, using the cascade-chain (blue) maintains correct execution for larger $n$, mitigating this issue.

Figure 7 reports the center node's average local-register value. With direct writes to the local register (red), the algorithm succeeds for small $n$, but the register value rapidly decreases as $n$ grows and becomes non-positive for $n \geq 9$, at which point execution fails. In a star graph, the center has $n$ neighbors and hence $n$ communication slots, so increasing $n$ increases the number of templates writing to the same target bit and therefore the number of collisions. These collisions eventually flip the sign of the local register. In contrast, the cascade-chain prevents this failure. The slight decrease in the local-register value under the cascade chain is due to normalization of the binary vector, which typically shrinks individual entries as the dimension increases.

## D. Proof of Theorem 5.1: Learnability of LOCAL model

We present all the arguments, lemmas, and theorems required to prove the formal statement of Theorem 5.1. The entire section is structured as follows: First, in Appendix D.1, we present the details of the graph template matching framework. Then, in Appendix D.2, we show that the local template matching function within our proposed graph template matching framework is Turing complete when executed in a loop. Next, in Appendix D.3, we prove that the graph template matching framework can simulate a LOCAL model. Finally, in Appendix D.4, we prove that the GNN defined in Equation (2) can learn to execute the LOCAL model by learning the graph template matching framework that simulates it.

### D.1. Graph Template Matching Framework

For reference, we reintroduce the *graph template-matching framework* and thoroughly discuss the details of this framework.

---

**Model 3** Graph template matching framework

---

1: **Initialization:**
2: Set $\boldsymbol{x}_{u_C}^{(0)}$ (initial local state) and $\boldsymbol{x}_{u_M}^{(0)}$ (initial neighbor messages) for all $u \in V$.
3: **for** round $\ell' \in [L']$ **do**
4:     compute: $(\boldsymbol{x}_{u_C}^{(\ell')}, \boldsymbol{x}_{u_M}^{(\ell')}) = f\big(\underbrace{\boldsymbol{x}_{u_C}^{(\ell'-1)} \oplus \boldsymbol{x}_{u_M}^{(\ell'-1)}}_{\boldsymbol{x}_u^{(\ell'-1)}}\big)$
5:     aggregate: $\boldsymbol{x}_{u_M}^{(\ell')} = \sum_{v \in \mathcal{N}_u^+} \boldsymbol{x}_{v_M}^{(\ell')}$
6: **end for**
7: **Return** $\boldsymbol{x}_u^{(L')}, \forall u \in V$

---

We assume that we are working with a network whose corresponding graph has maximum degree $D$. Each node has a local ID that might not be unique in the entire network, but it distinguishes the node in its 2-hop neighborhood, i.e. no two nodes among the 2-hop neighbours of a node (including itself) have the same local ID. Such a local ID can be assigned to the nodes by a greedy 2-hop graph coloring where the local IDs are in $\{1, 2, \ldots, D^2 + 1\}$. Our graph template-matching framework is a vectorized distributed computation model that closely resembles the LOCAL model of distributed computation (Model 1). Later, we will show that the introduced framework can simulate a LOCAL model.

In Model 3, node $u$ is represented by a vector $\boldsymbol{x}_u$, which constitutes the memory accessible to the node. We impose several assumptions on the structure of this vector. Specifically, the vectors are binary and consist of $k$ elements; formally, $\boldsymbol{x}_u \in \{0, 1\}^k$. The elements are grouped into $b$ disjoint blocks. Each block $B_i$, for $i \in [b]$, has size $s_i \in \mathbb{N}$. Although block sizes may vary, each $s_i$ is assumed to be bounded and $\mathcal{O}(1)$. We denote by $\boldsymbol{x}_u[B_i] \in \{0, 1\}^{s_i}$ the subvector corresponding to block $B_i$, that is, the bits of $\boldsymbol{x}_u$ belonging to that block.

Furthermore, $\boldsymbol{x}_u$ is divided into two main parts: $\boldsymbol{x}_{u_C}$ and $\boldsymbol{x}_{u_M}$. The component $\boldsymbol{x}_{u_C}$ is used to store bits required for local computation, while $\boldsymbol{x}_{u_M}$ is used to send and receive message bits. The vector $\boldsymbol{x}_{u_M}$ consists of $D^2 + 1$ slots, corresponding to all possible local IDs within a 2-hop neighborhood. Each slot has the same size and may contain multiple bits. A node may write information only to the slot corresponding to its own local ID, denoted by $u_{\mathcal{L}}$.

For node $u$, the vector $\boldsymbol{x}_{u_C}$ may be initialized with the IDs and node attributes of $u$ and its immediate neighbors, as well as the attributes of incoming edges. The vector $\boldsymbol{x}_{u_M}$ may be initialized either with incoming messages or with all zeros (lines 1 and 2 in Model 3).

In each round, two steps occur. First, the function $f : \{0, 1\}^k \to \{0, 1\}^k$ computes $\boldsymbol{x}_{u_C}$ for the next iteration and fills $\boldsymbol{x}_{u_M}$ with the message it intends to send, placing it in the node's dedicated slot (line 4). This step is analogous to the computation step in the LOCAL model. Second, $\boldsymbol{x}_{u_M}$ is aggregated with all $\boldsymbol{x}_{v_M}$ for $v \in \mathcal{N}$ (line 5), where $\mathcal{N}$ denotes the set of neighboring nodes. This step serves the role of message passing in the LOCAL model. Note that since node $u$ writes information only to its own dedicated slot in $\boldsymbol{x}_{u_M}$, the aggregation does not distort bits originating from different nodes. Once the aggregation is complete, $\boldsymbol{x}_{u_M}$ contains the messages from the neighbors in their original, undistorted form.

Function $f$ in Model 3 is defined as follows:

$$f(\hat{\boldsymbol{x}}) := \bigvee_{i=1}^{b} f_i(\hat{\boldsymbol{x}}[B_i]). \tag{8}$$

Here, $\bigvee$ denotes the bitwise disjunction (logical OR). For each $i \in [b]$, the function $f_i : \{0, 1\}^{s_i} \to \{0, 1\}^k$ operates on the content of block $i$. Associated with each block $B_i$ is a finite set of templates $\mathcal{T}_i \subseteq \{0, 1\}^{s_i} \times \{0, 1\}^k$, where each template $(\boldsymbol{z}, \boldsymbol{y})$ maps a block configuration $\boldsymbol{z}$ to an output vector $\boldsymbol{y} \in \{0, 1\}^k$. This form of block-level template matching is adopted from (Back de Luca et al., 2025a). Throughout the paper, we may also refer to such block-configuration-output pairs as *instructions*. The mapping is functional, meaning that no block configuration is associated with more than one output. Consequently, the cardinality of $\mathcal{T}_i$ is at most $2^{s_i}$.

Given $\mathcal{T}_i$, the function $f_i$ is defined as

$$f_i(\boldsymbol{z}) := \begin{cases} \boldsymbol{y} & \text{if } (\boldsymbol{z}, \boldsymbol{y}) \in \mathcal{T}_i, \\ \boldsymbol{0} & \text{otherwise,} \end{cases} \tag{9}$$

where $\boldsymbol{0}$ denotes the all-zero vector in $\{0, 1\}^k$. Each template-matching function is applied independently and simultaneously to its corresponding block.

COLLISIONS IN THE OUTPUTS OF THE TEMPLATES

Let $\mathcal{T}$ denote the set of all templates in $f$, and let a single template be denoted by $T = (\boldsymbol{z}, \boldsymbol{y})$. In our setting, a collision occurs when two or more distinct templates write a nonzero value to the same output coordinate. Formally, for an output coordinate $j \in [k]$, define the set of writers as $W_j = \{ T \in \mathcal{T} \mid \boldsymbol{y}_j = 1 \}$, where $\boldsymbol{y}_j$ denotes the $j$-th coordinate of the output label of template $T$. The collision count at coordinate $j$ is defined as $c_j = \max\{0, |W_j| - 1\}$, that is, the number of writers beyond the first.

We will quantify the maximum number of collisions over all output coordinates for any local template matching function. The reason is that there is a condition on this number when we later derive the learnability results for our graph template matching framework in Section D.4.

**D.2. Turing Completeness of $f$ in the Graph Template Matching Framework**

We begin this section by introducing the definition of a standard Turing machine in Appendix D.2.1. We then prove in Appendix D.2.2 that, when executed in a loop, the local template matching function $f$ defined in Appendix D.1 is Turing complete.

D.2.1. DEFINITION OF THE STANDARD DETERMINISTIC TURING MACHINE

A standard deterministic Turing machine is defined as a tuple $M = (Q, \Gamma, \delta, q_s, H)$, where $Q$ is a *finite* set of states, $q_s \in Q$ is the *initial state*, and $H \subseteq Q$ is the set of *halting states*. The set $\Gamma$ is a *finite* tape alphabet containing a distinguished blank symbol $\sqcup$. The transition function is $\delta : Q \times \Gamma \to Q \times \Gamma \times \{\mathrm{L}, \mathrm{R}\}$.

The machine operates on an infinite tape divided into cells, each containing a symbol from $\Gamma$. Initially, a finite input string $w \in (\Gamma \setminus \{\sqcup\})^*$ is written on an otherwise blank tape (this is referred to as *tape initialization*). The head is positioned on the leftmost symbol of $w$, and the machine is in the initial state $q_s$. At each step, if the current state $q \notin H$ and the symbol under the head is $\gamma \in \Gamma$, the machine computes $\delta(q, \gamma) = (q', \gamma', d)$, where $q' \in Q$ is the new state, $\gamma' \in \Gamma$ is the symbol written in the current cell, and $d \in \{\mathrm{L}, \mathrm{R}\}$ is the direction in which the head moves by one cell. If $\delta(q, \gamma)$ is undefined while $q \notin H$, the machine halts *irregularly*. If $q \in H$, the machine halts *regularly*; the tape contents, up to the rightmost non-blank symbol, are taken to be the output.

A (partial) function $\varphi : (\Gamma \setminus \{\sqcup\})^* \to (\Gamma \setminus \{\sqcup\})^*$ is said to be *Turing computable* if there exists a Turing machine $M$ such that, for every input string $w \in (\Gamma \setminus \{\sqcup\})^*$, if $\varphi(w)$ is defined, then $M$ halts regularly on input $w$ with output $\varphi(w)$; if $\varphi(w)$ is undefined, then $M$ does not halt regularly on input $w$ (i.e., it either runs forever or halts irregularly).

D.2.2. PROOF OF TURING COMPLETENESS FOR THE LOCAL TEMPLATE MATCHING FUNCTION $f$

**Lemma D.1** (Turing Completeness of the Template Matching Framework). *Given the template matching function $f$ defined in Appendix D.1, then for any Turing-computable function $\varphi$, there exists:*

1. *A vector of size $k = \mathcal{O}(m)$, $\hat{\boldsymbol{x}}$, where $m$ is the tape space (tape cells) needed by the TM to compute $\varphi$.*

2. *An encoding of inputs/outputs as binary vectors,*

3. *Template sets $\{\mathcal{T}_i\}_{i=1}^m$,*

*such that iterated application of $f$ on the binary vector computes $\varphi$.*

*Proof.* We prove this lemma constructively by providing templates of a template matching function that simulates the working dynamics of a TM.

**Simulation Setup**

Consider a Turing machine $M = (Q, \Gamma, \delta, q_s, H)$ that computes $\varphi$, where:

- $Q$ is a finite set of states (with initial state $q_s$ and halting states $H \subseteq Q$),

- $\Gamma$ is the tape alphabet (including a blank symbol $\sqcup$),

- $\delta : Q \times \Gamma \to Q \times \Gamma \times \{\mathrm{L}, \mathrm{R}\}$ is the transition function.

Note that $M$ can write only to the cell currently under the head, and at each time step it can move in one of the two directions on the tape, left or right.

In this construction, the template matching function operates on a binary input vector $\hat{\boldsymbol{x}} \in \{0, 1\}^k$ and returns a binary output vector of the same size and structure. This vector serves as the tape of the Turing machine in our simulation. Since the representation is binary, we use a binary encoding for each symbol and each state of $M$.

The vector is partitioned into $m$ blocks, each of size

$$L = |\Gamma| + 1 + |Q| \in \mathcal{O}(1).$$

Each block $B_i$ (for $i \in [m]$) corresponds to a tape cell of $M$ and consists of three sections. Each section contains a binary one-hot encoding of the following entities, in order:

- **Symbol**: $|\Gamma|$ bits, representing the tape symbol $\gamma_i \in \Gamma$ using a one-hot encoding. We assume an arbitrary but fixed ordering of $\Gamma$ for this encoding.

- **Head indicator**: 1 bit, where $h_i = 1$ if the head is at cell $i$, and $h_i = 0$ otherwise.

- **State**: $|Q|$ bits, representing the state $q_i \in Q$ using a one-hot encoding when the head is at cell $i$ (i.e., when $h_i = 1$). As with the symbol encoding, we assume an arbitrary but fixed ordering of $Q$.

Thus, the total vector length is $k = m \cdot L$, which accommodates the $m$ tape cells used by $M$ during computation. Although each block corresponds to a contiguous segment of the vector, for notational convenience we represent the contents of the $i$-th block as a triple

$$B_i = (\tilde{\boldsymbol{\gamma}}, h, \tilde{\boldsymbol{q}})_i,$$

where a tilde ($\sim$) denotes a one-hot binary encoding. We use the subscript $i$ to indicate that a variable belongs to the $i$-th block; for example, $\tilde{\boldsymbol{\gamma}}_i$ denotes the binary-encoded symbol in block $i$. Although the final section of each block is reserved for the state of $M$, it contains a state encoding only when $h_i = 1$.

A vector structured in this manner represents the tape of the Turing machine. Our goal is to define templates for each block $B_i$ over the binary vector such that the output of the template matching function simulates the behavior of the Turing machine $M$ at the $i$-th tape cell, including its transition to one of the adjacent cells. In the following, we define the templates required for each block.

**Block Templates**

Each block $B_i$ has a template set $\mathcal{T}_i$ that defines the function $f_i$ corresponding to the block. There are two cases for the head indicator $h_i$ in block $B_i$: if the Turing machine's head is on the $i$-th cell, then $h_i = 1$; otherwise, $h_i = 0$. For each case, we introduce a corresponding set of templates. Each template is a pair of binary input-output vectors, $(\boldsymbol{x}, \boldsymbol{y})$. Since each template is defined for a specific block of the input, when defining a template—by a slight abuse of notation—we specify only the contents of the intended block rather than the entire vector $\boldsymbol{x}$, with the understanding that all other elements are zero. We now introduce the templates:

1. NON-HEAD BLOCKS ($h_i = 0$)

In this case, both $h_i$ and $\tilde{\boldsymbol{q}}_i$ are zero. For all $\gamma \in \Gamma$, include the following template:

$$((\tilde{\boldsymbol{\gamma}}, 0, \boldsymbol{0})_i, \boldsymbol{y}), \tag{10}$$

where for $\boldsymbol{y}$ we have:

$$\begin{cases} B_i = (\tilde{\boldsymbol{\gamma}}, 0, \boldsymbol{0}) & \text{(the state is irrelevant)}, \\ B_j = \boldsymbol{0} & \text{for all } j \neq i. \end{cases}$$

This construction ensures that non-head blocks preserve their content. Since this template is defined for all $m$ blocks, a total of $m \cdot |\Gamma|$ templates are defined.

2. HEAD BLOCK ($h_i = 1$)

When the head is located on the current block, i.e., $h_i = 1$, the vector $\tilde{\boldsymbol{q}}_i$ encodes the current state of $M$. Let the current state and the symbol under the head of $M$ be $(q, \gamma)$, where $\gamma \in \Gamma$ and $q \in Q$. We define the domain of the transition function as

$$\text{dom}(\delta) = \{ (q, \gamma) \in Q \times \Gamma \mid \delta(q, \gamma) \text{ returns } (q', \gamma', d) \in Q \times \Gamma \times \{\text{L}, \text{R}\} \}.$$

First, consider the case where $(q, \gamma) \notin \text{dom}(\delta)$. The corresponding templates are defined as

$$((\tilde{\boldsymbol{\gamma}}, 1, \tilde{\boldsymbol{q}})_i, \boldsymbol{y}), \tag{11}$$

where in $\boldsymbol{y}$:

$$\begin{cases} B_i = (\tilde{\boldsymbol{\gamma}}, 0, \mathbf{0}), \\ B_j = \mathbf{0} \quad \text{for all } j \neq i. \end{cases}$$

This construction preserves the symbol in the head cell and sets the head indicator to zero, effectively halting the execution of the machine. Note that blocks corresponding to a halting state fall into this category; in particular, when $q \in H$, we have $(q, \gamma) \notin \mathrm{dom}(\delta)$.

Next, consider the case where $(q, \gamma) \in \mathrm{dom}(\delta)$. According to the transition function,

$$\delta(q, \gamma) = (q', \gamma', d),$$

where $d \in \{\mathrm{L}, \mathrm{R}\}$ denotes the head movement direction. For this case, we define the templates for the $i$-th block as

$$((\tilde{\boldsymbol{\gamma}}, 1, \tilde{\boldsymbol{q}})_i, \boldsymbol{y}), \quad \forall (q, \gamma) \in \mathrm{dom}(\delta), \tag{12}$$

where $\boldsymbol{y}$ is specified as follows, depending on the value of $d$:

- $d = \mathrm{R}$ (move right):

$$\begin{cases} B_i = (\tilde{\boldsymbol{\gamma}}', 0, \mathbf{0}), \\ B_{i+1} = (\mathbf{0}, 1, \tilde{\boldsymbol{q}}'), \\ B_j = \mathbf{0} \quad \text{for all other } j. \end{cases}$$

- $d = \mathrm{L}$ (move left):

$$\begin{cases} B_i = (\tilde{\boldsymbol{\gamma}}', 0, \mathbf{0}), \\ B_{i-1} = (\mathbf{0}, 1, \tilde{\boldsymbol{q}}'), \\ B_j = \mathbf{0} \quad \text{for all other } j. \end{cases}$$

The total number of templates required to handle the case $h_i = 1$ for all $i \in [m]$ is $m \cdot |Q| \cdot |\Gamma|$.

### Global Update Function

The templates above are defined separately for each block $i$. The functions $f_i$, constructed from the templates for the $i$-th block, update the vector based solely on the corresponding block. The global function $f$, defined in Equation (8) as the bitwise OR of the outputs of $f_i$ for all $i \in [m]$, ensures that a single iteration of the template-matching function satisfies the following properties: (i) the head moves correctly from the current head block (cell) to the adjacent block, (ii) the new state $q'$ propagates to the new head block, and (iii) non-head blocks retain their symbols.

### Simulation Correctness

We show that each iteration of the template-matching function $f$ exactly simulates one transition of the Turing machine $M$.

First, note that each block has fixed size

$$L = |\Gamma| + 1 + |Q|,$$

so it satisfies the requirements of the framework in Appendix D.1.

**Configuration encoding.** At any time step, the input vector represents a valid configuration of $M$: (i) each block $B_i$ contains exactly one symbol encoding, (ii) exactly one block has head indicator $h_i = 1$, and (iii) the state encoding is present only in the head block. The initial vector encodes the initial tape contents, with the head at cell 1 and state $q_s$.

**Single-step equivalence.** Suppose the current configuration of $M$ has the head at cell cur, reading symbol $\gamma_{\mathsf{cur}}$ and in state $q_{\mathsf{cur}}$.

- If $(q_{\mathsf{cur}}, \gamma_{\mathsf{cur}}) \in \mathrm{dom}(\delta)$ and $\delta(q_{\mathsf{cur}}, \gamma_{\mathsf{cur}}) = (q', \gamma', d)$, then the templates in Equation (12): (i) write $\gamma'$ to block $B_{\mathsf{cur}}$, (ii) remove the head indicator from $B_{\mathsf{cur}}$, and (iii) place the head and state $q'$ in the adjacent block $B_{\mathsf{cur}\pm 1}$ according to $d$.

- If $(q_{\text{cur}}, \gamma_{\text{cur}}) \notin \text{dom}(\delta)$ (including halting states), the templates in Equation (11) preserve the tape contents and remove the head indicator, yielding a halting configuration.

For all $i \neq \text{cur}$, the templates in Equation (10) ensure that $B_i$ remains unchanged.

**Global update.** The global function $f$ is defined as the bitwise OR of all blockwise updates. Since exactly one block contains the head at any time, at most one transition template applies per iteration, and the OR operation produces a unique, well-defined next configuration.

**Conclusion.** Therefore, one application of $f$ simulates exactly one transition of $M$. By induction, repeated application of $f$ simulates the full execution of $M$ until halting. Since $M$ is arbitrary and the vector length scales with the tape space it uses, the template-matching framework is Turing-complete.

$\square$

NUMBER OF COLLISIONS IN THE TEMPLATES OF THE CONSTRUCTION

Recall that a *collision* occurs when multiple templates write to the same bit position in the output vector. We analyze the maximum number of such collisions in the construction.

Consider an arbitrary block $(\tilde{\boldsymbol{\gamma}}, h, \tilde{\boldsymbol{q}})_z$. Only templates associated with block $z$ or its immediate neighbors can write to this block.

*Symbol bits.* Templates writing to the symbol section $\tilde{\boldsymbol{\gamma}}_z$ come only from $\mathcal{T}_z$. If $h_z = 0$, there are $|\Gamma|$ templates, and by one-hot encoding, at most one writes to any given bit. If $h_z = 1$, there are $|\Gamma| \cdot |Q|$ templates corresponding to state-symbol pairs. In the worst case, all such templates may write the same symbol. Thus, for any bit in $\tilde{\boldsymbol{\gamma}}_z$, the total number of possible writers is at most $|\Gamma| \cdot |Q| + 1$.

*Head indicator.* The head indicator $h_z$ can only be written by transition templates from neighboring blocks $z - 1$ and $z + 1$. Each contributes at most $|\Gamma| \cdot |Q|$ templates, yielding at most $2|\Gamma| \cdot |Q|$ possible writers.

*State bits.* Similarly, each bit of the state encoding $\tilde{\boldsymbol{q}}_z$ can only be written by transition templates from blocks $z - 1$ and $z + 1$, again giving at most $2|\Gamma| \cdot |Q|$ possible writers per bit.

*Remark* D.2. In the Turing machine simulation, the number of templates that can write to any single bit position is bounded by $2|\Gamma| \cdot |Q|$. Consequently, the maximum number of collisions in the entire construction is also bounded by $2|\Gamma| \cdot |Q|$.

## D.3. Simulation of the LOCAL Model

Here, we first introduce the *Local Turing Machine* (LTM) in Appendix D.3.1. The LTM is a standard Turing machine with a specific tape layout designed to execute the local computation performed at each node during a single round of the LOCAL model. Then, in Appendix D.3.2, we prove that our proposed graph template matching framework can simulate the LOCAL model. This proof is constructive and proceeds by explicitly building a corresponding graph template matching model. As part of the construction, the LTM is executed within the framework to perform the local computation at each node.

### D.3.1. LOCAL TURING MACHINE

In the LOCAL model, each node acts as a processor that performs local computation based on its internal state and the messages received from its neighbors. We assume each node is Turing complete and that the local computation step corresponds to a Turing-computable function, i.e., a function computable by a Turing machine in a finite number of steps. This assumption is necessary for the LOCAL model to advance between rounds.

To show that a LOCAL algorithm operating on a graph of maximum degree $D$ can be simulated by our graph template matching framework, we introduce a customized tape layout for a Turing machine that emulates the local computation at a node. We refer to this machine as the *LOCAL Turing Machine* (LTM). The LTM is a standard Turing machine initialized with the node's previous state and the messages received from its neighbors, and it computes (i) the node's new state and (ii) the message to be sent to its neighbors. The tape layout is designed to be compatible with the binary vector representation used in our framework and to allow efficient access to intermediate and output information. This construction is later used to prove that our framework can simulate the LOCAL model.

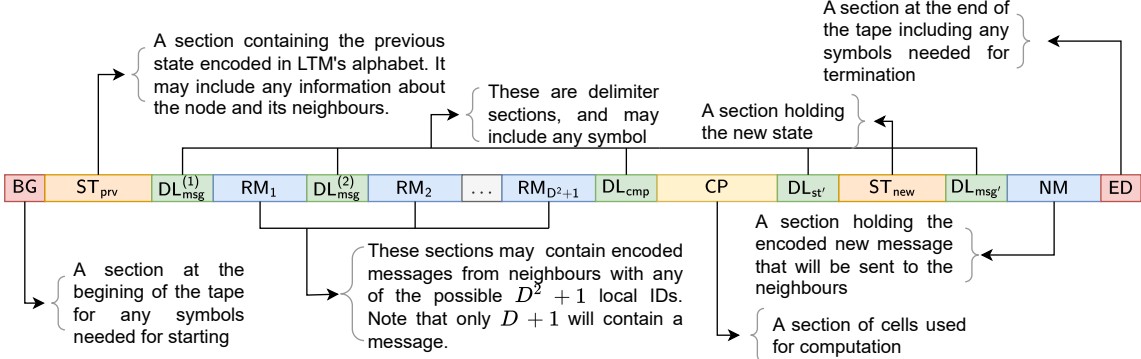

*Figure 8.* Tape layout for the LOCAL Turing machine (LTM). Each section has a fixed, predetermined size and may contain any symbol from the LTM alphabet, including marker and empty symbols.

Figure 8 illustrates the LTM tape layout. The tape is divided into multiple contiguous sections, each of fixed size known in advance for a given LOCAL algorithm. All information is encoded using symbols from the LTM alphabet, which includes the empty symbol and any marker symbols required for correct operation. We assume that the node's previous state and the received messages are already encoded using this alphabet and placed on the tape according to the layout described below. The number of tape cells required to encode a local state and a single message are bounded by $P_{\text{s.t.}}$ and $P_{\text{msg}}$, respectively. By *encoding*, we mean a representation that includes both the information content and any auxiliary symbols (e.g., delimiters or markers) needed by the LTM to process it.

At the beginning of the tape, the section BG contains auxiliary symbols that allow the LTM to initialize its execution. The LTM starts in its initial state $q_s$ with its head positioned at the first cell of BG. This is followed by the section $\text{ST}_{\text{prv}}$ of size $P_{\text{s.t.}}$, which encodes the node's previous local state. A delimiter section $\text{DL}_{\text{msg}}^{(1)}$ separates the state from the message sections.

The tape then contains $D^2 + 1$ message sections, corresponding to the message slots used in our graph template matching framework. Although only $D + 1$ of these sections can be non-empty (messages from the $D$ neighbors and the node itself), all $D^2 + 1$ slots are reserved for compatibility. Each message section $\text{RM}_i$ has size $P_{\text{msg}}$ and is separated from adjacent sections by delimiter blocks $\text{DL}_{\text{msg}}^{(i)}$. The section $\text{RM}_i$ stores the message received from the potential neighbor with local ID $i$. After the final message section, a delimiter $\text{DL}_{\text{cmp}}$ separates the input from the computation region.

The LTM performs its computation using an auxiliary work section CP. Once the new local state and outgoing message are computed, the results are encoded (including any required marker symbols) so that they can be directly reused for initialization in the next round. A delimiter section $\text{DL}_{\text{st}'}$ separates the computation region from the output state section $\text{ST}_{\text{new}}$, which has size $P_{\text{s.t.}}$. This is followed by another delimiter $\text{DL}_{\text{msg}'}$ and the outgoing message section NM of size $P_{\text{msg}}$. Finally, the section ED appears at the end of the tape and contains any symbols required for proper termination. By convention, the LTM halts only when its head is on the last cell of ED and the machine is in the halting configuration $(q_h, \gamma_h)$.

All section sizes and their order are fixed and known in advance for any specific LOCAL algorithm; hence, the total tape size of the LTM is a fixed constant. We emphasize that this construction imposes no restrictions on the internal structure of the encodings within each section. The LTM remains a standard Turing machine with a finite set of states $Q$ and alphabet $\Gamma$, and its internal operation is fully determined by its transition function. The LTM does not perform message aggregation; it solely computes the next local state and outgoing message based on the initialized tape contents.

### D.3.2. SIMULATING THE LOCAL MODEL VIA GRAPH TEMPLATE MATCHING

In this section, we present a lemma that formally establishes that the LOCAL model can be simulated by the graph template-matching framework introduced in Model 3. Specifically, we show that for a LOCAL algorithm running on a graph, the local state of a node $u$ at round $\ell$, denoted by $\boldsymbol{h}_u^{(\ell)}$, can be ggenerated within the graph template matching framework.

More precisely, there exists a round $\ell'$ in the graph template matching framework such that the node representation $\boldsymbol{x}_u^{(\ell')}$ in the framework contains $\boldsymbol{h}_u^{(\ell)}$ as a subrepresentation. This information is encoded using the alphabet symbols of an LTM that

is programmed to perform the local computation step of the LOCAL model. We denote this relationship by $\boldsymbol{h}_u^{(\ell)} \sqsubset \boldsymbol{x}_u^{(\ell')}$.

**Lemma D.3** (Expressivity of the graph template-matching framework)**.** *Let $\mathcal{A}$ be a LOCAL algorithm with a Turing computable computation step running on an attributed graph $G$ with maximum degree $D$, where the state at round $\ell$ for node $u \in V$ is represented by a binary vector $\boldsymbol{h}_u^{(\ell)}$ with an arbitrarily bounded size. Let us also denote the binary state of node $u$ in our graph template matching framework at round $\ell'$ by $\boldsymbol{x}_u^{(\ell')} \in \{0,1\}^k$. For the same attributed graph, there is an instance of our graph template matching framework, $\mathcal{M}$, with a local template matching function $f_{LOC}$, such that for every round $\ell \geq 0$ in the LOCAL model, there exists a corresponding round $\ell'$ in $\mathcal{M}$ for which, the following holds: $\boldsymbol{h}_u^{(\ell)} \sqsubset \boldsymbol{x}_u^{(\ell')}$, for every round $\ell$ and $u \in V$. Part of the templates of $f_{LOC}$, execute an LTM that has a program equivalent to the computation step of the LOCAL model. Assuming this LTM: (i) requires a tape of size $m$, (ii) uses at most $P_{s.t.}$ cells to encode the local state and $P_{msg}$ cells to encode the message of any node at any round of the LOCAL model, (iii) has an alphabet of size $|\Gamma|$, (iv) a cardinality of domain $|\operatorname{dom}(\delta)|$, (v) and takes $\tau$ steps to execute its algorithm. Then $f_{LOC}$ requires $2P_{s.t.}L_\Gamma + (3D^2 + 4)P_{msg}L_\Gamma + mL_\Gamma + m|\operatorname{dom}(\delta)| - m$ templates and the graph template matching round $\ell' = \ell(\tau + 4)$.*

*Proof.* We prove this lemma constructively by specifying (i) the structure of the node representation vector $\boldsymbol{x}_u$ and (ii) a collection of templates for the template-matching function $f_{\text{LOC}}$, within an instance of our graph template matching framework that simulates the LOCAL distributed computation model.

Consider an LTM with the tape layout shown in Figure 8. We assume that this LTM has internal state set $Q$ and an alphabet $\Gamma$, and executing the LTM according to its transition function $\delta(\cdot)$ for $\tau$ steps corresponds to a single computation step performed by each node in a LOCAL model. At a high level, the vector $\boldsymbol{x}_u$ in our construction is partitioned into multiple sections, including (but not limited to): a section storing the previous local state of the LOCAL algorithm, a section storing received messages, and a section encoding a binary representation of the LTM tape.

We construct a graph template matching framework that simulates each LOCAL round using four phases. In the first phase, a subset of templates in $f_{\text{LOC}}$ copies the previous local state and the received messages into the LTM tape, thereby forming the initial configuration. In the second phase, another subset of templates executes the LTM for $\tau$ iterations. In the third phase, a further subset of templates replaces the previous local state with the newly computed state and copies the newly generated message into the appropriate slot of the message section of the vector, denoted by $\boldsymbol{x}_{u_M}$. Finally, in the fourth phase, the aggregation step of the graph template matching framework aggregates $\boldsymbol{x}_{u_M}$ over all neighbors, thereby implementing the message-passing operation performed at the end of each LOCAL round.

We now formally define the structure of $\boldsymbol{x}_u$ and describe the workflow of each phase. In our construction, all information—including local states and messages—is encoded as sequences of LTM alphabet symbols. Each symbol is represented using $L_\Gamma$ bits.

NODE VECTOR STRUCTURE

We structure the node vector $\boldsymbol{x}_u = (\boldsymbol{x}_{u_C}, \boldsymbol{x}_{u_M})$ in our graph template matching framework as follows:

**A. Computation Part ($\boldsymbol{x}_{u_C}$).**  This part is partitioned into the following sections:

(a) **Local state section ($\mathsf{S}_{\text{s.t.}}$).** Size: $P_{\text{s.t.}}L_\Gamma$ bits. Here, $L_\Gamma = |\Gamma| - 1$ is the number of bits used to encode one LTM alphabet symbol, and $P_{\text{s.t.}}$ is the maximum number of LTM symbols used to encode a node's local state in the LOCAL algorithm. This section, which lies outside the LTM tape, stores the node's local state before and after LTM execution as a sequence of LTM alphabet symbols.

(b) **Load state flags section ($\mathsf{S}_{\text{LS}}$).** Size: $P_{\text{s.t.}}L_\Gamma$ bits. These bits act as control flags that trigger copying of the local state from $\mathsf{S}_{\text{s.t.}}$ onto the LTM tape. Each bit to be copied has a corresponding flag bit in this section.

(c) **Load message flags section ($\mathsf{S}_{\text{LM}}$).** Size: $(D^2 + 1)P_{\text{msg}}L_\Gamma$ bits. Here, $P_{\text{msg}}$ denotes the maximum number of LTM symbols used to encode any received message. There are $D^2 + 1$ communication slots, each capable of receiving a message encoded as $P_{\text{msg}}$ LTM alphabet symbols. This section contains one flag bit for each bit of the received messages, signaling their transfer onto the LTM tape.

(d) **Set tape flag section ($\mathsf{S}_{\text{t.flg}}$).** Size: 1 bit. This flag initializes fixed values on the LTM tape, such as the symbols in start, end, and delimiter subsections of the LTM tape illustrated in Figure 8.

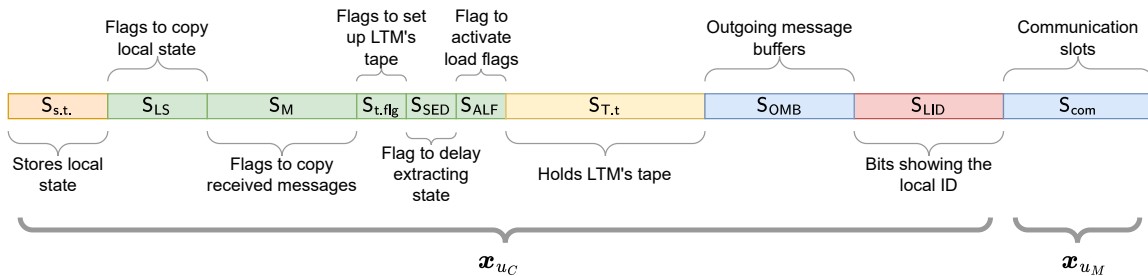

*Figure 9.* The figure shows the structure of the binary input/output vector used for the simulation of the LOCAL model with the graph template matching framework. Note that some sections have multiple subsections that have not been depicted here. For example, $S_{\text{T.t}}$ has its own inner partitions that have been illustrated in Figure 8.

(e) **Turing tape section** ($S_{\text{T.t}}$). This section contains all the bits that represent LTM's tape, with subsections arranged as shown in Figure 8. Each tape cell is represented by a block similar to that used in Lemma D.1, containing: (i) the binary encoded tape symbol, (ii) a head indicator bit, (iii) the binary encoded LTM state ($q \in Q$) when the head indicator is 1, and (iv) an erase flag bit. For subsections storing the output of the LTM computation—namely $ST_{\text{new}}$ and NM—the erase flag also serves as a signal to copy the output outside the tape. If the LTM uses at most $m$ tape cells, the size of this section is $m \cdot L_{\text{cell}}$, where $L_{\text{cell}} = |\Gamma| + |Q| + 1$.

(f) **Extract state delay section** ($S_{\text{ESD}}$). Size: 1 bit. This bit delays activation of the flags used to extract the new local state from the LTM tape by one iteration.

(g) **Activate load flag section** ($S_{\text{ALF}}$). Size: 1 bit. When set to 1, this flag activates the load flags in $S_{\text{LS}}$ and $S_{\text{LM}}$.

(h) **Outgoing message buffer section** ($S_{\text{OMB}}$). Size: $(D^2 + 1)P_{\text{msg}}L_\Gamma$ bits. This section contains $D^2 + 1$ subsections, each temporarily storing a copy of the message generated by the LTM after computation. It acts as a buffer prior to message transmission.

(i) **Local ID section** ($S_{\text{LID}}$). Size: $(D^2 + 1)P_{\text{msg}}L_\Gamma$ bits. This section acts as a selector that routes the message stored in $S_{\text{OMB}}$ to the correct communication slot. It contains $D^2 + 1$ subsections, each of size $P_{\text{msg}}L_\Gamma$. Only the subsection corresponding to the node's local ID is set to all ones.

**B. Communication Part ($\boldsymbol{x}_{u_M}$).**

- **Communication slots section** ($S_{\text{com}}$). Size: $(D^2 + 1)P_{\text{msg}}L_\Gamma$ bits. This section consists of $D^2 + 1$ slots. Node $u$ may write its outgoing message only to the slot corresponding to its local ID $u_{\mathcal{L}}$.

Figure 9 provides an overview of the full binary input/output vector structure. This structure is identical for both input and output vectors.

PHASE 1: MESSAGE INGESTION AND LTM INITIALIZATION (LOAD)

**Purpose:** Copy the local state from $S_{\text{s.t.}}$ and the received messages from $\boldsymbol{x}_{u_M}$ to their designated subsections on the Turing tape, namely $ST_{\text{prv}}$ and $RM_i$ for $i \in [D^2 + 1]$. This phase also initializes the start, end, and delimiter subsections of the tape.

**Trigger:** All bits in $S_{\text{LS}}$, $S_{\text{LM}}$, and $S_{\text{t.flg}}$ are equal to 1.

**Input vector status:** At the beginning of this phase, at most $D + 1$ slots of $\boldsymbol{x}_{u_M}$ are non-empty. These slots contain the messages generated at round $\ell - 1$ of the LOCAL algorithm by the neighbors of node $u$ (including $u$ itself), denoted $\{\boldsymbol{h}_{u \leftarrow v}^{(\ell-1)}\}_{v \in \mathcal{N}_u^+}$, where $\mathcal{N}_u^+$ is the set of neighbors of $u$ together with $u$ itself. Each message is represented as a sequence of symbols from the LTM alphabet. Every symbol is binary encoded using $L_\Gamma = |\Gamma| - 1$ bits: a one-hot encoding is used for all non-empty symbols, while the empty symbol is represented by the all-zero vector of length $L_\Gamma$. In addition, $S_{\text{s.t.}}$ contains the local state from round $\ell - 1$ of the LOCAL model, $\boldsymbol{h}_u^{(\ell-1)}$, also represented as a sequence of symbols from the LTM alphabet.

**Templates:** The local state, messages, and auxiliary information are all stored as sequences of LTM alphabet symbols in binary form. To index individual bits, we refer first to the symbol index and then to the bit index within the $L_\Gamma$-bit encoding of that symbol. Using this convention, we define the following template groups.

**Template group 1** : For all $p \in [L_\Gamma]$ and $s \in [P_{\text{s.t.}}]$:

If the $p$-th bit of the $s$-th symbol in $\mathsf{S}_{\text{s.t.}}$ and the corresponding flag bit in $\mathsf{S}_{\text{LC}}$ are both equal to 1;

Output $\boldsymbol{y}$: All zeros, except:

- the $p$-th bit of the $s$-th symbol in the $\mathsf{ST}_{\text{prv}}$ subsection of the Turing tape is set to 1.

**Template group 2** : For all $p \in [L_\Gamma]$, $s \in [P_{\text{msg}}]$, and $d \in [D^2 + 1]$:

If the $p$-th bit of the $s$-th symbol in the $d$-th communication slot of $\mathsf{S}_{\text{com}}$ and the corresponding flag bit in $\mathsf{S}_{\text{LM}}$ are both equal to 1;

Output $\boldsymbol{y}$: All zeros, except:

- the $p$-th bit of the $s$-th symbol in the $\mathsf{RM}_d$ subsection of the Turing tape is set to 1.

**Template group 3** : For the bit in $\mathsf{S}_{\text{t.flg}}$:

If the bit is equal to 1;

Output $\boldsymbol{y}$: All zeros, except:

- the bits corresponding to the required symbols in the BG, $\mathsf{DL}_{\text{msg}}^{(i)}$ for $i \in [D^2 + 1]$, $\mathsf{DL}_{\text{cmp}}$, $\mathsf{DL}_{\text{st}'}$, $\mathsf{DL}_{\text{msg}'}$, and ED subsections of the Turing tape are initialized as required by the LTM at the beginning of its execution.
- the head indicator bit $h$ in the first cell of BG is set to 1;
- the bits encoding the LTM state in the first cell of BG are set to $\tilde{\boldsymbol{q}}_s$, the binary encoding of the initial LTM state $q_s$.

We now count the number of templates introduced in this phase. Consider template group 1. Since one copy template is defined for each bit in $\mathsf{S}_{\text{s.t.}}$, the total number of templates in this group equals the number of bits in $\mathsf{S}_{\text{s.t.}}$, namely $P_{\text{s.t.}} L_\Gamma$. Similarly, template group 2, which copies the received messages, contains $(D^2 + 1) P_{\text{msg}} L_\Gamma$ templates. Template group 3 consists of a single template.

Although multiple templates are defined for this phase, they are all applied simultaneously within a single iteration of the template matching function. After this iteration, the sections $\mathsf{S}_{\text{LS}}$, $\mathsf{S}_{\text{LM}}$, $\mathsf{S}_{\text{t.flg}}$, $\mathsf{S}_{\text{s.t.}}$, and $\mathsf{S}_{\text{com}}$ are all set to zero.

PHASE 2: COMPUTATION (COMPUTE)

**Purpose:** Execute the LTM program on the initialized tape to compute the output of ALG (where ALG denotes the algorithm executed in the local computation step) at round $\ell$ of the LOCAL model, namely $\boldsymbol{h}_u^{(\ell)}$ and $\boldsymbol{h}_{u\rightarrow}^{(\ell)}$ (line 5 of Model 1).

**Trigger:** The head indicator bit in the first cell of the Turing tape is set to 1.

**Input vector status:** Before starting this phase, the Turing tape is properly initialized. The head indicator in the first cell is equal to 1, and the bits encoding the LTM state in that cell represent $\tilde{\boldsymbol{q}}_s$.

**Templates:** The templates and block structure introduced here are adapted from our construction in the proof of Lemma D.1, with additional components for erasing the tape content and copying the output once the LTM completes execution.

Each cell in all subsections of the LTM tape is represented as a 4-tuple $(\tilde{\boldsymbol{\gamma}}, h, \tilde{\boldsymbol{q}}, e)$, where $\tilde{\boldsymbol{\gamma}}$ is the binary encoding of the symbol stored in the cell, $h$ is the head indicator bit, $\tilde{\boldsymbol{q}}$ is the binary encoding of the current state (present only if the cell is the head cell), and $e$ is an erase flag. The erase flag is newly introduced in this construction.

Once the computation terminates, the erase flag is set to 1 in all cells except those in the $\mathsf{ST}_{\text{new}}$ subsection of the tape. When $e = 1$ in a cell, its contents are erased in the subsequent iteration. For cells in the $\mathsf{ST}_{\text{new}}$ and NM subsections of the tape, the erase flag additionally serves as a copy flag: when set to 1, it both erases the cell contents in the next iteration and triggers copying of the current symbol to designated locations. Below, we describe the templates defined for each cell of the Turing

tape.

**Template group 4** : For all $i \in [m]$ and all $\gamma \in \Gamma \setminus \{\text{empty symbol}\}$:

    If the $i$-th cell is $(\tilde{\boldsymbol{\gamma}}, 0, \mathbf{0}, 0)$;

    Output $\boldsymbol{y}$: All zeros, except:

      - the $i$-th cell is set to $(\tilde{\boldsymbol{\gamma}}, 0, \mathbf{0}, 0)$.

**Template group 5** : For all $i \in [m]$ and all $(q, \gamma) \in \mathrm{dom}(\delta) \setminus \{(q_h, \gamma_h)\}$:

    If the $i$-th cell is $(\tilde{\boldsymbol{\gamma}}, 1, \tilde{\boldsymbol{q}}, 0)$ and $\delta(q, \gamma) = (q', \gamma', d)$;

    Output $\boldsymbol{y}$: All zeros, except:

      - the $i$-th cell is set to $(\tilde{\boldsymbol{\gamma}}', 0, \mathbf{0}, 0)$.

      - if $d = \mathrm{L}$, the $(i-1)$-th cell is set to $(\mathbf{0}, 1, \tilde{\boldsymbol{q}}', 0)$; otherwise, if $d = \mathrm{R}$, the $(i+1)$-th cell is set to $(\mathbf{0}, 1, \tilde{\boldsymbol{q}}', 0)$.

**Template group 6** : For the last cell of the tape:

    If the cell is $(\tilde{\boldsymbol{\gamma}}_h, 1, \tilde{\boldsymbol{q}}_h, 0)$, where $q_h$ is the halting state;

    Output $\boldsymbol{y}$: All zeros, except:

      - set the erase flag $e$ to 1 in all cells except those in the $\mathsf{ST}_{\mathrm{new}}$ subsection of the tape

      - set the delay bit in $\mathsf{S}_{\mathrm{ESD}}$ to 1.

**Template group 7** : For all $i \in [P_{\mathrm{msg}}]$ and all $\gamma \in \Gamma \setminus \{\text{empty symbol}\}$:

    If cell $i$ in the NM subsection of the tape is $(\tilde{\boldsymbol{\gamma}}, 0, \mathbf{0}, 1)$;

    Output $\boldsymbol{y}$: All zeros, except:

      - the $i$-th symbol in each of the $D^2 + 1$ subsections of $\mathsf{S}_{\mathrm{OMB}}$ is set to $\tilde{\boldsymbol{\gamma}}$.

**Template group 8** : For the single delay bit in $\mathsf{S}_{\mathrm{ESD}}$:

    If the bit is equal to 1;

    Output $\boldsymbol{y}$: All zeros, except;

      - set the erase flag $e$ to 1 in all cells of the $\mathsf{ST}_{\mathrm{new}}$ subsection of the tape

      - set the flag in the $\mathsf{S}_{\mathrm{ALF}}$ to 1.

This phase begins with an initialized tape and the head indicator set to 1 in the first cell. Templates in group 4 ensure that tape symbols persist across iterations. Repeated applications of templates in group 5 execute the Turing machine program while moving the head as required. After $\tau$ iterations, the head reaches the final cell in the halting state. By template group 6, this sets the erase flag $e$ to 1 in all cells outside the $\mathsf{ST}_{\mathrm{new}}$ subsection of the tape and sets the delay bit in $\mathsf{S}_{\mathrm{ESD}}$ to 1.

For cells that do not belong to the subsections NM or $\mathsf{ST}_{\mathrm{new}}$ of the tape, setting $e = 1$ clears their contents in the next iteration, since no template is defined for this case. For cells in NM, setting $e = 1$ triggers the templates in Group 7, which copy the symbols in the NM cells to the message buffers in $\mathsf{S}_{\mathrm{OMB}}$. Recall that NM now contains the message that the node sends to its neighbors at the end of the $\ell$-th round of the LOCAL model, namely $\boldsymbol{h}_{u \to}^{(\ell)}$, encoded as a sequence of LTM alphabet symbols.

Simultaneously with the templates in Group 7, the template in Group 8 sets the $e$ flag to 1 in the cells of $\mathsf{ST}_{\mathrm{new}}$. Note that this occurs with a one-iteration delay compared to setting the $e$ flags of the other cells. This template also sets the flag bit in $\mathsf{S}_{\mathrm{ALF}}$ to 1.

We now count the number of templates defined above. Group 4 contains $m \cdot (|\Gamma| - 1)$ templates. Group 5 contains $m \cdot (|\mathrm{dom}(\delta)| - 1)$ templates. Group 6 contains a single template. Group 7 contains $P_{\mathrm{msg}} \cdot (|\Gamma| - 1)$ templates. Finally, Group 8 contains a single template.

PHASE 3: STORE OUTPUT OF THE LTM IN APPROPRIATE SECTIONS (OUTPUT)

**Purpose:** Route the outgoing messages stored in the $D^2 + 1$ buffers of $\mathsf{S}_{\mathsf{OMB}}$ to the node's dedicated slot in $\boldsymbol{x}_{u_M}$. Then, extract the updated local state and store it in $\mathsf{S}_{\mathsf{t.flg}}$, while also setting the flag bits required to initiate a new round of the LOCAL model.

**Trigger:** Before starting this phase, $D^2 + 1$ copies of the new message generated by the LTM are stored in $\mathsf{S}_{\mathsf{OMB}}$. The $e$ flags of the cells in the $\mathsf{ST}_{\mathsf{new}}$ subsection of the LTM tape containing $\boldsymbol{h}_u^{(\ell)}$ are all equal to 1. Additionally, the bit in $\mathsf{S}_{\mathsf{ALF}}$ that activates the load flags in $\mathsf{S}_{\mathsf{LS}}$ and $\mathsf{S}_{\mathsf{LM}}$ is set to 1.

**Templates:** Before introducing the templates for this phase, we first clarify the structure of the LOCAL ID section ($\mathsf{S}_{\mathsf{LID}}$). $\mathsf{S}_{\mathsf{LID}}$ determines which communication slot a node uses to share its message with its neighbors. It has the following form: $\boldsymbol{\delta}_{i_u} \otimes \mathbf{1}_{P_{\mathrm{msg}} L_\Gamma}$, where $\boldsymbol{\delta}_{i_u}$ denotes the $i_u$-th standard basis vector with $i_u \in [D^2 + 1]$ corresponding to the local ID of node $u$, $\otimes$ denotes the Kronecker product, and $\mathbf{1}_{P_{\mathrm{msg}} L_\Gamma}$ is an all-ones vector of length $P_{\mathrm{msg}} L_\Gamma$, which equals the number of bits in the outgoing message. Consequently, among the $D^2 + 1$ subsections in $S_{\mathsf{LID}}$, exactly one is an all-ones vector, and it remains unchanged from the beginning to the end of the process. We now present the templates for this phase.

**Template group 9** : For all $d \in [D^2 + 1]$, $s \in [P_{\mathrm{msg}}]$, and $p \in [L_\Gamma]$:

If the $p$-th bit of the $s$-th symbol in $d$-th subsection of $\mathsf{S}_{\mathsf{OMB}}$ is equal to 0, and the bit in $S_{\mathsf{LID}}$ corresponding to the $p$-th bit of the $s$-th symbol in $d$-th subsection is equal to 1;

Output $\boldsymbol{y}$: All zeros, except:

- the bit in $S_{\mathsf{LID}}$ corresponding to $p$-th bit of the $s$-th symbol in $d$-th subsection is set to 1.

**Template group 10** : For all $d \in [D^2 + 1]$, $s \in [P_{\mathrm{msg}}]$, and $p \in [L_\Gamma]$:

If the $p$-th bit of the $s$-th symbol in $d$-th subsection of $\mathsf{S}_{\mathsf{OMB}}$ is equal to 1, and the bit in $S_{\mathsf{LID}}$ corresponding to the $p$-th bit of the $s$-th symbol in $d$-th subsection is also equal to 1;

Output $\boldsymbol{y}$: All zeros, except:

- the $p$-th bit of $s$-th symbol in the $d$-th slot of the $\mathsf{S}_{\mathsf{com}}$ is set to 1.

- the bit in $S_{\mathsf{LID}}$ corresponding to $p$-th bit of the $s$-th symbol in $d$-th subsection is set to 1.

**Template group 11** : For all $i$ in $[P_{\mathrm{s.t.}}]$ and all $\gamma \in \Gamma \setminus \{\text{empty symbol}\}$:

If cell $i$ in the $\mathsf{ST}_{\mathsf{new}}$ subsection of the tape is $(\tilde{\boldsymbol{\gamma}}, 0, \mathbf{0}, 1)$;

Output $\boldsymbol{y}$: All zeros, except:

- the $i$-th symbol in $\mathsf{S}_{\mathsf{s.t.}}$ is set to $\tilde{\gamma}$

**Template group 12** : For the bit in $\mathsf{S}_{\mathsf{ALF}}$:

If the bit is equal to 1;

Output $\boldsymbol{y}$: All zeros, except:

- all the bits in $\mathsf{S}_{\mathsf{LS}}$, $\mathsf{S}_{\mathsf{LM}}$, and $\mathsf{S}_{\mathsf{t.flg}}$ are set to 1.

The $\mathsf{S}_{\mathsf{LID}}$ section is initialized before any iterations of the graph template matching framework. Templates in Group 9 ensure that when there is no message in $\mathsf{S}_{\mathsf{OMB}}$, the bits in the subsection of $\mathsf{S}_{\mathsf{LID}}$ corresponding to the local ID of the node remain set to 1. Templates in Group 10 copy the message in the subsection of $\mathsf{S}_{\mathsf{OMB}}$ determined by the active bits in $\mathsf{S}_{\mathsf{LID}}$ into the slot of $\mathsf{S}_{\mathsf{com}}$ that matches the local ID. This message contains $\boldsymbol{h}_{u\rightarrow}^{(\ell)}$; i.e., the information that the node sends to its neighbors in the $\ell$-th round of the LOCAL model.

Templates in Group 11 copy the newly computed local state from the $\mathsf{ST}_{\mathsf{new}}$ subsection of the LTM tape to $\mathsf{S}_{\mathsf{s.t.}}$. This subsection stores the local state $\boldsymbol{h}_u^{(\ell)}$, which is required for simulating the next round (round $\ell + 1$) of the LOCAL model. Finally, the template in Group 12 activates all flags required to start the simulation of the next round of the LOCAL model by setting all flag bits in $\mathsf{S}_{\mathsf{LS}}$, $\mathsf{S}_{\mathsf{LM}}$, and $\mathsf{S}_{\mathsf{t.flg}}$ to 1. The first two flags signal the loading of the local state and newly received messages onto the LTM tape, while the last flag initializes the fixed subsections of the tape for the beginning of a new LTM execution. Note that the templates in Groups 10, 11, and 12 are applied simultaneously at the beginning of this phase; therefore, this phase requires only a single iteration of the graph template matching framework.

We now count the number of templates defined above. In Group 9, there is one template for each pair of bits from $S_{\mathsf{OMB}}$ and $S_{\mathsf{LID}}$ with the same position. Thus, there are $(D^2 + 1)P_{\mathrm{msg}}L_\Gamma$ templates in total. Group 10 defines the same number of templates, namely $(D^2 + 1)P_{\mathrm{msg}}L_\Gamma$. In Group 11, a template is defined for each possible symbol, except the empty symbol, in the cells of $\mathsf{ST}_{\mathrm{new}}$; therefore, there are $P_{\mathrm{s.t.}} \cdot (|\Gamma| - 1)$ templates. Finally, Group 12 contains a single template.

PHASE 4: AGGREGATE $\boldsymbol{x}_{u_M}$ (MESSAGE AGGREGATION)

**Purpose:** In this phase, the $\boldsymbol{x}_{u_M}$ sections of the binary vectors of nodes within a neighborhood are aggregated. After aggregation, at most $D + 1$ slots contain a message: $D$ messages from neighboring nodes and one message from the node itself. The result of this aggregation becomes the new $\boldsymbol{x}_{u_M}$ for each node in the neighborhood.

As described in Model 3, aggregation is a mechanism that is separate from applying the template matching function. However, similar to the application of the function $f$, aggregation is performed in every iteration of the graph template matching framework. The key observation is that, in our simulation, the $\boldsymbol{x}_{u_M}$ section of each node's vector contains a message only during a single iteration, namely at the end of Phase 3. Therefore, aggregation has no effect in any other iteration, since $\boldsymbol{x}_{u_M}$ is an all-zero vector in those cases. Additionally, since each node shares its message with its neighbors in its own dedicated slot, the messages do not interfere with or distort one another.

**Binary vector status after aggregation:** After aggregation, the messages that the node receives from its neighbors at the $\ell$-th round of the LOCAL model, i.e. $\{\boldsymbol{h}^{(\ell)}_{u\leftarrow v}\}_{u\in\mathcal{N}^+_u}$ are all available in the slots of $\boldsymbol{x}_{u_M}$. The computation section, $\boldsymbol{x}_{u_C}$ is exactly the same as before aggregation. The local state $\boldsymbol{h}^{(\ell)}_u$ of the LOCAL model at the end of the round $\ell$ is stored in $S_{\mathrm{s.t.}}$, the load flags in $S_{\mathsf{LS}}$ and $S_{\mathsf{LM}}$ as well as the flag in $S_{\mathrm{t.flg}}$ are set to 1. This is the exact condition of the vector at the beginning of simulation of a new round of the LOCAL model, in this case the $(\ell + 1)$-th. Thus, our construction for the graph template matching framework successfully simulates a complete round of the LOCAL model. If we continue the constructed graph template matching framework for more iterations with the same templates defined above, the next rounds of the LOCAL model will be simulated accordingly.

**Binary vector status after aggregation:** After aggregation, the messages received by a node from its neighbors in the $\ell$-th round of the LOCAL model, namely $\{\boldsymbol{h}^{(\ell)}_{u\leftarrow v}\}_{v\in\mathcal{N}^+_u}$, are all available in the slots of $\boldsymbol{x}_{u_M}$.

The computation section $\boldsymbol{x}_{u_C}$ remains unchanged by the aggregation process. As before aggregation, the local state of node $u$ in the LOCAL model at the end of round $\ell$, denoted by $\boldsymbol{h}^{(\ell)}_u$, lies in $S_{\mathrm{s.t.}}$. Moreover, the load flags in $S_{\mathsf{LS}}$ and $S_{\mathsf{LM}}$, as well as the flag in $S_{\mathrm{t.flg}}$, are equal to 1.

Taken together, this configuration of $\boldsymbol{x}_{u_C}$ and $\boldsymbol{x}_{u_M}$ exactly matches the state of the vector at the beginning of the simulation of a new round of the LOCAL model, namely the $(\ell + 1)$-th round.

Consequently, our construction within the graph template matching framework successfully simulates a complete round of the LOCAL model. If the framework is executed for additional iterations using the same templates defined above, subsequent rounds of the LOCAL model are simulated accordingly.

CONCLUSION OF THE CONSTRUCTION AND CORRECTNESS OF THE LEMMA

Above, we constructed a graph template matching framework whose input binary vector was initialized with the encoded $\boldsymbol{h}^{(\ell-1)}_u$ and $\{\boldsymbol{h}^{(\ell-1)}_{u\leftarrow v}\}_{v\in\mathcal{N}^+_u}$. By applying

$$2P_{\mathrm{s.t.}}L_\Gamma + (3D^2 + 4)P_{\mathrm{msg}}L_\Gamma + mL_\Gamma + m|\operatorname{dom}(\delta)| - m$$

templates over $\tau + 4$ iterations, the framework computes the local state and the message to be sent to the neighbors at the end of the $\ell$-th round of the LOCAL model, i.e., $\boldsymbol{h}^{(\ell)}_u$ and $\boldsymbol{h}^{(\ell)}_{u\to}$, encoded as a sequence of LTM alphabet symbols in $\boldsymbol{x}_u$.

Now, assume that the LOCAL model starts from round 0 with the initial local state $\boldsymbol{h}^{(0)}_u$ and initial messages $\{\boldsymbol{h}^{(0)}_{u\leftarrow v}\}_{v\in\mathcal{N}^+_u}$. If the binary vector of the graph template matching framework is properly initialized with the binary encoding of $\boldsymbol{h}^{(0)}_u$ and $\{\boldsymbol{h}^{(0)}_{u\leftarrow v}\}_{v\in\mathcal{N}^+_u}$, then, by induction, we can show that for any round $\ell$ of the LOCAL model, there exists a round $\ell'$ of the graph template matching framework such that the binary vector at the end of round $\ell'$ contains the local state of the node at the end of round $\ell$ of the LOCAL model, encoded as a sequence of LTM alphabet symbols. Formally,

$$\boldsymbol{h}^{(\ell)}_u \sqsubset \boldsymbol{x}^{(\ell')}_u, \quad \text{for every round } \ell \text{ and every } u \in V,$$

where $\ell' = \ell(\tau + 4)$. This completes the proof of the lemma. $\qquad\square$

Collisions in the Templates of $f_{\mathrm{LOC}}$

We also need to determine the maximum possible number of collisions and ensure that it is a constant value. Let us start with the bits on the LTM's tape. Consider a bit of a symbol in a cell of the $\mathsf{ST}_{\mathsf{prv}}$ subsection of the LTM tape. One template from group 1, one template from group 4, and, in the worst case, $|\operatorname{dom}(\delta)| - 1$ templates from group 5 write into this bit position, resulting in a total of $|\operatorname{dom}(\delta)| + 1$ collisions. The same number of collisions occurs for a bit position of a symbol in the cells of the $\mathsf{RM}_d$ subsection of the LTM tape for $d \in [D^2 + 1]$: one template from group 2, one template from group 4, and in the worst case $|\operatorname{dom}(\delta)| - 1$ templates from group 5.

For a bit position of the symbols in cells of the BG, $\mathsf{DL}_{\mathsf{msg}}^{(i)}$ for $i \in [D^2 + 1]$, $\mathsf{DL}_{\mathsf{cmp}}$, $\mathsf{DL}_{\mathsf{st}'}$, $\mathsf{DL}_{\mathsf{msg}'}$, and ED subsections of the LTM tape, the maximum number of collisions is also $|\operatorname{dom}(\delta)| + 1$. Specifically, one template from group 3, one template from group 4, and $|\operatorname{dom}(\delta)| - 1$ templates might write into a bit position of the symbols in these subsections.

For cells in subsections of the LTM tape other than the ones already analyzed, there are at most $|\operatorname{dom}(\delta)|$ collisions: one template from group 4 and, in the worst case, $|\operatorname{dom}(\delta)| - 1$ templates from group 5.

The maximum number of collisions for the bits representing the state of the LTM in the cells is $2 \cdot (|\operatorname{dom}(\delta)| - 1)$, corresponding to $|\operatorname{dom}(\delta)| - 1$ templates from group 5 for each of the right and left neighbors of a cell that write in the cell's state. The maximum number of collisions for the head indicators is 2, and for any erase flag bit, there is only one template that writes into it. We omit further details of this analysis, as it follows the same principles discussed at the end of Appendix D.2.2, leading to Remark D.2.

Next, consider the bits in $\mathsf{S}_{\mathsf{s.t.}}$. Only one template from group 11 writes into a bit position of a symbol in $\mathsf{S}_{\mathsf{s.t.}}$. Similarly, only one template from group 12 writes into flag bits of the $\mathsf{S}_{\mathsf{LS}}$, $\mathsf{S}_{\mathsf{MS}}$, and $\mathsf{S}_{\mathsf{t.flg}}$ subsections. Likewise, one template from group 6 and one from group 8 write into the bits in $\mathsf{S}_{\mathsf{ESD}}$ and $\mathsf{S}_{\mathsf{ALF}}$, respectively.

Regarding $\mathsf{S}_{\mathsf{OMB}}$, only one template from group 7 writes into a bit position of a symbol in any of the $D^2 + 1$ subsections of $\mathsf{S}_{\mathsf{OMB}}$. The same holds for the bits in $\mathsf{S}_{\mathsf{com}}$, except that the template writing into each bit position is from group 10. However, for $S_{\mathsf{LID}}$, two templates—one from group 9 and one from group 10—write into a bit, resulting in two collisions.

The following remark summarizes the discussion above regarding the number of collisions in our construction for the simulation of the LOCAL model.

*Remark* D.4. The template matching function constructed in the simulation of the LOCAL model in the proof of Lemma D.3 has, in the worst case, at most $2 \cdot (|\operatorname{dom}(\delta)| - 1)$ collisions. This value is constant and depends only on the algorithm run in the LOCAL model.

## D.4. The GNN in Equation (2) can Learn LOCAL

In this section, we present the final component of the proof of Theorem D.5. Specifically, we leverage the results established in the preceding sections to show that the GNN defined in Equation (2) can execute algorithms in the LOCAL model. Since our arguments rely on core techniques from (Back de Luca et al., 2025a), we adopt their input encoding for the architecture. We first briefly introduce this encoding and then explain how it is incorporated into our construction in Lemma D.3.

### D.4.1. Input specification

Recall that the input to $f$ is a binary vector of dimension $k$, structured as blocks of bits, where each block has a set of defined templates (defined formally in Appendix D.1). Assume that the input vector to $f$ consists of $b$ blocks. To train MLPs to exactly learn $f$, a training dataset is constructed based on the templates of $f$. In this dataset, each template-label pair constitutes a training sample; however, a specific encoding is applied to the template inputs.

The encoded input vector has dimension

$$k' = \sum_{i=1}^{b} t_i,$$

where $t_i$, for $i \in [b]$, is the number of templates for the $i$-th block in the original (unencoded) vector. The encoded vector consists of $b$ blocks. Each block $i$ in the unencoded vector is mapped to its corresponding block in the encoded vector, which

has $t_i$ bits. Within this $t_i$-bit block, each bit corresponds to one of the $t_i$ templates defined for block $i$ in the unencoded vector. If the configuration of the $i$-th block in the unencoded input matches the $j$-th template, then the $j$-th bit of the $i$-th block in the encoded vector is set to 1. Since only one template can match a block at a time, all other bits of the $i$-th block in the encoded vector are set to 0.

Effectively, in the encoded vector, each block is represented by an orthonormal basis, where the basis vector $\mathbf{e}_j^{(t_i)}$ corresponds to the $j$-th template of the $i$-th block.

In the training dataset, each input template together with its corresponding output forms a training sample. Let us denote the $j$-th template-output pair of the $i$-th block as $(\boldsymbol{x}_j^{(i)}, \boldsymbol{y}_j^{(i)})$. This pair constitutes a feature-label sample for training, where the input $\boldsymbol{x}_j^{(i)}$ is encoded as:

$$\boldsymbol{q}_j^{(i)} = (\mathbf{0}^{(t_1)}, \mathbf{0}^{(t_2)}, \dots, \boldsymbol{e}_j^{(t_i)}, \dots, \mathbf{0}^{(t_b)}).$$

Here, $\mathbf{0}^{(t_i)}$ denotes a block of all zeros of length $t_i$ bits. The output $\boldsymbol{y}_j^{(i)}$ is kept in its original form as the label. Hence, the set of training input features is: $\mathcal{X} = \{\boldsymbol{q}_j^{(i)} : i \in [b], j \in [t_i]\}$, and the set of output labels is: $\mathcal{Y} = \{\boldsymbol{y}_j^{(i)} : i \in [b], j \in [t_i]\}$.

At inference time, i.e., when executing each step of the algorithm, multiple blocks of the vector may be non-zero depending on the current step. However, each block contains at most one non-zero bit. The input vector to the MLP during inference is:

$$\hat{\boldsymbol{x}} = \frac{1}{\sqrt{N_{\hat{x}}}}\big[\hat{\boldsymbol{x}}^{(1)}, \hat{\boldsymbol{x}}^{(2)}, \dots, \hat{\boldsymbol{x}}^{(b)}\big],$$

where, depending on the algorithmic step, each $\hat{\boldsymbol{x}}^{(i)}$ is either $\mathbf{0}^{(t_i)}$ or $\boldsymbol{e}_j^{(t_i)}$ for some $j \in [t_i]$, and $N_{\hat{x}}$ is the total number of non-zero bits in $\hat{\boldsymbol{x}}$.

### D.4.2. EXACT LEARNABILITY OF THE TEMPLATE MATCHING FRAMEWORK

For a point $\hat{\boldsymbol{x}}$ at inference time, the NTK predictor computes $\mu(\hat{\boldsymbol{x}}) = \Theta(\hat{\boldsymbol{x}}, \mathcal{X})\,\Theta(\mathcal{X}, \mathcal{X})^{-1}\,\mathcal{Y}$. Due to the orthogonal encoding introduced earlier, this value becomes a weighted sum of the training labels. Specifically, for training samples whose inputs match one of the blocks of $\hat{\boldsymbol{x}}$, the corresponding labels are summed with weight $w^1 \equiv w^1(\hat{\boldsymbol{x}})$. For training samples whose inputs do not match any block, the labels are summed with weight $w^0 \equiv w^0(\hat{\boldsymbol{x}})$.

**Theorem D.5** (NTK predictor behavior (Back de Luca et al., 2025a)). *Consider the template-matching function $f$ as described in Appendix D.1 and its templates encoded into a training set $(\mathcal{X}, \mathcal{Y}) \subseteq \mathbb{R}^{k'} \times \mathbb{R}^k$ as described in Appendix D.4.1. Then, under the assumption that the number of unwanted collisions is less than the ratio $-w^1/w^0$, the mean of the limiting NTK distribution $\mu(\hat{\boldsymbol{x}}) = \Theta(\hat{\boldsymbol{x}}, \mathcal{X})\Theta(\mathcal{X}, \mathcal{X})^{-1}\mathcal{Y}$ for any test input $\hat{\boldsymbol{x}} \in \mathbb{R}^{k'}$ contains sign-based information about the ground-truth output, namely for each coordinate of the output $i = 1, \dots, k$, $\mu(\hat{\boldsymbol{x}})_i \leq 0$ if the ground-truth bit at position $i$, $f(\hat{\boldsymbol{x}})_i$[3], is set, and $\mu(\hat{\boldsymbol{x}})_i > 0$ if the ground-truth bit at position $i$, $f(\hat{\boldsymbol{x}})_i$, is not set.*

Based on the above theorem, we can conclude the following lemma about the exact learnability of the template matching function that we defined to simulate the LOCAL model in Lemma D.3, namely $f_{\text{LOC}}$.

**Lemma D.6** (Learnability of $f_{\text{LOC}}$ for the NTK predictor). *Let $f_{LOC}$ denote the template-matching function used to simulate the LOCAL model. There exists a training dataset $(\mathcal{X}, \mathcal{Y}) \subseteq \mathbb{R}^{k'} \times \mathbb{R}^k$, where inputs are encoded as specified in Appendix D.4.1, with*

$$k' = (2 \cdot |\operatorname{dom}(\delta)| - 1) \cdot n_T, \tag{13}$$

*and*

$$k = 2P_{s.t.}L_\Gamma + 4(D^2 + 1)P_{msg}L_\Gamma + mL_\Gamma + m|Q| + 2m + 3,$$

*such that, for any test input $\hat{\boldsymbol{x}} \in \mathbb{R}^{k'}$, the mean of the limiting NTK predictor, $\mu(\hat{\boldsymbol{x}}) = \Theta(\hat{\boldsymbol{x}}, \mathcal{X})\Theta(\mathcal{X}, \mathcal{X})^{-1}\mathcal{Y}$ contains sign-based information about the ground-truth output of $f_{LOC}$. In the expressions above, $L_\Gamma$ denotes the number of bits required to represent symbols of the LTM alphabet, $Q$ is the set of LTM states, $\operatorname{dom}(\delta)$ is the domain of the LTM transition function, $D$ is the maximum degree of the underlying graph, $P_{s.t.}$ and $P_{msg}$ are the numbers of LTM cells used to store the local state and outgoing messages, respectively, $m$ is the total number of LTM cells, and $n_T = 2P_{s.t.}L_\Gamma + (3D^2 + 4)P_{msg}L_\Gamma + mL_\Gamma + m|\operatorname{dom}(\delta)| - m$ is the number of templates of $f_{LOC}$.*

---

[3]Note that there is some abuse of notation here, $\hat{\boldsymbol{x}}$ as the argument of $f$ in $f(\hat{\boldsymbol{x}})_i$ is the unencoded while it is encoded elsewhere

*Proof.* The correctness of the lemma follows from Theorem D.5. Consider a training dataset constructed from

$$n_T = 2P_{\text{s.t.}}L_\Gamma + (3D^2 + 4)P_{\text{msg}}L_\Gamma + mL_\Gamma + m|\text{dom}(\delta)| - m$$

templates of $f_{\text{LOC}}$, as specified in Appendix D.4.1. By Theorem D.5, the mean of the limiting NTK predictor trained on such a dataset contains sign-based information about the ground-truth output of $f_{\text{LOC}}$, provided that the number of collisions is less than $-w_1(\hat{\boldsymbol{x}})/w_0(\hat{\boldsymbol{x}})$.

To ensure that this condition is satisfied, we construct a dataset of size and input dimension

$$k' = \big(2 \cdot |\text{dom}(\delta)| - 1\big) n_T,$$

by padding the existing orthogonalized training dataset of size $n_T$ with additional training examples that are never matched and whose output label is $\mathbf{0}$

Following (Back de Luca et al., 2025a), we observe that for every normalized input $\hat{\boldsymbol{x}}$, the ratio $w_1(\hat{\boldsymbol{x}})/w_0(\hat{\boldsymbol{x}})$ is a decreasing function of $N_{\hat{x}}$ and is strictly greater than $2 \cdot \big(|\text{dom}(\delta)| - 1\big)$ for $N_{\hat{x}} \leq n_T$ (the maximum number of matched templates). Since the number of collisions is at most $2 \cdot \big(|\text{dom}(\delta)| - 1\big)$, the ratio condition is satisfied, and the result follows.

$\square$

As the lemma above states, the NTK predictor can learn to exactly execute the $f_{\text{LOC}}$, by correctly predicting the sign of each bit in the output vector, negative for a ground-truth zero bit and positive for a ground-truth bit equal to 1. Thus, passing the NTK predictor's output through an entry-wise Heaviside step function (denoted by $\Psi_H(\cdot)$) will give the desired output. More importantly, one can conclude from the lemma that the averaged output of an ensemble of independently trained 2-layer MLPs as introduced in Appendix B, trained on the training set built based on the templates of $f_{\text{LOC}}$ using the orthogonal encoding mentioned earlier, can approximate the NTK predictor up to arbitrarily high accuracy, as long as the ensemble size is large enough. We refer to the number of MLPs needed to achieve the desired post-Heaviside accuracy as *ensemble complexity*. According to Lemma 6.1 of (Back de Luca et al., 2025a), to maintain an arbitrarily chosen level of accuracy, the ensemble complexity grows like $\mathcal{O}\left(k' \cdot n_T + k' \log(nL')\right)$, where $n_T$ is the number of templates, $k'$ is the training dataset size, $n$ is the number of vertices in the graph, and $L'$ is the number of iterations required. With that, we have all the necessary pieces to state and prove the main theorem of our work, which we present below.

**Theorem D.7** (Learnability and execution of any LOCAL model using the GNN in Equation (2))**.** *Consider the GNN architecture defined in Equation (2) with an ensemble of infinite-width MLPs. There exists a training dataset of size*

$$k' = \mathcal{O}\left(|\text{dom}(\delta)| \cdot \big((P_{s.t.} + D^2 P_{msg} + m)L_\Gamma + m|\text{dom}(\delta)| + m\big)\right)$$

*and an ensemble size*

$$K = \mathcal{O}\left(|\text{dom}(\delta)| \cdot n_T^2 + |\text{dom}(\delta)| \cdot n_T \log(n\tau L)\right),$$

*such that the resulting GNN can learn to* exactly *simulate an $L$-round LOCAL model with arbitrarily high probability. The simulation is achieved by placing the GNN inside a loop of $L' = (\tau + 4)L$ iterations provided that each node in the LOCAL model has bounded memory, all exchanged messages have bounded size, and the underlying graph $G$ has maximum degree $D$. In the above expressions, $L_\Gamma$ denotes the number of bits required to encode symbols of the LTM alphabet, $Q$ is the set of LTM states, and $\text{dom}(\delta)$ is the domain of the LTM transition function. Moreover, $P_{s.t.}$ and $P_{msg}$ denote the numbers of LTM cells used to store the local state and outgoing messages, respectively, $m$ is the total number of LTM cells, $n$ is the number of vertices in the graph, and $\tau$ is the number of execution steps required for the LTM to reach its halting state. Lastly, $n_T = 2P_{s.t.}L_\Gamma + (3D^2 + 4)P_{msg}L_\Gamma + mL_\Gamma + m|\text{dom}(\delta)| - m$ is the number of constructed templates.*

*Proof.* Assume that the computation performed in each round of the LOCAL model is executed by a Local Turing Machine (LTM) as defined in Appendix D.3.1. By Lemma D.3, there exists a graph template matching framework with a template matching function $f_{\text{LOC}}$ that simulates an $L$-round LOCAL model in $L' = (\tau + 4)L$ iterations.

We show that the GNN architecture defined in Equation (2) can learn to execute this graph template matching framework. By Lemma D.6, the NTK predictor of a two-layer MLP followed by a Heaviside activation can exactly reproduce the output of $f_{\text{LOC}}$ for any input, provided that the training dataset has size $k'$ as stated in that lemma and repeated in the statement of this theorem. Moreover, as discussed earlier, the average output of an ensemble of 2-layer MLPs followed by a Heaviside

activation converges, with arbitrarily high probability, to the post-Heaviside output of the corresponding NTK predictor, where the probability is controlled by the ensemble size.

Therefore, the node update function

$$F_{\text{node}}(X) = \Psi_H(\hat{\mu}(\Psi_{\text{Enc}}(X)))$$

in Equation (2) can recover the output of $f_{\text{LOC}}$ exactly for all nodes in all iterations, with arbitrarily high probability. As mentioned earlier, this requires an ensemble size scaling as $\mathcal{O}\left(k' \cdot n_T + k' \log(nL')\right)$. Substituting $k'$ from Equation (13) and $L' = (\tau + 4)L$ yields the stated bound on $K$ in the theorem.

We now analyze how the remaining components of the GNN architecture implement the aggregation step of the graph template matching framework. According to Equation (2), the output of one GNN iteration is

$$F_{\text{node}}(X)P_C + AF_{\text{node}}(X)P_M,$$

where $A$ is the binary adjacency matrix of the graph and the projection matrices $P_C$ and $P_M$ are defined as

$$P_C = \begin{pmatrix} I_{k_C} & \mathbf{0} \\ \mathbf{0} & \mathbf{0} \end{pmatrix}, \qquad P_M = \begin{pmatrix} \mathbf{0} & \mathbf{0} \\ \mathbf{0} & I_{k_M} \end{pmatrix},$$

with $k$ denoting the dimension of each node's binary feature vector, and $k_C$ and $k_M$ denoting the dimensions of the $\boldsymbol{x}_C$ and $\boldsymbol{x}_M$ components, respectively (at the output).

Consider first the term $F_{\text{node}}(X)P_C$. This operation preserves the first $k_C$ columns of $F_{\text{node}}(X)$ and sets all remaining columns to zero. Consequently, it retains the $\boldsymbol{x}_C$ component of each node's binary vector.

Next, consider the term $AF_{\text{node}}(X)P_M$. The multiplication by $P_M$ preserves only the last $k_M$ columns of $F_{\text{node}}(X)$, corresponding to the $\boldsymbol{x}_M$ component. Left-multiplication by the adjacency matrix $A$ then aggregates these $\boldsymbol{x}_M$ components over the neighbors of each node in $G$. This exactly matches the message aggregation step of the constructed graph template matching framework.

Summing the two terms therefore produces a matrix in which, for each node, the first $k_C$ entries correspond to its updated $\boldsymbol{x}_C$ component, while the remaining $k_M$ entries correspond to the aggregated $\boldsymbol{x}_M$ components of the node and its neighbors. This is precisely the binary node representation obtained after one iteration of the graph template matching framework constructed in Lemma D.3.

Since this framework simulates an $L$-round LOCAL model in $L' = (\tau + 4)L$ iterations, it follows that the GNN in Equation (2), when unrolled for $(\tau + 4)L$ iterations and initialized with the same binary node features, exactly simulates the LOCAL model with arbitrarily high probability. This completes the proof. $\qquad\square$

# E. Proof of Theorems 6.1-6.4

We first give the proof of Theorem 6.1, which is similar in nature to the proof of Theorem D.7 and follows the proof of Theorem D.5 in (Back de Luca et al., 2025a).

Given the graph template matching instructions for the local computation of the Message Flooding algorithm in Appendix E.2, we construct an orthogonal training dataset according to the encoding discussed in Appendix D.4.1. Since there are $\mathcal{O}(l \cdot D^2)$ templates and the number of collisions (at most 2) does not depend on the algorithm parameters, the dataset size (and hence the embedding dimension $k'$ as well) is $\mathcal{O}(l \cdot D^2)$. Thus, since the number of iterations is $R = \mathcal{O}(D^2 \cdot d_G)$, an ensemble of $K = \mathcal{O}(k'^2 + k'R) = \mathcal{O}(l^2 \cdot D^4 + l \cdot D^4 \cdot d_G)$ MLPs can learn to execute exactly with arbitrarily high probability (after Heaviside activation) each local computation step of the Message Passing algorithm. Thus, after $\mathcal{O}(D^2 \cdot d_G)$ iterations, the GNN of Equation (2) with an ensemble size $K = \mathcal{O}(l^2 \cdot D^4 + l \cdot D^4 \cdot d_G)$ can execute the entire algorithm with arbitrarily high probability. The proofs of Theorems 6.2-6.4 follow using the same argument since the number of collisions is independent of the algorithm parameters in every case (at most 5 for BFS and DFS, and at most 4 for Bellman-Ford).

**Instructions of applications:** The following sections describe how to implement Message Flooding, Breadth First Search (BFS), Depth First Search (DFS), and Bellman-Ford (BF) within our graph template matching framework. We devote one subsection to each algorithm. In each subsection, we briefly introduce the algorithm and provide its pseudocode. We then present the template matching instructions that define the local function $f$ and execute the algorithm on any graph with maximum degree $D$. We also report the number of conflicts induced by the templates for each algorithm. Finally, we derive

a sufficient upper bound on the number of iterations required by the graph template matching framework and thus the GNN to execute the full algorithm on a test case. Before that, we explain our template presentation format and how to construct the dataset needed to train the NTK predictor or the ensemble of MLPs used in our GNN architecture

### E.1. Building a Training Dataset from Instructions

The input binary vector of $f$ is composed of groups of bits where each group represent a variable in the algorithm. We give a name to each group. This should not be confused with *blocks*. Remember that a block is an exclusive group of bits for which a set of template is defined in $f$. To present the templates, we will first introduce all the variable in the input binary vector of $f$ with their names and briefly discuss the purpose the variable and its bit structure. We will then present all templates for $f$ in the form of human-readable instructions. Consider the following toy example. Assume the binary vector of $f$, $\boldsymbol{x}$, is structured as follows:

$$\boldsymbol{x} = \underbrace{\boxed{a_1 \mid a_2}}_{a} \underbrace{\boxed{b_1 \mid b_2}}_{b} \underbrace{\boxed{c_1 \mid c_2}}_{c}$$

Here, $\boldsymbol{x}$ has three named variables $a$, $b$, and $c$. Each variable has two bits. $a_i$ denotes the $i$-th bit of the variable $a$. In the instructions we will denote the $i$-th bit of this variable as a[i]. Before we present the instructions for $f$, first we will define the variables with the indexing for their constituent bits:

```
Variable definition:
a[i] for i ∈ [2]
b[i] for i ∈ [2]
c[i] for i ∈ [2]
```

Then we will define the instructions of $f$. The following are only illustrative instructions:

```
INPUT:
a[i]=1 AND b[i]=0

OUTPUT:
c[i]=1
```

```
INPUT:
a[i]=1 AND b[i]=1

OUTPUT:
b[i]=1
```

```
INPUT:
c[i]=1

OUTPUT:
b[i]=1
```

Consider the first instruction box. It specifies the instruction for the $i$-th bit of variables $a$ and $b$. To avoid repetition, we adopt the convention that a single instruction box applies to *all* bit indices (here, for all $i \in [2]$) unless stated otherwise. The second box specifies the instruction for another bit configuration of the same variable as in the first box. The Third box, specifies the instructions for the bits of $c$. As a result, in total there are six instructions or templates in this example. The input statement of the above instructions also determines the blocks. Based on the first and second boxes, the $i$-bit of the $a$ and $b$ together are a block. We will denote this block as (a[i],b[i]). Since $i \in [2]$, there are two such blocks. Moreover, based on the third box, the $i$-th bit of $c$ is also a block that we will write it as (c[i]). Similarly, there are two such blocks. Thus, overall, the are four blocks in the above example.

Each template $T$ is an input-output pair of binary vectors $(\boldsymbol{x}, \boldsymbol{y})$. In an instruction box, the input statement determines $\boldsymbol{x}$ and the output statement determines $\boldsymbol{y}$. Let us provide the input output vectors of the templates corresponding to the instructions of this example. For convenience in referring to the instructions, we give the instructions in the first box number 1 and 2, the instructions in second box 3 and 4, and the instructions in the last box 5 and 6. The template corresponding to $j$-th

instruction will be denoted as $T_j$ for $j \in [6]$ instruction. The input and output vector of the template corresponding to the instructions are as follows:

$$T_1: \quad \boldsymbol{x} = \overbrace{\boxed{1 \mid 0}}^{a}\overbrace{\boxed{0 \mid 0}}^{b}\overbrace{\boxed{0 \mid 0}}^{c} \quad , \boldsymbol{y} = \overbrace{\boxed{0 \mid 0}}^{a}\overbrace{\boxed{0 \mid 0}}^{b}\overbrace{\boxed{1 \mid 0}}^{c}$$

$$T_2: \quad \boldsymbol{x} = \boxed{0 \mid 1 \mid 0 \mid 0 \mid 0 \mid 0} \quad , \boldsymbol{y} = \boxed{0 \mid 0 \mid 0 \mid 0 \mid 0 \mid 1}$$

$$T_3: \quad \boldsymbol{x} = \boxed{1 \mid 0 \mid 1 \mid 0 \mid 0 \mid 0} \quad , \boldsymbol{y} = \boxed{0 \mid 0 \mid 1 \mid 0 \mid 0 \mid 0}$$

$$T_4: \quad \boldsymbol{x} = \boxed{0 \mid 1 \mid 0 \mid 1 \mid 0 \mid 0} \quad , \boldsymbol{y} = \boxed{0 \mid 0 \mid 0 \mid 1 \mid 0 \mid 0}$$

$$T_5: \quad \boldsymbol{x} = \boxed{0 \mid 0 \mid 0 \mid 0 \mid 1 \mid 0} \quad , \boldsymbol{y} = \boxed{0 \mid 0 \mid 1 \mid 0 \mid 0 \mid 0}$$

$$T_6: \quad \boldsymbol{x} = \boxed{0 \mid 0 \mid 0 \mid 0 \mid 0 \mid 1} \quad , \boldsymbol{y} = \boxed{0 \mid 0 \mid 0 \mid 1 \mid 0 \mid 0}$$

In order to make a training dataset from the templates above, the input vectors of the templates should be encoded using the procedure explained with details in Appendix D.4.1. We encode each block using an orthonormal basis. Let us denote the encoded input vector with $\hat{\boldsymbol{x}}$. This vector is structured as follows:

$$\hat{\boldsymbol{x}} = \underbrace{\boxed{T_1 \mid T_3}}_{(a[1],\, b[1])}\underbrace{\boxed{T_2 \mid T_4}}_{(a[2],\, b[2])}\underbrace{\boxed{T_5}}_{(c[1])}\underbrace{\boxed{T_6}}_{(c[2])} \tag{14}$$

In the encoded vector each block's dimension is equal to the number of instructions for that block. Only one bit can be equal to 1 as instructions are written for different configurations of the block that do not happen simultaneously. For example if the input configuration of the block (a[1], b[1]) is (1, 1), template $T_3$ matches this block. Thus, the bit corresponding to $T_3$ in $\hat{\boldsymbol{x}}$ will is 1 and the bit corresponding to $T_1$ will be set to 0. Based on this encoding principle for the input vectors of the templates, the samples of the training dataset are as follows:

$$T_1: \quad \hat{\boldsymbol{x}} = \overbrace{\boxed{1 \mid 0}}^{(a[1], b[1])}\overbrace{\boxed{0 \mid 0}}^{(a[2], b[2])}\overbrace{\boxed{0}}^{(c[1])}\overbrace{\boxed{0}}^{(c[2])} \quad , \boldsymbol{y} = \overbrace{\boxed{0 \mid 0}}^{a}\overbrace{\boxed{0 \mid 0}}^{b}\overbrace{\boxed{1 \mid 0}}^{c}$$

$$T_2: \quad \hat{\boldsymbol{x}} = \boxed{0 \mid 0 \mid 1 \mid 0 \mid 0 \mid 0} \quad , \boldsymbol{y} = \boxed{0 \mid 0 \mid 0 \mid 0 \mid 0 \mid 1}$$

$$T_3: \quad \hat{\boldsymbol{x}} = \boxed{0 \mid 1 \mid 0 \mid 0 \mid 0 \mid 0} \quad , \boldsymbol{y} = \boxed{0 \mid 0 \mid 1 \mid 0 \mid 0 \mid 0}$$

$$T_4: \quad \hat{\boldsymbol{x}} = \boxed{0 \mid 0 \mid 0 \mid 1 \mid 0 \mid 0} \quad , \boldsymbol{y} = \boxed{0 \mid 0 \mid 0 \mid 1 \mid 0 \mid 0}$$

$$T_5: \quad \hat{\boldsymbol{x}} = \boxed{0 \mid 0 \mid 0 \mid 0 \mid 1 \mid 0} \quad , \boldsymbol{y} = \boxed{0 \mid 0 \mid 1 \mid 0 \mid 0 \mid 0}$$

$$T_6: \quad \hat{\boldsymbol{x}} = \boxed{0 \mid 0 \mid 0 \mid 0 \mid 0 \mid 1} \quad , \boldsymbol{y} = \boxed{0 \mid 0 \mid 0 \mid 1 \mid 0 \mid 0}$$

After an NTK predictor or a large enough ensemble of MLPs is trained on this dataset, given an encoded test input, the output after applying the Heaviside function will be the bitwise OR of the output vector of all the templates that match the test point. Consider the following illustrative test input before encoding:

$$\boldsymbol{x}_{test} = \underbrace{\boxed{1 \mid 0}}_{a}\underbrace{\boxed{0 \mid 0}}_{b}\underbrace{\boxed{0 \mid 1}}_{c}$$

In this input vector, template $T_1$ matches the block (a[1], b[1]) and template $T_6$ matches the block (c[2]). Thus, in the encoded vector with the structure as in Equation (14) the bits corresponding to $T_1$ and $T_6$ should be set to 1 an the rest of the bits should be 0. However, the whole vector should also be normalized before passing to the NTK predictor. The encoded normalized vector is as follows:

$$\hat{\boldsymbol{x}}_{test} = \underbrace{\boxed{\frac{1}{\sqrt{2}} \mid 0}}_{(a[1], b[1])}\underbrace{\boxed{0 \mid 0}}_{(a[2], b[2])}\underbrace{\boxed{0}}_{(c[1])}\underbrace{\boxed{\frac{1}{\sqrt{2}}}}_{(c[2])}$$

The output of the NTK predictor after passing through the Heaviside function will be the bitwise OR of the output vectors of $T_1$ and $T_6$ as follows:

$$\boldsymbol{y}_{test} = \underbrace{\boxed{0 \mid 0}}_{a}\underbrace{\boxed{0 \mid 1}}_{b}\underbrace{\boxed{1 \mid 0}}_{c}$$

For real graph algorithms that we will discuss in the next subsections, we will provide the instructions that can be converted to a dataset as we did above and be used to train an NTK predictor or the ensemble of MLPs in our GNN architecture. This in fact executes computation steps of the graph template matching framework on the binary vector of a node. At inference time, the binary vector of all nodes will be initialized with necessary information, e.g. IDs or initial values of the variables. These vectors will be encoded and passed through the NTK predictor simultaneously. Each vector has a communication section. After the NTK calculates the output vector, the communication sections of neighboring nodes will go through the aggregation step. Then, the whole vector will be encoded again based on the matching templates and fed back to the NTK predictor for the next iteration. The iterations of GNN will continue until the desired output is achieved. Now that we have clarified the procedure, in the following, let us discuss the four graph algorithms mentioned earlier, i.e. flooding, BFS, DFS, BF one by one.

### E.2. Message Flooding Algorithm

For the Message Flooding algorithm, we have chosen one of the simplest versions where a message is first stored in the message-register of a source node. Then, this source node sends this message to all of its immediate neighbors. Any node that receives the message, also sends it to its own immediate neighbors and so on. Algorithm 4 provides a pseudo-code of the Message Flooding algorithm. Note that to implement this algorithm with binary instructions that meet the requirements of our graph template matching framework, we will need more variables than those in the pseudo-code. Additionally, basic operations denoted simply by symbols like $\leftarrow$ involve more substeps when implemented in the binary domain of our graph template matching framework.

---

**Algorithm 4** Flooding ($G$, $s$, $Message$)

---

1: **for** each vertex $u \in G.V \setminus \{s\}$ **do**
2:     $u.message \leftarrow 0$
3: **end for**
4: $s.message \leftarrow Message$
5: **while** true **do**
6:     **for** $u \in G.V$ **in parallel do**
7:         **for** each $v \in G.Adj[u]$ **in parallel do**
8:             $v.message \leftarrow Message$
9:         **end for**
10:     **end for**
11: **end while**

---

### E.2.1. FLOODING INSTRUCTIONS

In what follows, we will provide the binary instructions for the template matching function $f$ of the graph template matching framework that executes the Message Flooding algorithm on any attributed graph with maximum degree $D$.

Let us first define the binary variables used in the binary implementation of the algorithm and the indexing system used for the bits of each variable. The purpose and structure of each variable is explained below it.

```
Index definition:
Let i ∈ [l] be the index over the message bits.
Let s ∈ [D² + 1] be the index for communication slots.

Variable definition:
  message-register ∈ {0,1}ˡ
  # holds the message locally in each node. Bits are indexed like message-register[i].

  transmission-buffer ∈ {0,1}^((D²+1)×l)
  # a 2D array with bits indexed as transmission-buffer[s][i], to temporarily hold
    copies of the outgoing message

  slot-selector ∈ {0,1}^((D²+1)×l)
  # a 2D array with bits indexed as slot-selector[s][i] where for one of the s
```

```
   corresponding to local ID bits are all 1.

reception-pipeline ∈ {0,1}^{(D²+1)×l}
# a 2D array with bits indexed as reception-pipeline[s][i], receives the message from
  communication channels

communication-channel ∈ {0,1}^{(D²+1)×l}
# a 2D array with bits indexed as communication-channel[s][i], holdding the message
  shared by the node. This is the only variable in the x_M section of the binary vector
```

Now we will present the instructions and briefly explain the purpose of each instruction below its box:

**Box 1**

```
INPUT:
message-register[i]=1

OUTPUT:
transmission-buffer[ALL][i]←1 AND message-register[i]←1
```

The instructions above create $D^2 + 1$ copies of the message at the register in transmission-buffer.

**Box 2**

```
INPUT:
transmission-buffer[s][i]=1 AND slot-selector[s][i]=1

OUTPUT:
communication-channel[s][i]←1 AND slot-selector[s][i]←1
```

The instructions above copy one of the same messages in transmission-buffer to the communication channel that the slot-selector determines. It also preserves the values in slot-selector for next iterations.

**Box 3**

```
INPUT:
transmission-buffer[s][i]=0 AND slot-selector[s][i]=1

OUTPUT:
slot-selector[s][i]←1
```

The above instruction is simply to preserve the values in slot-selector for the next iterations.

**Box 4**

```
INPUT:
communication-channel[s][i]=1

OUTPUT:
reception-pipeline[s][i]←1
```

For each communication channel, there is a corresponding section in reception-pipeline. The instructions above copy a message from any of the communication channels to the corresponding section in reception-pipeline.

**Box 5**

```
INPUT:
reception-pipeline[s][i]=1

OUTPUT:
(s < D²+1)  reception-pipeline[s+1][i]←1

OUTPUT:
(s = D²+1)  message-register[i]←1
```

The reception-pipeline acts as a cascade chain discussed in Appendix C.3. Once there is a message in one of the sections of reception-pipeline, the instructions above pass it to the next section, until it reaches $D^2 + 1$-th section. At this point the message is copied to the local message-register.

### E.2.2. THE SPECIFICATIONS OF THE TRAINING DATA

Based on the binary variables defined, one can calculate the dimension of the binary vector before the input encoding, $k$. By simply counting the number of bits we obtain $k = 4l(D^2 + 1) + l$. This value is the dimension of the output vector as the output is not in the encoded form. The variable in communication section $\boldsymbol{x}_M$ is `communication-channel` with $l(D^2 + 1)$ bits. Thus, the $k_M$ in Equation (3) is $l(D^2 + 1)$.

We obtain the total number of instructions, denoted by $n_T$, by simply counting the number of instructions in each box (each box represents the instructions for the entire range of the indexing variables). This yields: $n_T = 4l(D^2 + 1) + l$. Since each instruction is converted into a training sample, the number of training samples and the input embedding dimension before augmentation are both equal to $n_T$.

### E.2.3. THE MAXIMUM NUMBER OF COLLISIONS

The maximum number of collisions in this application is two. There are many bit position that have two collisions, however to just give and example consider the $i$-th bit for $i \in [l]$ in `message-register`. There is one instruction from Box 1 and one instruction from Box 5 that write into this bit position. Since the number of collisions is constant, the size of the training dataset after augmentation with idle samples as well as the input embedding dimension, denoted by $k'$, are proportional to $n_T$. Thus, similar to $n_T$, we have $k' = \mathcal{O}(l \cdot D^2)$.

### E.2.4. INITIALIZATION

To run the algorithm, some variables need to be initialized properly. Let us name the binary message that is going to be flooded $Message$, denote the local ID of node $u$ with $u_{\mathcal{L}}$, and denote the source node with $s$ while other nodes are denoted by $u$.

For the source node the variables are initialized as follows:

```
Source Intitialization
message-register[i]← Message[i]
slot-selector[s_L][ALL]← 1
```

For the other nodes, only the `slot-selector` needs to be initialized.

```
Non-Source Intitialization
slot-selector[u_L][ALL]← 1
```

### E.2.5. SUFFICIENT UPPER BOUND ON THE NUMBER OF ITERATIONS

Here we will provide a sufficient upper bound on the number of iterations that the graph template matching framework with the instructions above will require to complete the algorithm. Let us start right after an aggregation step where a message appears in one of the communication channels of a node. It takes one iteration for instructions in Box 4 to copy it to `reception-pipeline`, in worst case $D^2 + 1$ iteration for instructions in Box 5 to convey the message to the `message-register`, one iteration for the instructions in Box 1 to create copies of the message in `transmission-buffer`, and one iteration for the instructions in Box 2 to copy the message in `transmission-buffer` to the correct slot in `communication-channel`. Note that this is a moment that after aggregation step, the neighboring nodes of the current node will have the message in one of the communication channels and will go through the same iterations to send it to their neighbors. The worst case number of iteration from receiving the message up to the point that the message is shared with the neighbors is $D^2 + 4$. Let us denote the diameter of the graph with $d_G$. Repeating the iteration above $d_G + 1$ times guarantees completion of the algorithm. Thus the a sufficient upper bound on the total number of iteration to complete this algorithm is $(d_G + 1) \cdot (D^2 + 4)$.

### E.3. Breadth first search instructions

In this section, we provide instructions that define the function $f$ in our graph template matching framework for executing the Breadth-First Search (BFS) algorithm on graphs with maximum degree $D$. We use $l$ bits to represent all variables in our implementation, including the global identifiers of nodes. Consequently, at test time, the maximum graph size is limited to $2^l - 1$ nodes. Algorithm 5 presents the pseudocode of the implemented BFS algorithm.

---

**Algorithm 5** BFS($G, s$)

---

1: **for** each vertex $u \in G.V \setminus \{s\}$ **do**
2:     $u.white \leftarrow$ true
3:     $u.gray \leftarrow$ false
4:     $u.dist \leftarrow \infty$
5:     $u.\pi \leftarrow$ NIL
6:     $u.id \leftarrow$ id
7:     $u.priority \leftarrow NIL$
8: **end for**
9: $pointer \leftarrow 1$
10: $s.id \leftarrow s\_id$
11: $s.white \leftarrow$ false
12: $s.gray \leftarrow$ true
13: $s.dist \leftarrow 0$
14: $s.\pi \leftarrow$ NIL
15: $s.priority \leftarrow 1$
16: $queue \leftarrow 2$
17: **while** true **do**
18:     $u \leftarrow$ vertex $v \in G.V$ such that $v.priority = pointer$
19:     **for** each $v \in G.Adj[u]$ **do**
20:         **if** $v.white =$ true **then**
21:             $v.priority \leftarrow queue$
22:             $queue \leftarrow queue + 1$
23:             $v.white \leftarrow$ false
24:             $v.gray \leftarrow$ true
25:             $v.dist \leftarrow u.dist + 1$
26:             $v.\pi \leftarrow u.id$
27:         **end if**
28:     **end for**
29:     $pointer \leftarrow pointer + 1$
30: **end while**

---

#### E.3.1. BFS INSTRUCTIONS

We will begin by defining all the variables used in the instructions. Note that we adopt the little-endian convention for the binary representation of variables.

```
Variable definition:

  u-priority ∈ {0,1}ˡ
  # Any variable with a "-u-", "-u", or "u-" in its name holds information about the
    node itself. u-priority holds the node's priority in the queue to be expanded. Bits
    are indexed as u-priority[i] for i ∈ [l].

  q-pointer ∈ {0,1}ˡ
  # The algorithm has a variable called a ''pointer'' that indicates which node should
    be expanded next. A node learns the value of this pointer from the most recent
    message it receives. The pointer is stored in a variable named q-pointer. Its bits
    are indexed as q-pointer[i] for i ∈ [l].
```

compare-u-priority $\in \{0,1\}^l$
# Acts as a signal to start comparing bits of u-priority and q-pointer for potential matches. Bits are indexed as compare-u-priority[i] for i $\in [l]$.

priority-comparison-counter $\in \{0,1\}^{(l+1)}$
# Checking bit-by-bit equality for q-pointer and u-priority takes $l+1$ iterations. The active bit in the priority-comparison-counter indicates the current iteration number. Bits are indexed as priority-comparison-counter[i] for i $\in [l+1]$.

u-matches-pointer $\in \{0,1\}^l$
# The $i$-th bit in u-matches-pointer holds the result of comparing the $i$-th bit of the q-pointer and u-priority. Bits are indexed as compare-u-priority[i] for i $\in [l]$.

u-matches-counter $\in \{0,1\}^l$
# Checking that all bits in u-matches-pointer are set to 1 takes $l$ iterations. u-matches-counter shows the iteration number. Bits are indexed as u-matches-pointer[i] for i $\in [l]$.

reset-u-matches $\in \{0,1\}^l$
# The reset-u-matches signal resets u-matches-pointer after the comparison iterations are completed. Bits are indexed as reset-u-matches[i] for i $\in [l]$.

u-matched $\in \{0,1\}^1$
# keeps the final result of comparing u-priority and q-pointer. It has only one bit.

v-turn $\in \{0,1\}^D$
# Any variable with "-v-", "v-", or "-v" in its name holds information that nodes keep about their $D$ neighbours. Thus, if it is an array, one of its dimensions will be $D$. The variable v-turn determines which neighbour's turn it is to be discovered. Bits are indexed as v-turn[d] for d $\in [D]$.

v-white $\in \{0,1\}^D$
# Holds a flag for each neighbor of the node showing if it has been already discovered. Bits are indexed as v-white[d] for $i \in [D]$.

v-white-overwrite $\in \{0,1\}^D$
# Bits are indexed as v-white-overwrite[d] for d$\in [D]$. When the $d$-th bit is set to 1, it overwrites the corresponding v-white bit.

accept-last-in-q $\in \{0,1\}^l$
# The node becomes aware of the last number to be added to the queue through the latest message it receives. accept-last-in-q is a flag to accept the last number in the queue (last-in-q) received by the node. Bits are indexed as accept-last-in-q[i] for i $\in [l]$.

u-last-in-q $\in \{0,1\}^l$
# The received last-in-q is first stored in u-last-in-q. Bits are indexed as u-last-in-q[i] for i $\in [l]$.

v-set-priority $\in \{0,1\}^{D \times l}$
# Is a flag to allow setting priority number for a discovered neighbor. Bits are indexed as v-set-priority[d][i] for d $\in [D]$ and for i $\in [l]$.

v-last-in-q $\in \{0,1\}^{D \times l}$
# Keeps an up-to-date last-in-queue number for each neighbour that can be assigned as its priority when its turn comes up in the discovery loop. Bits are indexed as v-last-in-q[d][i] for d $\in [D]$ and for i $\in [l]$.

v-priority $\in \{0,1\}^{D \times l}$
# Keeps the priority assigned to each neighbour. Bits are indexed as v-priority[d][i] for d $\in [D]$ and i $\in [l]$.

upload-v-priority $\in \{0,1\}^{D \times l}$
# After the node is expanded, it sends a message to update its neighbours with the new
   changes. upload-v-priority is a flag used to copy the neighbours' priorities into a
   buffer. Later, information from the buffer will be included in the outgoing message.
   Bits are indexed as upload-v-priority[d][i] for $d \in [D]$ and $i \in [l]$.

v-last-in-q-augend $\in \{0,1\}^{D \times l}$
# Once a priority is assigned for a neighbour, the last-in-q number should be
   incremented. v-last-in-q-augend holds the augend of the addition process. Bits are
   indexed as v-last-in-q-augend[d][i] for $d \in [D]$ and $i \in [l]$.

v-last-in-q-addend $\in \{0,1\}^{D \times l}$
# This holds the addend value used to increment the last-in-q value for the next
   discovery. Bits are indexed as v-last-in-q-addend[d][i] for $d \in [D]$ and $i \in [l]$.

to-copy-last-in-q-augend $\in \{0,1\}^{D \times l}$
# This is a flag that allows copying the last-in-q value, after incrementing it, to
   the next neighbor's value in v-last-in-q. Bits are indexed as to-copy-last-in-q-
   augend[d][i] for $d \in [D]$ and $i \in [l]$.

v-last-in-q-carry $\in \{0,1\}^{D \times l}$
# This is a variable used to handle the carry value in the binary addition process of
   incrementing the last-in-q value. Bits are indexed as v-last-in-q-carry[d][i] for $d$
   $\in [D]$ and $i \in [l]$.

v-last-in-q-sum-counter $\in \{0,1\}^{D \times 2l}$
# This is a counter to determine when the iterations required for the binary process
   of incrementing the last-in-q value are completed. Bits are indexed as v-last-in-q-
   sum-counter[d][j] for $d \in [D]$ and $j \in [2l]$.

u-dist $\in \{0,1\}^{l}$
# Stores the distance of the node from the source. Bits are indexed as u-dist[i] for $i$
   $\in [l]$.

update-v-dist-augend $\in \{0,1\}^{l}$
# Acts as a flag to copy the value of u-dist into a variable named u-dist-augend,
   which will be used in the process of incrementing the distance. Bits are indexed as
   update-v-dist-augend[i] for $i \in [l]$.

v-dist-augend $\in \{0,1\}^{l}$
# The u-dist should be incremented to be assigned as the distance of neighbours. v-
   dist-augend holds a copy of u-dist for the addition process. Bits are indexed as v-
   dist-augend[i] for $i \in [l]$.

v-dist-addend $\in \{0,1\}^{l}$
# Holds the binary value to be added to the current distance. Bits are indexed as v-
   dist-addend[i] for $i \in [l]$

to-copy-dist-augend $\in \{0,1\}^{l}$
# Acts as a flag to transfer $D$ copies of the result of the distance addition into a
   buffer named v-dist-incremented. Bits are indexed as to-copy-dist-augend[i] for $i \in [l]$
   .

v-dist-carry $\in \{0,1\}^{l}$
# A variable used to handle the carry bit in the binary addition process of distance.
   Bits are indexed as v-dist-carry[i] for $i \in [l]$.

v-dist-sum-counter $\in \{0,1\}^{2l}$
# A variable used to control the required number of iterations for the addition
   process of the distance to be completed. Bits are indexed as v-dist-sum-counter[j]
   for $j \in [2l]$.

v-set-dist-counter $\in \{0,1\}^{D \times 2l}$

```
# A counter that makes sure the flag to set the distance of the discovered neighbours
  is not activated before the new distance has been computed. Bits are indexed as v-set
  -dist-counter[d][j] for d ∈ [D] and for j ∈ [2l].
```

v-set-dist $\in \{0,1\}^{D \times l}$
```
# A flag that allows assigning distance to the discovered neighbours. Bits are indexed
  as v-set-dist[d][i] for d ∈ [D] and for i ∈ [l].
```

v-dist-incremented $\in \{0,1\}^{D \times l}$
```
# Holds D copies of the distance calculated by incrementing the distance of the
  current node. Bits are indexed as v-dist-incremented[d][i] for d ∈ [D] and for i ∈ [l].
```

v-dist $\in \{0,1\}^{D \times l}$
```
# Hold the distance of each neighbour from the source. Bits are indexed as v-dist[d][i
  ] for d ∈ [D] and for i ∈ [l].
```

upload-v-dist $\in \{0,1\}^{D \times l}$
```
# Acts as a flag to copy the v-dist into a buffer that later will be included in the
  outgoing message. Bits are indexed as upload-v-dist[d][i] for d ∈ [D] and for i ∈ [l]
```

u-id $\in \{0,1\}^{l}$
```
# Holds the global ID of the current node. Bits are indexed as u-id[i] for i ∈ [l].
```

u-id-asparent $\in \{0,1\}^{D \times l}$
```
# Holds D copies of the u-id to assign it as the parent ID to the discovered neighbour
  nodes. Bits are indexed as u-id-asparent[d][i] for d ∈ [D] and for i ∈ [l].
```

v-set-$\pi$ $\in \{0,1\}^{D \times l}$
```
# Acts as a flag that allows setting the parent ID of the discovered nodes. Bits are
  indexed as v-set-π[d][i] for d ∈ [D] and for i ∈ [l].
```

v-$\pi$ $\in \{0,1\}^{D \times l}$
```
# Holds the parent ID of each of the neighbours. Bits are indexed as v-π[d][i] for
  d ∈ [D] and for i ∈ [l].
```

upload-v-$\pi$ $\in \{0,1\}^{D \times l}$
```
# Acts as a flag to copy v-π into a buffer whose content will be included in the
  outgoing message. Bits are indexed as upload_v_pi[d][i] for d ∈ [D] and for i ∈ [l].
```

u-white $\in \{0,1\}^{1}$
```
# This is a single bit that, if active, shows the node has not been discovered yet.
```

u-white-overwrite $\in \{0,1\}^{1}$
```
# This is a single bit that, once activated, overwrites u-white
```

u-gray $\in \{0,1\}^{1}$
```
# This is a single bit that tells if the node is already discovered.
```

u-$\pi$ $\in \{0,1\}^{l}$
```
# Stores the parent of each node. Bits are indexed as u-π[i] for i ∈ [l].
```

v-gray $\in \{0,1\}^{D}$
```
# Tells if each neighbor has already been discovered. Bits are indexed as v-gray[d]
  for d ∈ [D]
```

upload-v-gray $\in \{0,1\}^{D}$
```
# Acts as a flag to copy v-gray into a buffer whose content will be included in the
  outgoing message. Bits are indexed as upload-v-gray[d] for d ∈ [D].
```

upload-message-flag $\in \{0,1\}^{1}$
```
# The outgoing message has a flag that notifies the receiver node of an incoming
  message. The upload-message-flag is a single bit that creates D² + 1 copies of the
  flag in a buffer with D² + 1 slots. Later, the flag from the slot that matches the
```

```
    local ID of the node will be included in the outgoing message.
```

pointer-augend $\in \{0,1\}^l$
```
# After a node is expanded, it should increment the pointer and include it in its
  outgoing message. pointer-augend is a copy of the current pointer for the addition
  process. Bits are indexed as pointer-augend[i] for i ∈ [l].
```

pointer-addend $\in \{0,1\}^l$
```
# This is the added value to the pointer. Bits are indexed as pointer-addend[i] for i
  ∈ [l].
```

upload-pointer-augend $\in \{0,1\}^l$
```
# Acts as a flag to make copies of the incremented pointer in a buffer whose content
  will be included in the outgoing message. Bits are indexed as upload-pointer-augend[i
  ] for i ∈ [l].
```

pointer-carry $\in \{0,1\}^l$
```
# Handles the carry bit in the binary addition of the pointer. Bits are indexed as
  pointer-carry[i] for i ∈ [l].
```

pointer-sum-counter $\in \{0,1\}^l$
```
# Counts the number of required iterations for the addition process of the pointer to
  be complete. Bits are indexed as pointer-sum-counter[i] for i ∈ [l].
```

after-loop-last-in-q $\in \{0,1\}^l$
```
# Holds the last-in-queue value after all neighbors are processed in the discovery
  loop. This value will be included in the outgoing message. Bits are indexed as after-
  loop-last-in-q[i] for i ∈ [l].
```

upload-after-loop-last-in-q $\in \{0,1\}^l$
```
# Acts as a flag to make copies of after-loop-last-in-q in a buffer whose content will
   be included in the outgoing message. Bits are indexed as upload-after-loop-last-in-q
  [i] for i ∈ [l].
```

determine-slot $\in \{0,1\}^{D^2+1}$
```
# The variables that will be included in x_{u_M} for message-passing all have one
  dimension of size D² + 1, with coordinates referred to as slots. The node can modify
  only one of the slots for a variable that will be in x_{u_M}. Based on the node's local
  ID, one of the D² + 1 bits in determine-slot is set to 1. The active bit in determine
  -slot determines which slot the node should use for message-passing. For convenience,
   for the other variables in x_{u_C} that have this dimension, the term slot will be used
  as well. Note that for variables in x_{u_C}, the node can modify all slots if required.
   Note that the variables that are defined, including determine-slot, are in x_{u_C}. We
  will explicitly mention in the definition if this is not the case, i.e., if the
  variable is in x_{u_M}. The bits in determine-slot are indexed as determine-slot[s] for
  s ∈ [D² + 1].
```

determine-pointer-slot $\in \{0,1\}^{(D^2+1)\times l}$
```
# Tells which slot the node should use to include the latest pointer in the outgoing
  message. Bits are indexed as determine-pointer-slot[s][i] for s ∈ [D² + 1] and for i
  ∈ [l].
```

pointer-placeholder $\in \{0,1\}^{(D^2+1)\times l}$
```
# This holds D² + 1 copies of the outgoing pointer value. Only one slot, determined by
  determine-pointer-slot, will be used in the outgoing message. Bits are indexed as
  pointer-placeholder[s][i] for s ∈ [D² + 1] and for i ∈ [l].
```

determine-last-in-q-slot $\in \{0,1\}^{(D^2+1)\times l}$
```
# Tells which slot the node should use to include the latest last-in-q in the outgoing
   message. Bits are indexed as determine-last-in-q-slot[s][i] for s ∈ [D² + 1] and for i
  ∈ [l].
```

last-in-q-placeholder $\in \{0,1\}^{(D^2+1)\times l}$
*# This holds $D^2+1$ copies of the outgoing last-in-q value. Only one slot, determined by determine-last-in-q-slot, will be used in the outgoing message. Bits are indexed as last-in-q-placeholder[s][i] for $s \in [D^2+1]$ and for $i \in [l]$.*

determine-v-gray-slot $\in \{0,1\}^{(D^2+1)\times D}$
*# Tells which slot the node should use to include the latest v-gray in the outgoing message. Bits are indexed as determine-v-gray-slot[s][d] for $s \in [D^2+1]$ and for $d \in [D]$.*

v-gray-placeholder $\in \{0,1\}^{(D^2+1)\times D}$
*# This holds $D^2+1$ copies of the outgoing v-gray values. Only one slot, determined by determine-v-gray-slot, will be used in the outgoing message. Bits are indexed as v-gray-placeholder[s][d] for $s \in [D^2+1]$ and for $d \in [D]$.*

determine-v-dist-slot $\in \{0,1\}^{(D^2+1)\times D\times l}$
*# Tells which slot the node should use to include the latest v-dist in the outgoing message. Bits are indexed as determine-v-dist-slot[s][d][i] for $s \in [D^2+1]$, for $d \in [D]$, and for $i \in [l]$.*

v-dist-placeholder $\in \{0,1\}^{(D^2+1)\times D\times l}$
*# This holds $D^2+1$ copies of the outgoing v-dist values. Only one slot, determined by determine-v-dist-slot, will be used in the outgoing message. Bits are indexed as v-dist-placeholder[s][d][i] for $s \in [D^2+1]$, for $d \in [D]$, and for $i \in [l]$.*

determine-v-$\pi$-slot $\in \{0,1\}^{(D^2+1)\times D\times l}$
*# Tells which slot the node should use to include the latest v-$\pi$ in the outgoing message. Bits are indexed as determine-v-$\pi$-slot[s][d][i] for $s \in [D^2+1]$, for $d \in [D]$, and for $i \in [l]$.*

v-$\pi$-placeholder $\in \{0,1\}^{(D^2+1)\times D\times l}$
*# This holds $D^2+1$ copies of the outgoing v-$\pi$ values. Only one slot, determined by determine-v-$\pi$-slot, will be used in the outgoing message. Bits are indexed as v-$\pi$-placeholder[s][d][i] for $s \in [D^2+1]$, for $d \in [D]$, and for $i \in [l]$.*

determine-v-id-slot $\in \{0,1\}^{(D^2+1)\times D\times l}$
*# Tells which slot the node should use to include the latest v-id in the outgoing message. Bits are indexed as determine-v-id-slot[s][d][i] for $s \in [D^2+1]$, for $d \in [D]$, and for $i \in [l]$.*

v-id-placeholder $\in \{0,1\}^{(D^2+1)\times D\times l}$
*# This holds $D^2+1$ copies of the outgoing v-id values (these are IDs of the neighbours). Only one slot, determined by determine-v-id-slot, will be used in the outgoing message. Bits are indexed as v-id-placeholder[s][d][i] for $s \in [D^2+1]$, for $d \in [D]$, and for $i \in [l]$.*

determine-v-priority-slot $\in \{0,1\}^{(D^2+1)\times D\times l}$
*# Tells which slot the node should use to include the latest v-priority in the outgoing message. Bits are indexed as determine-v-priority-slot[s][d][i] for $s \in [D^2+1]$, for $d \in [D]$, and for $i \in [l]$.*

v-priority-placeholder $\in \{0,1\}^{(D^2+1)\times D\times l}$
*# This holds $D^2+1$ copies of the outgoing v-priority values. Only one slot, determined by determine-v-priority-slot, will be used in the outgoing message. Bits are indexed as v-priority-placeholder[s][d][i] for $s \in [D^2+1]$, for $d \in [D]$, and for $i \in [l]$.*

determine-message-flag-slot $\in \{0,1\}^{D^2+1}$
*# Tells which slot the node should use to include the message-flag in the outgoing message. Bits are indexed as determine-message-flag-slot[s] for $s \in [D^2+1]$*

message-flag-placeholder $\in \{0,1\}^{D^2+1}$
*# This holds $D^2+1$ copies of the outgoing message-flag values. Only one slot, determined by determine-message-flag-slot, will be used in the outgoing message. Bits are indexed as message-flag-placeholder[s] for $s \in [D^2+1]$.*

pointer-slot $\in \{0,1\}^{(D^2+1)\times l}$
*# This is the variable in $\boldsymbol{x}_{u_M}$ that the node uses to share the latest pointer, writing it exclusively to the slot corresponding to the local ID. Also, after the aggregation step, the rest of the slots might contain pointers that other nodes have shared. Bits are indexed as pointer-slot[s][i] for $s \in [D^2+1]$ and for $i \in [l]$.*

last-in-q-slot $\in \{0,1\}^{(D^2+1)\times l}$
*# This is the variable in $\boldsymbol{x}_{u_M}$ that the node uses to share the latest last-in-q, writing it exclusively to the slot corresponding to the local ID. Also, after the aggregation step, the rest of the slots might contain last-in-q values that other nodes have shared. Bits are indexed as last-in-q-slot[s][i] for $s \in [D^2+1]$ and $i \in [l]$.*

v-gray-slot $\in \{0,1\}^{(D^2+1)\times D}$
*# This is the variable in $\boldsymbol{x}_{u_M}$ that the node uses to share the latest v-gray, writing it exclusively to the slot corresponding to the local ID. Also, after the aggregation step, the rest of the slots might contain v-gray that other nodes have shared. Bits are indexed as v-gray-slot[s][d] for $s \in [D^2+1]$ and for $d \in [D]$.*

v-dist-slot $\in \{0,1\}^{(D^2+1)\times D \times l}$
*# This is the variable in $\boldsymbol{x}_{u_M}$ that the node uses to share the latest v-dist, writing it exclusively to the slot corresponding to the local ID. Also, after the aggregation step, the rest of the slots might contain v-dists that other nodes have shared. Bits are indexed as v-dist-slot[s][d][i] for $s \in [D^2+1]$, for $d \in [D]$, and for $i \in [l]$.*

v-id-slot $\in \{0,1\}^{(D^2+1)\times D \times l}$
*# This is the variable in $\boldsymbol{x}_{u_M}$ that the node uses to share the latest v-id, writing it exclusively to the slot corresponding to the local ID. Also, after the aggregation step, the rest of the slots might contain v-ids that other nodes have shared. Bits are indexed as v-id-slot[s][d][i] for $s \in [D^2+1]$ and for $d \in [D]$ and for $i \in [l]$*

v-$\pi$-slot $\in \{0,1\}^{(D^2+1)\times D \times l}$
*# This is the variable in $\boldsymbol{x}_{u_M}$ that the node uses to share the latest v-$\pi$, writing it exclusively to the slot corresponding to the local ID. Also, after the aggregation step, the rest of the slots might contain v-$\pi$ that other nodes have shared. Bits are indexed as v-$\pi$-slot[s][d][i] for $s \in [D^2+1]$, for $d \in [D]$, and for $i \in [l]$.*

v-priority-slot $\in \{0,1\}^{(D^2+1)\times D \times l}$
*# This is the variable in $\boldsymbol{x}_{u_M}$ that the node uses to share the latest v-priority, writing it exclusively to the slot corresponding to the local ID. Also, after the aggregation step, the rest of the slots might contain v-priority values that other nodes have shared. Bits are indexed as v-priority-slot[s][d][i] for $s \in [D^2+1]$, for $d \in [D]$, and for $i \in [l]$.*

message-flag-slot $\in \{0,1\}^{D^2+1}$
*# This is the variable in $\boldsymbol{x}_{u_M}$ that the node uses to share message-flag, writing it exclusively to the slot corresponding to the local ID. Also, after the aggregation step, the rest of the slots might contain message-flag that other nodes have shared. Bits are indexed as message-flag-slot[s] for $s \in [D^2+1]$.*

persistent-last-pointer $\in \{0,1\}^{(D^2+1)\times l}$
*# Holds $D^2+1$ copies of the latest pointer that the node is aware of. These are used for comparison with pointers in incoming messages. Bits are indexed as persistent-last-pointer[s][i] for $s \in [D^2+1]$ and for $i \in [l]$.*

overwrite-last-pointer $\in \{0,1\}^{(D^2+1)\times l}$
*# Acts as a flag to overwrite the current value of pointer in persistent-last-pointer.*

```
    Bits are indexed as overwrite-last-pointer[s][i] for s ∈ [D² + 1] and for i ∈ [l].
```

```
temp-last-in-q-slot ∈ {0,1}^{(D²+1)×l}
# When a new value is received in one of the slots of last-in-q-slot, it is
  immediately copied to temp-last-in-q-slot outside of x_{u_M} for further processing.
  Bits are indexed as temp-last-in-q-slot[s][i] for s ∈ [D² + 1] and for i ∈ [l].
```

```
to-accept-temp-last-in-q ∈ {0,1}^{(D²+1)×l}
# Acts as a flag to allow further processing of the value in temp-last-in-q-slot.
  Activated when the pointer of the incoming message is larger than the current pointer
  , showing a new message. Bits are indexed as to-accept-temp-last-in-q[s][i] for s
  ∈ [D² + 1] and for i ∈ [l].
```

```
overwrite-temp-last-in-q ∈ {0,1}^{(D²+1)×l}
# Acts as a flag to overwrite the value in temp-last-in-q for the next incoming
  messages. Bits are indexed as overwrite-temp-last-in-q[s][i] for s ∈ [D² + 1] and for i
  ∈ [l].
```

```
temp-v-gray-slot ∈ {0,1}^{(D²+1)×D}
# When a new value is received in one of the slots of v-gray-slot, it is immediately
  copied to temp-v-gray-slot outside of x_{u_M} for further processing. Bits are indexed
  as temp-v-gray-slot[s][d] for s ∈ [D² + 1] and for d ∈ [D].
```

```
to-accept-temp-v-gray ∈ {0,1}^{(D²+1)×D}
# Acts as a flag to allow further processing of the value in temp-v-gray-slot.
  Activated when the pointer of the incoming message is larger than the current pointer
  , indicating a new message. Bits are indexed as to-accept-temp-v-gray[s][d] for s
  ∈ [D² + 1] and for d ∈ [D].
```

```
overwrite-temp-v-gray ∈ {0,1}^{(D²+1)×D}
# Acts as a flag to overwrite the value in temp-v-gray for the next incoming messages.
   Bits are indexed as overwrite-temp-v-gray[s][d] for s ∈ [D² + 1] and for d ∈ [D].
```

```
temp-v-dist-slot ∈ {0,1}^{(D²+1)×D×l}
# When a new value is received in one of the slots of v-dist-slot, it is immediately
  copied to temp-v-dist-slot outside of x_{u_M} for further processing. Bits are indexed
  as temp-v-dist-slot[s][d][i] for s ∈ [D² + 1], for d ∈ [D], and for i ∈ [l].
```

```
to-accept-temp-v-dist ∈ {0,1}^{(D²+1)×D×l}
# Acts as a flag that allows further processing of the value in temp-v-dist-slot. It
  is activated when the pointer of the incoming message is larger than the current
  pointer, indicating a new message. Bits are indexed as to-accept-temp-v-dist[s][d][i]
   for s ∈ [D² + 1], for d ∈ [D], and for i ∈ [l].
```

```
overwrite-temp-v-dist ∈ {0,1}^{(D²+1)×D×l}
# Acts as a flag to overwrite the value in temp-v-dist for the next incoming messages.
   Bits are indexed as overwrite-temp-v-dist[s][d][i] for s ∈ [D² + 1], d ∈ [D], and i ∈ [l]
   .
```

```
temp-v-id-slot ∈ {0,1}^{(D²+1)×D×l}
# When a new value is received in one of the slots of v-id-slot, it is immediately
  copied to temp-v-id-slot outside of x_{u_M} for further processing. Bits are indexed as
  temp-v-id-slot[s][d][i] for s ∈ [D² + 1], for d ∈ [D], and for i ∈ [l].
```

```
to-accept-temp-v-id ∈ {0,1}^{(D²+1)×D×l}
# Acts as a flag to allow further processing of the value in temp-v-id-slot. It is
  activated when the pointer of the incoming message is larger than the current pointer
  , indicating a new message. Bits are indexed as to-accept-temp-v-id[s][d][i] for s
  ∈ [D² + 1], for d ∈ [D], and for i ∈ [l].
```

```
overwrite-temp-v-id ∈ {0,1}^{(D²+1)×D×l}
```

```
# Acts as a flag to overwrite the value in temp-v-id for the next incoming messages.
  Bits are indexed as overwrite-temp-v-id[s][d][i] for s ∈ [D² + 1], d ∈ [D], and i ∈ [l].
```

`temp-v-π-slot` $\in \{0,1\}^{(D^2+1) \times D \times l}$
```
# When a new value is received in one of the slots of the v-π-slot, it is immediately
  copied to temp-v-π-slot outside of x_{u_M} for further processing. Bits are indexed as
  temp-v-π-slot[s][d][i] for s ∈ [D² + 1], d ∈ [D], and i ∈ [l].
```

`to-accept-temp-v-π` $\in \{0,1\}^{(D^2+1) \times D \times l}$
```
# Acts as a flag that allows further processing of the value in temp-v-π-slot. It is
  activated when the pointer of the incoming message is larger than the current pointer
  , indicating a new message. Bits are indexed as to-accept-temp-v-π[s][d][i] for s
  ∈ [D² + 1], d ∈ [D], and i ∈ [l].
```

`overwrite-temp-v-π` $\in \{0,1\}^{(D^2+1) \times D \times l}$
```
# Acts as a flag to overwrite the value in temp-v-π for the next incoming messages.
  Bits are indexed as overwrite-temp-v-π[s][d][i] for s ∈ [D² + 1], for d ∈ [D], and for i
  ∈ [l].
```

`temp-v-priority-slot` $\in \{0,1\}^{(D^2+1) \times D \times l}$
```
# When a new value is received in one of the slots of v-priority-slot, it is
  immediately copied to temp-v-priority-slot outside of x_{u_M} for further processing.
  Bits are indexed as temp-v-priority-slot[s][d][i] for s ∈ [D² + 1], for d ∈ [D], and for
  i ∈ [l].
```

`to-accept-temp-v-priority` $\in \{0,1\}^{(D^2+1) \times D \times l}$
```
# Acts as a flag to allow further processing of the value in temp-v-priority-slot.
  Activated when the pointer of the incoming message is larger than the current pointer
  , indicating a new message. Bits are indexed as to-accept-temp-v-priority[s][d][i]
  for s ∈ [D² + 1], for d ∈ [D], and for i ∈ [l].
```

`overwrite-temp-v-priority` $\in \{0,1\}^{(D^2+1) \times D \times l}$
```
# Acts as a flag to overwrite the value in temp-v-priority for the next incoming
  messages. Bits are indexed as overwrite-temp-v-priority[s][d][i] for s ∈ [D² + 1], for
  d ∈ [D], and for i ∈ [l].
```

`return-message-flag` $\in \{0,1\}^{D^2+1}$
```
# Acts as a flag indicating that the incoming message is new. It is activated when the
  pointer of the incoming message is larger than the current pointer, indicating a new
  message. Bits are indexed as return-message-flag[s] for s ∈ [D² + 1].
```

`compare-last-pointer` $\in \{0,1\}^{(D^2+1) \times l}$
```
# Acts as a flag to allow bit-by-bit comparison between the pointer in the incoming
  message and the last local pointer. Bits are indexed as compare-last-pointer[s][i]
  for s ∈ [D² + 1] and for i ∈ [l].
```

`temp-pointer-slot` $\in \{0,1\}^{(D^2+1) \times l}$
```
# Holds a copy of the newly received pointer in any of the slots. It is used for bit-
  by-bit comparison with the last local pointer. Bits are indexed as temp-pointer-slot[
  s][i] for s ∈ [D² + 1] and i ∈ [l].
```

`last-pointer` $\in \{0,1\}^{(D^2+1) \times l}$
```
# Holds a copy of the last local pointer in each of its slots for comparison with the
  pointer of an incoming message in that slot. Bits are indexed as last-pointer[s][i]
  for s ∈ [D² + 1] and for i ∈ [l].
```

`to-accept-temp-pointer` $\in \{0,1\}^{(D^2+1) \times l}$
```
# Acts as a flag to allow further processing of the pointer value in temp-pointer-slot
  . Activated when the incoming message is deemed new. Bits are indexed as to-accept-
  temp-pointer[s][i] for s ∈ [D² + 1] and for i ∈ [l].
```

overwrite-temp-pointer $\in \{0,1\}^{(D^2+1)\times l}$
*# Acts as a flag to overwrite the pointer value in temp-pointer-slot. Bits are indexed*
  *as overwrite-temp-pointer[s][i] for $s \in [D^2+1]$ and for $i \in [l]$.*

new-message $\in \{0,1\}^{(D^2+1)\times l}$
*# Acts as a flag, showing a new message has arrived in any of the slots. At a bit-by-*
  *bit comparison of an incoming pointer value and the last local pointer in a slot, if*
  *the incoming pointer is larger, one of the bits in the same slot of new-message will*
  *be activated. Bits are indexed as new-message[s][i] for $s \in [D^2+1]$ and $i \in [l]$.*

old-message $\in \{0,1\}^{(D^2+1)\times l}$
*# Acts as a flag, showing that the message received in any of the slots is not new. A*
  *bit-by-bit comparison of an incoming pointer value and the last local pointer in a*
  *slot is performed; if the incoming pointer is equal to or smaller, one of the bits in*
  *the same slot of old-message will be activated. Bits are indexed as old-message[s][i*
  *] for $s \in [D^2+1]$ and for $i \in [l]$.*

delay-new-last-in-q $\in \{0,1\}^{(D^2+2)\times l}$
*# When a message received in any of the slots is deemed new, the last-in-q value*
  *inside it is copied from the temporary buffer into the corresponding slot of delay-*
  *new-last-in-q. Bits are indexed as delay-new-last-in-q[s][i] for $s \in [D^2+2]$ and for i*
  *$\in [l]$. The variables with the prefix 'delay-' act as a cascade chain, where the value*
  *moves from one slot to the next until it reaches the last slot. At that point, the*
  *value is copied into a destination variable. This helps avoid having multiple*
  *instructions all writing into the destination variable, which can lead to a growing*
  *number of collisions.*

shutdown-last-in-q-delay $\in \{0,1\}^{(D^2+2)\times l}$
*# Acts as a flag to remove the values in all slots of the delay-new-last-in-q. Bits*
  *are indexed as shutdown-last-in-q-delay[s][i] for $s \in [D^2+2]$ and for $i \in [l]$.*

delay-new-pointer $\in \{0,1\}^{(D^2+2)\times l}$
*# When a message received in any of the slots is deemed new, the pointer value inside*
  *it is copied from the temporary buffer into the corresponding slot of delay-new-*
  *pointer. Bits are indexed as delay-new-pointer[s][i] for $s \in [D^2+2]$ and $i \in [l]$.*

shutdown-pointer-delay $\in \{0,1\}^{(D^2+2)\times l}$
*# Acts as a flag to remove the values in all slots of the delay-new-pointer. Bits are*
  *indexed as shutdown-pointer-delay[s][i] for $s \in [D^2+2]$ and for $i \in [l]$.*

delay-new-v-gray $\in \{0,1\}^{(D^2+2)\times D}$
*# When a message received in any of the slots is deemed new, the v-gray value inside*
  *it is copied from the temporary buffer into the corresponding slot of delay-new-v-*
  *gray. Bits are indexed as delay-new-v-gray[s][d] for $s \in [D^2+2]$ and for $d \in [D]$.*

shutdown-v-gray-delay $\in \{0,1\}^{(D^2+2)\times D}$
*# Acts as a flag to remove the values in all slots of the delay-new-v-gray. Bits are*
  *indexed as shutdown-v-gray-delay[s][d] for $s \in [D^2+2]$ and for $d \in [D]$.*

delay-new-v-priority $\in \{0,1\}^{(D^2+2)\times D\times l}$
*# When a message received in any of the slots is deemed new, the v-priority value*
  *inside it is copied from the temporary buffer into the corresponding slot of delay-*
  *new-v-priority. Bits are indexed as delay-new-v-priority[s][d][i] for $s \in [D^2+2]$, for*
  *$d \in [D]$, and for $i \in [l]$.*

shutdown-v-priority-delay $\in \{0,1\}^{(D^2+2)\times D\times l}$
*# Acts as a flag to remove the values in all slots of the delay-new-v-priority. Bits*
  *are indexed as shutdown-v-priority-delay[s][d][i] for $s \in [D^2+2]$, for $d \in [D]$, and for*
  *$i \in [l]$.*

delay-new-v-dist $\in \{0,1\}^{(D^2+2)\times D\times l}$

```
# When a message received in any of the slots is deemed new, the v-dist value inside
  it is copied from the temporary buffer into the corresponding slot of delay-new-v-
  dist. Bits are indexed as delay-new-v-dist[s][d][i] for s ∈ [D² + 2], for d ∈ [D], and
  for i ∈ [l].
```

```
shutdown-v-dist-delay ∈ {0, 1}^(D²+2)×D×l
# Acts as a flag to remove the values in all slots of the delay-new-v-dist. Bits are
  indexed as shutdown-v-dist-delay[s][d][i] for s ∈ [D² + 2], for d ∈ [D], and for i ∈ [l].
```

```
delay-new-v-id ∈ {0, 1}^(D²+2)×D×l
# When a message received in any of the slots is deemed new, the v-id value inside it
  is copied from the temporary buffer into the corresponding slot of delay-new-v-id.
  Bits are indexed as delay-new-v-id[s][d][i] for s ∈ [D² + 2], for d ∈ [D], and for i ∈ [l]
  .
```

```
shutdown-v-id-delay ∈ {0, 1}^(D²+2)×D×l
# Acts as a flag to remove the values in all slots of the delay-new-v-id. Bits are
  indexed as shutdown-v-id-delay[s][d][i] for s ∈ [D² + 2], for d ∈ [D], and for i ∈ [l].
```

```
delay-new-v-π ∈ {0, 1}^(D²+2)×D×l
# When a message is received in any of the slots and deemed new, the v-π value inside
  it is copied from the temporary buffer into the corresponding slot of delay-new-v-π.
  Bits are indexed as delay-new-v-π[s][d][i] for s ∈ [D² + 2], for d ∈ [D], and for i ∈ [l].
```

```
shutdown-v-π-delay ∈ {0, 1}^(D²+2)×D×l
# Acts as a flag to remove the values in all slots of the delay-new-v-π. Bits are
  indexed as shutdown-v-π-delay[s][d][i] for s ∈ [D² + 2], for d ∈ [D], and for i ∈ [l].
```

```
delay-new-message-flag ∈ {0, 1}^(D²+2)
# When a message received in any of the slots is deemed new, the new-message-flag is
  copied into the corresponding slot of delay-new-message-flag. Bits are indexed as
  variable[s] for s ∈ [D² + 2].
```

```
shutdown-message-flag-delay ∈ {0, 1}^(D²+2)
# Acts as a flag to remove the values in all slots of the delay-new-message-flag. Bits
  are indexed as shutdown-message-flag-delay[s] for s ∈ [D² + 2].
```

```
v-id ∈ {0, 1}^D×l
# Store the ID of the neighbours of the node. Bits are indexed as v-id[d][i] for d
  ∈ [D] and for i ∈ [l].
```

```
upload-v-id ∈ {0, 1}^D×l
# Acts as a flag to create copies of v-ids in a buffer, whose content will be shared
  during message-passing. The bits are indexed as upload-v-id[d][i] for d ∈ [D] and for
  i ∈ [l].
```

```
received-new-message ∈ {0, 1}^1
# A flag bit to indicate a new message has been received and its content has passed
  the delay mechanism.
```

```
received-v-ids ∈ {0, 1}^D×l
# Holds the ID of the nodes for which the received message (accepted new message) has
  new information. Bits are indexed as received-v-ids[k][i] for k ∈ [D] and for i ∈ [l].
```

```
u-id2check-with-received-v-ids ∈ {0, 1}^D×l
# Holds copies of the node's ID to compare with each of the IDs in the received
  message. Bits are indexed as u-id2check-with-received-v-ids[k][i] for k ∈ [D] and for
  i ∈ [l].
```

```
check-u-id-with-received-v-ids ∈ {0, 1}^D×l
# Acts as a flag to allow bit-by-bit comparison of u-id copies and the IDs in the
  received message for a potential match. Bits are indexed as check-u-id-with-received-
```

```
      v-ids[k][i] for k ∈ [D] and for i ∈ [l].
```

received-v-ids2check-with-v-ids $\in \{0,1\}^{D \times D \times l}$
```
# Holds D copies of each of the IDs in the received message for comparison with the
  IDs of the node's neighbours for a potential match. Bits are indexed as received-v-
  ids2check-with-v-ids[d][k][i] for d ∈ [D], for k ∈ [D], and for i ∈ [l].
```

v-ids2check-with-received-v-ids $\in \{0,1\}^{D \times D \times l}$
```
# Holds D copies of each of the node's neighbors' IDs to compare with the received IDs
   for a potential match. Bits are indexed as v-ids2check-with-received-v-ids[d][k][i]
  for d ∈ [D], for k ∈ [D], and for i ∈ [l].
```

check-v-ids-with-received-v-ids $\in \{0,1\}^{D \times D \times l}$
```
# Acts as a flag to allow bit-by-bit comparison of copies of the node's neighbours'
  IDs and the IDs in the received message for a potential match. Bits are indexed as
  check-v-ids-with-received-v-ids[d][k][i] for d ∈ [D], for k ∈ [D], and for i ∈ [l].
```

u-equals-received-v-id $\in \{0,1\}^{D \times l}$
```
# Holds the per-bit result of the comparison between the u-id and the received IDs.
  Bits are indexed as u-equals-received-v-id[k][i] for k ∈ [D] and for i ∈ [l].
```

u-is-received-v-id-counter $\in \{0,1\}^{D \times l}$
```
# Counts the number of matching bits when comparing u-id with each of the received IDs
  . Bits are indexed as u-is-received-v-id-counter[k][i] for k ∈ [D] and for i ∈ [l].
```

reset-u-equals-v $\in \{0,1\}^{D \times l}$
```
# Acts as a flag to reset the results of comparisons in u-equals-received-v-id and u-
  is-received-v-id-counter for future comparisons. Bits are indexed as reset-u-equals-v
  [k][i] for k ∈ [D] and for i ∈ [l].
```

v-equals-received-v-id $\in \{0,1\}^{D \times D \times l}$
```
# Holds the per-bit result of the comparison between each neighbor ID and each of the
  received IDs. Bits are indexed as v-equals-received-v-id[d][k][i] for d ∈ [D], for k
  ∈ [D], and for i ∈ [l].
```

v-is-received-v-id-counter $\in \{0,1\}^{D \times D \times l}$
```
# Counts the number of matching bits for the comparison of each neighbour ID with each
   of the received IDs. Bits are indexed as v-is-received-v-id-counter[d][k][i] for d
  ∈ [D], for k ∈ [D], and for i ∈ [l].
```

reset-v-equals-v $\in \{0,1\}^{D \times D \times l}$
```
# Acts as a flag to reset the results of comparisons in v-equals-received-v-id and v-
  is-received-v-id-counter for future comparisons. Bits are indexed as reset-v-equals-v
  [d][k][i] for d ∈ [D], for k ∈ [D], and for i ∈ [l].
```

u-id-comparison-counter $\in \{0,1\}^{D \times (l+1)}$
```
# Checks the number of iterations required for the comparison between u-id and the
  received IDs to be complete. The bits are indexed as u-id-comparison-counter[k][j]
  for k ∈ [D] and for j ∈ [l + 1].
```

v-id-comparison-counter $\in \{0,1\}^{D \times D \times (l+1)}$
```
# Checks the number of iterations required for the comparison between neighbour IDs
  and the received IDs to be complete. Bits are indexed as v-id-comparison-counter[d][k
  ][j] for d ∈ [D], for k ∈ [D], and for j ∈ [l + 1].
```

update-pointer-counter $\in \{0,1\}^{D+l+2}$
```
# Acts as a counter that, upon reaching the end, updates a local pointer within the
  node. This pointer is compared with the node's priority (u-priority), and if they are
   equal, the node is expanded. Bits are indexed as update-pointer-counter[t] for t
  ∈ [D + l + 2].
```

received-v-priority $\in \{0,1\}^{D \times l}$
```
# Holds the v-priority from the received message after passing through the delay-new-v
```

```
    -priority. Bits are indexed as received-v-priority[k][i] for k ∈ [D] and for i ∈ [l].
```

renew-received-v-priority $\in \{0,1\}^{D \times l}$
```
# Acts as a flag to reset the value in received-v-priority to zero to clear the way
  for future received messages. Bits are indexed as renew-received-v-priority[k][i] for
  k ∈ [D] and for i ∈ [l].
```

u-priority-update $\in \{0,1\}^{D \times l}$
```
# Acts as a flag that allows updating the node's priority if there is a match in the
  received message. Bits are indexed as u-priority-update[k][i] for k ∈ [D] and for i
  ∈ [l].
```

received-v-priority4u $\in \{0,1\}^{D \times l}$
```
# Holds a copy of the v-priority in the received message to update u-priority in case
  there is a match. Bits are indexed as received-v-priority4u[k][i] for k ∈ [D] and for
  i ∈ [l].
```

u-priority-delay $\in \{0,1\}^{D \times l}$
```
# When the update flag for u-priority is activated because of a matching node in the
  received message, the priority of the matching node is placed in u-priority-delay.
  The value moves along the first dimension. When it reaches the D-th coordinate, it
  is copied into u-priority. The goal of the cascade chain mechanism is to avoid
  growing collision in the destination variable. Bits are indexed as u-priority-delay[k
  ][i] for k ∈ [D] and for i ∈ [l].
```

received-v-dist $\in \{0,1\}^{D \times l}$
```
# Holds the v-dist from the received message after passing through the delay-new-v-
  dist. Bits are indexed as received-v-dist[k][i] for k ∈ [D] and for i ∈ [l].
```

renew-received-v-dist $\in \{0,1\}^{D \times l}$
```
# Acts as a flag to reset the value in received-v-dist to zero to clear the way for
  future received messages. Bits are indexed as renew-received-v-dist[k][i] for k ∈ [D]
  and for i ∈ [l].
```

u-dist-update $\in \{0,1\}^{D \times l}$
```
# Acts as a flag that allows updating the node's distance if there is a match in the
  received message. Bits are indexed as u-dist-update[k][i] for k ∈ [D] and for i ∈ [l].
```

received-v-dist4u $\in \{0,1\}^{D \times l}$
```
# Holds a copy of the v-dist from the received message to update u-dist in case there
  is a match. Bits are indexed as received-v-dist4u[k][i] for k ∈ [D] and for i ∈ [l]
```

u-dist-delay $\in \{0,1\}^{D \times l}$
```
# When the update flag for u-dist is activated because of a matching node in the
  received message, the distance of the matching node is placed in u-dist-delay. The
  value moves along the first dimension. When it reaches the D-th coordinate, it is
  copied into u-dist. Bits are indexed as u-dist-delay[k][i] for k ∈ [D] and for i ∈ [l].
```

received-v-π $\in \{0,1\}^{D \times l}$
```
# Holds the v-π from the received message after passing through the delay-new-v-π.
  Bits are indexed as received-v-π[k][i] for k ∈ [D] and for i ∈ [l].
```

renew-received-v-π $\in \{0,1\}^{D \times l}$
```
# Acts as a flag to reset the value in received-v-π to zero to clear the way for
  future received messages. Bits are indexed as renew-received-v-π[k][i] for k ∈ [D] and
  for i ∈ [l].
```

u-π-update $\in \{0,1\}^{D \times l}$
```
# Acts as a flag that allows updating the node's parent if there is a match in the
  received message. Bits are indexed as u-π-update[k][i] for k ∈ [D] and for i ∈ [l].
```

received-v-π4u $\in \{0,1\}^{D \times l}$
```
# Holds a copy of the v-π from the received message to update u-π in case there exists
```

```
      a match. Bits are indexed as received-v-π4u[k][i] for k ∈ [D] and for i ∈ [l].
```

```
u-π-delay ∈ {0,1}^{D×l}
# When the update flag for u-π is activated because of a matching node in the received
  message, the parent of the matching node is placed in u-π-delay. The value moves
  along the first dimension. When it reaches the D-th coordinate, it is copied into u-
  π. Bits are indexed as u-π-delay[k][i] for k ∈ [D] and i ∈ [l].
```

```
received-v-gray ∈ {0,1}^D
# Holds the v-gray from the received message after passing through the delay-new-v-
  gray. Bits are indexed as received-v-gray[k] for k ∈ [D].
```

```
new-received-v-gray ∈ {0,1}^D
# Acts as a flag to reset the value in received-v-gray to zero to clear the way for
  future received messages. Bits are indexed as new-received-v-gray[k] for k ∈ [D].
```

```
u-gray-update ∈ {0,1}^D
# Acts as a flag that allows updating the node's gray if there is a match in the
  received message. Bits are indexed as u-gray-update[k] for k ∈ [D].
```

```
received-v-grays4u ∈ {0,1}^D
# Holds a copy of the v-gray from the received message to update u-gray in case there
  exists a match. Bits are indexed as received-v-grays4u[k] for k ∈ [D].
```

```
u-gray-delay ∈ {0,1}^D
# When the update flag for u-gray is activated because of a matching node in the
  received message, the gray of the matching node is placed in u-gray-delay. The value
  moves along the first dimension. When it reaches the D-th coordinate, it is copied
  into u-gray. Bits are indexed as u-gray-delay[k] for k ∈ [D].
```

```
received-pointer ∈ {0,1}^l
# Holds the pointer from the received message after passing through the delay-new-
  pointer. Bits are indexed as received-pointer[i] for i ∈ [l].
```

```
update-pointer ∈ {0,1}^l
# Acts as a flag to update the value of the q-pointer with the pointer in received-
  pointer. Bits are indexed as update-pointer[i] for i ∈ [l].
```

```
received-last-in-q ∈ {0,1}^l
# Holds the last-in-q from the received message after passing through the delay-new-
  last-in-q. Bits are indexed as received-last-in-q[i] for i ∈ [l].
```

```
update-last-in-q ∈ {0,1}^l
# Acts as a flag to update the value of u-last-in-q with the value of last-in-q in
  received-last-in-q. Bits are indexed as update-last-in-q[i] for i ∈ [l].
```

```
v-priority-update ∈ {0,1}^{D×D×l}
# Acts as a flag that allows updating the node's neighbors' priority if there are
  matches in the received message. Bits are indexed as v-priority-update[d][k][i] for d
  ∈ [D], for k ∈ [D], and for i ∈ [l].
```

```
received-v-priority4v ∈ {0,1}^{D×D×l}
# Holds a copy of the v-priority from the received message to update neighbors' v-
  priority in case there are matches. Bits are indexed as received-v-priority4v[d][k][i
  ] for d ∈ [D], for k ∈ [D], and for i ∈ [l].
```

```
v-dist-update ∈ {0,1}^{D×D×l}
# Acts as a flag that allows updating the node's neighbors' distance if there are
  matches in the received message. Bits are indexed as v-dist-update[d][k][i] for d
  ∈ [D], for k ∈ [D], and for i ∈ [l].
```

```
received-v-dist4v ∈ {0,1}^{D×D×l}
# Holds a copy of the v-dist from the received message to update neighbors' v-dist in
```

```
    case there are matches. Bits are indexed as received-v-dist4v[d][k][i] for d ∈ [D],
      for k ∈ [D], and for i ∈ [l].
```

```
  v-π-update ∈ {0,1}^{D×D×l}
  # Acts as a flag that allows updating the node's neighbors' parent if there are
    matches in the received message. Bits are indexed as v-π-update[d][k][i] for d ∈ [D],
      for k ∈ [D], and for i ∈ [l].
```

```
  received-v-π4v ∈ {0,1}^{D×D×l}
  # Holds a copy of the v-π from the received message to update neighbors' v-π in case
    there are matches. Bits are indexed as received-v-π4v[d][k][i] for d ∈ [D], for k ∈ [D]
    , and for i ∈ [l].
```

```
  v-gray-update ∈ {0,1}^{D×D}
  # Acts as a flag that allows updating the node's neighbors' gray if there are matches
    in the received message. Bits are indexed as v-gray-update[d][k] for d ∈ [D] and for k
      ∈ [D].
```

```
  received-v-gray4v ∈ {0,1}^{D×D}
  # Holds a copy of the v-gray from the received message to update neighbors' v-gray in
    case there are matches. Bits are indexed as received-v-gray4v[d][k] for d ∈ [D] and
      for k ∈ [D].
```

```
  v-priority-delay ∈ {0,1}^{D×D×l}
  # When the update flag for the priority of a neighbor is activated because of a
    matching node in the received message, the priority of the matching node is placed in
     v-priority-delay at the coordinates of that neighbor. The value moves along the
    second dimension. When it reaches the D-th coordinate, it is copied into v-priority
    at the coordinate of that neighbor. Bits are indexed as v-priority-delay[d][k][i] for
      d ∈ [D], k ∈ [D], and i ∈ [l].
```

```
  v-dist-delay ∈ {0,1}^{D×D×l}
  # When the update flag for the distance of a neighbor is activated because of a
    matching node in the received message, the distance of the matching node is placed in
     v-dist-delay at the coordinates of that neighbor. The value moves along the second
    dimension. When it reaches the D-th coordinate, it is copied into v-dist at the
    coordinate of that neighbor. Bits are indexed as v-dist-delay[d][k][i] for d ∈ [D],
      for k ∈ [D], and for i ∈ [l].
```

```
  v-π-delay ∈ {0,1}^{D×D×l}
  # When the update flag for the parent of a neighbor is activated because of a matching
      node in the received message, the parent of the matching node is placed in v-π-delay
     at the coordinates of that neighbor. The value moves along the second dimension.
    When it reaches the D-th coordinate, it is copied into v-π at the coordinate of that
     neighbor. Bits are indexed as v-π-delay[d][k][i] for d ∈ [D], for k ∈ [D], and for i
      ∈ [l].
```

```
  v-gray-delay ∈ {0,1}^{D×D}
  # When the update flag for the gray of a neighbor is activated because of a matching
      node in the received message, the gray of the matching node is placed in v-gray-delay
     at the coordinates of that neighbor. The value moves along the second dimension.
    When it reaches the D-th coordinate, it is copied into v-gray at the coordinate of
    that neighbor. Bits are indexed as v-gray-delay[d][k] for d ∈ [D] and for k ∈ [D].
```

Now we will present the instructions. We have divided the instructions into a number of sections. Each section includes the instructions that together form a meaningful algorithmic subroutine.

**Compare the pointer with the node's priority to trigger the for loop**

Input blocks used in instruction definitions:

```
Index definition:
Let i ∈ [l] and j ∈ [l + 1]
```

```
Block definition:
  (u-priority[i], q-pointer[i], compare-u-priority[i])
  (u-matches-pointer[i], u-matches-counter[i], reset-u-matches[i])
  (priority-comparison-counter[j])
```

Instructions:

**Box 1**

```
INPUT:
u-priority[i]=1 AND q-pointer[i]=1 AND compare-u-priority[i]=1

OUTPUT:
(i=1) u-matches-pointer[i]←1 AND u-matches-counter[i]←1 AND
u-priority[i]←1 AND pointer-augend[i]←1

OUTPUT:
(i>1) u-matches-pointer[i]←1 AND u-priority[i]←1 AND
pointer-augend[i]←1
```

**Box 2**

```
INPUT:
u-priority[i]=0 AND q-pointer[i]=0 AND compare-u-priority[i]=1

OUTPUT:
(i=1) u-matches-pointer[i]←1 AND u-matches-counter[i]←1

OUTPUT:
(i>1) u-matches-pointer[i]←1
```

**Box 3**

```
INPUT:
u-priority[i]=1 AND q-pointer[i]=0 AND compare-u-priority[i]=1

OUTPUT:
u-priority[i]←1 AND pointer-augend[i]←1
```

**Box 4**

```
INPUT:
u-priority[i]=1 AND q-pointer[i]=0 AND compare-u-priority[i]=0

OUTPUT:
u-priority[i]←1
```

**Box 5**

```
INPUT:
priority-comparison-counter[j]=1

OUTPUT:
(j < l) priority-comparison-counter[j+1]←1

OUTPUT:
(j = l) priority-comparison-counter[j+1]←1 AND update-last-in-q[ALL]←1

OUTPUT:
(j = l+1) reset-u-matches[ALL]←1
```

**Box 6**

```
INPUT:
u-matches-pointer[i]=1 AND u-matches-counter[i]=1 AND
reset-u-matches[i]=0

OUTPUT:
(i< l) u-matches-counter[i+1]←1

OUTPUT:
(i= l) u-matched←1 AND accept-last-in-q[ALL]←1 AND
update-v-dist-augend[ALL]←1
```

**Box 7**

```
INPUT:
u-matches-pointer[i]=1 AND u-matches-counter[i]=0 AND
reset-u-matches[i]=0

OUTPUT:
u-matches-pointer[i]←1
```

Each node u has a priority denoted by `u-priority`, which indicates when it should be expanded. The algorithm has a variable called pointer that indicates which priority number is to be expanded. Each node becomes aware of the current pointer from the most recent message it receives; here, `q-pointer` is the received pointer. The instructions in this section collectively check whether the node's priority matches the current pointer, but only when the `compare-u-priority` flag is set to 1.

There are also instructions that preserve the value of `u-priority` and store a copy in `pointer-augend`; this copy will later be incremented by 1 to become the new pointer of the algorithm. When all bits of `u-priority` and `q-pointer` match, `u-matched` is set to 1.

Additionally, the algorithm has a variable that indicates the latest number added to the queue. The node also becomes aware of this through the latest message it receives; this variable is named `last-in-q`. Thus, when a match is verified, the flag to accept it, `accept-last-in-q`, is also set.

In this implementation, each node u keeps information about its neighbors. In the case of a match, the flag to update the distance of neighbors from the source node, `update-v-dist-augend`, is activated. A flag to start incrementing the current pointer, `update-pointer-augend`, is also set to 1.

Finally, after the required iterations for checking the match are completed, all variables related to the checking process are reset to clear the way for the next messages, since a match does not necessarily occur for the first received `q-pointer`.

**Set up the parameters for the loop where neighbor nodes are discovered**

Input blocks used in instruction definitions:

```
Index definition:
Let d ∈ [D] be the index over the neighbors.

Block definition:
   (u-matched)
   (v-turn[d], v-white[d], v-white-overwrite[d])
```

Instructions:

**Box 8**

```
INPUT:
u-matched=1

OUTPUT:
v-turn[1]←1 AND v-dist-addend[1]←1 AND
v-dist-sum-counter[1]←1
```

**Box 9**

```
INPUT:
v-turn[d]=1 AND v-white[d]=1 AND v-white-overwrite[d]=0

OUTPUT:
v-gray[d]←1 AND v-last-in-q-sum-counter[d][1]←1 AND v-last-in-q-addend[d][1]←1 AND v-
    set-π[d][ALL]←1 AND v-set-priority[d][ALL]←1 AND v-set-dist-counter[d][1]←1
```

**Box 10**

```
INPUT:
v-turn[d]=0 AND v-white[d]=1 AND v-white-overwrite[d]=0

OUTPUT:
v-white[d]←1
```

**Box 11**

```
INPUT:
v-turn[d]=1 AND v-white[d]=0 AND v-white-overwrite[d]=0

OUTPUT:
v-last-in-q-sum-counter[d][1]←1
```

The instruction in Box 8 sets the bit corresponding to the first neighbor in `v-turn` to 1. It also sets the first bit of `v-dist-addend` and `v-dist-sum-counter`, which will increment the current node's distance from the source. The new distance will then be given to the discovered neighbors. Instructions in Boxes 9, 10, and 11 collectively check a neighbor node, when its turn comes, to see if it has already been discovered. If not, they set `v-gray` for that node to 1, indicating it is discovered. Additionally, some flags are activated to set the discovered node's parent, its priority in the queue, and its distance from the source. Variables such as `v-last-in-q-sum-counter` and `v-last-in-q-addend` are also set so that the last-in-queue number is incremented for the next neighbor to be discovered. There are also instructions to preserve the values when the neighbor's turn has not yet arrived.

**Assign priority to white nodes and increment it**

Input blocks used in instruction definitions:

```
Index definition:
Let i ∈ [l],  j ∈ [2l].
Let d ∈ [D]

Block definition:
  (accept-last-in-q[i], u-last-in-q[i])
  (v-set-priority[d][i], v-last-in-q[d][i])
  (v-priority[d][i], upload-v-priority[d][i])
  (v-last-in-q-augend[d][i], v-last-in-q-addend[d][i],
   to-copy-augend[d][i])
  (v-last-in-q-carry[d][i])
  (v-last-in-q-sum-counter[d][j])
```

Instructions:

**Box 11**

```
INPUT:
accept-last-in-q[i]=1 AND u-last-in-q[i]=1

OUTPUT:
v-last-in-q[1][i]←1 AND v-last-in-q-augend[1][i]←1
```

**Box 12**

```
INPUT:
v-set-priority[d][i]=1 AND v-last-in-q[d][i]=1

OUTPUT:
v-priority[d][i]←1
```

**Box 13**

```
INPUT:
v-set-priority[d][i]=0 AND v-last-in-q[d][i]=1

OUTPUT:
v-last-in-q[d][i]←1
```

**Box 14**

```
INPUT:
v-priority[d][i]=1 AND upload-v-priority[d][i]=0

OUTPUT:
v-priority[d][i]←1
```

**Box 15**

```
INPUT:
v-priority[d][i]=1 AND upload-v-priority[d][i]=1

OUTPUT:
v-priority[d][i]←1 AND v-priority-placeholder[ALL][d][i]←1
```

**Box 16**

```
INPUT:
v-last-in-q-augend[d][i]=1 AND v-last-in-q-addend[d][i]=0 AND to-copy-last-in-q-augend[d
    ][i]=0

OUTPUT:
v-last-in-q-augend[d][i]←1
```

**Box 17**

```
INPUT:
v-last-in-q-augend[d][i]=1 AND v-last-in-q-addend[d][i]=1 AND to-copy-last-in-q-augend[d
    ][i]=0

OUTPUT:
v-last-in-q-carry[d][i]←1
```

**Box 18**

```
INPUT:
v-last-in-q-augend[d][i]=0 AND v-last-in-q-addend[d][i]=1 AND to-copy-last-in-q-augend[d
    ][i]=0

OUTPUT:
v-last-in-q-augend←1
```

**Box 19**

```
INPUT:
v-last-in-q-carry[d][i]=1
```

```
OUTPUT:
(i< l) v-last-in-q-addend[d][i+1]←1
```

```
OUTPUT:
(i= l) v-last-in-q-carry[d][i]←1
```

**Box 20**

```
INPUT:
v-last-in-q-sum-counter[d][j]=1
```

```
OUTPUT:
(j< 2l) v-last-in-q-sum-counter[d][j+1]←1
```

```
OUTPUT:
(j= 2l AND d < D) to-copy-last-in-q-augend[d][ALL]←1 AND
v-turn[d+1]←1
```

```
OUTPUT:
(j= 2l AND d = D) to-copy-last-in-q-augend[d][ALL]←1 AND
pointer-addend[1]←1 AND pointer-sum-counter[1]←1
```

**Box 21**

```
INPUT:
v-last-in-q-augend[d][i]=1 AND v-last-in-q-addend[d][i]=0 AND to-copy-last-in-q-augend[d
    ][i]=1
```

```
OUTPUT:
(d < D) v-last-in-q[d+1][i]←1 AND v-last-in-q-augend[d+1][i]←1
```

```
OUTPUT:
(d = D) after-loop-last-in-q[i]←1
```

The instructions in Box 11 copy the `u-last-in-q` received by the node into `v-last-in-q` for the first neighbor, if the accept flag allows it. Boxes 12 and 13 assign the neighbors' priority number (`v-priority`) if the set flag allows it; otherwise, they preserve `v-last-in-q` until the set flag allows assignment. Instructions in Boxes 14 and 15 copy the neighbors' priorities into `v-priority-placeholder` if the upload flag allows it; otherwise, they preserve the priorities until the upload flag becomes active. The content of `v-priority-placeholder` will later be included in the outgoing message. The instructions in Boxes 16 to 21 collectively increment the value of last-in-q so that in the next neighbor discovery, the updated priority can be assigned. When the priority of the last discovered neighbor is assigned (i.e., after all neighbors are processed in the discovery loop), last-in-q is incremented once more. This value is stored in `after-loop-last-in-q` and included in the outgoing message so that the receiving nodes know the next priority number they can assign.

**Increment distance and assign white nodes the new distance**

Input blocks used in instruction definitions:

```
Index definition:
Let i ∈ [l], j ∈ [2l].
Let d ∈ [D]

Block definition:
  (u-dist[i], update-v-dist-augend[i])
  (v-dist-augend[i], v-dist-addend[i], to-copy-dist-augend[i])
  (v-dist-carry[i])
  (v-dist-sum-counter[j])
  (v-set-dist-counter[d][j])
  (v-set-dist[d][i], v-dist-incremented[d][i])
  (v-dist[d][i], upload-v-dist[d][i])
```

Instructions:

**Box 22**

```
INPUT:
u-dist[i]=1 AND update-v-dist-augend[i]=0

OUTPUT:
u-dist[i]←1
```

**Box 23**

```
INPUT:
u-dist[i]=1 AND update-v-dist-augend[i]=1

OUTPUT:
v-dist-augend[i]←1 AND u-dist[i]←1
```

**Box 24**

```
INPUT:
v-dist-augend[i]=1 AND v-dist-addend[i]=0 AND to-copy-dist-augend[i]=0

OUTPUT:
v-dist-augend[i]←1
```

**Box 25**

```
INPUT:
v-dist-augend[i]=0 AND v-dist-addend[i]=1 AND to-copy-dist-augend[i]=0

OUTPUT:
v-dist-augend[i]←1
```

**Box 26**

```
INPUT:
v-dist-augend[i]=1 AND v-dist-addend[i]=1 AND to-copy-dist-augend[i]=0

OUTPUT:
v-dist-carry[i]←1
```

**Box 27**

```
INPUT:
v-dist-carry[i]=1

OUTPUT:
(i< l) v-dist-addend[i+1]←1

OUTPUT:
(i= l) v-dist-carry[i]←1
```

**Box 28**

```
INPUT:
v-dist-sum-counter[j]=1

OUTPUT:
(j< 2l) v-dist-sum-counter[j+1]←1

OUTPUT:
(j= 2l) to-copy-dist-augend[ALL]←1
```

**Box 29**

```
INPUT:
v-dist-augend[i]=1 AND v-dist-addend[i]=0 AND to-copy-dist-augend[i]=1

OUTPUT:
v-dist-incremented[ALL][i]←1
```

**Box 30**

```
INPUT:
v-set-dist-counter[d][j]=1

OUTPUT:
(j< 2l) v-set-dist-counter[d][j+1]←1

OUTPUT:
(j= 2l) v-set-dist[d][ALL]←1
```

**Box 31**

```
INPUT:
v-set-dist[d][i]=1 AND v-dist-incremented[d][i]=1

OUTPUT:
v-dist[d][i]←1
```

**Box 32**

```
INPUT:
v-set-dist[d][i]=0 AND v-dist-incremented[d][i]=1

OUTPUT:
v-dist-incremented[d][i]←1
```

**Box 33**

```
INPUT:
v-dist[d][i]=1 AND upload-v-dist[d][i]=1

OUTPUT:
v-dist[d][i]←1 AND v-dist-placeholder[ALL][d][i]←1
```

**Box 34**

```
INPUT:
v-dist[d][i]=1 AND upload-v-dist[d][i]=0

OUTPUT:
v-dist[d][i]←1
```

Instructions in Boxes 22 and 23 preserve the current node's distant `u-dist` and, when the update flag allows, also copy the distance into `v-dist-augend` so that it can be incremented. Instructions in Boxes 24 to 28 perform the binary addition process to calculate the new distance. Then, instructions in Box 29 create $D$ copies of the computed distance in `v-dist-incremented`. Instructions in Box 30 ensure that the flag to set the new distance for discovered neighbors is not activated until the addition is completed. Instructions in Boxes 31 and 32 copy the calculated distance in `v-dist-incremented` into `v-dist` if the set flag allows; otherwise, they preserve the values in `v-dist-incremented` until the set flag is activated. Instructions in Boxes 33 and 34 preserve the `v-dist` values and, when the upload flag is activated, copy distances to `v-dist-placeholder`. This content will be included in the outgoing message.

**Set the parent of the white nodes equal to the current node**

Input blocks used in instruction definitions:

```
Index definition:
Let i ∈ [l]
Let d ∈ [D]

Block definition:
   (u-id[i])
   (set-v-π[d][i], u-id-asparent[d][i])
   (v-π[d][i], upload-v-π[d][i])
```

Instructions:

**Box 35**

```
INPUT:
u-id[i]=1

OUTPUT:
u-id-asparent[ALL][i]←1 AND u-id[i]←1 AND u-id2check-with-received-v-ids[ALL][i]←1
```

**Box 36**

```
INPUT:
v-set-π[d][i]=1 AND u-id-asparent[d][i]=1

OUTPUT:
v-π[d][i]←1
```

**Box 37**

```
INPUT:
v-π[d][i]=1 AND upload-v-π[d][i]=0

OUTPUT:
v-π[d][i]←1
```

**Box 38**

```
INPUT:
v-π[d][i]=1 AND upload-v-π[d][i]=1

OUTPUT:
v-π[d][i]←1 AND v-π-placeholder[ALL][d][i]←1
```

Instructions in Box 35 preserve the current node's ID as well as create $D$ copies of it in `u-id-asparent` and `u-id2check-with-received-v-ids`. The copies in `u-id-asparent` are used to set the parent ID of the discovered neighbors. The messages a node receives include information about the neighbors of the node that sent the message, which might include the receiver node (u) itself. To determine which node each piece of information in the incoming message belongs to, an ID is sent alongside each node's information. The copies in `u-id2check-with-received-v-ids` are used to compare node u's ID with the IDs in an incoming message. If there is a match, the instructions that we will see in later sections are activated and update node u based on the matching information. Instructions in Box 36 set the parent ID of the discovered nodes when the set flag allows it. Instructions in Boxes 37 and 38 preserve parent IDs in v-π and also copy them into v-π-placeholder when the upload flag is activated. The content of v-π-placeholder will later be included in the outgoing message.

### Preserve this node's colors and parent

Input blocks used in instruction definitions:

```
Index definition:
Let i ∈ [l] be the index over the bits.
```

```
Block definition:
   (u-white, u-white-overwrite)
   (u-gray)
   (u-π[i])
```

Instructions:

**Box 39**

```
INPUT:
u-white=1 AND u-white-overwrite=0

OUTPUT:
u-white←1
```

**Box 40**

```
INPUT:
u-gray=1

OUTPUT:
u-gray←1
```

**Box 41**

```
INPUT:
u-π[i]=1

OUTPUT:
u-π[i]←1
```

The instruction in Box 39 preserves the undiscovered status of the node if it has not yet been discovered, unless the overwrite flag is activated. The instruction in Box 40 preserves the discovered status of the node if it is already discovered. The instructions in Box 41 preserve the parent ID of the node.

**Preserve and upload gray color of neighbor nodes**

Input blocks used in instruction definitions:

```
Index definition:
Let d ∈ [D] be the index over the neighbors.

Block definition:
   (v-gray[d], upload-v-gray[d])
```

Instructions:

**Box 42**

```
INPUT:
v-gray[d]=1 AND upload-v-gray[d]=0

OUTPUT:
v-gray[d]←1
```

**Box 43**

```
INPUT:
v-gray[d]=1 AND upload-v-gray[d]=1

OUTPUT:
v-gray[d]←1 AND v-gray-placeholder[ALL][d]←1
```

The instructions in Boxes 42 and 43 preserve v-gray values, which indicate whether the neighboring nodes have already been discovered. Moreover, when the upload flag is activated, $D^2 + 1$ copies of v-gray are transferred to v-gray-placeholder. Later, one of these copies will be included in the outgoing message, i.e., the one that is in the slot corresponding to the local ID of the node.

### Upload message flag

Input blocks used in instruction definitions:

```
Block definition:
   (upload-message-flag)
```

Instructions:

**Box 44**

```
INPUT:
upload-message-flag=1

OUTPUT:
message-flag-placeholder[ALL]←1
```

The instruction in Box 44 transfers $D^2+1$ copies of a flag named message-flag into message-flag-placeholder. Later, one of these copies will be included in the outgoing message to notify the receiver node of an incoming message.

### Increment the pointer and set upload flags

Input blocks used in instruction definitions:

```
Index definition:
Let i ∈ [l] be the index over the bits.

Block definition:
   (pointer-augend[i], pointer-addend[i], upload-pointer-augend[i])
   (pointer-carry[i])
   (pointer-sum-counter[i])
   (after-loop-lastinq[i], upload-after-loop-lastinq[i])
```

Instructions:

**Box 45**

```
INPUT:
pointer-augend[i]=1 AND pointer-addend[i]=0 AND
upload-pointer-augend[i]=0

OUTPUT:
pointer-augend[i]←1
```

**Box 46**

```
INPUT:
pointer-augend[i]=0 AND pointer-addend[i]=1 AND
upload-pointer-augend[i]=0

OUTPUT:
pointer-augend[i]←1
```

**Box 47**

```
INPUT:
pointer-augend[i]=1 AND pointer-addend[i]=1 AND
upload-pointer-augend[i]=0
```

```
OUTPUT:
pointer-carry[i]←1
```

**Box 48**

```
INPUT:
pointer-carry[i]=1

OUTPUT:
(i< l) pointer-addend[i+1]←1

OUTPUT:
(i= l) pointer-carry[i]←1
```

**Box 49**

```
INPUT:
pointer-sum-counter[j]=1

OUTPUT:
(j< 2l) pointer-sum-counter[j+1]←1

OUTPUT:
(j= 2l) upload-pointer-augend[ALL]←1 AND upload-v-gray[ALL]←1 AND
upload-v-π[ALL][ALL]←1 AND upload-v-dist[ALL][ALL]←1 AND
upload-v-id[ALL][ALL]←1 AND upload-v-priority[ALL][ALL]←1 AND
upload-after-loop-last-in-q[ALL]←1 AND
overwrite-last-pointer[ALL][ALL]←1 AND upload-message-flag←1
```

**Box 50**

```
INPUT:
pointer-augend[i]=1 AND pointer-addend[i]=0 AND
upload-pointer-augend[i]=1

OUTPUT:
pointer-placeholder[ALL][i]←1 AND persistent-last-pointer[ALL][i]←1
```

**Box 51**

```
INPUT:
after-loop-last-in-q[i]=1 AND upload-after-loop-last-in-q[i]=1

OUTPUT:
last-in-q-placeholder[ALL][i]←1
```

**Box 52**

```
INPUT:
after-loop-last-in-q[i]=1 AND upload-after-loop-last-in-q[i]=0

OUTPUT:
after-loop-last-in-q[i]←1
```

The instructions in Boxes 45 to 48 perform binary addition to increment the pointer. This new pointer is included in the outgoing message. The instructions in Box 49, once the addition is completed, activate the upload flags, which start the process in which the node includes its updated information about itself and its neighbors in the outgoing message. Each upload flag, when activated, creates copies of its corresponding information in a buffer that has "placeholder" in its name. The contents of these buffers are later included in the outgoing message. The instructions in Box 50 make $D^2 + 1$ copies of the new pointer in `pointer-placeholder`, whose content is included in the outgoing message. Because of the

distributed nature of the implementation, older messages might still circulate in the graph until they reach every node at least once. Thus, each node also keeps additional copies of the latest pointer it is aware of. These copies are compared against the pointer values in incoming messages in any of the $D^2+1$ slots to reject older messages, i.e., those with a smaller pointer. The instructions in Box 50 create $D^2+1$ copies of the new pointer in `persistent-last-pointer` for this purpose. The instructions in Boxes 51 and 52 make copies of `after-loop-last-in-q` in `loop-last-in-q-placeholder`, whose content is later included in the outgoing message. Otherwise, the value of `after-loop-last-in-q` is preserved until the upload flag is activated.

### Guide the placeholder content to the appropriate sending slot

Input blocks used in instruction definitions:

```
Index definitions:
Let s ∈ [D² + 1] be the index over the slots used in message-passing.
Let d ∈ [D]
Let i ∈ [l]

Block definitions:
  (determine-slot[s])
  (determine-pointer-slot[s][i], pointer-placeholder[s][i])
  (determine-afterloop-lastinq-slot[s][i], afterloop-lastinq-placeholder[s][i])
  (determine-v-white-slot[s][d], white-placeholder[s][d])
  (determine-v-gray-slot[s][d], v-gray-placeholder[s][d])
  (determine-v-dist-slot[s][d][i], v-dist-placeholder[s][d][i])
  (determine-v-π-slot[s][d][i], v-π-placeholder[s][d][i])
  (determine-v-id-slot[s][d][i], v-id-placeholder[s][d][i])
  (determine-v-priority-slot[s][d][i], v-priority-placeholder[s][d][i])
  (determine-message-flag-slot[s], message-flag-placeholder[s])
```

Instructions:

**Box 52**

```
INPUT:
determine-slot[s]=1

OUTPUT:
determine-slot[s]←1 AND determine-v-gray-slot[s][ALL]←1 AND determine-v-dist-slot[s][
    ALL][ALL]←1 AND determine-v-π-slot[s][ALL][ALL]←1 AND determine-v-id-slot[s][ALL][
    ALL]←1 AND determine-v-priority-slot[s][ALL][ALL]←1 AND determine-pointer-slot[s][
    ALL]←1 AND determine-message-flag-slot[s]←1
```

**Box 53**

```
INPUT:
determine-pointer-slot[s][i]=1 AND pointer-placeholder[s][i]=1

OUTPUT:
pointer-slot[s][i]←1 AND determine-pointer-slot[s][i]←1
```

**Box 54**

```
INPUT:
determine-pointer-slot[s][i]=1 AND pointer-placeholder[s][i]=0

OUTPUT:
determine-pointer-slot[s][i]←1
```

**Box 55**

```
INPUT:
determine-last-in-q-slot[s][i]=1 AND last-in-q-placeholder[s][i]=1
```

```
OUTPUT:
last-in-q-slot[s][i]←1 AND determine-last-in-q-slot[s][i]←1 1
```

**Box 56**

```
INPUT:
determine-last-in-q-slot[s][i]=1 AND last-in-q-placeholder[s][i]=0

OUTPUT:
determine-last-in-q-slot[s][i]←1
```

**Box 57**

```
INPUT:
determine-v-gray-slot[s][d]=1 AND v-gray-placeholder[s][d]=1

OUTPUT:
gray-slot[s][d]←1 AND determine-v-gray-slot[s][d]←1
```

**Box 58**

```
INPUT:
determine-v-gray-slot[s][d]=1 AND v-gray-placeholder[s][d]=0

OUTPUT:
determine-v-gray-slot[s][d]←1
```

**Box 59**

```
INPUT:
determine-v-dist-slot[s][d][i]=1 AND v-dist-placeholder[s][d][i]=1

OUTPUT:
v-dist-slot[s][d][i]←1 AND determine-v-dist-slot[s][d][i]←1
```

**Box 60**

```
INPUT:
determine-v-dist-slot[s][d][i]=1 AND v-dist-placeholder[s][d][i]=0

OUTPUT:
determine-v-dist-slot[s][d][i]←1
```

**Box 61**

```
INPUT:
determine-v-π-slot[s][d][i]=1 AND v-π-placeholder[s][d][i]=1

OUTPUT:
v-π-slot[s][d][i]←1 AND determine-v-π-slot[s][d][i]←1
```

**Box 62**

```
INPUT:
determine-v-π-slot[s][d][i]=1 AND v-π-placeholder[s][d][i]=0

OUTPUT:
determine-v-π-slot[s][d][i]←1
```

**Box 63**

```
INPUT:
determine-v-id-slot[s][d][i]=1 AND v-id-placeholder[s][d][i]=1

OUTPUT:
v-id-slot[s][d][i]←1 AND determine-v-id-slot[s][d][i]←1
```

**Box 64**

```
INPUT:
determine-v-id-slot[s][d][i]=1 AND v-id-placeholder[s][d][i]=0

OUTPUT:
determine-v-id-slot[s][d][i]←1
```

**Box 65**

```
INPUT:
determine-v-priority-slot[s][d][i]=1 AND v-priority-placeholder[s][d][i]=1

OUTPUT:
v-priority-slot[s][d][i]←1 AND determine-v-priority-slot[s][d][i]←1
```

**Box 66**

```
INPUT:
determine-v-priority-slot[s][d][i]=1 AND v-priority-placeholder[s][d][i]=0

OUTPUT:
determine-v-priority-slot[s][d][i]←1
```

**Box 67**

```
INPUT:
determine-message-flag-slot[s]=1 AND message-flag-placeholder[s]=1

OUTPUT:
message-flag-slot[s]←1 AND determine-message-flag-slot[s]←1
```

**Box 68**

```
INPUT:
determine-message-flag-slot[s]=1 AND message-flag-placeholder[s]=0

OUTPUT:
determine-message-flag-slot[s]←1
```

The instructions in Box 52 preserve `determine-slot`, which in general shows which slot the node should use for message passing based on the node's local ID. Using `determine-slot`, the instructions also set up variables named with the following pattern `determine-name-slot`. These variables show which slot should be used to share the corresponding information in the message-passing. For each node, they all point to the same slot. For example, `determine-pointer-slot` shows which slot should be used to share the latest pointer of the node in the message-passing. The instructions in Boxes 53 and 54 ensure that when there are $D^2 + 1$ copies of the latest pointer in `pointer-placeholder`, only one is transferred into the proper slot in `pointer-slot`. The pointer-slot is the variable in $x_{u_M}$ used in message-passing (aggregation step) to share the latest pointer that the node is aware of. Thus, the node can only write into one of the slots in `pointer-slot`, in particular the one that corresponds to the node's local ID and is determined by `determine-pointer-slot`. Also, `determine-pointer-slot` is preserved. The remaining instructions in Boxes 55 to 68 do the same as the instructions in Boxes 53 and 54 for their corresponding variables.

**Copy incoming messages to temporary slots while checking whether the pointer in them is new**

Input blocks used in instruction definitions:

```
Index definitions:
Let s ∈ [D² + 1]
Let d ∈ [D]
Let i ∈ [l]

Block definitions:
   (pointer-slot[s][i])
   (last-in-q-slot[s][i])
   (v-gray-slot[s][d])
   (v-dist-slot[s][d][i])
   (v-id-slot[s][d][i])
   (v-π-slot[s][d][i])
   (v-priority-slot[s][d][i])
   (message-flag-slot[s])
```

Instructions:

### Box 69

```
INPUT:
pointer-slot[s][i]=1

OUTPUT:
temp-pointer-slot[s][i]←1
```

### Box 70

```
INPUT:
last-in-q-slot[s][i]=1

OUTPUT:
temp-last-in-q-slot[s][i]←1
```

### Box 71

```
INPUT:
v-gray-slot[s][d]=1

OUTPUT:
temp-v-gray-slot[s][d]←1
```

### Box 72

```
INPUT:
v-dist-slot[s][d][i]=1

OUTPUT:
temp-v-dist-slot[s][d][i]←1
```

### Box 73

```
INPUT:
v-id-slot[s][d][i]=1

OUTPUT:
temp-v-id-slot[s][d][i]←1
```

### Box 74

```
INPUT:
v-π-slot[s][d][i]=1
```

```
OUTPUT:
temp-v-π-slot[s][d][i]←1
```

**Box 75**

```
INPUT:
v-priority-slot[s][d][i]=1

OUTPUT:
temp-v-priority-slot[s][d][i]←1
```

**Box 76**

```
INPUT:
message-flag-slot[s]=1
OUTPUT:
compare-last-pointer[s][l]←1
```

The variable `pointer-slot` is in $x_{u_M}$. After the aggregation step, it might contain pointer values that other nodes have shared. The instructions in Box 69 make sure that, as soon as a new pointer value is shared through any of the slots, it is copied to a temporary buffer outside $x_{u_M}$, `temp-pointer-slot`, for further processing. The instructions in Boxes 70 to 75 do the same for their corresponding variables. Note that when a node wants to send a message, it shares all the information (i.e., pointer, last-in-q, and so on) simultaneously. The instructions in Box 76 activate a flag to start a process in which the incoming pointer in a slot is compared to the last pointer stored locally at the node to determine whether it is a new message or not.

**Preserve data in temporary buffers until the new message check is complete.**

Input blocks used in instruction definitions:

```
Index definitions:
Let i ∈ [l]
Let d ∈ [D]
Let s ∈ [D² + 1]

Block definitions:
  (persistent-last-pointer[s][i], overwrite-last-pointer[s][i])
  (temp-last-in-q-slot[s][i], to-accept-temp-last-in-q[s][i], ...
  overwrite-temp-last-in-q[s][i])
  (temp-v-gray-slot[s][d], to-accept-temp-v-gray[s][d], ...
  overwrite-temp-v-gray[s][d])
  (temp-v-dist-slot[s][d][i], to-accept-temp-v-dist[s][d][i], ...
  overwrite-temp-v-dist[s][d][i])
  (temp-v-id-slot[s][d][i], to-accept-temp-v-id[s][d][i], overwrite-temp-v-id[s][d][i])
  (temp-v-π-slot[s][d][i], to-accept-temp-v-π[s][d][i], overwrite-temp-v-π[s][d][i])
  (temp-v-priority-slot[s][d][i], to-accept-temp-v-priority[s][d][i], ...
  overwrite-temp-v-priority[s][d][i])
  (return-message-flag[s])
```

Instructions:

**Box 77**

```
INPUT:
persistent-last-pointer[s][i]=1 AND overwrite-last-pointer[s][i]=0

OUTPUT:
last-pointer[s][i]←1 AND persistent-last-pointer[s][i]=1←1
```

**Box 78**

```
INPUT:
```

```
persistent-last-pointer[s][i]=1 AND overwrite-last-pointer[s][i]=1
```

**OUTPUT**:
```
last-pointer[s][i]←1
```

---

**Box 79**

**INPUT**:
```
temp-last-in-q-slot[s][i]=1 AND to-accept-temp-last-in-q[s][i]=0 AND overwrite-temp-last
    -in-q[s][i]=0
```

**OUTPUT**:
```
temp-last-in-q-slot[s][i]←1
```

---

**Box 80**

**INPUT**:
```
temp-last-in-q-slot[s][i]=1 AND to-accept-temp-last-in-q[s][i]=1 AND overwrite-temp-last
    -in-q[s][i]=0
```

**OUTPUT**:
```
delay-new-last-in-q[s][i]←1
```

---

**Box 81**

**INPUT**:
```
temp-v-gray-slot[s][d]=1 AND to-accept-temp-v-gray[s][d]=0 AND overwrite-temp-v-gray[s][
    d]=0
```

**OUTPUT**:
```
temp-v-gray-slot[s][d]←1
```

---

**Box 82**

**INPUT**:
```
temp-v-gray-slot[s][d]=1 AND to-accept-temp-v-gray[s][d]=1 AND overwrite-temp-v-gray[s][
    d]=0
```

**OUTPUT**:
```
delay-new-v-gray[s][d]←1
```

---

**Box 83**

**INPUT**:
```
temp-v-dist-slot[s][d][i]=1 AND to-accept-temp-v-dist[s][d][i]=0 AND overwrite-temp-v-
    dist[s][d][i]=0
```

**OUTPUT**:
```
temp-v-dist-slot[s][d][i]←1
```

---

**Box 84**

**INPUT**:
```
temp-v-dist-slot[s][d][i]=1 AND to-accept-temp-v-dist[s][d][i]=1 AND overwrite-temp-v-
    dist[s][d][i]=0
```

**OUTPUT**:
```
delay-new-v-dist[s][d][i]←1
```

---

**Box 85**

**INPUT**:

```
temp-v-id-slot[s][d][i]=1 AND to-accept-temp-v-id[s][d][i]=0 AND overwrite-temp-v-id[s][
    d][i]=0

OUTPUT:
temp-v-id-slot[s][d][i]←1
```

**Box 86**

```
INPUT:
temp-v-id-slot[s][d][i]=1 AND to-accept-temp-v-id[s][d][i]=1 AND overwrite-temp-v-id[s][
    d][i]=0

OUTPUT:
delay-new-v-id[s][d][i]←1
```

**Box 87**

```
INPUT:
temp-v-π-slot[s][d][i]=1 AND to-accept-temp-v-π[s][d][i]=0 AND overwrite-temp-v-π[s][d][
    i]=0

OUTPUT:
temp-v-π-slot[s][d][i]←1
```

**Box 88**

```
INPUT:
temp-v-π-slot[s][d][i]=1 AND to-accept-temp-v-π[s][d][i]=1 AND overwrite-temp-v-π[s][d][
    i]=0

OUTPUT:
delay-new-v-π[s][d][i]←1
```

**Box 89**

```
INPUT:
temp-v-priority-slot[s][d][i]=1 AND to-accept-temp-v-priority[s][d][i]=0 AND overwrite-
    temp-v-priority[s][d][i]=0

OUTPUT:
temp-v-priority-slot[s][d][i]←1
```

**Box 90**

```
INPUT:
temp-v-priority-slot[s][d][i]=1 AND to-accept-temp-v-priority[s][d][i]=1 AND overwrite-
    temp-v-priority[s][d][i]=0

OUTPUT:
delay-new-v-priority[s][d][i]←1
```

**Box 91**

```
INPUT:
return-message-flag[s]=1

OUTPUT:
delay-new-message-flag[s]←1
```

The instructions in Boxes 77 and 78 preserve the last pointer in `persistent-last-pointer` until the overwrite flag is activated. This is done to replace the value with the new pointer. Also, the instruction stores a copy of the last pointer in

last-pointer. This copy is used to compare the latest pointer that the node is aware of with the pointers in incoming messages, in order to detect new messages. Instructions in Boxes 79 and 80 preserve the received last-in-q values stored in temp-last-in-q-slot until the accept flag is activated, i.e., when the message is verified to be new. At this point, a copy of the last-in-q value is created in delay-new-last-in-q, in the same slot as in temp-last-in-q-slot, for further processing. Also, in the case that the overwrite flag is activated, the value in temp-last-in-q-slot is removed. The instructions in Boxes 81 to 90 do the same as the instructions in Boxes 79 and 80 for their corresponding variables. The flag return-message-flag is activated for a slot when a new message is received. The instructions in Box 91 place the flag in delay-new-message-flag to carry it forward in time, as it will be used later to initiate some processes.

**Check if the incoming pointer is new**

Input blocks used in instruction definitions:

```
Index definitions:
Let  i ∈ [l]
Let  s ∈ [D² + 1]

Block definitions:
  (compare-last-pointer[s][i], temp-pointer-slot[s][i], last-pointer[s][i], to-accept-
    temp-pointer[s][i], overwrite-temp-pointer[s][i])
  (new-message[s][i])
  (old-message[s][i])
```

Instructions:

**Box 92**

```
INPUT:
compare-last-pointer[s][i]=1 AND temp-pointer-slot[s][i]=1 AND last-pointer[s][i]=1 AND
    to-accept-temp-pointer[s][i]=0 AND overwrite-temp-pointer[s][i]=0

OUTPUT:
(i = 1) overwrite-temp-pointer[s][ALL]←1 AND overwrite-temp-last-in-q[s][ALL]←1 AND
    overwrite-temp-v-priority[s][ALL][ALL]←1 AND overwrite-temp-v-π[s][ALL][ALL]←1 AND
    overwrite-temp-v-id[s][ALL][ALL]←1 AND overwrite-temp-v-dist[s][ALL][ALL]←1 AND
    overwrite-temp-v-gray[s][ALL]←1

OUTPUT:
(i>1) compare-last-pointer[s][i-1]←1 AND temp-pointer-slot[s][i]←1
```

**Box 93**

```
INPUT:
compare-last-pointer[s][i]=1 AND temp-pointer-slot[s][i]=0 AND last-pointer[s][i]=0 AND
    to-accept-temp-pointer[s][i]=0 AND overwrite-temp-pointer[s][i]=0

OUTPUT:
(i=1) overwrite-temp-pointer[s][ALL]←1 AND overwrite-temp-last-in-q[s][ALL]←1 AND
    overwrite-temp-v-priority[s][ALL][ALL]←1 AND overwrite-temp-v-π[s][ALL][ALL]←1 AND
    overwrite-temp-v-id[s][ALL][ALL]←1 AND overwrite-temp-v-dist[s][ALL][ALL]←1 AND
    overwrite-temp-v-gray[s][ALL]←1

OUTPUT:
(i>1) compare-last-pointer[s][i-1]←1
```

**Box 94**

```
INPUT:
compare-last-pointer[s][i]=1 AND temp-pointer-slot[s][i]=1 AND last-pointer[s][i]=0 AND
    to-accept-temp-pointer[s][i]=0 AND overwrite-temp-pointer[s][i]=0

OUTPUT:
new-message[s][i]←1 AND temp-pointer-slot[s][i]←1
```

**Box 95**

```
INPUT:
compare-last-pointer[s][i]=1 AND temp-pointer-slot[s][i]=0 AND last-pointer[s][i] = 1
    AND to-accept-temp-pointer[s][i]=0 AND overwrite-temp-pointer[s][i]=0:

OUTPUT:
old-message[s][i] ← 1
```

**Box 96**

```
INPUT:
old-message[s][i]=1:

OUTPUT:
(i > 1) old-message[s][i-1] ← 1

OUTPUT:
(i = 1) overwrite-temp-pointer[s][ALL] ← 1 AND overwrite-temp-last-in-q[s][ALL] ← 1 AND
    overwrite-temp-v-priority[s][ALL][ALL] ← 1 AND overwrite-temp-v-π[s][ALL][ALL] ← 1
    AND overwrite-temp-v-id[s][ALL][ALL] ← 1 AND overwrite-temp-v-dist[s][ALL][ALL] ←
    1 AND overwrite-temp-v-gray[s][ALL] ← 1
```

**Box 97**

```
INPUT:
new-message[s][i]=1

OUTPUT:
(i > 1) new-message[s][i-1]←1

OUTPUT:
(i = 1) to-accept-temp-pointer[s][ALL]←1 AND to-accept-temp-last-in-q[s][ALL]←1 AND to-
    accept-temp-v-gray[s][ALL]←1 AND to-accept-temp-v-dist[s][ALL][ALL]←1 AND to-accept
    -temp-v-
π[s][ALL][ALL]←1 AND to-accept-temp-v-id[s][ALL][ALL]←1 AND to-accept-temp-v-priority[s
    ][ALL][ALL]←1 AND return-message-flag[s]←1
```

**Box 98**

```
INPUT:
compare-last-pointer[s][i]=0 AND temp-pointer-slot[s][i]=1 AND last-pointer[s][i]=1 AND
    to-accept-temp-pointer[s][i]=0 AND overwrite-temp-pointer[s][i]=0

OUTPUT:
temp-pointer-slot[s][i]←1
```

**Box 99**

```
INPUT:
compare-last-pointer[s][i]=0 AND temp-pointer-slot[s][i]=1 AND last-pointer[s][i]=0 AND
    to-accept-temp-pointer[s][i]=0 AND overwrite-temp-pointer[s][i]=0

OUTPUT:
temp-pointer-slot[s][i]←1
```

**Box 100**

```
INPUT:
compare-last-pointer[s][i]=0 AND temp-pointer-slot[s][i]=1 AND last-pointer[s][i]=0 or 1
    AND to-accept-temp-pointer[s][i]=1 AND overwrite-temp-pointer[s][i]=0

OUTPUT:
delay-new-pointer[s][i]←1
```

**Box 101**

```
INPUT:
compare-last-pointer[s][i]=0 AND temp-pointer-slot[s][i]=1 AND last-pointer[s][i]=1 or 1
    AND to-accept-temp-pointer[s][i]=1 AND overwrite-temp-pointer[s][i]=0

OUTPUT:
delay-new-pointer[s][i]←1
```

The instructions in Boxes 92 to 96, collectively perform a bit-by-bit comparison between an incoming pointer in any of the slots and the last local pointer, when `compare-last-pointer` is set to 1. If the incoming pointer in any of the messages is larger, it means a new message has arrived. Thus, one of the bits in the same slot of `new-message` is activated. The instructions in Box 97, after a few iterations, activate the accept flags for the variable in the incoming message of that slot. Otherwise, the bits in the same slot of `old-message` get activated. In this case, instructions in Box 96, after a few iteration, will turn on the overwrite flag for all the variable in the incoming message of that slot, clearing the path for next incoming messages. The instructions in Boxes 98 and 99 preserve the `temp-pointer-slot` until the comparison flag is activated. The instructions in Boxes 100 and 101 put the incoming pointer into a new variable `delay-new-pointer` for further processing when the accept flag for pointer gets activated.

The instructions in Boxes 92 to 96 collectively perform a bit-by-bit comparison between an incoming pointer in any of the slots and the last local pointer, when `compare-last-pointer` is set to 1. If the incoming pointer in any of the messages is larger, it means that a new message has arrived. Thus, one of the bits in the same slot of `new-message` is activated. The instructions in Box 97, after a few iterations, activate the accept flags for the variable in the incoming message of that slot. Otherwise, the bits in the same slot of `old-message` get activated. In this case, the instructions in Box 96, after a few iterations, will turn on the overwrite flag for all the variables in the incoming message of that slot, clearing the path for the next incoming messages. The instructions in Boxes 98 and 99 preserve the `temp-pointer-slot` until the comparison flag is activated. When the accept flag for the pointer is activated, the instructions in Boxes 100 and 101 store the incoming pointer in a new variable, `delay-new-pointer`, for further processing.

**Take new messages from the -delays and put them into the buffers for received messages, as well as put them back into the placeholder**

Input blocks used in instruction definitions:

```
Index definitions:
Let i ∈ [l]
Let d ∈ [D]
Let s ∈ [D² + 2]

Block definitions:
  (delay-new-last-in-q[s][i], shutdown-last-in-q-delay[s][i])
  (delay-new-pointer[s][i], shutdown-pointer-delay[s][i])
  (delay-new-v-gray[s][d], shutdown-v-gray-delay[s][d])
  (delay-new-v-priority[s][d][i], shutdown-v-priority-delay[s][d][i])
  (delay-new-v-dist[s][d][i], shutdown-v-dist-delay[s][d][i])
  (delay-new-v-id[s][d][i], shutdown-v-id-delay[s][d][i])
  (delay-new-v-π[s][d][i], shutdown-v-π-delay[s][d][i])
  (delay-new-message-flag[s], shutdown-message-flag-delay[s])
```

Instructions:

**Box 102**

```
INPUT:
delay-new-last-in-q[s][i]=1 AND shutdown-last-in-q-delay[s][i]=0

OUTPUT:
(s< D² + 2) delay-new-last-in-q[s+1][i]←1

OUTPUT:
(s= D² + 2) received-last-in-q[i]←1 AND last-in-q-placeholder[ALL][i]←1
```

**Box 103**

```
INPUT:
delay-new-pointer[s][i]=1 AND shutdown-pointer-delay[s][i]=0

OUTPUT:
(s< D² + 2) delay-new-pointer[s+1][i]←1

OUTPUT:
(s= D² + 2) received-pointer[i]←1 AND pointer-placeholder[ALL][i]←1 AND persistent-last-
    pointer[ALL][i]←1
```

**Box 104**

```
INPUT:
delay-new-v-gray[s][d]=1 AND shutdown-v-gray-delay[s][d]=0

OUTPUT:
(s< D² + 2) delay-new-v-gray[s+1][d]←1

OUTPUT:
(s= D² + 2) received-v-gray[d]←1 AND v-gray-placeholder[ALL][d]←1
```

**Box 105**

```
INPUT:
delay-new-v-priority[s][d][i]=1 AND shutdown-v-priority-delay[s][d][i]=0

OUTPUT:
(s< D² + 2) delay-new-v-priority[s+1][d][i]←1

OUTPUT:
(s= D² + 2) received-v-priority[d][i]←1 AND v-priority-placeholder[ALL][d][i]←1
```

**Box 106**

```
INPUT:
delay-new-v-dist[s][d][i]=1 AND shutdown-v-dist-delay[s][d][i]=0

OUTPUT:
(s< D² + 2) delay-new-v-dist[s+1][d][i]←1

OUTPUT:
(s= D² + 2) received-v-dist[d][i]←1 AND v-dist-placeholder[ALL][d][i]←1
```

**Box 107**

```
INPUT:
delay-new-v-id[s][d][i]=1 AND shutdown-v-id-delay[s][d][i]=0

OUTPUT:
(s< D² + 2) delay-new-v-id[s+1][d][i]←1

OUTPUT:
(s= D² + 2) received-v-id[d][i]←1 AND v-id-placeholder[ALL][d][i]←1
```

**Box 108**

```
INPUT:
delay-new-v-π[s][d][i]=1 AND shutdown-v-π-delay[s][d][i]=0

OUTPUT:
(s< D² + 2) delay-new-v-π[s+1][d][i]←1
```

```
OUTPUT:
(s= D² + 2) received-v-π[d][i]←1 AND v-π-placeholder[ALL][d][i]←1
```

**Box 109**

```
INPUT:
delay-new-message-flag[s]=1 AND shutdown-message-flag-delay[s]=0

OUTPUT:
(s< D² + 1) delay-new-message-flag[s+1]←1

OUTPUT:
(s= D² + 1) delay-new-message-flag[s+1] ← 1 AND overwrite-last-pointer[ALL][ALL] ← 1 AND
    shutdown-message-flag-delay[for all s< D² + 2]← 1 AND shutdown-pointer-delay[for all
    s< D² + 2][ALL]← 1 AND shutdown-v-gray-delay[for all s< D² + 2][ALL]← 1 AND shutdown
    -v-dist-delay[for all s< D² + 2][ALL][ALL]← 1 AND shutdown-v-id-delay[for all s
    < D² + 2][ALL][ALL]← 1 AND shutdown-v-π-delay[for all s< D² + 2][ALL][ALL]← 1 AND
    shutdown-v-priority-delay[for all s< D² + 2][ALL][ALL]← 1 AND shutdown-last-in-q-
    delay[for all s< D² + 2][ALL]← 1

OUTPUT:
(s= D² + 2) message-flag-placeholder[ALL] ← 1 AND received-new-message ← 1 AND check-u-
    id-with-received-v-ids [ALL][ALL] ← 1 AND u-id-comparison-counter[ALL][1] ← 1
```

The instructions in Box 102 make sure that once a new last-in-q value is copied into one of the slots of `delay-new-last-in-q`, it moves from one slot to the next, like a cascade chain. Once it reaches the last slot, it is copied into a destination variable, `received-last-in-q`. At this point, $D^2 + 1$ copies of the value are also made in `last-in-q-placeholder`, which from there will be moved to one of the slots in `last-in-q-slot` to be shared with the neighbors. The reason is that each node needs to relay any new messages it receives to other nodes in its neighborhood. In the case that `shutdown-last-in-q-delay` is activated, all the values in `delay-new-last-in-q` are removed. The purpose of this cascade chain mechanism is to avoid having a growing number of instructions directly writing into the destination variable, which would result in a growing number of collisions. The instructions in Boxes 103 to 108 do the same for their corresponding variables.

The instructions in Box 109 also make sure that once a new-message-flag is copied into any of the slots in `delay-new-message-flag`, it moves from one slot to the next until it reaches the end. Once the flag arrives at the $D^2 + 1$-th slot, the instructions activate the overwrite flag of the last local pointer (it needs to be replaced with the new received one). They also activate shutdown flags. Once it reaches the $D^2 + 2$-th slot, the instructions copy new-message-flag to a destination variable, `received-new-message`. They also create $D^2 + 1$ copies of the flag in `message-flag-placeholder`, from where it will be moved into one of the slots in `message-flag-slot` to be included in the relayed message. The instructions also activate a flag variable, `check-u-id-with-received-v-ids`, which allows bit-by-bit comparison of u-id with v-ids in the new message for a potential match. The counter for the comparison is also initiated.

**Check local ids with ids in the message and set update flags for the matching ones**

Input blocks used in instruction definitions:

```
Index definitions:
Let i ∈ [l]
Let j ∈ [l + 1]
Let k ∈ [D]
Let d ∈ [D]
Let t ∈ [D + l + 2]

Block definitions:
  (v-id[d][i], upload-v-id[d][i])
  (received-new-message)
  (received-v-ids[k][i], u-id2check-with-received-v-ids[k][i], check-u-id-with-received-
    v-ids[k][i])
```

```
    (received-v-ids2check-with-v-ids[d][k][i], v-ids2check-with-received-v-ids[d][k][i],
      check-v-ids-with-received-v-ids[d][k][i])
    (u-equals-received-v-id[k][i], u-is-received-v-id-counter[k][i], ...
    reset-u-equals-v[k][i])
    (v-equals-received-v-id[d][k][i], v-is-received-v-id-counter[d][k][i], ...
    reset-v-equals-v[d][k][i])
    (u-id-comparison-counter[k][j])
    (v-id-comparison-counter[d][k][j])
    (update-pointer-counter[t])
```

The instructions:

**Box 110**

```
INPUT:
v-id[d][i]=1 AND upload-v-id[d][i]=0

OUTPUT:
v-id[d][i]←1 AND v-ids2check-with-received-v-ids[d][ALL][i]←1
```

**Box 111**

```
INPUT:
v-id[d][i]=1 AND upload-v-id[d][i]=1

OUTPUT:
v-id[d][i]←1 AND v-ids2check-with-received-v-ids[d][ALL][i]←1 AND v-id-placeholder[ALL
    ][d][i]←1
```

**Box 112**

```
INPUT:
received-new-message=1

OUTPUT:
check-v-ids-with-received-v-ids[ALL][ALL][ALL]←1 AND update-pointer-counter[1 (=j, j in
    [l+2+D] )], v-id-comparison-counter[ALL][ALL][1]
```

**Box 113**

```
INPUT:
received-v-id[k][i]=1 AND u-id2check-with-received-v-ids[k][i]=1 AND check-u-id-with-
    received-v-ids[k][i]=1

OUTPUT:
(i=1) u-equals-received-v-id[k][i]←1 AND received-v-ids2check-with-v-ids[ALL][k][i]←1
    AND u-is-received-v-id-counter[k][i]←1

OUTPUT:
(i > 1) u-equals-received-v-id[k][i]←1 AND received-v-ids2check-with-v-ids[ALL][k][i]←1
```

**Box 114**

```
INPUT:
received-v-id[k][i]=0 AND u-id2check-with-received-v-ids[k][i]=0 AND check-u-id-with-
    received-v-ids[k][i]=1

OUTPUT:
(i=1) u-equals-received-v-id[k][i]←1 AND u-is-received-v-id-counter[k][i]←1

OUTPUT:
(i > 1) u-equals-received-v-id[k][i]←1
```

**Box 115**

```
INPUT:
received-v-id[k][i]=1 AND u-id2check-with-received-v-ids[k][i]=0 AND check-u-id-with-
    received-v-ids[k][i]=1

OUTPUT:
received-v-ids2check-with-v-ids[ALL][k][i]←1
```

**Box 116**

```
INPUT:
received-v-ids2check-with-v-ids[d][k][i]=1 AND v-ids2check-with-received-v-ids[d][k][i
    ]=1 AND check-v-ids-with-received-v-ids[d][k][i]=1

OUTPUT:
(i=1) v-equals-received-v-id[d][k][i]←1 AND v-is-received-v-id-counter[d][k][i]←1

OUTPUT:
(i > 1) v-equals-received-v-id[d][k][i]←1
```

**Box 117**

```
INPUT:
received-v-ids2check-with-v-ids[d][k][i]=0 AND v-ids2check-with-received-v-ids[d][k][i
    ]=0 AND check-v-ids-with-received-v-ids[d][k][i]=1

OUTPUT:
(i=1) v-equals-received-v-id[d][k][i]←1 AND v-is-received-v-id-counter[d][k][i]←1

OUTPUT:
(i > 1) v-equals-received-v-id[d][k][i]←1
```

**Box 118**

```
INPUT:
u-id-comparison-counter[k][j]=1

OUTPUT:
(j < l+1) u-id-comparison-counter[k][j+1]←1

OUTPUT:
(j = l+1) reset-u-equals-v[k][ALL]←1
```

**Box 119**

```
INPUT:
u-equals-received-v-id[k][i]=1 AND u-is-received-v-id-counter[k][i]=1 AND reset-u-equals
    -v[k][i]=0

OUTPUT:
(i < l) u-is-received-v-id-counter[k][i+1]←1

OUTPUT:
(i = l) u-dist-update[k][ALL]←1 AND, u-π-update[k][ALL]←1 AND u-priority-update[k][ALL]
    ←1 AND u-gray-update[k]←1
```

**Box 120**

```
INPUT:
u-equals-received-v-id[k][i]=1 AND u-is-received-v-id-counter[k][i]=0 AND reset-u-equals
    -v[k][i]=0

OUTPUT:
u-equals-received-v-id[k][i]←1
```

**Box 121**

```
INPUT:
v-id-comparison-counter[d][k][j]=1

OUTPUT:
(j < l + 1) v-id-comparison-counter[d][k][j+1]←1

OUTPUT:
(j = l + 1) reset-v-equals-v[d][k][ALL]←1
```

**Box 122**

```
INPUT:
v-equals-received-v-id[d][k][i]=1 AND v-is-received-v-id-counter[d][k][i]=1 AND reset-v-
    equals-v[d][k][i]=0

OUTPUT:
(i < l) v-is-received-v-id-counter[d][k][i+1]←1

OUTPUT:
(i = l) v-dist-update[d][k][ALL]←1 AND v-π-update[d][k][ALL]←1 AND v-priority-update[d][
    k][ALL]←1 AND v-gray-update[d][k]←1
```

**Box 123**

```
INPUT:
v-equals-received-v-id[d][k][i]=1 AND v-is-received-v-id-counter[d][k][i]=0 AND reset-v-
    equals-v[d][k][i]=0

OUTPUT:
v-equals-received-v-id[d][k][i]←1
```

**Box 124**

```
INPUT:
update-pointer-counter[t]=1

OUTPUT:
(t < l + D + 1) update-pointer-counter[t+1]←1

OUTPUT:
(t = l + D + 1) update-pointer[ALL]←1 AND update-pointer-counter[t+1]←1

OUTPUT:
(t = l + D + 2) compare-u-priority[ALL]←1 AND priority-comparison-counter[1]←1 AND renew-
    received-v-dist[ALL][ALL]←1 AND renew-received-v-π[ALL][ALL]←1 AND renew-received-v
    -priority[ALL][ALL]←1 AND renew-received-v-gray[ALL]←1
```

The instructions in Boxes 110 and 111 preserve `v-id` and, when the upload flag is set, make copies of it in `v-id-placeholder`, whose content will be shared through one of the slots in the outgoing message. The instructions also create $D$ copies of each neighbor ID and put them in `v-ids2check-with-received-v-ids`. These will be used for comparison with IDs in the received message for potential matches. Once the new-message flag passes the delay variable and is copied into `received-new-message`, the instruction in Box 112 activates `check-v-ids-with-received-v-ids` and initiates `v-id-comparison-counter`, which allow bit-by-bit comparison between the neighbor IDs and the IDs in the received message, and check the number of iterations required for the comparison to be completed, respectively. It also initiates `update-pointer-counter`, which, once it reaches the end, allows `q-pointer` to be updated.

The instructions in Boxes 114, 115, 118, 119, and 120 collectively compare the u-id to each of the IDs in the received message. If there is a match, it means u has been discovered before. This activates update flags for the node's variables, such as its distance, $\pi$, priority, and color (indicating discovery status). These variables are then updated with information

from the received message. After the required iterations for comparison are completed, the variable holding the results of the bit-by-bit comparisons is cleared to prevent affecting future comparisons.

The instructions in Boxes 116, 117, 121, 122, and 123 compare the node's neighbors' IDs with the IDs in the received message. If there is a match, it means the node has been discovered before. Thus, the update flags for the variables of the matching neighbors are activated. At the end of the required iterations for comparison, the results of the bit-by-bit comparisons are removed to clear the way for future comparisons.

The instructions in Box 124 advance the counter to set up the update flag for `q-pointer` based on the pointer in the received message. When the counter reaches the end, it also activates the compare flag, which allows `u-priority` to be compared to `q-pointer`, and initiates the comparison counter for it as well. Additionally, it activates flags to reset the values in the variables that hold the information of the received message to zero, clearing the way for information from future received messages.

**Update the current node's distance, priority, parent, white and gray**

Input blocks used in instruction definitions:

```
Index definitions:
Let i ∈ [l]
Let k ∈ [D]

Block definitions:
  (received-v-priority[k][i], renew-received-v-priority[k][i])
  (u-priority-update[k][i], received-v-priority4u[k][i])
  (u-priority-delay[k][i])
  (received-v-dist[k][i], renew-received-v-dist[k][i])
  (u-dist-update[k][i], received-v-dist4u[k][i])
  (u-dist-delay[k][i])
  (received-v-π[k][i], renew-received-v-π[k][i])
  (u-π-update[k][i], received-v-π4u[k][i])
  (u-π-delay[k][i])
  (received-v-gray[k], new-received-v-gray[k])
  (u-gray-update[k], received-v-grays4u[k])
  (u-gray-delay[k])
  (received-pointer[i], update-pointer[i])
  (received-last-in-q[i], update-last-in-q[i])
```

The instructions:

**Box 125**

```
INPUT:
received-v-priority[k][i]=1 AND renew-received-v-priority[k][i]=0

OUTPUT:
received-v-priority[k][i]←1 AND received-v-priority4u[k][i]←1 , received-v-priority4v[
    ALL][k][i]←1
```

**Box 126**

```
INPUT:
u-priority-update[k][i]=1 AND received-v-priority4u[k][i]=1

OUTPUT:
u-priority-delay[k][i]←1
```

**Box 127**

```
INPUT:
u-priority-delay[k][i]=1

OUTPUT:
```

```
(k < D) u-priority-delay[k+1][i]←1
```

**OUTPUT:**
```
(k = D) u-priority[i]←1
```

---

**Box 128**

**INPUT:**
```
received-v-dist[k][i]=1 AND renew-received-v-dist[k][i]=0
```

**OUTPUT:**
```
received-v-dist[k][i]←1 AND received-v-dist4u[k][i]←1 , received-v-dist4v[ALL][k][i]←1
```

---

**Box 129**

**INPUT:**
```
u-dist-update[k][i]=1 AND received-v-dist4u[k][i]=1
```

**OUTPUT:**
```
u-dist-delay[k][i]←1
```

---

**Box 130**

**INPUT:**
```
u-dist-delay[k][i]=1
```

**OUTPUT:**
```
(k < D) u-dist-delay[k+1][i]←1
```

**OUTPUT:**
```
(k = D) u-dist[i]←1
```

---

**Box 131**

**INPUT:**
```
received-v-π[k][i]=1 AND renew-received-v-π[k][i]=0
```

**OUTPUT:**
```
received-v-π[k][i]←1 AND received-v-π4u[k][i]←1 AND received-v-π4v[ALL][k][i]←1
```

---

**Box 132**

**INPUT:**
```
u-π-update[k][i]=1 AND received-v-π4u[k][i]=1
```

**OUTPUT:**
```
u-π-delay[k][i]←1
```

---

**Box 133**

**INPUT:**
```
u-π-delay[k][i]=1
```

**OUTPUT:**
```
(k< D) u-π-delay[k+1][i]←1
```

**OUTPUT:**
```
(k = D) u-π[i]←1
```

---

**Box 134**

**INPUT:**

```
received-v-gray[k]=1 AND renew-received-v-gray[k]=0

OUTPUT:
received-v-gray[k]←1 AND received-v-gray4u[k]←1 , received-v-gray4v[ALL][k]←1
```

**Box 135**

```
INPUT:
u-gray-update[k]=1 AND received-v-gray4u[k]=1

OUTPUT:
u-gray-delay[k]←1
```

**Box 136**

```
INPUT:
u-gray-delay[k]=1

OUTPUT:
(k< D) u-gray-delay[k+1]←1

OUTPUT:
(k = D) u-gray←1 AND u-white-overwrite←1
```

**Box 137**

```
INPUT:
received-pointer[i]=1 AND update-pointer[i]=0

OUTPUT:
received-pointer[i]←1
```

**Box 138**

```
INPUT:
received-pointer[i]=1 AND update-pointer[i]=1

OUTPUT:
q-pointer[i]←1
```

**Box 139**

```
INPUT:
received-last-in-q[i]=1 AND update-last-in-q[i]=0

OUTPUT:
received-last-in-q[i]←1
```

**Box 140**

```
INPUT:
received-last-in-q[i]=1 AND update-last-in-q[i]=1

OUTPUT:
u-last-in-q[i]←1
```

The instructions in Box 125 preserve `received-v-priority` until the renew flag sets it to zero again. They also make a copy of the priorities in the received message for the purpose of updating the priority of u (`received-v-priority4u`) and the neighbors (`received-v-priority4v`) for the case that there are matches with nodes in the received message. In the case that the ID of one of the nodes in the received message matches `u-id`, the flag `u-priority-update` is activated to update the priority of u.

The instructions in Box 126 place a copy of the priority of the matching node from the received message into `u-priority-delay`. Here, `u-priority-delay` acts as a cascade chain. The instructions in Box 127 ensure that the new priority value moves along the first dimension until it reaches the last coordinate $D$. At this point, the value of `u-priority` is updated. This cascade mechanism prevents growing collisions in `u-priority`. The instructions in Boxes 128 to 136 perform the same function for other variables.

The instructions in Boxes 137 and 138 preserve the `received-pointer` until the update flag is activated. At that point, `q-pointer` is updated with the new value. The instructions in Boxes 139 and 140 preserve the `received-last-in-q` until the update flag is activated. At that point, `u-last-in-q` is updated.

### Update the adjacent nodes' distance, parent, white, and gray

Input blocks used in instruction definitions:

```
Index definition:
Let i ∈ [l]
Let k ∈ [D]
Let d ∈ [D]

Block definition:
   (v-priority-update[d][k][i], received-v-priority4v[d][k][i])
   (v-dist-update[d][k][i], received-v-dist4v[d][k][i])
   (v-π-update[d][k][i], received-v-π4v[d][k][i])
   (v-gray-update[d][k], received-v-gray4v[d][k])
   (v-priority-delay[d][k][i])
   (v-dist-delay[d][k][i])
   (v-π-delay[d][k][i])
   (v-gray-delay[d][k])
```

Instructions:

**Box 141**

```
INPUT:
v-priority-update[d][k][i]=1 AND received-v-priority4v[d][k][i]=1

OUTPUT:
v-priority-delay[d][k][i]←1
```

**Box 142**

```
INPUT:
v-priority-delay[d][k][i]=1

OUTPUT:
(k< D) v-priority-delay[d][k+1][i]←1

OUTPUT:
(k = D) v-priority[d][i]←1
```

**Box 143**

```
INPUT:
v-dist-update[d][k][i]=1 AND received-v-dist4v[d][k][i]=1

OUTPUT:
v-dist-delay[d][k][i]←1
```

**Box 144**

```
INPUT:
v-dist-delay[d][k][i]=1
```

```
OUTPUT:
(k< D) v-dist-delay[d][k+1][i]←1

OUTPUT:
(k = D) v-dist[d][i]←1
```

**Box 145**

```
INPUT:
v-π-update[d][k][i]=1 AND received-v-π4v[d][k][i]=1

OUTPUT:
v-π-delay[d][k][i]←1
```

**Box 146**

```
INPUT:
v-π-delay[d][k][i]=1

OUTPUT:
(k < D) v-π-delay[d][k+1][i]←1

OUTPUT:
(k = D) v-π[d][i]←1
```

**Box 147**

```
INPUT:
v-gray-update[d][k]=1 AND received-v-gray4v[d][k]=1

OUTPUT:
v-gray-delay[d][k]←1
```

**Box 148**

```
INPUT:
v-gray-delay[d][k]=1

OUTPUT:
(k < D) v-gray-delay[d][k+1]←1

OUTPUT:
(k = D) v-gray[d]←1 AND v-white-overwrite[d]←1
```

In the case that the ID of one of the nodes in the received message matches any of the neighbors' IDs, the flag `v-priority-update` activates to update the priority of that neighbor. The instructions in Box 141 place a copy of the priority of the matching node from the received message into `v-priority-delay` at the coordinates of that neighbor. Here, `v-priority-delay` acts as a cascade chain. The instructions in Box 142 ensure that the new priority value moves along the second dimension until it reaches the last coordinate $D$. At this point, the value of v-priority gets updated for that neighbor. This cascade mechanism is used to avoid growing collisions in `v-priority`. The instructions in Boxes 143 to 148 perform the same operations for the other variables.

E.3.2. The Specifications of the Training Data

Based on the defined binary variables, one can calculate the dimension of the binary vector before input encoding, denoted by $k$. By simply counting the number of bits, we obtain

$$k = 8D^3 + 11D^2 + 22D + 58l + 84Dl + 38D^2l + 32D^3l + 18.$$

This value is the dimension of the output vector, as the output is not in encoded form. The variables in the message section, $\boldsymbol{x}_M$, are `pointer-slot`, `last-in-q-slot`, `v-gray-slot`, `v-dist-slot`, `v-pi-slot`, `v-id-slot`,

`v-priority-slot`, and `message-flag-slot`, with a total of

$$4D^3l + D^3 + 2D^2l + D^2 + 4Dl + D + 2l + 1$$

bits. Thus, $k_M$ in Equation (3) is equal to this value.

We obtain the total number of instructions, denoted by $n_T$, by simply counting the number of instructions in each box (each box represents the instructions for the entire range of the indexing variables). This yields:

$$n_T = 6D^3 + 9D^2 + 17D + 57l + 65Dl + 33D^2l + 24D^3l + 15.$$

Since each instruction is converted into a training sample, the number of training samples and the input embedding dimension before augmentation are both equal to $n_T$.

### E.3.3. MAXIMUM NUMBER OF COLLISIONS

The maximum number of collisions in the instructions of this algorithm is 5. Consider the bits in `temp-pointer-slot`. One instruction from Boxes 69, 92, 94, 98, and 99 may set each bit in `temp-pointer-slot` to 1. Thus, there are 5 instructions that collide on each bit. Since the number of collisions is constant, the size of the training dataset after augmentation with idle samples as well as the input embedding dimension, denoted by $k'$, are proportional to $n_T$. Thus, similar to $n_T$, we have $k' = \mathcal{O}(l \cdot D^3)$.

### E.3.4. INITIALIZATION

To run the algorithm, certain variables must be properly initialized at each node. Let the local ID of a node $u$ be denoted by $u_\mathcal{L}$, and its global ID by $u_\mathcal{G}$. We denote the source node by $s$, while all other nodes are denoted by $u$. The neighbors of a node are denoted by $v_i$ for $i \in [D]$. The notation $\mathrm{Bin}(z, l)$ denotes the binary representation of the decimal number $z$ using $l$ bits.

Below, we first describe the initialization of variables for the source node $s$, and then for any other node $u$. Note that all variables that are not explicitly mentioned are initialized to zero.

```
Source Intitialization
q-pointer ← Bin(1,l)
priority-comparison-counter ← Bin(1,l+1)
compare-u-priority[ALL] ← 1
u-id ← Bin(s_G,l)
u_priority ← Bin(1,l)
u-gray ← 1
v_last_in_q_augend[1][2] ← 1
v_last_in_q[1][2] ← 1
determine-slot[s_L] ← 1
v-id[i] ← Bin(v_iG,l) for i ∈ [D]
v-white[i] ← 1 for i ∈ [D]
```

```
Non-Source Intitialization
q-pointer ← Bin(1,l)
priority-comparison-counter ← Bin(1,l+1)
compare-u-priority[ALL] ← 1
u-id ← Bin(u_G,l)
determine-slot[u_L] ← 1
v-id[i] ← Bin(v_iG,l) for i ∈ [D]
v-white[i] ← 1 for i ∈ [D] if v_i ≠ s
v-gray[i] ← 1 for i ∈ [D] if v_i = s
```

### E.3.5. SUFFICIENT UPPER BOUND ON THE NUMBER OF ITERATIONS

Here, we provide a sufficient upper bound on the number of iterations required by the graph template matching framework with the instructions described above to complete the algorithm. We begin immediately after an aggregation step, where new information appears through one of the message-passing slots. We assume that the node's priority matches the incoming pointer, and we count all iterations until the node places its own message into the message section of the output vector.

It takes one iteration for the instruction in Box 76 to activate `compare-last-pointer`, which enables the comparison between the most recent local pointer and the pointer in the incoming message. Then, it takes $l + 1$ iterations for the instructions in Boxes 92 to 97 to compare the pointers and activate `return-message-flag`, as well as the accept flag for the incoming information. Next, it takes one iteration for the instructions in Boxes 79 to 91 to place the information into the delay variables, including `delay-new-message-flag`. Based on the instructions in Boxes 102 to 109, it takes at most $D^2 + 2$ iterations for the information to pass through the delay mechanism and be stored in the destination variables whose names start with ``received-'', including `received-new-message`, and to be placed into placeholders for relaying to neighboring nodes. The total number of iterations up to this point is $D^2 + l + 4$.

As a side note, in the following iteration, the information stored in the placeholders is copied to the slots in the message section of the output vector and shared with the neighbors. Thus, it takes $D^2 + l + 5$ iterations for a node to simply relay a newly received message to its immediate neighbors.

When `received-new-message` is activated, it takes one iteration for the instruction in Box 112 to initiate `update-pointer-counter`. After an additional $D + l + 2$ iterations, the instructions in Box 124 activate the bits of `compare-u-priority`, which allow the comparison between the node's priority and the incoming pointer. It then takes $l + 1$ iterations for the comparison to be completed and for `u-matched` to be activated. At this point, it takes $D \cdot (l + 1) + 1$ iterations for the instructions in Boxes 8 to 21 to assign priorities to the newly discovered neighbors one by one and to initiate the process of incrementing the current pointer. Subsequently, it takes $2l$ iterations for the instructions in Boxes 45 to 49 to increment the pointer and activate the upload flags. This is followed by one iteration to copy the variables to their corresponding placeholders, and one additional iteration for the instructions in Boxes 53 to 68 to place the contents of the placeholders into the corresponding slots. At this stage, the node shares its own message with its neighbors.

Therefore, from the moment a node receives a message whose pointer value matches its priority to the moment it sends out a message with an updated pointer, the total number of iterations required is at most

$$2Dl + D^2 + 2D + 5l + 12.$$

In the worst-case scenario, we may assume that the message received by a node originates from a node at distance $d_G$, where $d_G$ denotes the diameter of the graph. In this case, the message requires $d_G - 1$ relay steps to reach the node. Consequently, when a message circulating in the graph has a pointer equal to the priority of a node, it may take up to $(d_G - 1)(D^2 + l + 5)$ iterations for it to reach that node, followed by an additional $2Dl + D^2 + 2D + 5l + 12$ iterations for the node to be expanded and to send out a message with updated information and pointer.

Since this process is repeated for all $n$ nodes, a sufficient upper bound on the total number of iterations required for the completion of the algorithm is

$$n \left( (d_G - 1)(l + D^2 + 5) + (2Dl + D^2 + 2D + 5l + 12) \right).$$

Note that, since our objective is to derive a sufficient upper bound, we do not treat the source node differently.

## E.4. Depth-first Search

In this section, we provide instructions that define the function $f$ in our graph template matching framework for executing the Depth-First Search (DFS) algorithm on graphs with maximum degree $D$. We use $l$ bits to represent all variables in our implementation, including the global identifiers of nodes. Consequently, at test time, the maximum graph size is limited to $2^l - 1$ nodes. Algorithm 6 presents the pseudocode of the implemented DFS algorithm.

### E.4.1. INSTRUCTIONS OF DFS

In this section, we present the instructions for implementing DFS in our framework. For the sake of brevity, we do not explain the purpose of each variable. However, we have chosen the variable names to be as self-explanatory as possible. We divide the instructions into several sections. Each section includes instructions that together form a meaningful algorithmic subroutine. Each section starts by introducing the blocks for which the instructions in that section are defined, followed by the instructions themselves. Similarly, for brevity, we avoid providing explanations for the instructions. Nonetheless, the explanations given for BFS can help to better understand the working dynamics here as well.

**Checking if the node id is equal to "current"**

---

**Algorithm 6** DFS$(G, s)$

---

1: **for each** vertex $u \in G.V$ **do**
2:     $u.white \leftarrow true$
3:     $u.gray \leftarrow false$
4:     $u.parent \leftarrow 0$
5:     $u.next \leftarrow 1$
6:     $u.d \leftarrow \infty$
7: **end for**
8: $s.white \leftarrow false$
9: $s.gray \leftarrow true$
10: $s.parent \leftarrow 0$
11: $current \leftarrow s.id$
12: $s.d \leftarrow 0$
13: $stamp \leftarrow 1$
14: **while** true **do**
15:     $u \leftarrow$ vertex $v \in G.V$ such that $v.id = current$
16:     **if** $u.next < D + 1$ **then**
17:         $v \leftarrow G.Adj[u][u.next]$
18:         $u.next \leftarrow u.next + 1$
19:         **if** $v.white = true$ **then**
20:             $v.white \leftarrow false$
21:             $v.gray \leftarrow true$
22:             $v.parent \leftarrow u.id$
23:             $v.d \leftarrow u.d + 1$
24:             $stamp \leftarrow stamp + 1$
25:             $current \leftarrow v.id$
26:         **end if**
27:     **else**
28:         $current \leftarrow u.parent$
29:     **end if**
30: **end while**

---

Input blocks used in instruction definitions:

```
Index definition:
Let i ∈ [l]
Let j ∈ [l + 1]

Block definition:
   (u-id[i], current[i])
   (id-equals[i], id-equals-counter[i])
   (comparison-counter[j])
```

Instructions:

```
INPUT:
u-id[i]=1 AND current[i]=1 AND compare-current[i]=1:

OUTPUT:
(i= 1) id-equals[i]←1 AND u-id-asparent[ALL][i]←1 AND u-id[i]←1 AND id-equals-counter[i
   ]←1 AND u-ids2check-with-received-v-ids[ALL][i]←1

OUTPUT:
(i> 1) id-equals[i]←1 AND u-id-asparent[ALL][i]←1 AND u-id[i]←1 AND u-ids2check-with-
   received-v-ids[ALL][i]←1
```

```
INPUT:
u-id[i]=1 AND current[i]=1 AND compare-current[i]=0:

OUTPUT:
u-id-asparent[ALL][i]←1 AND u-id[i]←1 AND u-ids2check-with-received-v-ids[ALL][i]←1
```

```
INPUT:
u-id[i]=1 AND current[i]=0 AND compare-current[i]=1:

OUTPUT:
u-id-asparent[ALL][i]←1 AND u-id[i]←1 AND u-ids2check-with-received-v-ids[ALL][i]←1
```

```
INPUT:
u-id[i]=1 AND current[i]=0 AND compare-current[i]=0:

OUTPUT:
u-id-asparent[ALL][i]←1 AND u-id[i]←1 AND u-ids2check-with-received-v-ids[ALL][i]←1
```

```
INPUT:
u-id[i]=0 AND current[i]=0 AND compare-current[i]=1:

OUTPUT:
(i= 1) id-equals[i]←1 AND  id-equals-counter[i]←1

OUTPUT:
(i> 1) id-equals[i]←1
```

```
INPUT:
comparison-counter[j]=1:

OUTPUT:
(j < l) comparison-counter[j+1]←1

OUTPUT:
(j= l) comparison-counter[j+1]←1 AND update-stamp[ALL]←1

OUTPUT:
(j = l + 1) reset-is-equal[ALL]←1
```

```
INPUT:
id-equals[i]=1 AND id-equals-counter[i]=0 AND reset-is-equal[i]=0:

OUTPUT:
id-equals[i]←1 AND
```

```
INPUT:
id-equals[i]=1 AND id-equals-counter[i]=1 AND reset-is-equal[i]=0:

OUTPUT:
(i< l − 1) id-equals-counter[i+1]←1

OUTPUT:
(i= l) u-is-current[ALL]←1 AND accept-stamp[ALL]←1
```

### Get the message time stamp value and increment it and activate end of loop flags

Input blocks used in instruction definitions:

```
Index definition:
Let i ∈ [l], j ∈ [2l].
Let d ∈ [D]

Block definition:
   (stamp[i])
   (u-stamp-addend-delay[d])
   (u-stamp-augend[i], u-stamp-addend[i], to-copy-stamp-augend[i])
   (u-stamp-carry[i])
   (u-stamp-sum-counter[j])
```

Instructions:

```
INPUT:
stamp[i]=1 AND accept-stamp[i]=1

OUTPUT:
u-stamp-augend[i]←1
```

```
INPUT:
u-stamp-addend-delay[d]=1:

OUTPUT:
(d< D) u-stamp-addend-delay[d+1]←1

OUTPUT:
(d=D) u-stamp-addend[1]←1 AND u-stamp-sum-counter[1]
```

```
INPUT:
u-stamp-augend[i]=1 AND u-stamp-addend[i]=0 AND to-copy-dist-augend[i]=0:

OUTPUT:
u-stamp-augend[i]←1
```

```
INPUT:
u-stamp-augend[i]=0 AND u-stamp-addend[i]=1 AND to-copy-dist-augend[i]=0:

OUTPUT:
u-stamp-augend[i]←1
```

```
INPUT:
u-stamp-augend[i]=1 AND u-stamp-addend[i]=1 AND to-copy-stamp-augend[i]=0:

OUTPUT:
u-stamp-carry[i]←1
```

```
INPUT:
u-stamp-carry[i]=1:

OUTPUT:
(i< l) u-stamp-addend[i+1]←1

OUTPUT:
(i= l) u-stamp-carry[i]←1
```

```
INPUT:
u-stamp-sum-counter[j]=1:

OUTPUT:
(j< 2l) u-stamp-sum-counter[j+1]←1

OUTPUT:
(j= 2l) to-copy-stamp-augend[ALL]←1 AND upload-v-π[ALL][ALL]←1 AND upload-u-current[ALL]
    ←1 AND upload-v-dist[ALL][ALL]←1 AND upload-v-id[ALL][ALL]←1 AND upload-v-white[ALL
    ]←1 AND upload-v-gray[ALL]←1 AND overwrite-last-stamp[ALL][ALL]←1 AND upload-
    message-flag←1
```

```
INPUT:
u-stamp-augend[i]=1 AND u-stamp-addend[i]=0 AND to-copy-stamp-augend[i]=1:

OUTPUT:
u-stamp-placeholder[ALL][i]←1 AND persistent-last-stamp[ALL][i]←1
```

**Checking which neighboring node should be attended and activate flags to update its features**

Input blocks used in instruction definitions:

```
Index definition:
Let d ∈ [D]

Block definition:
  (v-turn[d], u-is-current[d])
  (attend2v[d], v-white[d])
```

Instructions:

```
INPUT:
v-turn[d]=1 AND u-is-current[d]=1:

OUTPUT:
(d< D + 1) attend2v[d]←1 AND v-turn[d+1]←1

OUTPUT:
(d= D + 1) u-stamp-addend[1]←1 AND u-stamp-sum-counter[1] AND set-current-back[ALL]←1
```

```
INPUT:
v-turn[d]=1 AND u-is-current[d]=0:

OUTPUT:
v-turn[d]=1
```

```
INPUT:
attend2v[d]=1 AND v-white[d]=1 AND v-white-overwrite[d]=0:

OUTPUT:
v-gray[d]←1 AND v-set-π[d][ALL]←1 AND v-dist-addend[d][1]←1 AND v-dist-sum-counter[d
    ][1] AND u-stamp-addend-delay[d]←1 AND set-current[d][ALL]←1
```

```
INPUT:
attend2v[d]=1 AND v-white[d]=0 AND v-white-overwrite[d]=0:

OUTPUT:
u-is-current[d+1]←1
```

```
INPUT:
attend2v[d]=0 AND v-white[d]=1 AND v-white-overwrite[d]=0:

OUTPUT:
v-white[d]←1
```

**Setting the current index to the next unexplored neighbor or the parent if all is explored**

Input blocks used in instruction definitions:

```
Index definition:
Let i ∈ [l]
Let d ∈ [D]

Block definition:
   (set-current-back[i], u-π[i])
   (set-current[d][i], v-id[d][i], upload-v-id[d][i])
   (u-current-delay[d][i])
```

Instructions:

```
INPUT:
set-current-back[i]=1 AND u-π[i]=1::

OUTPUT:
u-current[i]←1 AND u-π[i]←1
```

```
INPUT:
set-current-back[i]=0 AND u-π[i]=1::

OUTPUT:
u-π[i]←1
```

```
INPUT:
set-current[d][i]=1 AND v-id[d][i]=1 AND upload-v-id[d][i]=0:

OUTPUT:
u-current-delay[d][i]←1 AND v-id[d][i]←1 AND v-ids2check-with-received-v-ids[d][ALL][i]
    ←1
```

```
INPUT:
set-current[d][i]=0 AND v-id[d][i]=1 AND upload-v-id[d][i]=0:

OUTPUT:
v-id[d][i]←1 AND v-ids2check-with-received-v-ids[d][ALL][i]←1
```

```
INPUT:
set-current[d][i]=0 AND v-id[d][i]=1 AND upload-v-id[d][i]=1:

OUTPUT:
v-id[d][i]←1 AND v-id-placeholder[ALL][d][i]←1 AND v-ids2check-with-received-v-ids[d][
    ALL][i]←1
```

```
INPUT:
u-current-delay[d][i]=1:

OUTPUT:
(d< D) u-current-delay[d+1][i]←1

OUTPUT:
(d=D) u-current[i]←1
```

### Getting this node's distance and incrementing and assigning it to the next node to be explored

Input blocks used in instruction definitions:

```
Index definition:
Let i ∈ [l], j ∈ [2l].
Let d ∈ [D]

Block definition:
  (u-dist[i], update-v-dist-augend[i])
  (v-dist-augend[d][i], v-dist-addend[d][i], to-copy-dist-augend[d][i])
  (v-dist-sum-counter[d][j])
  (v-dist[d][i], upload-v-dist[d][i])
```

Instructions:

```
INPUT:
u-dist[i]=1 AND update-v-dist-augend[i]=1:

OUTPUT:
v-dist-augend[ALL][i]←1 AND u-dist[i]←1
```

```
INPUT:
u-dist[i]=1 AND update-v-dist-augend[i]=0:

OUTPUT:
u-dist[i]←1
```

```
INPUT:
v-dist-augend[d][i]=1 AND v-dist-addend[d][i]=0 AND to-copy-dist-augend[d][i]=0:

OUTPUT:
v-dist-augend[d][i]←1
```

```
INPUT:
v-dist-augend[d][i]=0 AND v-dist-addend[d][i]=1 AND to-copy-dist-augend[d][i]=0:

OUTPUT:
v-dist-augend[d][i]←1
```

```
INPUT:
v-dist-augend[d][i]=1 AND v-dist-addend[d][i]=1 AND to-copy-dist-augend[d][i]=0:

OUTPUT:
v-dist-carry[d][i]←1
```

```
INPUT:
v-dist-carry[d][i]=1:

OUTPUT:
(i< l) v-dist-addend[d][i+1]←1

(i= l) v-dist-carry[d][i]←1
```

```
INPUT:
v-dist-sum-counter[d][j]=1:

OUTPUT:
(j< 2l) v-dist-sum-counter[d][j+1]←1

OUTPUT:
(j= 2l) to-copy-dist-augend[d][ALL]←1
```

```
INPUT:
v-dist-augend[d][i]=1 AND v-dist-addend[d][i]=0 AND to-copy-dist-augend[d][i]=1:

OUTPUT:
v-dist[d][i]←1
```

```
INPUT:
v-dist[d][i]=1 AND upload-v-dist[d][i]=1:

OUTPUT:
v-dist[d][i]←1 AND v-dist-placeholder[ALL][d][i]←1
```

```
INPUT:
v-dist[d][i]=1 AND upload-v-dist[d][i]=0:

OUTPUT:
v-dist[d][i]←1
```

## Assigning the node's ID as parent to the next node to explored

Input blocks used in instruction definitions:

```
Index definition:
Let i ∈ [l].
Let d ∈ [D]

Block definition:
  (v-set-π[d][i], u-id-asparent[d][i])
  (v-π[d][i]=1, upload-v-π[d][i])
```

Instructions:

```
INPUT:
v-set-π[d][i]=1 AND u-id-asparent[d][i]=1

OUTPUT:
v-π[d][i]←1
```

```
INPUT:
v-π[d][i]=1 AND upload-v-π[d][i]=0

OUTPUT:
v-π[d][i]←1
```

```
INPUT:
v-π[d][i]=1 AND upload-v-π[d][i]=1

OUTPUT:
v-π[d][i]←1 AND v-π-placeholder[ALL][d][i]←1
```

### Uploading this node's color as well as the neighbours' colors

Input blocks used in instruction definitions:

```
Index definition:
Let d ∈ [D]

Block definition:
   (v-gray[d], upload-v-gray[d])
```

Instructions:

```
INPUT:
v-gray[d]=1 AND upload-v-gray[d]=0

OUTPUT:
v-gray[d]←1
```

```
INPUT:
v-gray[d]=1 AND upload-v-gray[d]=1

OUTPUT:
v-gray[d]←1 AND v-gray-placeholder[ALL][d]←1
```

```
INPUT:
u-gray =1:

OUTPUT:
u-gray←1
```

```
INPUT:
u-white =1 AND u-white-overwrite =0:

OUTPUT:
u-white←1
```

### Uploading the next node ID to be explored and a flag to declare outgoing message

Input blocks used in instruction definitions:

```
Index definition:
Let i ∈ [l]

Block definition:
   (u-current[i], upload-u-current[i])
   (upload-message-flag)
```

Instructions:

```
INPUT:
u-current[i]=1 AND upload-u-current[i]=0

OUTPUT:
u-current[i]←1
```

```
INPUT:
u-current[i]=1 AND upload-u-current[i]=1

OUTPUT:
u-current-placeholder[ALL][i]←1
```

```
INPUT:
upload-message-flag =1:

OUTPUT:
message-flag-placeholder[ALL]←1
```

## Guide the placeholder content to appropriate slots

Input blocks used in instruction definitions:

```
Index definition:
Let s ∈ [D² + 1], d ∈ [D], i ∈ [l].

Block definition:
   (determine-slot[s])
   (determine-v-gray-slot[s][d], v-gray-placeholder[s][d])
   (determine-v-dist-slot[s][d][i], v-dist-placeholder[s][d][i])
   (determine-v-π-slot[s][d][i], v-π-placeholder[s][d][i])
   (determine-v-id-slot[s][d][i], v-id-placeholder[s][d][i])
   (determine-u-stamp-slot[s][i], u-stamp-placeholder[s][i])
   (determine-u-current-slot[s][i], u-current-placeholder[s][i])
   (determine-message-flag-slot[s], message-flag-placeholder[s])
```

Instructions:

```
INPUT:
determine-slot[s]=1:

OUTPUT:
determine-slot[s]←1 AND determine-v-gray-slot[s][ALL]←1 AND determine-v-dist-slot[s][
    ALL][ALL]←1 AND determine-v-pi-slot[s][ALL][ALL]←1 AND determine-v-id-slot[s][ALL][
    ALL]←1 AND determine-u-current-slot[s][ALL]←1 AND determine-u-stamp-slot[s][ALL]←1
    AND determine-message-flag-slot[s]←1
```

```
INPUT:
determine-v-gray-slot[s][d]=1 AND v-gray-placeholder[s][d]=1:

OUTPUT:
gray-slot[s][d]←1 AND determine-v-gray-slot[s][d]←1
```

```
INPUT:
determine-v-gray-slot[s][d]=1 AND v-gray-placeholder[s][d]=0:

OUTPUT:
determine-v-gray-slot[s][d]←1
```

```
INPUT:
determine-v-dist-slot[s][d][i]=1 AND v-dist-placeholder[s][d][i]=1:

OUTPUT:
v-dist-slot[s][d][i]←1 AND determine-v-dist-slot[s][d][i]←1
```

```
INPUT:
determine-v-dist-slot[s][d][i]=1 AND v-dist-placeholder[s][d][i]=0:
```

```
OUTPUT:
determine-v-dist-slot[s][d][i]←1
```

```
INPUT:
determine-v-π-slot[s][d][i]=1 AND v-π-placeholder[s][d][i]=1:

OUTPUT:
v-π-slot[s][d][i]←1 AND determine-v-π-slot[s][d][i]←1
```

```
INPUT:
determine-v-π-slot[s][d][i]=1 AND v-π-placeholder[s][d][i]=0:

OUTPUT:
determine-v-π-slot[s][d][i]←1
```

```
INPUT:
determine-v-id-slot[s][d][i]=1 AND v-id-placeholder[s][d][i]=1:

OUTPUT:
v-id-slot[s][d][i]←1 AND determine-v-id-slot[s][d][i]←1
```

```
INPUT:
determine-v-id-slot[s][d][i]=1 AND v-id-placeholder[s][d][i]=0:

OUTPUT:
determine-v-id-slot[s][d][i]←1
```

```
INPUT:
determine-v-id-slot[s][d][i]=1 AND v-id-placeholder[s][d][i]=1:

OUTPUT:
v-id-slot[s][d][i]←1 AND determine-v-id-slot[s][d][i]←1
```

```
INPUT:
determine-u-current-slot[s][i]=1 AND u-current-placeholder[s][i]=0:

OUTPUT:
determine-u-current-slot[s][i]←1
```

```
INPUT:
determine-u-current-slot[s][i]=1 AND u-current-placeholder[s][i]=1:

OUTPUT:
u-current-slot[s][i]←1 AND determine-u-current-slot[s][i]←1
```

```
INPUT:
determine-u-stamp-slot[s][i]=1 AND u-stamp-placeholder[s][i]=0:

OUTPUT:
determine-u-stamp-slot[s][i]←1
```

```
INPUT:
determine-u-stamp-slot[s][i]=1 AND u-stamp-placeholder[s][i]=1:

OUTPUT:
u-stamp-slot[s][i]←1 AND determine-u-stamp-slot[s][i]←1
```

```
INPUT:
determine-message-flag-slot[s]=1 AND message-flag-placeholder[s]=0:

OUTPUT:
determine-message-flag-slot[s]←1
```

```
INPUT:
determine-message-flag-slot[s]=1 AND message-flag-placeholder[s]=1:

OUTPUT:
message-flag-slot[s]←1 AND determine-message-flag-slot[s]←1
```

### Copy incoming messages to temporary slots while checking if stamp is new

Input blocks used in instruction definitions:

```
Index definition:
Let i ∈ [l],  s ∈ [D² + 1].
Let d ∈ [D]

Block definition:
   (v-gray-slot[s][d])
   (v-dist-slot[s][d][i])
   (v-π-slot[s][d][i])
   (v-id-slot[s][d][i])
   (u-current-slot[s][i])
   (u-stamp-slot[s][i])
   (message-flag-slot[s])
```

Instructions:

```
INPUT:
v-gray-slot[s][d]=1:

OUTPUT:
temp-v-gray-slot[s][d]←1
```

```
INPUT:
v-dist-slot[s][d][i]=1:

OUTPUT:
temp-v-dist-slot[s][d][i]←1
```

```
INPUT:
v-π-slot[s][d][i]=1:

OUTPUT:
temp-v-π-slot[s][d][i]←1
```

```
INPUT:
v-id-slot[s][d][i]=1:

OUTPUT:
temp-v-id-slot[s][d][i]←1
```

```
INPUT:
u-current-slot[s][i]=1:

OUTPUT:
temp-u-current-slot[s][i]←1
```

```
INPUT:
u-stamp-slot[s][i]=1:

OUTPUT:
temp-u-stamp-slot[s][i]←1
```

```
INPUT:
message-flag-slot[s]=1:

OUTPUT:
compare-last-stamp[s][l]←1
```

## Preserve temp data till the new message check is complete

Input blocks used in instruction definitions:

```
Index definition:
Let i ∈ [l], d ∈ [D], s ∈ [D² + 1].

Block definition:
  (temp-v-gray-slot[s][d], to-accept-temp-v-gray[s][d], overwrite-temp-v-gray[s][d])
  (temp-v-dist-slot[s][d][i], to-accept-temp-v-dist[s][d][i], overwrite-temp-v-dist[s][d
    ][i])
  (temp-v-π-slot[s][d][i], to-accept-temp-v-π[s][d][i], overwrite-temp-v-π[s][d][i])
  (temp-v-id-slot[s][d][i], to-accept-temp-v-id[s][d][i], overwrite-temp-v-id[s][d][i])
  (temp-u-current-slot[s][i], to-accept-temp-u-current[s][i], overwrite-temp-u-current[s
    ][i])
  (temp-u-stamp-slot[s][i], to-accept-temp-u-stamp[s][i], overwrite-temp-u-stamp[s][i])
  (persisiting-last-stamp[s][i], overwrite-last-stamp[s][i])
  (return-message-flag[s])
```

Instructions:

```
INPUT:
temp-v-gray-slot[s][d]=1 AND to-accept-temp-v-gray[s][d]=0 AND overwrite-temp-v-gray[s][
    d]=0:

OUTPUT:
temp-v-gray-slot[s][d]←1
```

```
INPUT:
temp-v-gray-slot[s][d]=1 AND to-accept-temp-v-gray[s][d]=1 AND overwrite-temp-v-gray[s][
    d]=0:

OUTPUT:
delay-new-v-gray[s][d]←1
```

```
INPUT:
temp-v-dist-slot[s][d][i]=1 AND to-accept-temp-v-dist[s][d][i]=0 AND overwrite-temp-v-
    dist[s][d][i]=0:

OUTPUT:
temp-v-dist-slot[s][d][i]←1
```

```
INPUT:
temp-v-dist-slot[s][d][i]=1 AND to-accept-temp-v-dist[s][d][i]=1 AND overwrite-temp-v-
    dist[s][d][i]=0:

OUTPUT:
delay-new-v-dist[s][d][i]←1
```

```
INPUT:
temp-v-π-slot[s][d][i]=1 AND to-accept-temp-v-π[s][d][i]=0 AND overwrite-temp-v-π[s][d][
    i]=0:

OUTPUT:
temp-v-π-slot[s][d][i]←1
```

```
INPUT:
temp-v-π-slot[s][d][i]=1 AND to-accept-temp-v-π[s][d][i]=1 AND overwrite-temp-v-π[s][d][
    i]=0:

OUTPUT:
delay-new-v-π-slot[s][d][i]←1
```

```
INPUT:
temp-v-id-slot[s][d][i]=1 AND to-accept-temp-v-id[s][d][i]=0 AND overwrite-temp-v-id[s][
    d][i]=0:

OUTPUT:
temp-v-id-slot[s][d][i]←1
```

```
INPUT:
temp-v-id-slot[s][d][i]=1 AND to-accept-temp-v-id[s][d][i]=1 AND overwrite-temp-v-id[s][
    d][i]=0:

OUTPUT:
delay-new-v-id[s][d][i]←1
```

```
INPUT:
temp-u-current-slot[s][i]=1 AND to-accept-temp-u-current[s][i]=0 AND overwrite-temp-u-
    current[s][i]=0:

OUTPUT:
temp-u-current-slot[s][i]←1
```

```
INPUT:
temp-u-current-slot[s][i]=1 AND to-accept-temp-u-current[s][i]=1 AND overwrite-temp-u-
    current[s][i]=0:

OUTPUT:
delay-new-current[s][i]←1
```

```
INPUT:
persistent-last-stamp[s][i]=1 AND overwrite-last-stamp[s][i]=0:

OUTPUT:
last-stamp[s][i]←1 AND persistent-last-stamp[s][i]=1←1
```

```
INPUT:
persistent-last-stamp[s][i]=1 AND overwrite-last-stamp[s][i]=1:

OUTPUT:
last-stamp[s][i]←1
```

```
INPUT:
return-message-flag[s]=1

OUTPUT:
delay-new-message-flag[s]←1
```

**Check if the stamp is new**

Input blocks used in instruction definitions:

```
Index definition:
Let i ∈ [l].
Let s ∈ [D² + 1]

Block definition:
   (compare-last-stamp[s][i], temp-u-stamp-slot[s][i], last-stamp[s][i],
    to-accept-temp-u-stamp[s][i], overwrite-temp-u-stamp[s][i])
   (new-message[s][i])
   (old-message[s][i])
```

Instructions:

```
INPUT:
compare-last-stamp[s][i]=1 AND temp-u-stamp-slot[s][i]=1 AND last-stamp[s][i]=1 AND to-
    accept-temp-u-stamp[s][i]=0 AND overwrite-temp-u-stamp[s][i]=0:

OUTPUT:
(i=1) overwrite-temp-u-stamp[s][ALL]←1 AND overwrite-temp-u-current[s][ALL]←1 AND
    overwrite-temp-v-dist[s][ALL][ALL]←1 AND overwrite-temp-v-π[s][ALL][ALL]←1 AND
    overwrite-temp-v-id[s][ALL][ALL]←1 AND overwrite-temp-v-gray[s][ALL]←1

OUTPUT:
(i>1) compare-last-stamp[s][i-1]←1 AND temp-u-stamp-slot[s][i]←1
```

```
INPUT:
compare-last-stamp[s][i]=1 AND temp-u-stamp-slot[s][i]=0 AND last-stamp[s][i]=0 AND to-
    accept-temp-u-stamp[s][i]=0 AND overwrite-temp-u-stamp[s][i]=0:

OUTPUT:
(i=1) overwrite-temp-u-stamp[s][ALL]←1 AND overwrite-temp-u-current[s][ALL]←1 AND
    overwrite-temp-v-dist[s][ALL][ALL]←1 AND overwrite-temp-v-π[s][ALL][ALL]←1 AND
    overwrite-temp-v-id[s][ALL][ALL]←1 AND overwrite-temp-v-gray[s][ALL]←1

OUTPUT:
(i>1) compare-last-stamp[s][i-1]←1
```

```
INPUT:
compare-last-stamp[s][i]=1 AND temp-u-stamp-slot[s][i]=1 AND last-stamp[s][i]=0 AND to-
    accept-temp-u-stamp[s][i]=0 AND overwrite-temp-u-stamp[s][i]=0:

OUTPUT:
new-message[s][i]←1 AND temp-u-stamp-slot[s][i]←1
```

```
INPUT:
compare-last-stamp[s][i]=1 AND temp-u-stamp-slot[s][i]=0 AND last-stamp[s][i] = 1 AND to
    -accept-temp-u-stamp[s][i]=0 AND overwrite-temp-u-stamp[s][i]=0:

OUTPUT:
old-message[s][i] ← 1
```

```
INPUT:
old-message[s][i]=1:

OUTPUT:
(i > 1) old-message[s][i-1] ← 1

OUTPUT:
```

```
(i = 1) overwrite-temp-u-stamp[s][ALL(for i)] ← 1 AND overwrite-temp-u-current[s][ALL(
    for i)] ← 1 AND overwrite-temp-v-dist[s][ALL (for d)][ALL (for i)] ← 1 AND
    overwrite-temp-v-π[s][ALL (for d)][ALL (for i)] ← 1 AND overwrite-temp-v-id[s][ALL(
    for d)][ALL(for i)] ← 1 AND overwrite-temp-v-gray[s][ALL (for d)] ← 1
```

```
INPUT:
new-message[s][i]=1:

OUTPUT:
(i > 1) new-message[s][i-1]←1

OUTPUT:
(i = 1) to-accept-temp-u-stamp[s][ALL (for i)]←1 AND to-accept-temp-u-current[s][ALL (
    for i)]←1 AND to-accept-temp-v-gray[s][ALL (for i)]←1 AND to-accept-temp-v-dist[s][
    ALL(for d)][ALL(for i)]←1 AND to-accept-temp-v-
π[s][ALL(for d)][ALL(for i)]←1 AND to-accept-temp-v-id[s][ALL(for d)][ALL(for i)]←1 AND
    return-message-flag[s]←1
```

```
INPUT:
compare-last-stamp[s][i]=0 AND temp-u-stamp-slot[s][i]=1 AND last-stamp[s][i]=1 AND to-
    accept-temp-u-stamp[s][i]=0 AND overwrite-temp-u-stamp[s][i]=0:

OUTPUT:
temp-u-stamp-slot[s][i]←1
```

```
INPUT:
compare-last-stamp[s][i]=0 AND temp-u-stamp-slot[s][i]=1 AND last-stamp[s][i]=0 AND to-
    accept-temp-u-stamp[s][i]=0 AND overwrite-temp-u-stamp[s][i]=0:

OUTPUT:
temp-u-stamp-slot[s][i]←1
```

```
INPUT:
compare-last-stamp[s][i]=0 AND temp-u-stamp-slot[s][i]=1 AND last-stamp[s][i]=0 AND to-
    accept-temp-u-stamp[s][i]=1 AND overwrite-temp-u-stamp[s][i]=0:

OUTPUT:
delay-new-stamp[s][i]←1
```

```
INPUT:
compare-last-stamp[s][i]=0 AND temp-u-stamp-slot[s][i]=1 AND last-stamp[s][i]=1 AND to-
    accept-temp-u-stamp[s][i]=1 AND overwrite-temp-u-stamp[s][i]=0:

OUTPUT:
delay-new-stamp[s][i]←1
```

**Take new messages from delays and put them into received messages as well as put them back in to placeholder**

Input blocks used in instruction definitions:

```
Index definition:
Let i ∈ [l], d ∈ [D], s ∈ [D² + 2].

Block definition:
  (delay-new-stamp[s][i], shutdown-stamp-delay[s][i])
  (delay-new-v-gray[s][d], shutdown-v-gray-delay[s][d])
  (delay-new-v-dist[s][d][i], shutdown-v-dist-delay[s][d][i])
  (delay-new-v-id[s][d][i], shutdown-v-id-delay[s][d][i])
  (delay-new-v-π[s][d][i], shutdown-v-π-delay[s][d][i])
  (delay-new-current[s][i], shutdown-current-delay[s][i])
  (delay-new-message-flag[s], shutdown-message-flag-delay[s])
```

Instructions:

```
INPUT:
delay-new-stamp[s][i]=1 AND shutdown-stamp-delay[s][i]=0:

OUTPUT:
(s< D² + 2) delay-new-stamp[s+1][i]←1

OUTPUT:
(s= D² + 2) received-stamp[i]←1 AND u-stamp-placeholder[ALL][i]←1 AND persistent-last-
    stamp[ALL][i]←1
```

```
INPUT:
delay-new-v-gray[s][d]=1 AND shutdown-v-gray-delay[s][d]=0:

OUTPUT:
(s< D² + 2) delay-new-v-gray[s+1][d]←1

OUTPUT:
(s= D² + 2) received-v-gray[d]←1 AND v-gray-placeholder[ALL][d]←1
```

```
INPUT:
delay-new-v-dist[s][d][i]=1 AND shutdown-v-dist-delay[s][d][i]=0:

OUTPUT:
(s< D² + 2) delay-new-v-dist[s+1][d][i]←1

OUTPUT:
(s= D² + 2) received-v-dist[d][i]←1 AND v-dist-placeholder[ALL][d][i]←1
```

```
INPUT:
delay-new-v-id[s][d][i]=1 AND shutdown-v-id-delay[s][d][i]=0:

OUTPUT:
(s< D² + 2) delay-new-v-id[s+1][d][i]←1

OUTPUT:
(s= D² + 2) received-v-id[d][i]←1 AND v-id-placeholder[ALL][d][i]←1
```

```
INPUT:
delay-new-v-π[s][d][i]=1 AND shutdown-v-π-delay[s][d][i]=0:

OUTPUT:
(s< D² + 2) delay-new-v-π[s+1][d][i]←1

OUTPUT:
(s= D² + 2) received-v-π[d][i]←1 AND v-π-placeholder[ALL][d][i]←1
```

```
INPUT:
delay-new-current[s][i]=1 AND shutdown-current-delay[s][i]=0:

OUTPUT:
(s< D² + 2) delay-new-current[s+1][i]←1

OUTPUT:
(s= D² + 2) received-current[i]←1 AND u-current-placeholder[ALL][i]←1
```

```
INPUT:
delay-new-message-flag[s]=1 AND shutdown-message-flag-delay[s]=0:

OUTPUT:
```

```
(s< D² + 1) delay-new-message-flag[s+1]←1
```

```
OUTPUT:
(s= D² + 1) delay-new-message-flag[s+1]←1 AND overwrite-last-stamp[ALL][ALL]←1 AND
    shutdown-message-flag-delay[for all s< D² + 2]←1 AND shutdown-stamp-delay[for all s
    < D² + 2][ALL]←1 AND shutdown-v-gray-delay[for all s< D² + 2][ALL]←1 AND shutdown-v-
    dist-delay[for all s< D² + 2][ALL][ALL]←1 AND shutdown-v-id-delay[for all s< D² + 2][
    ALL][ALL]←1 AND shutdown-v-π-delay[for all s< D² + 2][ALL][ALL]←1 AND shutdown-
    current-delay[for all s< D² + 2][ALL]←1
OUTPUT:
(s= D² + 2) message-flag-placeholder[ALL]←1 AND received-new-message←1 AND check-u-id-
    with-received-v-ids [ALL][ALL]←1 AND u-id-comparison-counter[ALL][1]←1
```

**Check local IDs with IDs in the message and set update flags for the matching ones**

Input blocks used in instruction definitions:

```
Index definition:
Let i ∈ [l],  j ∈ [l + 1],  t ∈ [D + l + 2].
Let k ∈ [D],  d ∈ [D].

Block definition:
  (received-new-message)
  (received-v-ids[k][i], u-id2check-with-received-v-ids[k][i], check-u-id-with-received-
    v-ids[k][i])
  (received-v-ids2check-with-v-ids[d][k][i], v-ids2check-with-received-v-ids[d][k][i],
    check-v-ids-with-received-v-ids[d][k][i])
  (u-equals-received-v-id[k][i], u-is-received-v-id-counter[k][i], reset-u-equals-v[k][i
    ])
  (v-equals-received-v-id[d][k][i], v-is-received-v-id-counter[d][k][i], reset-v-equals-
    v[d][k][i])
  (u-id-comparison-counter[k][j])
  (v-id-comparison-counter[d][k][j])
  (update-current-counter[t])
```

The instructions:

```
INPUT:
received-new-message=1:

OUTPUT:
check-v-ids-with-received-v-ids[ALL][ALL][ALL]←1 AND update-current-counter[1] AND v-id-
    comparison-counter[ALL][ALL][1]
```

```
INPUT:
received-v-id[k][i]=1 AND u-id2check-with-received-v-ids[k][i]=1 AND check-u-id-with-
    received-v-ids[k][i]=1:

OUTPUT:
(i =1) u-equals-received-v-id[k][i]←1 AND, received-v-ids2check-with-v-ids[ALL][k][i]←1
     AND u-is-received-v-id-counter[k][i]←1

OUTPUT:
(i > 1) u-equals-received-v-id[k][i]←1 AND, received-v-ids2check-with-v-ids[ALL][k][i]←
    1
```

```
INPUT:
received-v-id[k][i]=0 AND u-id2check-with-received-v-ids[k][i]=0 AND check-u-id-with-
    received-v-ids[k][i]=1:

OUTPUT:
(i =1) u-equals-received-v-id[k][i]←1 AND u-is-received-v-id-counter[k][i]←1
```

```
OUTPUT:
(i > 1) u-equals-received-v-id[k][i]←1
```

```
INPUT:
received-v-id[k][i]=1 AND u-id2check-with-received-v-ids[k][i]=0 AND check-u-id-with-
    received-v-ids[k][i]=1:

OUTPUT:
received-v-ids2check-with-v-ids[ALL][k][i]←1
```

```
INPUT:
received-v-ids2check-with-v-ids[d][k][i]=1 AND v-ids2check-with-received-v-ids[d][k][i
    ]=1 AND check-v-ids-with-received-v-ids[d][k][i]=1:

OUTPUT:
(i =1) v-equals-received-v-id[d][k][i]←1 AND v-is-received-v-id-counter[d][k][i]←1

OUTPUT:
(i > 1) v-equals-received-v-id[d][k][i]←1
```

```
INPUT:
received-v-ids2check-with-v-ids[d][k][i]=0 AND v-ids2check-with-received-v-ids[d][k][i
    ]=0 AND check-v-ids-with-received-v-ids[d][k][i]=1:

OUTPUT:
(i =1) v-equals-received-v-id[d][k][i]←1 AND v-is-received-v-id-counter[d][k][i]←1

OUTPUT:
(i > 1) v-equals-received-v-id[d][k][i]←1
```

```
INPUT:
u-id-comparison-counter[k][j]=1

OUTPUT:
(j < l + 1) u-id-comparison-counter[k][j+1]←1

OUTPUT:
(j = l + 1) reset-u-equals-v[k][ALL]←1
```

```
INPUT:
u-equals-received-v-id[k][i]=1 AND u-is-received-v-id-counter[k][i]=1 AND reset-u-equals
    -v[k][i]=0:

OUTPUT:
(i < l) u-is-received-v-id-counter[k][i+1]←1

OUTPUT:
(i = l) u-dist-update[k][ALL]←1 AND u-π-update[k][ALL]←1 AND u-gray-update[k]←1
```

```
INPUT:
u-equals-received-v-id[k][i]=1 AND u-is-received-v-id-counter[k][i]=0 AND reset-u-equals
    -v[k][i]=0:

OUTPUT:
u-equals-received-v-id[k][i]←1
```

```
INPUT:
v-id-comparison-counter[d][k][j]=1

OUTPUT:
(j < l + 1) v-id-comparison-counter[d][k][j+1]←1
```

```
OUTPUT:
(j = l + 1) reset-v-equals-v[d][k][ALL]←1
```

```
INPUT:
v-equals-received-v-id[d][k][i]=1 AND v-is-received-v-id-counter[d][k][i]=1 AND reset-v-
    equals-v[d][k][i]=0:

OUTPUT:
(i < l) v-is-received-v-id-counter[d][k][i+1]←1

OUTPUT:
(i = l) v-dist-update[d][k][ALL]←1 AND v-π-update[d][k][ALL]←1 AND v-gray-update[d][k]←
    1
```

```
INPUT:
v-equals-received-v-id[d][k][i]=1 AND v-is-received-v-id-counter[d][k][i]=0 AND reset-v-
    equals-v[d][k][i]=0:

OUTPUT:
v-equals-received-v-id[d][k][i]←1
```

```
INPUT:
update-current-counter[j]=1:

OUTPUT:
(j < l + D + 1) update-current-counter[j+1]←1

OUTPUT:
(j = l + D + 1) update-current[ALL]←1 AND update-current-counter[j+1]←1

OUTPUT:
(j = l + D + 2) compare-current[ALL]←1 AND comparison-counter[1 (=i, i ∈ [l + 1])]←1 AND
    renew-received-v-dist[ALL][ALL]←1 AND renew-received-v-π[ALL][ALL]←1 AND renew-
    received-v-gray[ALL]←1 AND update-v-dist-augend[ALL]←1
```

**Update the current node's distance, parent, white and gray flag**

Input blocks used in instruction definitions:

```
Index definition:
Let i ∈ [l], k ∈ [D].

Block definition:
  (received-v-dist[k][i], renew-received-v-dist[k][i])
  (u-dist-update[k][i], received-v-dist4u[k][i])
  (u-dist-delay[k][i])
  (received-v-π[k][i], renew-received-v-π[k][i])
  (u-π-update[k][i], received-v-π4u[k][i])
  (u-π-delay[k][i])
  (received-v-gray[k], new-received-v-gray[k])
  (u-gray-update[k], received-v-grays4u[k])
  (u-gray-delay[k])
  (received-stamp[i], update-stamp[i])
  (received-current[i], update-current[i])
```

The instructions:

```
INPUT:
received-v-dist[k][i]=1 AND renew-received-v-dist[k][i]=0:

OUTPUT:
received-v-dist[k][i]←1 AND received-v-dist4u[k][i]←1 , received-v-dist4v[ALL][k][i]←1
```

```
INPUT:
u-dist-update[k][i]=1 AND received-v-dist4u[k][i]=1:

OUTPUT:
u-dist-delay[k][i]←1
```

```
INPUT:
u-dist-delay[k][i]=1:

OUTPUT:
(k < D) u-dist-delay[k+1][i]←1

OUTPUT:
(k = D) u-dist[i]←1
```

```
INPUT:
received-v-π[k][i]=1 AND renew-received-v-π[k][i]=0:

OUTPUT:
received-v-π[k][i]←1 AND received-v-π4u[k][i]←1 AND received-v-π4v[ALL][k][i]←1
```

```
INPUT:
u-π-update[k][i]=1 AND received-v-π4u[k][i]=1:

OUTPUT:
u-π-delay[k][i]←1
```

```
INPUT:
u-π-delay[k][i]=1:

OUTPUT:
(k< D) u-π-delay[k+1][i]←1

OUTPUT:
(k = D) u-π[i]←1
```

```
INPUT:
received-v-gray[k]=1 AND renew-received-v-gray[k]=0:

OUTPUT:
received-v-gray[k]←1 AND received-v-gray4u[k]←1 , received-v-gray4v[ALL][k]←1
```

```
INPUT:
u-gray-update[k]=1 AND received-v-gray4u[k]=1:

OUTPUT:
u-gray-delay[k]←1
```

```
INPUT:
u-gray-delay[k]=1:

OUTPUT:
(k< D) u-gray-delay[k+1]←1

OUTPUT:
(k = D) u-gray←1 AND u-white-overwrite←1
```

```
INPUT:
received-stamp[i]=1 AND update-stamp[i]=0:
```

```
OUTPUT:
received-stamp[i]←1
```

```
INPUT:
received-stamp[i]=1 AND update-stamp[i]=1:

OUTPUT:
stamp[i]←1
```

```
INPUT:
received-current[i]=1 AND update-current[i]=0:

OUTPUT:
received-current[i]←1
```

```
INPUT:
received-current[i]=1 AND update-current[i]=1:

OUTPUT:
current[i]←1
```

### Update the adjacent nodes' distance, parent, and white and gray flags

Input blocks used in instruction definitions:

```
Index definition:
Let i ∈ [l], k ∈ [D].
Let d ∈ [D]

Block definition:
  (v-dist-update[d][k][i]=1, received-dist4v[d][k][i])
  (v-π-update[d][k][i]=1, received-πs4v[d][k][i])
  (v-gray-update[d][k]=1, received-gray4v[d][k])
  (v-dist-delay[d][k][i])
  (v-π-delay[d][k][i])
  (v-gray-delay[d][k])
```

Instructions:

```
INPUT:
v-dist-update[d][k][i]=1 AND received-dist4v[d][k][i]=1:

OUTPUT:
v-dist-delay[d][k][i]←1
```

```
INPUT:
v-dist-delay[d][k][i]=1:

OUTPUT:
(k< D) v-dist-delay[d][k+1][i]←1

OUTPUT:
(k = D) v-dist[d][i]←1
```

```
INPUT:
v-π-update[d][k][i]=1 AND received-πs4v[d][k][i]=1:

OUTPUT:
v-π-delay[d][k][i]←1
```

```
INPUT:
v-π-delay[d][k][i]=1:

OUTPUT:
(k < D) v-π-delay[d][k+1][i]←1

OUTPUT:
(k = D) v-π[d][i]←1
```

```
INPUT:
v-gray-update[d][k]=1 AND received-gray4v[d][k]=1:

OUTPUT:
v-gray-delay[d][k]←1
```

```
INPUT:
v-gray-delay[d][k]=1:

OUTPUT:
(k < D) v-gray-delay[d][k+1]←1

OUTPUT:
(k = D) v-gray[d]←1 AND v-white-overwrite[d]←1
```

### E.4.2. THE SPECIFICATIONS OF THE TRAINING DATA

Based on the defined binary variables, one can calculate the dimension of the binary vector before input encoding, denoted by $k$. By simply counting the number of bits, we obtain

$$k = 24D^3l + 8D^3 + 35D^2l + 11D^2 + 63Dl + 25D + 52l + 19.$$

This value is the dimension of the output vector, as the output is not in encoded form. The variables in the message section, $\boldsymbol{x}_M$, are u-stamp-slot, v-id-slot, v-dist-slot, v-pi-slot, v-gray-slot, u-current-slot, and message-flag-slot, with a total of

$$2lD^2 + 2l + 3lD^3 + 3lD + D^3 + D + D^2 + 1$$

bits. Thus, $k_M$ in Equation (3) is equal to this value.

We obtain the total number of instructions, denoted by $n_T$, by simply counting the number of instructions in each box (each box represents the instructions for the entire range of the indexing variables). This yields:

$$n_T = 18D^3l + 6D^3 + 31D^2l + 9D^2 + 49Dl + 20D + 51l + 14.$$

Since each instruction is converted into a training sample, the number of training samples and the input embedding dimension before augmentation are both equal to $n_T$.

### E.4.3. MAXIMUM NUMBER OF COLLISIONS

The maximum number of collisions in the instructions is 5 and occurs for the bits in temp-u-stamp-slot. Since the number of collisions is constant, the size of the training dataset after augmentation with idle samples as well as the input embedding dimension, denoted by $k'$, are proportional to $n_T$. Thus, similar to $n_T$, we have $k' = \mathcal{O}(l \cdot D^3)$.

### E.4.4. INITIALIZATION

To run the algorithm, certain variables must be properly initialized at each node. Let the local ID of a node $u$ be denoted by $u_{\mathcal{L}}$, and its global ID by $u_{\mathcal{G}}$. We denote the source node by $s$, while all other nodes are denoted by $u$. The neighbors of a node are denoted by $v_i$ for $i \in [D]$. The notation $\mathrm{Bin}(z, l)$ denotes the binary representation of the decimal number $z$ using $l$ bits.

Below, we first describe the initialization of variables for the source node $s$, and then for any other node $u$. Note that all variables that are not explicitly mentioned are initialized to zero.

```
Source Intitialization
current[1] ← 1
comparison-counter[1] ← 1
compare-current[ALL] ← 1
v-turn[1] ← 1
u-id ← Bin(s_G, l)
u-gray ← 1
determine-slot[s_L] ← 1
v-id[i] ← Bin(v_{i_G}, l) for i ∈ [D]
v-white[i] ← 1 for i ∈ [D]
```

```
Non-Source Intitialization
current[1] ← 1
comparison-counter[1] ← 1
compare-current[ALL] ← 1
v-turn[1] ← 1
u-id ← Bin(u_G, l)
u-white ← 1
determine-slot[u_L] ← 1
v-id[i] ← Bin(v_{i_G}, l) for i ∈ [D]
v-white[i] ← 1 for i ∈ [D] if v_i ≠ s
v-gray[i] ← 1 for i ∈ [D] if v_i = s
```

### E.4.5. SUFFICIENT UPPER BOUND ON THE NUMBER OF ITERATIONS

Here, we provide a sufficient upper bound on the number of iterations required by the graph template matching framework, under the instructions described above, to complete the algorithm. From the time a node receives a message indicating that it should be visited (or processed) until the time it sends a message to one of its neighbors indicating that it is that neighbor's turn to be processed next, the algorithm takes at most $l + D^2 + 3D + 12$ iterations. For brevity, we omit the detailed derivation of this value, but the approach follows a similar line of reasoning as in our analysis of BFS iterations. When backtracking occurs, in the worst case, the node goes through the same sequence of iterations to notify its next neighbor to be visited.

Each node has at most $D$ neighbors; thus, this sequence of iterations can repeat at most $D$ times per node. Since our goal is to derive a sufficient upper bound, we assume that all nodes undergo the dive-and-return process with the same worst-case number of iterations before the algorithm completes. Therefore, a sufficient upper bound on the total number of iterations is:

$$nD(5l + D^2 + 3D + 12).$$

### E.5. Bellman-Ford

In this section, we provide instructions that define the function $f$ in our graph template matching framework for executing the Bellman-Ford algorithm on graphs with maximum degree $D$, assuming all edge weights are non-negative. We use $l$ bits to represent all variables in our implementation, including the global node identifiers. Consequently, at test time, the maximum number of nodes in a graph is bounded by $2^l - 1$. Moreover, the length of any path in the graph cannot exceed $2^l - 1$. Algorithm 7 presents the pseudo-code of the implemented Bellman-Ford algorithm.

### E.5.1. INSTRUCTIONS OF BELLMAN-FORD

In this section, we present the instructions for implementing Bellman-Ford in our framework. For the sake of brevity, we do not explain the purpose of each variable. However, we have chosen the variable names to be as self-explanatory as possible. We divide the instructions into several sections. Each section includes instructions that together form a meaningful algorithmic subroutine. Each section starts by introducing the blocks for which the instructions in that section are defined, followed by the instructions themselves. Similarly, for brevity, we avoid providing explanations for the instructions. Nonetheless, the explanations given for BFS can help to better understand the working dynamics here as well.

**Creating required copies of the value of $|V|$**

**Algorithm 7** Bellman-Ford$(G, s)$

1: **for each** vertex $u \in G.V$ **do**
2:     $u.dist \leftarrow \infty$
3:     $u.\pi \leftarrow$ NIL
4: **end for**
5: $s.dist \leftarrow 0$
6: $s.\pi \leftarrow 0$
7: $round \leftarrow 1$
8: $pointer \leftarrow 1$
9: **while** round $< |V|$ **do**
10:     $u \leftarrow$ vertex $v \in G.V$ such that $v.id = pointer$
11:     **for** each $v \in G.Adj[u]$ **do**
12:         **if** $u.dist > v.dist + w(u, v)$ **then**
13:             $u.dist \leftarrow v.dist + w(u, v)$
14:             $u.\pi \leftarrow v.id$
15:         **end if**
16:     **end for**
17:     **if** $pointer = |V|$ **then**
18:         $pointer \leftarrow 1$
19:         $round \leftarrow round + 1$
20:     **else**
21:         $pointer \leftarrow pointer + 1$
22:     **end if**
23: **end while**

Input blocks used in instruction definitions:

```
Index definition:
Let i∈ [l]

Block definition:
   (num-nodes[i])
```

Instructions:

```
INPUT:
num-nodes[i]=1

OUTPUT:
num-nodes←1 AND n4round←1 AND n4pointer←1
```

### Check if the round is smaller than $|V|$

Input blocks used in instruction definitions:

```
Index definition:
Let i∈ [l]

Block definition:
   (compare-round[i], n4round[i], round[i])
   (round-smaller-than-num[i])
```

Instructions:

```
INPUT:
compare-round[i]=1 AND n4round[i]=1 AND round[i]=1

OUTPUT:
```

```
(i>1) compare-round[i-1]←1
```

```
INPUT:
compare-round[i]=1 AND num-nodes[i]=0 AND round[i]=0

OUTPUT:
(i>1) compare-round[i-1]←1
```

```
INPUT:
compare-round[i]=1 AND num-nodes[i]=1 AND round[i]=0

OUTPUT:
round-smaller-than-num[i]←1
```

```
INPUT:
round-smaller-than-num[i]=1

OUTPUT:
(i > 1) round-smaller-than-num[i-1]←1

OUTPUT:
(i = 1) update-pointer[ALL]←1  AND u-id-comparison-counter[1]←1
```

### Check if ID matches the pointer

Input blocks used in instruction definitions:

```
Index definition:
Let i∈ [l],  j∈ [l + 2]

Block definition:
   (u-id[i], pointer[i], compare-u-id[i], upload-u-id[i])
   (u-matches-pointer[i], u-matches-counter[i], reset-u-matches[i])
   (u-id-comparison-counter[j])
```

Instructions:

```
INPUT:
u-id[i]=1 AND pointer[i]=1 AND compare-u-id[i]=1 AND upload-u-id[i]=0

OUTPUT:
(i=1) u-matches-pointer[i]←1 AND u-matches-counter[i]←1 AND u-id[i]←1 AND u-pointer[i]
    ←1 AND pointer-augend[i]←1

OUTPUT:
(i>1) u-matches-pointer[i]←1 AND u-id[i]←1 AND u-pointer[i]←1 AND pointer-augend[i]←1
```

```
INPUT:
u-id[i]=0 AND pointer[i]=0 AND compare-u-id[i]=1 AND upload-u-id[i]=0

OUTPUT:
(i=1) u-matches-pointer[i]←1 AND u-matches-counter[i]←1

OUTPUT:
(i>1) u-matches-pointer[i]←1
```

```
INPUT:
u-id[i]=1 AND pointer[i]=0 AND compare-u-id[i]=1 AND upload-u-id[i]=0

OUTPUT:
u-id[i]←1 AND u-pointer[i]←1 AND pointer-augend[i]←1
```

```
INPUT:
u-id[i]=1 AND pointer[i]=0 AND compare-u-id[i]=0 AND upload-u-id[i]=0

OUTPUT:
u-id[i]←1 AND u-pointer[i]←1
```

```
INPUT:
u-id[i]=1 AND pointer[i]=0 AND compare-u-id[i]=0 AND upload-u-id[i]=1

OUTPUT:
u-id[i]←1 AND u-pointer[i]←1 AND u-id-placeholder[ALL][i]←1
```

```
INPUT:
u-id-comparison-counter[j]=1

OUTPUT:
(j < l+2) u-id-comparison-counter[j+1]←1

OUTPUT:
(j = l+2) reset-u-matches[ALL]←1
```

```
INPUT:
u-matches-pointer[i]=1 AND  u-matches-counter[i]=1 AND reset-u-matches[i]=0

OUTPUT:
(i< l) u-matches-counter[i+1]←1

OUTPUT:
(i= l) u-matched←1 AND check-pointer[ALL]←1 AND check-pointer-counter[1]←1 AND setup-
    round-augend[ALL]←1
```

```
INPUT:
u-matches-pointer[i]=1 AND  u-matches-counter[i]=0 AND reset-u-matches[i]=0

OUTPUT:
u-matches-pointer[i]←1
```

**Setting up the neighboring node distances to be aggregated with the weights**

Input blocks used in instruction definitions:

```
Index definition:
Let i∈ [l]
Let d ∈ [D]

Block definition:
   (v-dist[d][i], fill-candidate-dist-augend[d][i], overwrite-v-dist[d][i])
   (v-inf-backup[d][i], overwrite-v-inf[d][i])
```

Instructions:

```
INPUT:
v-dist[d][i]=1 AND fill-candidate-dist-augend[d][i]=1 AND overwrite-v-dist[d][i]=0

OUTPUT:
v-dist[d][i]←1 AND candidate-dist-augend[d][i]←1
```

```
INPUT:
v-dist[d][i]=1 AND fill-candidate-dist-augend[d][i]=0 AND overwrite-v-dist[d][i]=0

OUTPUT:
v-dist[d][i]←1
```

```
INPUT:
v-inf-backup[d][i]=1 AND overwrite-v-inf[d][i]=0

OUTPUT:
v-inf-backup[d][i]←1 AND v-inf[d][i]←1
```

## Setting up the edge weights to be aggregated with the neighboring distances

Input blocks used in instruction definitions:

```
Index definition:
Let i∈ [l]
Let d ∈ [D]

Block definition:
   (weight[d][i], fill-candidate-dist-addend[d][i], v-inf[d][i])
```

Instructions:

```
INPUT:
weight[d][i]=1 AND fill-candidate-dist-addend[d][i]=1 AND v-inf[d][i]=0

OUTPUT:
candidate-dist-addend[d][i]←1 AND weight[d][i]←1
```

```
INPUT:
weight[d][i]=1 AND fill-candidate-dist-addend[d][i]=0 AND v-inf[d][i]=0

OUTPUT:
weight[d][i]←1
```

```
INPUT:
weight[d][i]=1 AND fill-candidate-dist-addend[d][i]=1 AND v-inf[d][i]=1

OUTPUT:
weight[d][i]←1
```

```
INPUT:
weight[d][i]=1 AND fill-candidate-dist-addend[d][i]=0 AND v-inf[d][i]=1

OUTPUT:
weight[d][i]←1
```

## Setting up the node's current distances for comparison

Input blocks used in instruction definitions:

```
Index definition:
Let i∈ [l]
Let d∈ [D]

Block definition:
   (u-dist[i], fill-incumbent-dist[i], upload-u-dist[i])
   (backup-incumbent-dist[d][i], keep-incumbent-dist[d][i], ...
     reset-backup-incumbent-dist[d][i])
   (overwrite-u-inf-pipe[d][i])
   (u-inf-backup[i], overwrite-u-inf[i], upload-u-inf[i])
```

Instructions:

```
INPUT:
u-dist[i]=1 AND fill-incumbent-dist[i]=1 AND upload-u-dist[i]=0

OUTPUT:
incumbent-dist[1][i]←1 AND backup-incumbent-dist[1][i]←1
```

```
INPUT:
u-dist[i]=1 AND fill-incumbent-dist[i]=0 AND upload-u-dist[i]=0

OUTPUT:
u-dist[i]←1
```

```
INPUT:
u-dist[i]=1 AND fill-incumbent-dist[i]=0 AND upload-u-dist[i]=1

OUTPUT:
u-dist[i]←1 AND u-dist-placeholder[ALL][i]←1
```

```
INPUT:
backup-incumbent-dist[d][i]=1 AND keep-incumbent-dist[d][i]=0 AND reset-backup-incumbent
    -dist[d][i]=0

OUTPUT:
incumbent-dist[d][i]←1 AND backup-incumbent-dist[d][i]←1
```

```
INPUT:
backup-incumbent-dist[d][i]=1 AND keep-incumbent-dist[d][i]=1 AND reset-backup-incumbent
    -dist[d][i]=0

OUTPUT:
```
$(d < D)$ `backup-incumbent-dist[d+1][i]←1 AND incumbent-dist[d+1][i]←1`
```
OUTPUT:
```
$(d = D)$ `u-dist[d][i]←1`

```
INPUT:
overwrite-u-inf-pipe[d][i]=1

OUTPUT:
```
$(d < D)$ `overwrite-u-inf-pipe[d+1][i]←1`
```
OUTPUT:
```
$(d = D)$ `overwrite-u-inf[i]←1`

```
INPUT:
u-inf[i]=1 AND overwrite-u-inf[i]=0 AND upload-u-inf[i]=0

OUTPUT:
u-inf[i]←1
```

```
INPUT:
u-inf[i]=1 AND overwrite-u-inf[i]=0 AND upload-u-inf[i]=1

OUTPUT:
u-inf[i]←1 AND u-inf-placeholder[ALL][i]←1
```

### Set up the loop parameters

Input blocks used in instruction definitions:

```
Index definition:
Let j∈ [2l + 2]
Let d ∈ [D]

Block definition:
    (u-matched)
    (v-turn[d])
    (candidate-dist-sum-counter[j])
```

Instructions:

```
INPUT:
u-matched=1

OUTPUT:
v-turn[1 ]←1
```

```
INPUT:
v-turn[d]=1

OUTPUT:
(d =1) fill-candidate-dist-augend[ALL][ALL]←1 AND fill-candidate-dist-addend[ALL][ALL]←
    1 AND fill-incumbent-dist[ALL]←1  AND candidate-dist-sum-counter[1]←1

OUTPUT:
(1< d <D+1) compare-u-dist[d][l]←1

OUTPUT:
(d = D + 1) upload-next-pointer[ALL]←1 AND upload-next-round[ALL]←1 AND upload-u-id[ALL]
    ←1 AND upload-u-dist[ALL]←1 AND upload-u-inf[ALL]←1 AND reset-backup-incumbent-dist
    [ALL][ALL]←1 AND reset-backup-candidate-dist[ALL][ALL]←1  AND upload-message-flag←1
     AND overwrite-last-pointer[ALL][ALL]←1 AND overwrite-last-round[ALL][ALL]←1
```

```
INPUT:
candidate-dist-sum-counter[j]=1

OUTPUT:
(j< 2l + 1) candidate-dist-sum-counter[j+1]←1

OUTPUT:
(j= 2l + 1) candidate-dist-sum-counter[j+1]←1 AND to-copy-candidate-dist-augend[ALL][ALL]
    ←1

OUTPUT:
(j= 2l + 2) compare-u-dist[1 (=d, d ∈ [D])][l (=i, i ∈ [l])]←1
```

**Addition of the neighbors' distance with the edge weights**

Input blocks used in instruction definitions:

```
Index definition:
Let i∈ [l]
Let d ∈ [D]

Block definition:
  (candidate-dist-augend[d][i], candidate-dist-addend[d][i], to-copy-candidate-dist-
    augend[d][i])
  (candidate-dist-carry[d][i])
```

Instructions:

```
INPUT:
```

```
candidate-dist-augend[d][i]=1 AND candidate-dist-addend[d][i]=0 AND to-copy-candidate-
    dist-augend[d][i]=0
```

**OUTPUT**:
```
candidate-dist-augend[d][i]←1
```

---

**INPUT**:
```
candidate-dist-augend[d][i]=1 AND candidate-dist-addend[d][i]=1 AND to-copy-candidate-
    dist-augend[d][i]=0
```

**OUTPUT**:
```
candidate-dist-carry[d][i]←1
```

---

**INPUT**:
```
candidate-dist-augend[d][i]=0 AND candidate-dist-addend[d][i]=1 AND to-copy-candidate-
    dist-augend[d][i]=0
```

**OUTPUT**:
```
candidate-dist-augend←1
```

---

**INPUT**:
```
candidate-dist-carry[d][i]=1
```

**OUTPUT**:
```
(i< l) candidate-dist-addend[d][i+1]←1
```

**OUTPUT**:
```
(i= l) candidate-dist-carry[d][i]←1
```

---

**INPUT**:
```
candidate-dist-augend[d][i]=1 AND candidate-dist-addend[d][i]=0 AND to-copy-candidate-
    dist-augend[d][i]=1
```

**OUTPUT**:
```
candidate-dist[d][i]←1 AND backup-candidate-dist[d][i]←1
```

### Set up the next incumbent distance with candidate distance if the latter is smaller

Input blocks used in instruction definitions:

**Index definition**:
**Let** i∈ $[l]$
**Let** $d \in [D]$

**Block definition**:
```
  (backup-candidate-dist[d][i], choose-candidate-dist[d][i], reset-backup-candidate-dist
    [d][i])
```

Instructions:

**INPUT**:
```
backup-candidate-dist[d][i]=1 AND choose-candidate-dist[d][i]=0 AND reset-backup-
    candidate-dist[d][i]=0
```

**OUTPUT**:
```
candidate-dist[d][i]←1 AND backup-candidate-dist[d][i]←1
```

---

**INPUT**:
```
backup-candidate-dist[d][i]=1 AND choose-candidate-dist[d][i]=1 AND reset-backup-
    candidate-dist[d][i]=0
```

```
OUTPUT:
(d < D) backup-incumbent-dist[d+1][i]←1 AND incumbent-dist[d+1][i]←1

OUTPUT:
(d = D) u-dist[i]←1
```

**Check if the nodes current distance is larger than candidate distances**

Input blocks used in instruction definitions:

```
Index definition:
Let i∈ [l]
Let d ∈ [D]

Block definition:
   (compare-u-dist[d][i], incumbent-dist[d][i], candidate-dist[d][i])
   (smaller-candidate-dist[d][i])
```

Instructions:

```
INPUT:
compare-u-dist[d][i]=1 AND incumbent-dist[d][i]=1 AND candidate-dist[d][i]=1

OUTPUT:
(i=1) keep-incumbent-dist[d][ALL]←1 AND v-turn[d+1]←1

OUTPUT:
(i>1) compare-u-dist[d][i-1]←1
```

```
INPUT:
compare-u-dist[d][i]=1 AND incumbent-dist[d][i]=0 AND candidate-dist[d][i]=1

OUTPUT:
smaller-incumbent-dist[d][i]←1
```

```
INPUT:
compare-u-dist[d][i]=1 AND incumbent-dist[d][i]=0 AND candidate-dist[d][i]=0

OUTPUT:
(i=1) keep-incumbent-dist[d][ALL]←1 AND v-turn[d+1]←1

OUTPUT:
(i>1) compare-u-dist[d][i-1]←1
```

```
INPUT:
compare-u-dist[d][i]=1 AND incumbent-dist[d][i]=1 AND candidate-dist[d][i]=0

OUTPUT:
smaller-candidate-dist[d][i]←1
```

```
INPUT:
smaller-candidate-dist[d][i]=1

OUTPUT:
(i > 1) smaller-candidate-dist[d][i-1]←1

OUTPUT:
(i = 1 ) choose-candidate-dist[d][ALL]←1 AND v-turn[d+1]←1 AND overwrite-u-inf-pipe[d][
    ALL]←1
```

```
INPUT:
smaller-incumbent-dist[d][i]=1

OUTPUT:
(i > 1) smaller-incumbent-dist[d][i-1]←1

OUTPUT:
(i = 1 ) keep-incumbent-dist[d][ALL]←1 AND v-turn[d+1]←1
```

### Check if the pointer has reached the last node

TInput blocks used in instruction definitions:

```
Index definition:
Let i∈ [l],  j∈ [l + 1],  t∈ [2l]

Block definition:
   (u-pointer[i], n4pointer[i], check-pointer[i])
   (pointer-matches-n[i], pointer-matches-n-counter[i])
   (check-pointer-counter[j])
   (decide-pointer, reset-pointer)
   (pointer-sum-counter[t])
   (round-sum-counter[t])
```

Instructions:

```
INPUT:
u-pointer[i]=1 AND n4pointer[i]=1 AND check-pointer[i]=1

OUTPUT:
(i=1) pointer-matches-n[i]←1 AND pointer-matches-n-counter[i]←1

OUTPUT:
(i>1) pointer-matches-n[i]←1
```

```
INPUT:
u-pointer[i]=0 AND n4pointer[i]=0 AND check-pointer[i]=1

OUTPUT:
(i=1) pointer-matches-n[i]←1 AND pointer-matches-n-counter[i]←1

OUTPUT:
(i>1) pointer-matches-n[i]←1
```

```
INPUT:
pointer-matches-n[i]=1 AND  pointer-matches-n-counter[i]=1

OUTPUT:
(i< l) pointer-matches-n-counter[i+1]←1

OUTPUT:
(i= l) reset-pointer←1
```

```
INPUT:
pointer-matches-n[i]=1 AND  pointer-matches-n-counter[i]=0

OUTPUT:
pointer-matches-n[i]←1
```

```
INPUT:
check-pointer-counter[j]=1
```

```
OUTPUT:
(j < l + 1)  check-pointer-counter[j+1]←1
```

```
OUTPUT:
(j = l + 1)  decide-pointer←1  %reset-pointer-matches-last[ALL]←1
```

```
INPUT:
decide-pointer =1 AND reset-pointer=1
```

```
OUTPUT:
next-pointer[1]←1 AND round-addend[1]←1 AND round-sum-counter[1]←1
```

```
INPUT:
decide-pointer =1 AND reset-pointer=0
```

```
OUTPUT:
pointer-addend[1]←1 AND pointer-sum-counter[1]←1 AND round-sum-counter[1]←1
```

```
INPUT:
pointer-sum-counter[t]=1
```

```
OUTPUT:
(t< 2l)  pointer-sum-counter[t+1]←1
```

```
OUTPUT:
(t= 2l)  to-copy-pointer-augend[ALL]←1
```

```
INPUT:
round-sum-counter[t]=1
```

```
OUTPUT:
(t< 2l)  round-sum-counter[t+1]←1
```

```
OUTPUT:
(t= 2l)  to-copy-round-augend[ALL]←1 AND reset-round[ALL]←1
```

**Incrementing the pointer if it has not reached the last node**

Input blocks used in instruction definitions:

```
Index definition:
Let i∈ [l]

Block definition:
   (pointer-augend[i], pointer-addend[i], to-copy-pointer-augend[i])
   (pointer-carry[i])
```

Instructions:

```
INPUT:
pointer-augend[i]=1 AND pointer-addend[i]=0 AND to-copy-pointer-augend[i]=0
```

```
OUTPUT:
pointer-augend[i]←1
```

```
INPUT:
pointer-augend[i]=1 AND pointer-addend[i]=1 AND to-copy-pointer-augend[i]=0
```

```
OUTPUT:
pointer-carry[i]←1
```

```
INPUT:
pointer-augend[i]=0 AND pointer-addend[i]=1 AND to-copy-pointer-augend[i]=0

OUTPUT:
pointer-augend[i]←1
```

```
INPUT:
pointer-carry[i]=1

OUTPUT:
(i<l) pointer-addend[i+1]←1

OUTPUT:
(i=l) pointer-carry[i]←1
```

```
INPUT:
pointer-augend[i]=1 AND pointer-addend[i]=0 AND to-copy-pointer-augend[i]=1

OUTPUT:
next-pointer[i]←1
```

**Incrementing the round if the pointer has reached the last node**

Input blocks used in instruction definitions:

```
Index definition:
Let i∈[l]

Block definition:
   (round-augend[i], round-addend[i], to-copy-round-augend[i])
   (round-carry[i])
```

Instructions:

```
INPUT:
round-augend[i]=1 AND round-addend[i]=0 AND to-copy-round-augend[i]=0

OUTPUT:
round-augend[i]←1
```

```
INPUT:
round-augend[i]=1 AND round-addend[i]=1 AND to-copy-round-augend[i]=0

OUTPUT:
round-carry[i]←1
```

```
INPUT:
round-augend[i]=0 AND round-addend[i]=1 AND to-copy-round-augend[i]=0

OUTPUT:
round-augend[i]←1
```

```
INPUT:
round-carry[i]=1

OUTPUT:
(i<l) round-addend[i+1]←1

OUTPUT:
(i=l) round-carry[i]←1
```

```
INPUT:
round-augend[i]=1 AND round-addend[i]=0 AND to-copy-round-augend[i]=1

OUTPUT:
next-round[i]←1
```

## Upload the next pointer and next round

The blocks required:

```
Index definition:
Let i∈ [l]

Block definition:
   (next-pointer[i], upload-next-pointer[i])
   (next-round[i], upload-next-round[i])
   (upload-message-flag)
```

Instructions:

```
INPUT:
next-pointer[i]=1 AND upload-next-pointer[i]=1

OUTPUT:
pointer-placeholder[ALL][i]←1 AND persistent-last-pointer[ALL][i]←1
```

```
INPUT:
next-pointer[i]=1 AND upload-next-pointer[i]=0

OUTPUT:
next-pointer[i]←1
```

```
INPUT:
next-round[i]=1 AND upload-next-round[i]=1

OUTPUT:
round-placeholder[ALL][i]←1 AND persistent-last-round[ALL][i]←1
```

```
INPUT:
next-round[i]=1 AND upload-next-round[i]=0

OUTPUT:
next-round[i]←1
```

```
INPUT:
upload-message-flag=1

OUTPUT:
message-flag-placeholder[ALL]←1
```

## Guide the placeholder content to the appropriate sending slot

Input blocks used in instruction definitions:

```
Index definition:
Let i∈ [l]
Let s∈ [D² + 1]

Block definition:
   (determine-slot[s])
   (determine-pointer-slot[s][i], pointer-placeholder[s][i])
```

```
    (determine-round-slot[s][i], round-placeholder[s][i])
    (determine-u-dist-slot[s][i], u-dist-placeholder[s][i])
    (determine-u-id-slot[s][i], u-id-placeholder[s][i])
    (determine-u-inf-slot[s][i], u-inf-placeholder[s][i])
    (determine-message-flag-slot[s], message-flag-placeholder[s])
```

Instructions:

**INPUT**:
determine-slot[s]=1

**OUTPUT**:
determine-slot[s]←1 **AND** determine-u-dist-slot[s][**ALL**(**for** i)]←1 **AND** determine-u-inf-slot
    [s][All(**for** i)] ← 1 **AND** determine-u-id-slot[s][**ALL**(**for** i)]←1 **AND** determine-round-
    slot[s][**ALL**(**for** i)]←1 **AND** determine-pointer-slot[s][**ALL**(**for** i)]←1 **AND** determine-
    message-flag-slot[s]←1

---

**INPUT**:
determine-pointer-slot[s][i]=1 **AND** pointer-placeholder[s][i]=1

**OUTPUT**:
pointer-slot[s][i]←1 **AND** determine-pointer-slot[s][i]←1

---

**INPUT**:
determine-pointer-slot[s][i]=1 **AND** pointer-placeholder[s][i]=0

**OUTPUT**:
determine-pointer-slot[s][i]←1

---

**INPUT**:
determine-round-slot[s][i]=1 **AND** round-placeholder[s][i]=1

**OUTPUT**:
round-slot[s][i]←1 **AND** determine-round-slot[s][i]←1 1

---

**INPUT**:
determine-round-slot[s][i]=1 **AND** round-placeholder[s][i]=0

**OUTPUT**:
determine-round-slot[s][i]←1

---

**INPUT**:
determine-u-dist-slot[s][i]=1 **AND** u-dist-placeholder[s][i]=1

**OUTPUT**:
u-dist-slot[s][i]←1 **AND** determine-u-dist-slot[s][i]←1 1

---

**INPUT**:
determine-u-dist-slot[s][i]=1 **AND** u-dist-placeholder[s][i]=0

**OUTPUT**:
determine-u-dist-slot[s][i]←1

---

**INPUT**:
determine-u-id-slot[s][i]=1 **AND** u-id-placeholder[s][i]=1

**OUTPUT**:
u-id-slot[s][i]←1 **AND** determine-u-id-slot[s][i]←1 1

```
INPUT:
determine-u-id-slot[s][i]=1 AND u-id-placeholder[s][i]=0

OUTPUT:
determine-u-id-slot[s][i]←1
```

```
INPUT:
determine-u-inf-slot[s][i]=1 AND u-inf-placeholder[s][i]=1

OUTPUT:
u-inf-slot[s][i]←1 AND determine-u-inf-slot[s][i]←1 1
```

```
INPUT:
determine-u-inf-slot[s][i]=1 AND u-inf-placeholder[s][i]=0

OUTPUT:
determine-u-inf-slot[s][i]←1
```

```
INPUT:
determine-message-flag-slot[s]=1 AND message-flag-placeholder[s]=1

OUTPUT:
message-flag-slot[s]←1 AND determine-message-flag-slot[s]←1
```

```
INPUT:
determine-message-flag-slot[s]=1 AND message-flag-placeholder[s]=0

OUTPUT:
determine-message-flag-slot[s]←1
```

**Copy incoming messages to temporary slots while checking if the round or pointer are new**

Input blocks used in instruction definitions:

```
Index definition:
Let i∈ [l]
Let s∈ [D² + 1]

Block definition:
  (pointer-slot[s][i])
  (round-slot[s][i])
  (u-id-slot[s][i])
  (u-dist-slot[s][i])
  (u-inf-slot[s][i])
  (message-flag-slot[s])
```

Instructions:

```
INPUT:
pointer-slot[s][i]=1

OUTPUT:
temp-pointer-slot[s][i]←1 AND incoming-pointer[s][i]←1
```

```
INPUT:
round-slot[s][i]=1

OUTPUT:
temp-round-slot[s][i]←1 AND incoming-round[s][i]←1
```

```
INPUT:
u-id-slot[s][i]=1

OUTPUT:
temp-u-id-slot[s][i]←1
```

```
INPUT:
u-dist-slot[s][i]=1

OUTPUT:
temp-u-dist-slot[s][i]←1
```

```
INPUT:
u-inf-slot[s][i]=1

OUTPUT:
temp-u-inf-slot[s][i]←1
```

```
INPUT:
message-flag-slot[s]=1

OUTPUT:
compare-last-round[s][l]←1
```

**Preserve the temporary data until the new message check is complete**

Input blocks used in instruction definitions:

```
Index definition:
Let i∈ [l]
Let s∈ [D² + 1]

Block definition:
  (persistent-last-pointer[s][i], overwrite-last-pointer[s][i])
  (persistent-last-round[s][i], overwrite-last-round[s][i])
  (temp-pointer-slot[s][i], to-accept-temp-pointer[s][i], overwrite-temp-pointer[s][i])
  (temp-round-slot[s][i], to-accept-temp-round[s][i], overwrite-temp-round[s][i])
  (temp-u-id-slot[s][i], to-accept-temp-u-id[s][i], overwrite-temp-u-id[s][i])
  (temp-u-dist-slot[s][i], to-accept-temp-u-dist[s][i], overwrite-temp-u-dist[s][i])
  (temp-u-inf-slot[s][i], to-accept-temp-u-inf[s][i], overwrite-temp-u-inf[s][i])
  (return-message-flag[s])
```

Instructions:

```
INPUT:
persistent-last-pointer[s][i]=1 AND overwrite-last-pointer[s][i]=0

OUTPUT:
last-pointer[s][i]←1 AND persistent-last-pointer[s][i]=1←1
```

```
INPUT:
persistent-last-pointer[s][i]=1 AND overwrite-last-pointer[s][i]=1

OUTPUT:
last-pointer[s][i]←1
```

```
INPUT:
persistent-last-round[s][i]=1 AND overwrite-last-round[s][i]=0

OUTPUT:
last-round[s][i]←1 AND persistent-last-round[s][i]=1←1
```

```
INPUT:
persistent-last-round[s][i]=1 AND overwrite-last-round[s][i]=1

OUTPUT:
last-round[s][i]←1
```

```
INPUT:
temp-pointer-slot[s][i]=1 AND to-accept-temp-pointer[s][i]=0 AND overwrite-temp-pointer[
    s][i]=0

OUTPUT:
temp-pointer-slot[s][i]←1 AND incoming-pointer[s][i]←1
```

```
INPUT:
temp-pointer-slot[s][i]=1 AND to-accept-temp-pointer[s][i]=1 AND overwrite-temp-pointer[
    s][i]=0

OUTPUT:
delay-new-pointer[s][i]←1
```

```
INPUT:
temp-round-slot[s][i]=1 AND to-accept-temp-round[s][i]=0 AND overwrite-temp-round[s][i
    ]=0

OUTPUT:
temp-round-slot[s][i]←1 AND incoming-round[s][i]←1
```

```
INPUT:
temp-round-slot[s][i]=1 AND to-accept-temp-round[s][i]=1 AND overwrite-temp-round[s][i
    ]=0

OUTPUT:
delay-new-round[s][i]←1
```

```
INPUT:
temp-u-id-slot[s][i]=1 AND to-accept-temp-u-id[s][i]=0 AND overwrite-temp-u-id[s][i]=0

OUTPUT:
temp-u-id-slot[s][i]←1
```

```
INPUT:
temp-u-id-slot[s][i]=1 AND to-accept-temp-u-id[s][i]=1 AND overwrite-temp-u-id[s][i]=0

OUTPUT:
delay-new-u-id[s][i]←1
```

```
INPUT:
temp-u-dist-slot[s][i]=1 AND to-accept-temp-u-dist[s][i]=0 AND overwrite-temp-u-dist[s][
    i]=0

OUTPUT:
temp-u-dist-slot[s][i]←1
```

```
INPUT:
temp-u-dist-slot[s][i]=1 AND to-accept-temp-u-dist[s][i]=1 AND overwrite-temp-u-dist[s][
    i]=0

OUTPUT:
delay-new-u-dist[s][i]←1
```

```
INPUT:
temp-u-inf-slot[s][i]=1 AND to-accept-temp-u-inf[s][i]=0 AND overwrite-temp-u-inf[s][i
    ]=0

OUTPUT:
temp-u-inf-slot[s][i]←1
```

```
INPUT:
temp-u-inf-slot[s][i]=1 AND to-accept-temp-u-inf[s][i]=1 AND overwrite-temp-u-inf[s][i
    ]=0

OUTPUT:
delay-new-u-inf[s][i]←1
```

```
INPUT:
return-message-flag[s]=1

OUTPUT:
delay-new-message-flag[s]←1
```

### Check if the round in the message is new

Input blocks used in instruction definitions:

```
Index definition:
Let i∈ [l]
Let s∈ [D² + 1]

Block definition:
    (compare-last-round[s][i], incoming-round[s][i], last-round[s][i])
    (new-message[s][i])
```

Instructions:

```
INPUT:
compare-last-round[s][i]=1 AND incoming-round[s][i]=1 AND last-round[s][i]=1

OUTPUT:
(i=1) compare-last-pointer[s][l]←1

OUTPUT:
(i>1) compare-lasts-round[s][i-1]←1
```

```
INPUT:
compare-last-round[s][i]=1 AND incoming-round[s][i]=0 AND last-round[s][i]=0

OUTPUT:
(i=1) compare-last-pointer[s][l]←1

OUTPUT:
(i>1) compare-last-round[s][i-1]←1
```

```
INPUT:
compare-last-round[s][i]=1 AND incoming-round[s][i]=1 AND last-round[s][i]=0

OUTPUT:
new-message[s][i]←1
```

```
INPUT:
compare-last-round[s][i]=1 AND incoming-round[s][i]=0 AND last-round[s][i]=1:
```

```
OUTPUT:
old-message[s][i] ← 1
```

```
INPUT:
new-message[s][i]=1

OUTPUT:
(i > 1) new-message[s][i-1]←1

OUTPUT:
(i = 1) to-accept-temp-pointer[s][ALL(for i)]←1 AND to-accept-temp-round[s][ALL(for i)]
    ←1 AND to-accept-temp-u-id[s][ALL(for i)]←1 AND to-accept-temp-u-dist[s][ALL(for i)
    ]←1 AND to-accept-temp-u-inf[s][ALL(for i)]←1 AND return-message-flag[s]←1
```

```
INPUT:
old-message[s][i]=1:

OUTPUT:
(i > 1) old-message[s][i-1] ← 1

OUTPUT:
(i = 1) overwrite-temp-pointer[s][ALL] ← 1 AND overwrite-temp-round[s][ALL] ← 1 AND
    overwrite-temp-u-id[s][ALL] ← 1 AND overwrite-temp-u-dist[s][ALL] ← 1 AND overwrite
    -temp-u-inf[s][ALL] ← 1
```

**In the case that the round is not new, check if the pointer in the message is new**

Input blocks used in instruction definitions:

```
Index definition:
Let i∈ [l]
Let s∈ [D² + 1]

Block definition:
   (compare-last-pointer[s][i], incoming-pointer[s][i], last-pointer[s][i])
```

Instructions:

```
INPUT:
compare-last-pointer[s][i]=1 AND incoming-pointer[s][i]=1 AND last-pointer[s][i]=1

OUTPUT:
(i=1) overwrite-temp-pointer[s][ALL]←1 AND overwrite-temp-round[s][ALL]←1 AND overwrite
    -temp-u-id[s][ALL]←1 AND overwrite-temp-u-dist[s][ALL]←1 AND overwrite-temp-u-inf[s
    ][ALL]←1

OUTPUT:
(i>1) compare-last-pointer[s][i-1]←1
```

```
INPUT:
compare-last-pointer[s][i]=1 AND incoming-pointer[s][i]=0 AND last-pointer[s][i]=0

OUTPUT:
(i=1) overwrite-temp-pointer[s][ALL]←1 AND overwrite-temp-round[s][ALL]←1 AND overwrite
    -temp-u-id[s][ALL]←1 AND overwrite-temp-u-dist[s][ALL]←1 AND overwrite-temp-u-inf[s
    ][ALL]←1

OUTPUT:
(i>1) compare-last-pointer[s][i-1]←1
```

```
INPUT:
compare-last-pointer[s][i]=1 AND incoming-pointer[s][i]=1 AND last-pointer[s][i]=0

OUTPUT:
new-message[s][i]←1
```

```
INPUT:
compare-last-pointer[s][i]=1 AND incoming-pointer[s][i]=0 AND last-pointer[s][i]=1:

OUTPUT:
old-message[s][i] ← 1
```

### Take new messages from delays and put them into received messages, and also put them back in to placeholder

Input blocks used in instruction definitions:

```
Index definition:
Let i∈ [l]
Let s∈ [D² + 1]

Block definition:
  (delay-new-round[s][i], shutdown-round-delay[s][i])
  (delay-new-pointer[s][i], shutdown-pointer-delay[s][i])
  (delay-new-u-id[s][i], shutdown-u-id-delay[s][i])
  (delay-new-u-dist[s][i], shutdown-u-dist-delay[s][i])
  (delay-new-u-inf[s][i], shutdown-u-inf-delay[s][i])
  (delay-new-message-flag[s], shutdown-message-flag-delay[s])
```

Instructions:

```
INPUT:
delay-new-round[s][i]=1 AND shutdown-round-delay[s][i]=0

OUTPUT:
(s< D² + 2) delay-new-round[s+1][i]←1

OUTPUT:
(s= D² + 2) received-round[i]←1 AND round-placeholder[ALL][i]←1 AND persistent-last-
    round[ALL][i]←1
```

```
INPUT:
delay-new-pointer[s][i]=1 AND shutdown-pointer-delay[s][i]=0

OUTPUT:
(s< D² + 2) delay-new-pointer[s+1][i]←1

OUTPUT:
(s= D² + 2) received-pointer[i]←1 AND pointer-placeholder[ALL][i]←1 AND persistent-last-
    pointer[ALL][i]←1
```

```
INPUT:
delay-new-u-id[s][i]=1 AND shutdown-u-id-delay[s][i]=0

OUTPUT:
(s< D² + 2) delay-new-u-id[s+1][i]←1

OUTPUT:
(s= D² + 2) received-u-id[ALL][i]←1 AND u-id-placeholder[ALL][i]←1
```

```
INPUT:
delay-new-u-dist[s][i]=1 AND shutdown-u-dist-delay[s][i]=0
```

```
OUTPUT:
(s< D² + 2) delay-new-u-dist[s+1][i]←1
```

```
OUTPUT:
(s= D² + 2) received-u-dist[ALL][i]←1 AND u-dist-placeholder[ALL][i]←1
```

```
INPUT:
delay-new-u-inf[s][i]=1 AND shutdown-u-inf-delay[s][i]=0
```

```
OUTPUT:
(s< D² + 1) delay-new-u-inf[s+1][i]←1
```

```
OUTPUT:
(s= D² + 1) received-u-inf[ALL][i]←1 AND u-inf-placeholder[ALL][i]←1
```

```
INPUT:
delay-new-message-flag[s]=1 AND shutdown-message-flag-delay[s]=0
```

```
OUTPUT:
(s< D² + 1) delay-new-message-flag[s+1]←1
```

```
OUTPUT:
(s= D² + 1) delay-new-message-flag[s+1] ← 1 AND overwrite-last-pointer[ALL][ALL] ← 1 AND
    overwrite-last-round[ALL][ALL] ← 1 AND shutdown-message-flag-delay[for all s< D² + 2
    ]← 1 AND shutdown-pointer-delay[for all s< D² + 2][ALL]← 1 AND shutdown-round-delay[
    for all s< D² + 2][ALL]← 1 AND shutdown-u-id-delay[for all s< D² + 2][ALL]← 1 AND
    shutdown-u-dist-delay[for all s< D² + 2][ALL]← 1 AND shutdown-u-inf-delay[for all s
    < D² + 2][ALL]← 1
```

```
OUTPUT:
(s= D² + 2) message-flag-placeholder[ALL] ← 1 AND received-new-message ← 1 AND check-v-
    id-with-received-u-id [ALL][ALL] ← 1 AND v-id-comparison-counter[ALL][1] ← 1
```

**Check local IDs with the ID in the message and set the update flags for the matching one**

Input blocks used in instruction definitions:

```
Index definition:
Let i∈ [l],  j∈ [l + 1]
Let d∈ [D]

Block definition:
   (v-id[d][i])
   (received-new-message)
   (received-u-id[d][i], v-id2check[d][i], check-v-id-with-received-u-id[d][i])
   (v-id-comparison-counter[d][j])
   (v-equals-received-u-id[d][i], v-is-received-u-id-counter[d][i], reset-v-equals-u[d][i
     ])
   (update-round-counter[j])
```

The instructions:

```
INPUT:
v-id[d][i]=1
```

```
OUTPUT:
v-id[d][i]←1 AND v-id2check[d][i]←1
```

```
INPUT:
received-new-message=1
```

```
OUTPUT:
update-round-counter[1]
```

```
INPUT:
received-u-id[d][i]=1 AND v-id2check[d][i]=1 AND check-v-id-with-received-u-id[d][i]=1

OUTPUT:
(i =1) v-equals-received-u-id[d][i]←1 AND v-is-received-u-id-counter[d][i]←1 AND v-id[d
    ][i]←1

OUTPUT:
(i > 1) v-equals-received-u-id[d][i]←1 AND v-id[d][i]←1
```

```
INPUT:
received-u-id[d][i]=0 AND v-id2check[d][i]=0 AND check-v-id-with-received-u-id[d][i]=1

OUTPUT:
(i =1) v-equals-received-u-id[d][i]←1 AND v-is-received-u-id-counter[d][i]←1

OUTPUT:
(i > 1) v-equals-received-u-id[d][i]←1
```

```
INPUT:
v-id-comparison-counter[d][j]=1

OUTPUT:
(j < l+1) v-id-comparison-counter[d][j+1]←1

OUTPUT:
(j = l+1) reset-v-equals-u[d][ALL]←1
```

```
INPUT:
v-equals-received-u-id[d][i]=1 AND v-is-received-u-id-counter[d][i]=1 AND reset-v-equals
    -u[d][i]=0

OUTPUT:
(i < l) v-is-received-u-id-counter[d][i+1]←1

OUTPUT:
(i = l) overwrite-v-inf[d][ALL]←1 AND overwrite-v-dist[d][ALL]←1 AND v-dist-update[d][
    ALL]←1 AND v-inf-update[d][ALL]←1
```

```
INPUT:
v-equals-received-u-id[d][i]=1 AND v-is-received-u-id-counter[d][i]=0 AND reset-v-equals
    -u[d][i]=0

OUTPUT:
v-equals-received-u-id[d][i]←1
```

```
INPUT:
update-round-counter[j]=1

OUTPUT:
(j < l) update-round-counter[j+1]←1

OUTPUT:
(j = l) update-round[ALL]←1 AND update-pointer-counter[j+1]←1 AND reset-round [ALL]←1

OUTPUT:
(j = l+1) compare-round[l]←1 AND renew-received-u-dist[ALL][ALL]←1 AND renew-received-u
    -inf[ALL][ALL]←1
```

**Update the distance information that the node has about its neighbors**

Input blocks used in instruction definitions:

```
Index definition:
Let i∈ [l]
Let d ∈ [D]

Block definition:
   (received-u-dist[d][i], v-dist-update[d][i], renew-received-u-dist[d][i])
   (received-u-inf[d][i], v-inf-update[d][i], renew-received-u-inf[d][i])
```

The instructions:

```
INPUT:
received-u-dist[d][i]=1 AND v-dist-update[d][i]=0 AND renew-received-u-dist[d][i]=0

OUTPUT:
received-u-dist[d][i]←1
```

```
INPUT:
received-u-dist[d][i]=1 AND v-dist-update[d][i]=1 AND renew-received-u-dist[d][i]=0

OUTPUT:
v-dist[d][i]←1
```

```
INPUT:
received-u-inf[d][i]=1 AND v-inf-update[d][i]=0 AND renew-received-u-inf[d][i]=0

OUTPUT:
received-u-inf[d][i]←1
```

```
INPUT:
received-u-inf[d][i]=1 AND v-inf-update[d][i]=1 AND renew-received-u-inf[d][i]=0

OUTPUT:
v-inf-backup[d][i]←1
```

**Put the incoming round in the comparison blocks and update**

Input blocks used in instruction definitions:

```
Index definition:
Let i∈ [l] be the index over the message bits.

Block definition:
   (received-round[i], update-round[i])
```

Instructions:

```
INPUT:
received-round[i]=1 AND update-round[i]=0

OUTPUT:
received-round[i]←1
```

```
INPUT:
received-round[i]=1 AND update-round[i]=1

OUTPUT:
backup-round[i]←1 AND round[i]←1
```

**Setting up the round to check if it is less than $|V|$ and to increment it if the pointer has reached the end value**

The blocks required:

```
Index definition:
Let i∈ [l] be the index over the message bits.

Block definition:
   (backup-round[i], setup-round-augend[i], reset-round[i])
```

Instructions:

```
INPUT:
backup-round[i]=1 AND setup-round-augend[i]=1 AND reset-round[i]=0

OUTPUT:
backup-round[i]←1 AND round[i]←1 AND round-augend[i]←1
```

```
INPUT:
backup-round[i]=1 AND setup-round-augend[i]=0 AND reset-round[i]=0

OUTPUT:
backup-round[i]←1 AND round[i]←1
```

**Put the incoming pointer in the comparison blocks and update**

The blocks required:

```
Index definition:
Let i∈ [l]

Block definition:
   (received-pointer[i], update-pointer[i])
```

Instructions:

```
INPUT:
received-pointer[i]=1 AND update-pointer[i]=0

OUTPUT:
received-pointer[i]←1
```

```
INPUT:
received-pointer[i]=1 AND update-pointer[i]=1

OUTPUT:
pointer[i]←1 AND compare-u-id[i]←1
```

```
INPUT:
received-pointer[i]=0 AND update-pointer[i]=1

OUTPUT:
compare-u-id[i]←1
```

### E.5.2. THE SPECIFICATIONS OF THE TRAINING DATA

Based on the defined binary variables, one can calculate the dimension of the binary vector before input encoding, denoted by $k$. By simply counting the number of bits, we obtain

$$k = 7D^2 + 113l + 38Dl + 2D + 52D^2l + 21.$$

This value is the dimension of the output vector, as the output is not in encoded form. The variables in the message section, $\boldsymbol{x}_M$, are `pointer-slot`, `u-id-slot`, `u-inf-slot`, `round-slot`, `u-dist-slot`, and `message-flag-slot`, with a total of

$$5lD^2 + 5l + D^2 + 1$$

bits. Thus, $k_M$ in Equation (3) is equal to this value.

We obtain the total number of instructions, denoted by $n_T$, by simply counting the number of instructions in each box (each box represents the instructions for the entire range of the indexing variables). This yields:

$$n_T = 5D^2 + 100l + 33Dl + 2D + 44D^2l + 18.$$

Since each instruction is converted into a training sample, the number of training samples and the input embedding dimension before augmentation are both equal to $n_T$.

### E.5.3. MAXIMUM NUMBER OF COLLISIONS

The maximum number of collisions in the instructions is 4. An example is the bits in `u-id`. Since the number of collisions is constant, the size of the training dataset after augmentation with idle samples as well as the input embedding dimension, denoted by $k'$, are proportional to $n_T$. Thus, similar to $n_T$, we have $k' = \mathcal{O}(l \cdot D^2)$.

### E.5.4. INITIALIZATION

To run the algorithm, certain variables must be properly initialized at each node. Let the local ID of a node $u$ be denoted by $u_{\mathcal{L}}$, and its global ID by $u_{\mathcal{G}}$. We denote the source node by $s$, and all other nodes by $u$. The neighbors of a node are denoted by $v_i$ for $i \in [D]$. The weight of the edge between node $u$ and a neighbor $v_i$ will be denoted by $w(u, v_i)$. The total number of nodes is denoted by $n$. The notation $\mathrm{Bin}(z, l)$ denotes the binary representation of the decimal number $z$ using $l$ bits.

Below, we first describe the initialization of variables for the source node $s$, and then for any other node $u$. Note that all variables that are not explicitly mentioned are initialized to zero.

```
Source Intitialization
num-nodes = ← Bin(n, l)
n4round = ← Bin(n, l)
round[1] ← 1
backup-round[1] ← 1
received-pointer ← 1
compare-round[l] ← 1
v-inf-backup[ALL][ALL] ← 1
v-dist[ALL][ALL] ← 1
u-id ← Bin(s_G, l)
determine-slot[s_L] ← 1
v-id[i] ← Bin(v_{i_G}, l) for i ∈ [D]
weight[i] ← Bin(w(s, v_i), l) for i ∈ [D]
```

```
Non-Source Intitialization
num-nodes = ← Bin(n, l)
n4round = ← Bin(n, l)
round[1] ← 1
backup-round[1] ← 1
received-pointer ← 1
compare-round[l] ← 1
v-inf-backup[ALL][ALL] ← 1
v-dist[ALL][ALL] ← 1
u-id ← Bin(u_G, l)
u-inf[ALL] ← 1
u-dist[ALL] ← 1
determine-slot[u_L] ← 1
v-id[i] ← Bin(v_{i_G}, l) for i ∈ [D]
weight[i] ← Bin(w(u, v_i), l) for i ∈ [D]
```

### E.5.5. SUFFICIENT UPPER BOUND ON THE NUMBER OF ITERATIONS

Here, we provide a sufficient upper bound on the number of iterations required by the graph template matching framework, under the instructions described above, to complete the algorithm. In the worst case, during each relaxation round of the Bellman-Ford algorithm, the message that notifies a node to update its distance to the source based on information from its neighborhood may need to be relayed through other nodes $(d_G - 1)$ times before reaching the target node. Here, $d_G$ denotes the diameter of the graph when all edges are assumed to have unit length. Each such relay requires at most $(2l + D^2 + 6)$ iterations.

After receiving the message, a node requires at most $(D^2 + Dl + D + 7l + 16)$ iterations to update its distance and send a message to other nodes, which will eventually reach the next node to be relaxed. For the purpose of a sufficient upper bound analysis, we assume that in each relaxation round, all nodes incur the same worst-case number of iterations. Since the Bellman-Ford algorithm consists of $(n - 1)$ relaxation rounds, a sufficient upper bound on the total number of graph template matching iterations required for the algorithm to complete is given by

$$n(n - 1)\left((d_G - 1)(2l + D^2 + 6) + (D^2 + Dl + D + 7l + 16)\right).$$

