# OpenReview forum: "Learning to Execute Graph Algorithms Exactly with Graph Neural Networks"
_ICML.cc/2026/Conference — ICML 2026 spotlight_

### Official Review · Reviewer_M4cx · 2026-03-02

**Soundness:** 3
**Presentation:** 2
**Significance:** 3
**Originality:** 3
**Overall Recommendation:** 4
**Confidence:** 2

**Summary:**

This paper derives learnability guarantees for a particular choice of GNN architecture, for algorithms representable in the LOCAL model of distributed computation. For this, the paper extends the results of Back de Luca et al. (2025) to GNNs leveraging NTK theory. Empirical results validate the theory on small-scale synthetic instances for (i) infinite-width networks for several graph algorithms, as well as (ii) when using finite-width MLP ensembles for one graph algorithm (message flooding).

**Compliance With Llm Reviewing Policy:**

Affirmed.

**Final Justification:**

See my answer to the author's rebuttal.

**Key Questions For Authors:**

1. Given step-wise training as described in lines 211-218. How is training actually performed given Model 1 and that different applications of an algorithm to different problems leads to different number of instructions. Does the learnability result in Thm 5.1 only concern the application of an algorithm $\mathcal{A}$ to same sized problems (requiring the same number of local instructions)? How does this affect the learnability of $\mathcal{A}$ w.r.t. application to arbitrary-sized problems?
2. From my understanding, the proof of the formal main Thm. D.7 requires the MLPs used in the ensemble to be of infinite width. Is it correct to assume that given a finite-width architecture (incl. finite-width MLPs), the learnability result is not applicable? If yes, I think this should be clearly mentioned in the informal Thm 5.1.
3. W.r.t. which variable/parameter are Thm 6.1 - 6.4 high probability statements?
4. Given using an NTK predictor instead of MLP ensembles, why are the experiments limited to such very small graphs?

Given my questions and presentation-related weaknesses are addressed, I'm ready to raise my score. For raising my score, I'm not expecting the authors to "change" the applicability / practical aspects of the theory.

**Limitations:**

Yes w.r.t. including such a paragraph at the end of the paper. However, authors should be clear on the limitations of their learnability result to infinite-width models / NTK regime (as well as large ensemble sizes) when presenting their "informal" Theorems.

**Strengths And Weaknesses:**

**Strengths:**
* Except for the cases noted in the weaknesses, the work is well written and well-positioned w.r.t. prior work.
* Proving that certain algorithms can not only be theoretically expressed but actually learned is highly relevant and impactful.
* Building on Back de Luca et al. (2025) to derive (better) learnability results of graph algorithms using GNNs is an interesting extension leading to interesting results.

**Weaknesses:**
1. Results seem distant and not applicable to practical model architecture or learning practically sized problem instances
    1. Theory is only applicable to infinite-width networks in the NTK regime.
    2. Empirical (finite-width) results only concern Message Flooding due to the large required ensemble sizes.
    3. Empirical results provided only for very small graph instances with up to 7 nodes.
2. The dependence on the infinite-width assumption is not made clear in the introduction, nor mentioned in the main results in Section 5 & 6. This dependence should be made explicit in these Sections and not (only) mentioned at the very end of the paper. See also my Question 2.
3. The technical relation of the size of the ensemble and approximating the NTK predictor, as well as the implications of this, should be made clearer in the technical part of the main paper. This forms the main part of the empirical evaluation and should be discussed in more detail before.
3. The structure of the presentation of some important concepts to follow the main part of the paper could be improved:
    1. What is meant by templates and where to find a definition for them is not made clear until Section 6. The definition of \Phi_{Enc}, for which Section 4 refers to the appendix already assumes familiarity with this concept, as does the proof outline of the main result in line 218-219 Section 5.
    2. When introducing \Phi_{Enc} in Lines 151-153 in the paper, make a concrete references to its definition (as you note e.g. Lines 174-180).
3. Expanding on evaluating the theory empirically with NTK predictors, also for larger graphs, would the be interesting (see also Question 4).

---

> ### Author Rebuttal · Authors · 2026-03-31
>
> We thank the reviewer for the comments and address them below.
>
> Weakness Comments:
> >W1
>
> Our primary goal is to provide, to the best of our knowledge, the first theoretical exact learning guarantees for graph algorithms using GNNs.
>
> (a) We adopt the NTK framework in the infinite-width regime as it offers the clearest path to such guarantees without additional complexities from finite-width effects (e.g., higher-order kernels as in Section 3 of [1]). Analyzing finite-width behaviour can constitute an independent research direction. Empirically, sufficiently wide networks are expected to approximate infinite-width behaviour.
>
> (b) The purpose of our finite-width experiments is to demonstrate that the theoretical predictions extend to practical regimes when width is large. These observations are not specific to Message Flooding; however, we selected it due to its lower instruction complexity and computational feasibility as we have very limited computational resources.
>
> (c) For Message Flooding, we exhaustively test all labeled trees with n=7. As the number of trees grows exponentially, exhaustive testing for n≥8 was infeasible for us. To compensate, we evaluate 1000 random graphs with n∈[5,20], achieving 100% accuracy for sufficiently large ensembles (Appendix C.2, Fig. 6).
> >W2
>
> In the revised manuscript we will explicitly restate this assumption in the Introduction and Sections 5 and 6, in addition to its current mention in the limitations.
> >W3
>
> We will incorporate the following discussion to the main paper:
> Based on results in Appendix B, the mean output of an ensemble with size K has a Gaussian distribution. The goal is to derive a lower bound on K such that for a δ ∈ (0, 1), with probability 1-δ, i-th coordinate in ensemble mean, Z(i), is rounded correctly for all i. One can verify that if |Z(i) − μ(i)| < |μ(i)|/2, then Z(i) will be rounded correctly by the Heaviside. Applying the standard Gaussian concatenation bound shows K scales like O(k' log(k')) where k' is the input dimension. An application of union bound over the L iterations of an algorithm and n nodes, while considering that the number of all possible test inputs is O(2^(nT)), shows the ensemble size scales like O (k ′ · nT + k ′ log(n· L )), where nT is the number of templates and is proportional to k'.
>
> >W4
>
> We will improve clarity as follows:
> (i) Expand the definition of “template” in Section 5 and reference its formal definition in Appendix D.1.
> (ii) Expand the explanation of \Phi_{Enc} in Section 4 and reference its formal definition in Appendix D.4.1.
> >W5
>
> We evaluate Message Flooding on 1000 random graphs with n∈[5,20], using both the NTK predictor and MLP ensembles, achieving exact learning (Appendix C.2, Fig. 6).
> >Q1
>
> For example, assume an algorithm involves copying an L bit ID. There are L instructions, one for each bit. This is enough for copying all the 2^L possible ID's. Assuming the ID is a unique rank given to each node, the same set of instructions will work for graphs of size up to 2^L. Each instruction forms a training sample (see Appendix E.1), and models are trained via gradient descent with MSE loss. The size of the instructions for an algorithm depends only on bit precision and maximum degree D. This fixed set, will execution the algorithm on all graphs with sizes supported by the bit precision. Thus, Theorems 5.1 and 6.1–6.4 assume a fixed instruction set for a certain algorithm on problems with various graph sizes within the bit precision and the maximum degree constraints. We would like to note that this type of constraints on precision also exists when, for example, running an algorithms in an operating system.
> >Q2
>
> We will restate this assumption in informal Theorem 5.1. While our theory is derived in the infinite-width regime, sufficiently wide finite models empirically match this behaviour (Section 7.2, Fig. 4; Appendix C.2, Fig. 6). Extending guarantees to finite width via NTK is possible but requires handling higher-order kernels and can be a separate research work.
> >Q3
>
> The high-probability guarantees in Theorems 6.1–6.4 are over random initialization of MLP ensembles. Using the NTK predictor removes this randomness, yielding exact (deterministic) guarantees (Appendix C.1).
> >Q4
>
> We evaluate Message Flooding on 1000 random graphs (n∈[5,20]) using both NTK predictor and MLP ensembles (Appendix C.2, Fig. 6). While scaling experiments with NTK predictor is not limited by ensemble size, larger graphs increase the required iterations. Furthermore For BFS, DFS, and Bellman-Ford, input/output dimensions also scale, making experiments computationally expensive. Beyond our limited computational resources, there is no fundamental barrier to scaling.
>
> ---
> We hope that our responses address the reviewer's concerns and they kindly consider improving our score.
>
> Reference:
> [1] Gilmer, Justin, et al. "Neural message passing for quantum chemistry." International conference on machine learning. PMLR, 2017.

---

> > ### Author Rebuttal · Reviewer_M4cx · 2026-04-01
> >
> > I want to thank the authors for their answers. My concerns have mainly been addressed, however, only under the condition that the authors include the mentioned changes in the presentation, in particular, clearly stating the infinite-width assumption in the introduction and Section 5 & 6. While I think that the empirical results are quite unsatisfying (only being able to implement one single graph algorithm and that only on graphs of size at most 20), the theoretical results could be a first step towards more practical results, which could justify acceptance. Thus, I'm cautiously increasing to 4.

---

> > > ### Author Response · Authors · 2026-04-01
> > >
> > > We sincerely thank the reviewer for their time and their overall positive assessment of our work. Their feedback has been invaluable in helping us improve the quality of our paper. We would like to assure them that the suggested changes will be incorporated into the revised manuscript.

---

### Official Review · Reviewer_SGa6 · 2026-03-04

**Soundness:** 2
**Presentation:** 3
**Significance:** 2
**Originality:** 2
**Overall Recommendation:** 4
**Confidence:** 3

**Summary:**

This paper proves that Graph Neural Networks (GNNs) can learn to exactly execute distributed algorithms from the LOCAL model on bounded-degree graphs. Previous approaches show that neural networks can approximate Turing machines, but often suffer from error accumulation over iterations.

The proposed architecture consists of an ensemble of infinite bandwidth Multilayer Perceptrons (MLPs) trained locally on a shared synthetic dataset. These MLPs are then embedded into a GNN framework where the aggregation and communication steps are fixed and follow the input graph structure. The authors demonstrate that for any algorithm in the LOCAL model, there exists a training set of size quadratic in the maximum degree that guarantees exact computation with the same order of iterations.

The paper further validates this on fundamental distributed algorithms, BFS, DFS, Bellman-Ford, and Flooding, explicitly providing the specific training sets needed for the MLPs. For Flooding in particular, the architecture scales to arbitrary graph sizes provided the maximum degree is bounded, as it does not rely on global identifiers.

**Compliance With Llm Reviewing Policy:**

Affirmed.

**Final Justification:**

The authors' rebuttal successfully addressed my concerns. They agreed to clarify the infinite-width MLP assumption in the main theorems. While the theory appears solid, the practical significance is only fair. I have raised my score to Weak Accept.

**Key Questions For Authors:**

- What is the additional work required to prove the theorems in Section 6 given Theorem 5.1? Why didn’t you name them Corollaries?
- What is the difference of your dataset from the one used in Nerem et al. (2025)?

**Limitations:**

Yes.

**Strengths And Weaknesses:**

The presentation is good, effectively connecting previous work with this contribution. There are a few aspects that could be improved. Firstly, the citation ‘Back de Luca et al., 2025’ maps to two different papers in the bibliography. Secondly, the citation to Malach (2023) is not up to date. More importantly, the authors do not sufficiently argue for the significance of this line of work on exact learning.
While I can see that proving exact learnability of exact algorithms in GNNs can be a significant area of research, the paper could provide few lines to motivate it.

While the paper provides proof for all its statements in the appendix, it does not sufficiently explain the technical difficulty of its results in the main paper. This makes it difficult to evaluate the paper’s significance and understand the intuition behind the proofs.
Compared to previous work, this paper greatly improves the scalability of exact learnability. However, despite its significant contributions, the paper’s originality is limited. The GNN’s message passing approach naturally lends itself to distributed computing models such as LOCAL. Therefore, the main contribution is to provide exact learning of local node processes, which seems to rely mostly on the work of Back de Luca et al., 2025.

I am satisfied with the way the authors present the strengths and weaknesses of their architecture, with one exception. In fact, the use of infinite-width MLP is not mentioned in the theorems. This fundamental omission can mislead the reader.

---

> ### Author Rebuttal · Authors · 2026-03-31
>
> We thank the reviewer for the comments and address them below.
> > ...aspects that could be improved...
>
> In the revised version, references to “Back de Luca et al., 2025” will be separated into “2025a” and “2025b”, and Malach (2023) will be updated to its ICML version.
> > ... significance of this line of work ...
>
> We will add the following discussion to the introduction:
> A central question in machine learning is what an architecture can or cannot learn. Prior work such as [1] derived negative results regarding the limitations of GNNs in learning some classical graph algorithms and have also shown that message-passing GNNs are as expressive as the LOCAL model. But they fall short of providing positive exact learning results.
> Other works highlight the importance of exact learning for reasoning tasks [2,3]. We provide a training method and an architecture that guarantee exact learning of any LOCAL model using SGD, which to our knowledge, is the first such result.
>
> Prior work adopted exact algorithms and step-by-step executions to demonstrate the learning power of neural networks because: (i) execution is interpretable and errors are traceable; (ii) labels are precisely computable, avoiding noise; (iii)  exact computability of ground-truths for out-of-distribution inputs, facilitates reliable evaluation of generalization [4,5].
> > ...difficult to evaluate the paper’s significance ...
>
> Intuition and Technical Novelties:
> We propose a novel framework that, for the first time, extends exact instruction learning from non-graph feedforward networks to graph algorithms with GNNs. The framework introduces (i) a local template-matching function, f, and (ii) a new message-passing mechanism where nodes share partial features with neighbours via communication channels defined by distinct local IDs within each node’s 2-hop neighbourhood. This design is original and novel.
>
> Theorem 5.1 provides a general learning result for algorithms in the LOCAL model. Since LOCAL computations are Turing-complete, we introduce a local Turing machine (LTM) with a structured tape (Sec. D.3.1). We then construct instructions for f, which initializes the tape, executes the LTM step-by-step, and writes outputs back to storage and communication slots (Sec. D.3.2). We leverage Back de Luca et al. (2025) to show NTK predictors can learn f. Then we use concentration bounds to replace NTK with an ensemble of MLPs.
>
> We further present learning results for four fundamental graph algorithms (Theorems 6.1–6.4). Instead of reducing them to LOCAL+LTM, we directly construct bit-wise instructions (Secs. E.2–E.5), empirically verify correctness, and derive bounds on iteration complexity and ensemble size.
>
> Comparison with Prior Work:
> We do not improve scaling; rather, we provide the first known results on exact learning of graph algorithms with GNNs. Prior work (Back de Luca et al., 2025a) is limited to feedforward networks on single inputs and does not address graph algorithms. Thus, our work is an original contribution to the theory of exact learning with GNNs.
> > ... use of infinite-width MLP ...
>
> We will state this assumption earlier in the manuscript in addition to current mentioning of it in the limitations section.
> > ... name them Corollaries?
>
> Results in Section 6 are not based on LOCAL model of these algorithms (see Section 6, lines 248-258). For each algorithm, distinct instructions are constructed, as provided in Sections E.2- E.5 of the appendix. These instructions are part of the constructive proof for each theorem. Based on independent analysis of these instructions, the dataset size, dimensions, required iterations of the framework, and the ensemble complexities are derived for each Theorem.
> > ... Nerem et al. (2025)?
>
> The result in Nerem et al. (2025) is limited to Bellman-Ford (BF) shortest path. They pick an architecture that is highly aligned with the BF update rule (e.g., min aggregation). Thus, it might not work for all algorithms. Their approach to training is similar to dynamic programming: they select a limited set of simple graphs, introduce a specific L1-regularized loss, and show that if their model achieves zero loss on the training set, it will correctly execute BF on larger graphs. However, it falls short of providing a theoretical guarantee that training via gradient descent will reach that point.
> In contrast, our framework is not specific to any particular algorithm. The architecture in Eq. (2) remains the same for any algorithm. Our training dataset consists of instructions (see Appendix E.1 ), not graph-structured data. Our loss is a simple MSE. Crucially, using NTK theory, we provide theoretical guarantees for exact learning via gradient descent.
>
> ---
> We hope that our responses address the reviewer's concerns and they kindly consider improving our score.
>
> [References (click)](https://anonymous.4open.science/r/icml2026_rebuttal-0E0B/r3_ref.jpg)

---

> > ### Author Rebuttal · Reviewer_SGa6 · 2026-04-03
> >
> > I thank the authors for their rebuttal, which addresses my concerns. Provided that no substantial weaknesses remain after the discussion with the other reviewers, I will raise my score.

---

> > > ### Author Response · Authors · 2026-04-05
> > >
> > > We sincerely thank the reviewer for their time and for their overall positive assessment of our work. Their feedback has been invaluable in helping us improve the quality of our paper. We are pleased to note that all reviewers have confirmed that their concerns have been adequately addressed during the rebuttal process. Given that the reviewer kindly indicated they would raise their score provided that all concerns are resolved, we would be very grateful if they could now consider raising their score accordingly. We truly appreciate their thoughtful consideration and continued support.

---

### Official Review · Reviewer_yKJ1 · 2026-03-13

**Soundness:** 3
**Presentation:** 3
**Significance:** 2
**Originality:** 2
**Overall Recommendation:** 5
**Confidence:** 3

**Summary:**

The article focuses on GNNs' ability to implement certain algorithms (such as Message Flooding) exactly. The key point is that these algorithms can be mapped to the LOCAL computational model and are based on a functional approximation using MLP.

The claims are supported both theoretically and experimentally.

**Compliance With Llm Reviewing Policy:**

Affirmed.

**Final Justification:**

My questions have been answered. Given my already high rating and the valid concerns raised by other reviewers, I’ll keep it.

**Key Questions For Authors:**

The idea of using machine learning in combinatorial or algorithmic contexts is often offset by the need to generate training datasets using those same methods. Is it feasible to reuse a model trained on one algorithm for another algorithm with minimal retraining costs?

In an applied setting with a given task, how could we learn a dedicated LOCAL algorihm and then implement it without learning ?

Can an approximate version be derived ?

**Limitations:**

yes

**Strengths And Weaknesses:**

Strengths

The paper is well-written and well-structured.
The research question is interesting and has been thoroughly
explored from both theoretical and practical perspectives.
The limitations are also clearly stated, which is a plus.

Weakness
The approach relies on ensembles of MLPs for functional approximation, thereby combining the drawbacks of ensemble methods and MLPs in terms of memory requirements.

There is no reference to the existing GNN architecture literature, but rather a definition based on Equation (2). This point warrants clarification. Perhaps by stating which part of architecture replicate standard MPNN architecture

Note: Since the learning is focused at a preliminary level, the possibility of automatically associating an algorithm with a task or creating a “meta-algorithm” based on the presented work appears complex.

---

> ### Author Rebuttal · Authors · 2026-03-30
>
> We sincerely thank you for all your comments and your positive view of our work. please see our answers below:
> > ...relies on ensembles of MLPs...
>
> Please note that our main objective has been to provide _theoretical_ guarantees for exact learning graph algorithms, with GNNs, using the NTK theory. To empirically assess this theory, one has to approximate the NTK predictor with an ensemble of MLPs. Please note that even a single MLP can be used to approximate the NTK predictor in our architecture, and the execution might still be correct. An ensemble is required only to provide a desired level of probabilistic guarantee for exact execution, where the number of ensemble members depends on the desired probability threshold for correctness.
> >...reference to the existing GNN architecture ...
>
> In the revised manuscript we will add reference to the standard MPNN[1]. as well as a version of the following discussion:
>
> The update relation for the features of node u, x_u, in a standard MPNN for graphs with unattributed edges is as follows: (i) function f_1 is applied to the (x_u, x_v) pairs for all neighbours v, and the results are summed. The aggregated value is taken as the message, m_u. (ii) function f_2 is applied to (x_u, m_u) to update x_u.
> In our framework, x_u has two parts, which are treated differently: a part treated as a message, x_u_M, and a part that is not directly involved in message passing, x_u_C. The update rule is as follows: (i) the sum of the message parts of node u and all its neighbours is taken as the new x_u_M. (ii) the updated x_u is obtained by applying function f to (x_u_C, x_u_M). Note that the updated x_u contains the message that the node will send for aggregation.
> Similarities: Both the standard MPNN and our architecture have an aggregation mechanism that sums features from the neighbourhood. Both also have an update function that updates the local features based on a combination of the message and the local features from the previous round.
> > Since the learning is focused at a preliminary level, the possibility of automatically associating an algorithm with a task or creating a “meta-algorithm” based on the presented work appears complex.
>
> In our work, we provide a systematic way to build instructions for and learn any algorithm expressible in the form of a LOCAL computation model. The first step is to construct the local Turing machine (LTM) that computes the equivalent of the local state and the outgoing message in each round of the LOCAL implementation of the algorithm. The characteristics of such an LTM are provided in Section D.3.1 of the appendix. Once the corresponding LTM is defined, the instructions for learning to simulate the task are based on our construction in Section D.3.2 of the appendix.
> >...machine learning in combinatorial or algorithmic contexts ...?
>
> Classical methods generate a large corpus of training data by executing the algorithms. Nevertheless, as soon as a test case falls outside the training samples, correct inference-time execution is not guaranteed. Unlike the classical approach, in our method, we build a very efficient training dataset by turning the bitwise algorithm instructions into training samples. At inference, the model is able to execute the algorithm exactly for an exponentially larger set of test cases that were not in the training samples. For example, consider Message Flooding: we turn the per-bit copy instruction for each of the L bits into a training sample. At inference, the model works correctly for all possible 2^L messages, and for any graph shape and size as long as constraints such as the maximum degree are met.
> >... learn a dedicated LOCAL algorithm... ?
>
> Assume that an algorithm is expressed in the form of a LOCAL model. Let us denote the operations in the computation step of this model as ALG. The ALG has two outputs: (i) computing the new local state, and (ii) computing the outgoing message.
> To learn this, first an LTM should be defined that computes (i) and (ii). The instruction to execute the LTM are provided in our construction in Section D.3.2 of the appendix. Once the instructions are converted into training samples, the resulting dataset can be used to train the ensemble of MLPs with SGD. The weights should be stored. Later, to implement the model without retraining, the same number of ensembles can be initialized with the recorded weights and used in the architecture defined in Eq. (2).
> >Can an approximate version be derived?
>
> Most of the prior work in this context provides approximate learning results. The main advantage of our work that makes it distinct from prior work is that it guarantees exact execution with arbitrary high probability.
>
> ---
> We hope that our answers are to your satisfaction. Thank you again for your feedback, which helps us improve the quality of our work.
>
> References
>
> [1] Gilmer, Justin, et al. "Neural message passing for quantum chemistry." International conference on machine learning. PMLR, 2017.

---

> > ### Author Rebuttal · Reviewer_yKJ1 · 2026-04-03
> >
> > My questions have been answered.
> > Given my already high rating and the valid concerns raised by other reviewers, I’ll keep it.

---

> > > ### Author Response · Authors · 2026-04-03
> > >
> > > We sincerely thank the reviewer for their time and their positive assessment of our work. Their feedback has been invaluable in helping us improve the quality of our paper.

---

### Official Review · Reviewer_yRZa · 2026-03-14

**Soundness:** 3
**Presentation:** 3
**Significance:** 2
**Originality:** 3
**Overall Recommendation:** 4
**Confidence:** 4

**Summary:**

This paper studies the learnability of distributed graph algorithms in the classic LOCAL model of computation using Graph Neural Networks (GNNs). By mapping the LOCAL model of distributed computing to a "Graph Template Matching" framework, the authors demonstrate that an ensemble of MLPs, using Neural Tangent Kernel (NTK) theory and Heaviside step functions, can simulate Local Turing Machines (LTMs). They provide explicit binary instruction templates for a variety of problems including Message Flooding, BFS, DFS, and Bellman-Ford. They also establish bounds on required embedding dimensions, dataset sizes, and ensemble complexities. Overall, this paper looks into the computational expressivity of GNNs beyond probabilistic approximations. Overall, the work focuses on the concept of exact simulation via orthogonal basis encodings and infinite-width NTK limits.

**Compliance With Llm Reviewing Policy:**

Affirmed.

**Final Justification:**

The authors have carefully answered my concerns and I've raised my score. However, they have an error in their definition of the LOCAL model. I'd like to see them present the correct definition in their final draft.

**Key Questions For Authors:**

Questions:

1) The requirement of $\mathcal{O}(n \cdot D^2 \cdot d_G)$ iterations for BFS is much larger than classical distributed BFS algorithms in the LOCAL model. Is there any way to recover these classical bounds in your framework?

2) The bound on ensemble size $K$ seems quite loose. Is there any way to tighten it?

3) Is there any way to relax the constraint on bounded degree for graphs with a small number of high-degree nodes?

**Limitations:**

yes

**Strengths And Weaknesses:**

Strengths:

1) Investigating the expressivity of GNNs is an interesting problem. The formulation of the Local Turing Machine (LTM) operating on a simulated tape within the node feature vectors is an interesting theoretical bridge between classical distributed computing (the LOCAL model) and modern GNNs.

2) Moving from statistically meaningful probabilistic bounds to exact algorithmic execution using NTK predictor limits is an original approach to the algorithmic alignment literature.

3) The authors exhaustively detail and provide ample supplementary materials for the exact binary templates required for classical graph algorithms; they also rigorously prove each of their theoretical claims.

Weaknesses:

1) The round complexities of their approach is much greater than that of the original algorithms. Standard distributed LOCAL BFS completes in $\mathcal{O}(d_G)$ rounds. However, Theorem 6.2 states the GNN requires $\mathcal{O}(n(l \cdot D + l \cdot d_G + D^2 \cdot d_G))$ iterations. To avoid bit-write collisions, the authors artificially serialize the execution of the LTM across the message slots using a "cascade chain trick." This completely destroys the advantages you gain from simultaneous message passing in the LOCAL model, increasing the number of rounds to a point where the GNN is essentially executing a sequential CPU program.

2) The asymptotic complexity of the given bounds on the ensemble size is also large. For BFS (Theorem 6.2), the ensemble size scales as $\mathcal{O}(l^2 D^6)$. For a graph with a maximum degree of $D=100$ and bit precision $l=32$, this requires training trillions of MLPs to achieve concentration. This renders the "learnability" result purely a theoretical result rather than practical.

3) The results hold only for graphs bounded by maximum degree $D$. Because the message section $x_{u_M}$ explicitly reserves $D^2+1$ slots (derived from 2-hop coloring), the embedding dimension $k'$ scales polynomially with $D$. This approach completely breaks down for real-world graphs which often contain a few high-degree nodes.

---

> ### Author Rebuttal · Authors · 2026-03-31
>
> We thank the reviewer for the comments and address them below.
>
> Weakness Comments:
>
> >1.
>
> We would like to clarify that iteration/round comparison is not appropriate. Bounds in the LOCAL model capture only communication complexity. The entire local computation is abstracted into a single round, even if it requires multiple operations. In contrast, in our paper, local computations are done bitwise and take multiple iterations. Furthermore, each iteration of Eq (2) performs one communication step and only one computation step. To the best of our knowledge, reproducing the LOCAL's separation between communication and computation would require external control mechanisms, which we intentionally avoid. As a result, when a local computation requires multiple iterations, communication cannot be “turned off” during computation, so each iteration for computation also induces a communication.
>
> If we count only non-auxiliary message-passing steps, then our proof technique is tight. For example, the paper already includes a parallel algorithm, namely Message Flooding. In that case, the complexity of our implementation is tight in terms of non-auxiliary message-passing rounds, i.e. O(d_G). The additional iterations arise only because, in steps where the model is effectively carrying out computation, communication still takes place as an unavoidable auxiliary operation.
>
> We would also like to clarify that our construction does not implement the parallel LOCAL-model version of BFS. We implement serial versions of BFS, DFS, and Bellman–Ford. The parallel LOCAL model is used only to establish an expressivity result, as stated in the paper (lines 248–258, col. 2). If a parallel LOCAL version of BFS is implemented in our framework, then the complexity of non-auxiliary message-passing rounds will be O(d_G). This follows by a straightforward adaptation of the Message Flooding construction, with the addition of a few variables storing depth and parent identifiers, which does not change the number of non-auxiliary message-passing iterations.
>
> We would be happy to add these clarifications to the paper.
>
> >2.
>
> We would like to emphasize that our contribution is intentionally theoretical, and we do not view this as a limitation. While much of the prior literature has focused on the expressivity of GNNs, learning guarantees have received comparatively less attention. In particular, other works have expressed the importance of exact learning for reasoning tasks [1,2], and, to the best of our knowledge, ours is the first work to establish such guarantees for graph algorithms.
>
> >3.
>
> We would like to clarify that the dependence on D is not a weakness of the proof technique, but a consequence of working with any general message-passing architecture where neighbourhood information must be encoded into a bounded representation. Hence only finitely many collections of messages can be distinguished after aggregation. Beyond a certain degree, collisions are unavoidable by a pigeonhole argument, and computation may fail.
>
> Task-specific architectures can sometimes avoid this dependence. For example, min-based aggregation aligns naturally with shortest paths[5]. Our goal is instead to analyze a general sum-based message-passing framework capable of expressing multiple algorithms. Thus, some dependence on D is inevitable.
>
> This dependence also clarifies the advantage of GNNs over MLPs. An MLP must encode the whole graph into one vector and therefore cannot avoid dependence on N (number of nodes) even for bounded-degree graphs. A GNN avoids this, but complexity may scale with D; if D itself grows with N, that advantage disappears.
>
>
> Questions:
>
> >1.
>
> Please see our first response for details on iteration count. Briefly, BFS can also be implemented using the same message-flooding principles, and under that implementation, counting only the effective communication rounds yields the same O(d_G) bound.
>
> >2.
>
> Potentially, yes. A sharper bound could be obtained by directly analyzing the Gaussian survival function arising in the NTK regime, instead of using coordinate-wise estimates and a union bound. However, this would require lengthy calculations and is unlikely to yield an equally interpretable expression in which the roles of the main algorithmic parameters remain transparent.
>
> >3.
>
> To be efficient for specific settings such as the one suggested by the reviewer, the communication mechanism of the architecture needs to be changed and be specialized to the setting. Although doable, it might not be a viable approach to have specialized architectures for different settings.
>
> ---
> We hope that our answers address reviewers concerns and they kindly consider improving our score.
>
>
> [References (click)](https://anonymous.4open.science/r/icml2026_rebuttal-0E0B/r1_ref.jpg)

---

> > ### Author Rebuttal · Reviewer_yRZa · 2026-04-03
> >
> > Thanks to the authors for their detailed responses to my concerns. There is one misconception I'd like to clear up regarding the LOCAL model of computation. The authors state: "We would like to clarify that iteration/round comparison is not appropriate. Bounds in the LOCAL model capture only communication complexity." This statement is not true. The *only* measure of efficiency that the LOCAL model (as defined in the originally cited papers on distributed computation: Angluin, 1980; Linial, 1992; Naor & Stockmeyer, 1993) captures is **number of rounds of communication**. It does not capture communication complexity. In fact, the LOCAL model allows for unbounded message size. However, I do understand what the authors mean to say in this response and they have sufficiently answered my questions and concerns. I will increase my score.

---

> > > ### Author Response · Authors · 2026-04-05
> > >
> > > We sincerely thank the reviewer for their time and their overall positive assessment of our work. Their feedback has been invaluable in helping us improve the quality of our paper.

---

### Decision · Program_Chairs · 2026-04-30

**Decision:**

Accept (spotlight)

**Comment:**

The reviewers unanimously recommend acceptance of the paper with varying degrees of strength. I agree with their assessment and am happy to recommend acceptance of this paper. Several reviewers mentioned that the paper is taking an interesting approach and is well-written. The reviewers' concerns have largely been addressed in the discussion period. The reviews and the rebuttal have given rise to several interesting points and results that I strongly encourage the authors to include in their revised manuscript.